# Review of the land snails of the genus *Kora* from Brazil, with description of eight new species and a new related genus *Koltrora*, including comparison with two Andean *Neopetraeus* species (Gastropoda, Eupulmonata, Orthalicoidea)

**Luiz Ricardo L. Simone** *

Museu de Zoologia da Universidade de São Paulo, São Paulo, SP, Brazil

* lrsimone@usp.br, lrlsimone@gmail.com

**Data Availability Statement:** All relevant data are within the paper.

## Abstract

The orthalicoidean genus *Kora* Simone, 2012 is reviewed. Three of the four known species are redescribed, including their anatomy. These species are *K. corallina* (the type species), *K. nigra*, and *K. rupestris*. Eight new species are introduced, all of which occur in the region of the São Francisco River, from the northern of Minas Gerais to the southern of Bahia, Brazil. They are *K. tupan*, *K. ajar*, *K. aetheria*, *K. jimenezi*, *K. uhlei*, *K. kremerorum*, *K. vania*, and *K. curumim*. All of them are described, including anatomical features, except the last three, which are based solely on shell characters. Another related genus is also described, *Koltrora*, with a single new species, *K. pyrostoma*. They are compared to an Andean genus, *Neopetraeus* (*N. lobbii*, *N. tesselatus*), which also exhibit similarities. This detailed phenotypic study was performed in several comparative ways, including a morphological phylogenetic approach, using other orthalicoideans with the same level of phenotypic details known. The single objective is to justify the current taxonomic scheme, and to provide a brief comparison with recent results based on molecular approaches. According to these preliminary results, the triad *Neopetraeus-Koltrora-Kora* is monophyletic, supported by 10 synapomorphies. *Koltrora* and *Kora* are sister taxa, supported by 11 synapomorphies. *Kora* is monophyletic, with strong support from 25 synapomorphies. Discussions on classification, phylogeny, anatomy, and comparison with other recent orthalicoidean literature are also included. Some newly identified and diagnostic structures are described, such as the odontophore pair of muscles m8, the accessory albumen chamber, a different kind of spermatophore, the exclusive kind of radula of *Kora* and *Koltrora*, and a calcified epiphragm, a rare feature in South American snails. Register ZooBank: urn:lsid:zoobank.org:pub:2F13D53C-0A36-42FB-B936-D80E7C958259.

**Funding:** The author(s) received no specific funding for this work.

**Competing interests:** NO authors have competing interests

## Introduction

The Orthalicoidea Brazilian genus *Kora* Simone, 2012 [1] was introduced in 2012 to comprise a single species, *K. corallina* Simone, 2012 ([1]: 432), described only based on shell characteristics in specimens collected from Santa Maria da Vitória, Bahia. Three years later, three new species were described [2], including *K. nigra* Simone, 2015 (Carinhanha, Bahia). The other two species were afterwards transferred to another genus in a paper [3] that also described *K. rupestris* Salvador & Simone, 2016 (also from Carinhanha region, Bahia) and included an emended diagnosis of the genus.

The genus *Kora* was, thus, until recently attributed to the aforementioned three species that habit semidry regions of Bahia, Brazil. Morphologically, taxa in this genus are mainly characterized by a medium-sized, relatively monochromatic shell, with color varying from brown to beige-orange, with paler areas [1, 3]. The protoconch has about two whorls, it is initially smooth, grading into sinuous axial/colabral ribs. Teleoconch sculptured varying from growth lines up to axial ribs. Peristome expanded laterally, located slightly away from the uniform growth shell whorls. There is a clear oblique columellar lamella. The umbilicus is wide and open. The first anatomical information was published concurrently with the analysis presented in this paper [4], within a study describing a new species–*K. arnaldoi* Pena, 2024, and the anatomy of a *K.* cf. *rupestris* (see Discussion), both originating from the region of Itacarambi, Minas Gerais, Brazil.

Originally, *Kora* was described as belonging to the family Bulimulidae. However, a recent molecular study found that the genus forms a paraphyletic group with the genera *Thaumastus* Martens, 1860, and *Megaspira* Lea, 1836 [5]. The authors of that study designated this paraphyletic grouping as "Megaspiridae," and this classification has been adopted by other researchers (e.g., [6]).

Since 2016, several new specimens of land snails have been collected and brought to study by the team of collectors associated with the naturalist José Coltro Jr., in a project studying malacofauna from Bahia and Minas Gerais regions of Brazil in which the soil is mostly calcareous. A diversity of forms of several land snails has been collected. This includes some new *Kora* samples from known and unknown species and many with soft parts, permitting anatomical investigations. Similar collection and study of eupulmonates in the same project include description of the urocoptid *Habeas* Simone 2012, with eight species [7, 8], and the strophocheilid *Anthinus* Albers, 1850, now with eight species, and a, at that time, new genus *Catracca* Simone, 2022 [9]. The calcareous region of Brazil, free from the typical acidic soils, is home to numerous rivers, valleys, and caves that have isolated snail populations for millions of years, resulting in many endemic species. With ongoing economic exploitation in the area, it is crucial to describe these species promptly to highlight their endemism and the need for conservation.

Analysis of collected material from the Coltro team has enabled the redescription and the revision of the genus *Kora*, which is the main objective herein, including the redescription of the three of the four known (and the type) species, including a complete anatomical description of topotypes and other specimens. Six new species are introduced, four of which are also anatomically described. A correlated new species, which has some similarity with *Kora*, is also described in the new genus *Koltrora*, and includes anatomical characteristics necessary for diagnosis. The literature survey revealed that some specimens of *Kora* were already studied, and misidentified as *Neopetraeus* Martens, 1885 [10]. Unfortunately, that material is no longer available, destroyed by the fire at the National Museum of Rio de Janeiro in 2018, and both authors are deceased. Aiming to establish that *Neopetraeus* and *Kora* are different taxa, samples of two species of the former are here also included. *Neopetraeus* is an Andean genus,

which includes 15 valid species [6]. It has some resemblance to *Kora* in having a deep, open umbilicus and a projected peristome relatively far away from shell axis. However, its characteristic protoconch is differently sculptured and it has more shell colors [11], both features already set both genera apart. An initial phylogenetic analysis on a sample of Bulimulidae and allies are also performed. It is based on the data surveyed in the present paper and in papers possessing equivalent anatomical information.

## Material and methods

### Material

This study is based on material collected in expeditions commented above, now housed in MZSP, including dry shells and preserved samples in 70% EtOH. All listed specimens were used to compose the descriptions; those that are only shells (sh) were examined at a conchological level; all complete specimens (shell and soft parts) (spm) were extracted and dissected. Table 1 summarizes the total number of specimens for each studied species, including shells, dissected individuals, and measured shells. It is important to note that no statistical analysis is provided at this stage; the Table is intended solely to give an indication of size and proportions. For the phylogenetic analysis, it was necessary to categorize certain measurements. In these cases, the average value was used, and in some instances, the average was rounded to the nearest whole number, as indicated in the provided tables.

The material studied and described in this work was collected by a team working for Femorale, a private company [www.femorale.com; http://www.femorale.com/femorale/index. asp]. The places of collection are not within protected areas, and as such collection activity did not require special permits. Nonetheless, the collections were made under general/permanent license IBAMA-Sisbio #10560–2, which permits extraction of wildlife samples for scientific purposes. As most of the studied material was collected by non-scientific expeditions, no further data beyond coordinates and place names were available. Thus, details on vegetation, climate, soil, rainfall, etc., were not available, but, when possible and relevant, these data were extracted from the literature, digital online resources, or official websites.

**Table 1. Summary of number of specimens.**

| species | Shells examined | Specimens dissected | Shells measured* |
|---|---|---|---|
| *Kora corallina* | 106 | 6 | 15 |
| *Kora nigra* | 27 | 4 | 14 |
| *Kora rupestris* | 70 | 5 | 12 |
| *Kora tupan* | 5 | 4 | 5 |
| *Kora ajar* | 58 | 15 | 14 |
| *Kora aetheria* | 30 | 5 | 13 |
| *Kora jimenezi* | 115 | 2 | 15 |
| *Kora uhlei* | 23 | 8 | 12 |
| *Kora kremerorum* | 22 | — | 14 |
| *Kora vania* | 3 | — | 3 |
| *Kora curumim* | 1 | — | 1 |
| *Koltrora pyrostoma* | 55 | 11 | 15 |
| *Neopetraeus lobbi* | 2 | 1 | 2 |
| *Neopetr. tesselatus* | 2 | 2 | 2 |

*Just to give an idea of size for taxonomy, no statistical intentions except for basic average.

## Methods and scope

Photos were obtained by digital cameras, either hand-held or attached to the dissecting microscope. Shell measurements were obtained with digital caliper for a number of specimens reported in Table 1. In the reported measurements, the first parameter of each shell is the length, the second it the width. Specimens were dissected by standard techniques [12] under dissecting stereomicroscopes, with the specimen immersed under the fixative. All drawings were obtained with the aid of a camera lucida; initially penciled, afterwards inked; usually drawings produced for each species include data derived from several specimens, as they have exhibited minimal intraspecific variation. Thus, the anatomical drawings are a composite of all examined specimens. To draw specimens to scale, a ruler was positioned at the side of each specimen. The type and voucher material are mainly deposited in Museu de Zoologia da Universidade de São Paulo (MZSP) malacological collection, with some duplicates to other indicated museums. Specimens were usually easily extracted from their shells, except for few cases (e.g., holotype of *Koltrora pyrostoma* and *Neopetraeus cremnobates*), in those cases a 'cesarean' (a small window) needed to be excised in the last whorl of the shell using a small saw. For SEM work, the radula or jaw was cleaned with potassium hypochlorite 5%, glued to conductive double-sided tape, with stubs coated by gold, and examined in the Laboratory of Electron Microscopy of the MZSP [12]. Anatomical terminology, particularly of the odontophore muscles, follows Simone [12], which has been subsequently explained in Malacopedia project http://www.moluscos.org/malacopedia_previous.html. In the present descriptions the anterior genital structures like the penis and vagina, are examined and described in retracted condition, and thus the structures present in the internal lumen is called "internal", despite all of them are everted, and become external, during the copulation. For comparison of the presently studied species with those already known, the large MZSP collection was consulted. Collections of other European and American museums were also consulted while seeking type specimens, some of them illustrated in a catalogue [13]. The present paper has its style, model and disposal of items entirely based on a previous similar paper in this journal [9]. The degree of fusion of both odontophore cartilages are expressed in percentage, and are calculated comparing the length of the cartilages with the length of the fused portion.

The present paper is almost entirely performed in a comparative context. The descriptions compare each taxon with the first description, highlighting the differences, and only showcasing some more important similarities. The set of characters that defines each species is reported in the diagnoses. Furthermore, the main differences among the studied species (new or not) are synthetically exposed in Table 2, concerning conchology, and in Table 3, concerning anatomy. Moreover, the phylogenetic analysis, explained below, exposed and discussed in other sections, is also comparative approach. As explained above, its main concern is the comparison of the species studied herein with some others studied at the same level of anatomical detail, rather than aiming to be the "phylogeny of the orthalicoideans."

The Discussion section, therefore, presents a formal taxonomic comparison of the genera and studied species. However, it necessarily needs to be complemented by the above-mentioned texts and tables. No exclusive diagnostic character was obtained at the genus or at the species levels, except for few cases only reported in the Diagnoses. Actually, what is diagnostic is always a set of characters, and this set is reported in the respective diagnoses.

## Phylogenetic analysis

The phylogenetic methodology is the same as reported by Simone [9, 12, 14], that basically consists of the morphological matrix (in Nexus), analyzed by programs TNT and PAUP (details below). All analyses resulted in a single cladogram. The present preliminary phylogeny

**Table 2. Synoptic table of main conchological differences among studied species.**

| sp\| charact | shell size range mm* | longer than wide (average) | dorso-ventral flattened | aperture % length (average**) | aperture % width (average**)) | horizontal end in outer lip | middle fold in inner lip |
|---|---|---|---|---|---|---|---|
| *Kora corallina* | (35.5–*41.5*–46.2) / 45 | 2.3 | no | 44 | 70 | no | yes |
| *Kora nigra* | (31.5–*34.5*–40.3) / 30 | 1.6 | no | 48 | 60 | yes | yes |
| *Kora rupestris* | (44.2–*45.1*–48.0) / 45 | 2.3 | no | 50 | 66 | no | yes |
| *Kora tupan* | (47.0–*53.9*–56.8) / 55 | 1.9 | yes | 54 | 70 | yes | yes |
| *Kora ajar* | (43.0–*52.5*–54.1) / 55 | 1.7 | no | 54 | 71 | yes | yes |
| *Kora aetheria* | (28.7–*30.7*–33.6) / 30 | 2.0 | yes | 45 | 60 | no | no |
| *Kora jimenezi* | (38.6–*42.5*–43.8) / 45 | 2.0 | no | 45 | 70 | yes | yes |
| *Kora uhlei* | (38.8–*42.7*–43.6) / 45 | 2.0 | no | 50 | 65 | no | yes |
| *Kora kremerorum* | (40.8–*44.8*–48.6) / 45 | 1.9 | yes | 49 | 70 | yes | yes |
| *Kora vania* | (34.7–*36.1*–38.8) / 30 | 1.9 | no | 46 | 65 | yes | yes |
| *Kora curumim* | (27.7) / 30 | 1.7 | no | 50 | 65 | no | no |
| *Koltrora pyrostoma* | (27.2–*30.5*–32.5) / 30 | 1.8 | yes | 50 | 57 | no | no |
| *Neopetraeus lobbii* | (40.2–*42.3*–44.4) / 55 | 2.2 | no | 45 | 60 | no | no |
| *Neopetraeus tesselatus* | (36.9) / 45 | 1.5 | no | 57 | 65 | no | no |

* Above: average and range, below arbitrary size category for analysis; check N in Table 1.

** Rounding the average number to the next integer number

is based upon already published morphological data of 9 species listed below, as well as additional examination of their voucher material deposited in MZSP unstudied structures, such as, e.g., the odontophore. The list of characters is in **S1 Appendix**; and respective matrix in **S2 Appendix**. In **S1** Appendix each character starts with a descriptive sentence, followed by plesiomorphic and apomorphic states; in parenthesis the taxa that possess each apomorphic

**Table 3. Synoptic table of main anatomical differences among studied species.**

| sp| charact | mb folds | venation L from cv | anterior end of cv | strong venation L of cv | kidney lobe | insertions cl | insertions cr |
|---|---|---|---|---|---|---|---|
| *Kora corallina* | low | pair intercal | branched | near PN | several entire | 13 | 7 |
| *Kora nigra* | narrow pointed | pair post | simple | up to 1/2 pu | single broad surround | 7 | 7 |
| *Kora rupestris* | narrow pointed | pair intercal | simple | almost absent | 4 anterior | 7 | 7 |
| *Kora tupan* | wide point | pair intercal | branched | up to 1/2 pu | single broad surround | 8 | 6 |
| *Kora ajar* | wide point | pair intercal | branched | up to 1/2 pu | 3 anterior | 5 | 4 |
| *Kora aetheria* | wide projetct | pair intercal +Y | branched | up to 1/2 pu | several entire | 5 | 7 |
| *Kora jimenezi* | wide round | no | simple | near PN | several entire | 3 | 3 |
| *Kora uhlei* | wide round | pair intercal | branched | up to 1/2 pu | several entire | 3 | 3 |
| *Koltrora pyrostoma* | wide round | pair intercal | simple | up to 1/2 py | several entire | 4 | 2 |
| *Neopetraeus lobbii* | no | no | branched | up to 1/2 py | 4 anterior | 3 | 4 |
| *Neopetraeus tesselatus* | no | no | branched | near PN | 5 anterior | 3 | 4 |
| sp| charact | medial branch differentiated | m1l | m1v pairs | m3 | m2a | dp post duct dg | dd ant duct |
| *Kora corallina* | no | no | 2 | pair esoph | no | R branches | both branch |
| *Kora nigra* | yes | no | 2 | pair esoph | no | Y-shaped | R branch |
| *Kora rupestris* | yes | yes | 2 | pair m2 | no | R branches | both branch |
| *Kora tupan* | yes | yes | 0 | pair esoph | no | R branches | R branch |
| *Kora ajar* | yes, asymm | no | 0 | pair esoph | yes | R branches | R branch |
| *Kora aetheria* | yes | yes | 1 | pair esoph | no | R branches | both branch |
| *Kora jimenezi* | no | yes | 0 | 0 | yes | R branches | R branch |
| *Kora uhlei* | yes, post | yes | 0 | pair esoph | no | R branches | R branch |
| *Koltrora pyrostoma* | no | yes | 0 | 0 | no | single | double* |
| *Neopetraeus lobbii* | no | yes | 1 | 0 | no | single | R branch |
| *Neopetraeus tesselatus* | no | yes | 1 | 2 pairs longit | no | single | R branch |
| sp| charact | jaw | sa localiz | sa form | oc fusion % (average*) | m4-m5 | m7 | m8 |
| *Kora corallina* | central notch | middle 1/3 | 0 | 75 | m4 covering m5 | narrow, single origin | wide |
| *Kora nigra* | rectangular | middle 1/3 | 0 | 60 | m4 covering m5 | narrow, single origin | wide |
| *Kora rupestris* | central notch | middle 1/3 | 0 | 75 | m4 covering m5 | narrow, single origin | wide |
| *Kora tupan* | rectangular | post 1/3 | 0 | 75 | m4 as continuation of m5 | 3 bundles | wide |
| *Kora ajar* | rectangular | middle 1/3 | 0 | 75 | m4 covering m5 | narrow, single origin | wide |
| *Kora aetheria* | central notch | middle 1/3 | 0 | 75 | m4 covering m5 | narrow, single origin | wide |
| *Kora jimenezi* | arched | post 1/3 | papilla | 75 | m5 covering m4 | broad, thick | narrow |
| *Kora uhlei* | rectangular | middle 1/3 | papilla | 90 | m4 as continuation of m5 | filiform | wide |
| *Koltrora pyrostoma* | rectangular | post 1/3 | 0 | 100 | m5 covering m4 | 2 separated stripes | 0 |
| *Neopetraeus lobbii* | central notch | post 1/3 | zigzag fold | 50 | single mass | 0 | 0 |
| *Neopetraeus tesselatus* | central notch | post 1/3 | 0 | 70 | single mass | 0 | 0 |
| sp| charact | m10 | m11 | hd curved at end | ca-carrefour | ca-duct | ca bulged portion | ca insertion |
| *Kora corallina* | narrow | absent | yes | conic | wide-short | 0 | in ad |

(*Continued*)

**Table 3.** (Continued)

| sp\| charact | mb folds | venation L from cv | anterior end of cv | strong venation L of cv | kidney lobe | insertions cl | insertions cr |
|---|---|---|---|---|---|---|---|
| *Kora nigra* | narrow | absent | no | elongated | narrow-long | 0 | betw ad-ac |
| *Kora rupestris* | narrow | absent | yes | entire narrow | narrow-long | 0 | tip eo |
| *Kora tupan* | broad | absent | yes | conic | narrow-long | yes-narrow | betw ad-ac |
| *Kora ajar* | narrow | absent | yes | conic | narrow-short | 0 | in ac |
| *Kora aetheria* | broad | absent | yes | conic | narrow-long | 0 | betw ad-eo |
| *Kora jimenezi* | filiform | absent | yes | conic | narrow-short | yes-wide | in ac |
| *Kora uhlei* | narrow | absent | yes | conic | narrow-long | yes-narrow | betw ad-ac |
| *Koltrora pyrostoma* | broad | absent | no | entire narrow | narrow-long | 0 | tip eo |
| *Neopetraeus lobbii* | broad | present | yes | conic | narrow-long | yes-small | in ad |
| *Neopetraeus tesselatus* | broad | present | no | conic | narrow-long | 0 | in ad |

| sp\| charact | ac-alb chamb | as -accessory alb chamb | numb of sp | % of pt in eo (average**) | bu- musc bursa duct | % pe of eo (average**) | % bd of eo (average**) |
|---|---|---|---|---|---|---|---|
| *Kora corallina* | curve | present | 1 | 35 | yes | 70 | 90 |
| *Kora nigra* | sac | present | 2 | 35 | yes | 100 | 90 |
| *Kora rupestris* | sac | present | 1 | 50 | yes | 60 | 100 |
| *Kora tupan* | sac | present | 1 | 45 | yes | 85 | 100 |
| *Kora ajar* | curve | present | 1 | 45 | yes | 90 | 90 |
| *Kora aetheria* | sac | present | 1 | 35 | yes | 65 | 80 |
| *Kora jimenezi* | curve | present | 1 | 50 | no | 100 | 80 |
| *Kora uhlei* | curve | present | 1 | 45 | yes | 60 | 80 |
| *Koltrora pyrostoma* | sac | 0 | 1 | 25 | no | 50 | 70 |
| *Neopetraeus lobbii* | curve | present | 2 | 20 | no | 50 | 70 |
| *Neopetraeus tesselatus* | curve | present | 2 | 20 | no | 90 | 90 |

| sp\| charact | vd terminal curve | mp basal musc wall | pf pair longit | um umbrella fold | um # rods | % eh of pe (average***) | ei- epiph fold | pm insertion |
|---|---|---|---|---|---|---|---|---|
| *Kora corallina* | 0 | yes | yes-imbricated | 0 | 0 | 13 | yes | terminal |
| *Kora nigra* | yes | yes | yes-simple | 0 | 0 | 25 | 0 | terminal |
| *Kora rupestris* | 0 | yes | yes-simple | yes | 5 | 15 | 0 | base |
| *Kora tupan* | yes | yes | yes-wings | yes | 3 | 20 | yes | terminal |
| *Kora ajar* | yes | yes | yes-simple | yes | 5 | 25 | yes | base |
| *Kora aetheria* | 0 | yes | yes-imbricated | yes | 6 | 33 | 0 | terminal |
| *Kora jimenezi* | 0 | 0 | yes-simple | 0 | 0 | 25 | 0 | terminal |
| *Kora uhlei* | yes | yes | yes-imbricated | yes | 3 | 25 | 0 | base |
| *Koltrora pyrostoma* | 0 | yes | yes-fused | 0 | 0 | 25 | 0 | subterminal |
| *Neopetraeus lobbii* | 0 | 0 | 0 | 0 | 0 | 33 | yes | terminal |
| *Neopetraeus tesselatus* | 0 | 0 | 0 | 0 | 0 | 20 | yes | terminal |

\* Rounding the average number to the next integer number

\*\* Rounding the average number to the next nearest integer number

\*\*\* Rounding the average number to the next nearest integer number

state are listed, sometimes only a collective name is given, e.g., a genus. It is important to emphasize again that the shown phylogeny is not to be interpreted as "the phylogeny of the Bulimulidae". It has only the intention of demonstrating that the description of the new genus–*Koltrora*–is necessary, the genus *Kora*, helping to determine their placements in light of recent molecular scenarios reported at the Discussion. In the literature, a small set of

orthalicoideans has their anatomy known in sufficient details for an initial phylogenetic inference. They are:

1. *Drymaeus castilhensis* Simone & Amaral, 2018 [15];

2. *D. micropyrus* Simone & Amaral, 2018 [15];

3. *D. currais* Simone, Belz & Gernet, 2020 [16];

4. *Bulimulus sula* Simone & Amaral, 2018 [16];

5. *Sanniostracus carnavalescus* (Simone & Salvador, 2016) [17];

6. *Rhinus botocudus* Simone & Salvador, 2016 [17];

7. *Anctus angiostomus* (Wagner, 1827) [18];

As a remote outgroup, the morphological ground plan of Strophocheilidae, as obtained in Simone's phylogeny (2022 [9], fig 27, node 2), is used. The seven species mentioned above are considered close outgroups but are operationally analyzed as part of the ingroup, which consists of 11 anatomically studied species. This methodology has been previously applied in other studies [9, 12, 14]. In the case of *Anctus angiostomus*, the anatomical description provided by [18] contains some missing data, which are addressed later in this paper. Additionally, two other species, taxonomically distant from Orthalicoidea, are included as far outgroups (listed below) but are also operationally analyzed as part of the ingroup. This approach is adopted to test the monophyly of the orthalicoidean families, at least within the current assembly of species. It's important to note that this assembly does not encompass the full diversity of Orthalicoidea, but, as mentioned earlier, represents a preliminary step, primarily aimed at understanding the taxa studied in this paper, especially in light of the new classification based on molecular approaches [5], which will be examined in the context of the current morphological analysis in the Discussion. As far outgroups, the following taxa are included:

8. *Olympus nimbus* Simone 2010 (Solaropsidae) [19]

9. *Lavajatus moroi* Simone 2018 (Achatinidae/Subulininae) [20]

10. Strophocheilidae ground plan [9] rooting.

Regarding the phylogenetic analysis, both TNT and PAUP software were used to analyze the matrix constructed in Nexus format. The results were consistent across both programs. Similar algorithms were employed, including a random seed search with at least 100 replications, the tree bisection and reconnection (TBR) algorithm for branch swapping, and the retention of all trees found. Since the analysis produced a single cladogram, no additional procedures were required. The cladogram was then examined character by character in both programs to understand the contribution of each trait to the structure of the cladogram. Few ambiguous cases were reviewed individually, and the final cladogram reflects what appeared to be the most biologically plausible arrangement. Further details can be found in references [9, 12, 14].

## Nomenclatural acts

The electronic edition of this article conforms to the requirements of the amended International Code of Zoological Nomenclature, and hence the new names contained herein are available under that Code from the electronic edition of this article. This published work and the nomenclatural acts it contains have been registered in ZooBank. ZooBank LSIDs (Life Science Identifiers) can be resolved, and the associated information viewed through any standard web

browser by appending the LSID to the prefix "http://zoobank.org/". The LSID for this publication is: urn:lsid:zoobank.org:pub: FC4DD323-EF6A-404B-9755-F124F9DBB6D4. The electronic edition of this work was published in a journal with an ISSN, and has been archived and is available from the following digital repositories: PubMed Central, LOCKSS, ResearchGate. It is important to state that funders had no role in study design, data collection and analysis, decision to publish, or preparation of the manuscript. The author received no specific funding for this work.

## Abbreviations in figures

**aa**, anterior aorta; **ac**, albumen chamber; **ad**, albumen gland duct; **ag**, albumen gland; **an**, anus; **as**, accessory albumen chamber; **au**, auricle; **bc**, bursa copulatrix; **bd**, bursa copulatrix duct; **bg**, buccal ganglion; **bm**, buccal mass; **br**, subradular membrane; **bu**, muscular wall of bursa duct; **bv**, blood vessel; **ca**, carrefour; **cc**, cerebral commissure; **ce**, cerebral ganglion; **cd**, cerebral node; **cl**, left secondary columellar muscle; **cm**, columellar muscle; **cn**, cerebro-pedal and cerebro-pleural connectives; **co**, collar vessel; **cr**, right secondary columellar muscle; **cv**, pulmonary (efferent) vein; **da**, digestive gland anterior lobe; **dc**, dorsal chamber of buccal cavity; **dd**, anterior gastric duct to digestive gland; **df**, dorsal folds of buccal mass; **dg**, digestive gland posterior lobe; **di**, diaphragm or pallial floor; **dp**, posterior gastric duct to digestive gland; **ed**, esophageal dilatation; **ef**, esophageal fold; **eh**, epiphallus; **ei**, epiphallus inner longitudinal fold; **eo**, spermoviduct; **es**, esophagus; **ey**, eye; **fe**, female right lateral sulcus; **fg**, fecal groove; **fo**, free oviduct; **fp**, genital pore; **fs**, foot sole; **ft**, foot; **gm**, genital muscle; **go**, gonad; **gp**, pallial gland; **hd**, hermaphrodite duct; **if**, inner fold of pneumostome; **in**, intestine; **ir**, insertion of m4 in tissue on radula (to) and m7a; **iv**, intestinal transverse fold; **jw**, jaw; **ki**, kidney; **kl**, kidney lobe; **m1–m10**, extrinsic and intrinsic odontophore muscles; **mb**, mantle border (edge); **me**, ommatophore muscle; **mf**, mantle fold; **mi**, micro muscular pallial longitudinal fibers; **mj**, jaw and peribuccal muscles; **ml**, pallial muscle; **mo**, mouth; **mp**, muscular wall of penis; **mr**, membrane surrounding radular sac; **mt**, mantle; **mu**, prerectal muscle; **ne**, nephropore; **nr**, nerve ring; **oc**, odontophore cartilage; **od**, odontophore; **om**, ommatophore; **on**, optical nerve; **ou**, ommatophore muscle; **pb**, penis bulged portion; **pc**, pericardium; **pe**, penis; **pf**, penis inner fold(s); **pg** pedal gland; **pl**, pleural ganglia bridge; **pm**, penis muscle; **pn**, pneumostome; **pp**, pedal ganglion; **pr**, penis aperture; **ps**, penis shield; **pt**, prostate; **pu**, pulmonary cavity; **pv**, penis inner transverse fold; **ra**, radula; **rn**, radular nucleus; **rs**, radular sac; **rt**, rectum; **sa**, salivary gland aperture; **sd**, salivary gland duct; **se**, septum between esophagus and odontophore; **sg**, salivary gland; **sp**, sperm inner longitudinal fold; **sr**, seminal receptacle; **st**, stomach; **su**, anal sulcus; **sy**, statocyst; **tg**, integument; **tm**, tentacle muscle; **to**, tissue on radula et end of radular sac; **ty**, typhlosole; **ua**, urinary aperture; **ug**, external urinary gutter in head-foot; **um**, umbrella-like transverse penis fold; **un**, union of mantle border with nuchal surface; **up**, primary ureter; **ur**, urinary gutter; **us**, secondary ureter; **ut**, uterus; **va**, vagina; **vd**, vas deferens; **ve**, ventricle; **vf**, vaginal fold; **vg**, vagina; **vm**, visceral mass; **wo**, parasite worm.

Additionally in the text, the following abbreviations are used: **L**, length; **sh**, empty dry shell; **spm**, complete specimen (shell and soft parts); **W**, width. Institutions: **MNRJ**: Museu Nacional da Universidade Federal do Rio de Janeiro, Brazil; **MZSP**: Museu de Zoologia da Universidade de São Paulo, Brazil; **NHMUK**, Natural History Museum, London, UK; **USNM**: National Museum of Natural History, Smithsonian Institution, USA.

## Results

### Comparative conchology and anatomy

**Systematics.** Genus *Kora* Simone, 2012

*Neopetraeus*: Salgado & Coelho, 2003 [10]: 134 (non Martens, 1885).

*Kora* Simone, 2012 [1]: 432; 2015 [2]: 51; Salvador & Simone, 2016 [3]: 2.

**Diagnosis.**    Shell fusiform to obese; with brownish, relatively uniform color, subsutural pale band. Sculpture axial undulations only, with minute spiral aligned pits in some areas. Protoconch of ~2 smooth whorls, some axial sculpture in last whorl sometimes present. Umbilicus usually well-developed, resulted of columellar hollow area, producing inner lip usually with middle region bulged. Secondary pair of columellar muscles with medial differentiated branch. White pallial gland. Ureter is totally closed (tubular). Radular sac bulging posteriorly in buccal mass, covered by translucent membrane (mr). Radula as numerous hook-like teeth, with blunt tip, almost no difference among rachidian, lateral and marginal teeth. Odontophore pair m8 present; ventral tensor muscle of radula lost. Base of bursa copulatrix and penis thick muscular. Accessory albumen chamber present. Penis divided into compartments, with pair of longitudinal inner folds. Spermatophore with chitinous basal tube. Calcified epiphragm.

**List of included species.**    *Kora corallina* Simone, 2012 (type species by M & OD); *K. nigra* Simone, 2015; *K. rupestris* Salvador & Simone, 2016; *K. tupan* new species; *K. ajar* new species; *K. aetheria* new species; *K. jimenezi* new species; *K. kremerorum* new species; *K. curumim* new species; *K. vania* new species; *K. uhlei* new species; *K. arnaldoi* Pena, 2024.

**Taxonomic discussion.**    see Discussion.

***Kora corallina* Simone, 2012 Figs 1–6.**    *Kora corallina* Simone, 2012 [1]: 432–433 (fig 1–8); 2015 [2]: 51, 53–55 (fig 14); Salvador & Simone, 2016 [3]: 2–6 (Fig 2); Cavallari et al., 2016 [21]: 15 (Fig 5); MolluscaBase, 2023 [11].

**Types.**    Holotype MZSP 103910; Paratypes: MZSP 103911, 1 shell; MZSP 103912, 1 shell, USNM, 2 shells; MNRJ, 2 shells; NHMUK, 2 shells; MZSP 103913, 32 shells; all from type locality (all examined).

**Type locality.**    BRAZIL. Bahia; Santa Maria da Vitória, ~13˚24'S 44˚12'W, ~460 m of elevation (Coltro col., i/2012).

**Diagnosis.**    Size about 45 mm, ~2.3 times longer than wide; lacking dorso-ventral compression. Apex with light color. Subsutural lighter band present. Peristome white. Delicate spiral striae present. Aperture occupying ~44% of length and ~70% width. Implantation of outer lip slightly vertical. Inner lip with high middle fold. Umbilicus wide. Secondary columellar muscles with 13 insertions in left and 7 in right. Two pairs of m1v. Jaw with central notch. Odontophore cartilages ~75% fused. Pair m10 narrow. Carrefour duct wide and short, inserted in duct of albumen gland. Albumen chamber in curve. Penis ~70% of spermoviduct length, lacking umbrella-like fold; epiphallus ~13% of penis length; penis muscle at epiphallus tip.

**Redescription.**    *Shell.* (Figs 1 and 2A–2J) Proper description in [1]. Complement: length up to 45 mm, outline fusiform, elongated, ~2.3 longer than wide. Color yellowish-white (Fig 1A, 1B), beige (Fig 1F–1H and 1J–1N), or brown (Fig 2A–2E) in first whorls, gradually becoming darker towards past whorl, with brown pigment, particularly dark in last whorl (Fig 1G, 1K, 1M); peristome white. Protoconch (Figs 1H and 2F) with ~2 whorls, bluntly pointed; first whorl smooth, axial riblets gradually appearing in second whorl. Callus low, weak (Figs 1A, 1C, 1F, 1J, 1L and 2A). Aperture with inner lip bearing strong oblique, middle, wide fold (Figs 1D, 1N and 2C), almost forming stubby plica. Umbilicus broadly (Fig 1I) to narrow (Fig 2D) opened, partially covered by inferior half of inner lip.

**Epiphragm.**    Present in few specimens (Figs 1C–1E and 2J) calcified, thick, occluding entire aperture; dislocated posteriorly from peristome.

**Head-foot.**    (Fig 3B) Of normal shape. Color uniformly pale beige. Columellar muscle thick, 1.5 whorls in length. Inner arrangement of columellar annexed muscles relatively complex. Main columellar bundle (cm) occupying ventral floor of haemocoel, relatively flattened, wide ~3/4 of foot width. Pair of secondary columellar/cephalic muscles, each of which with

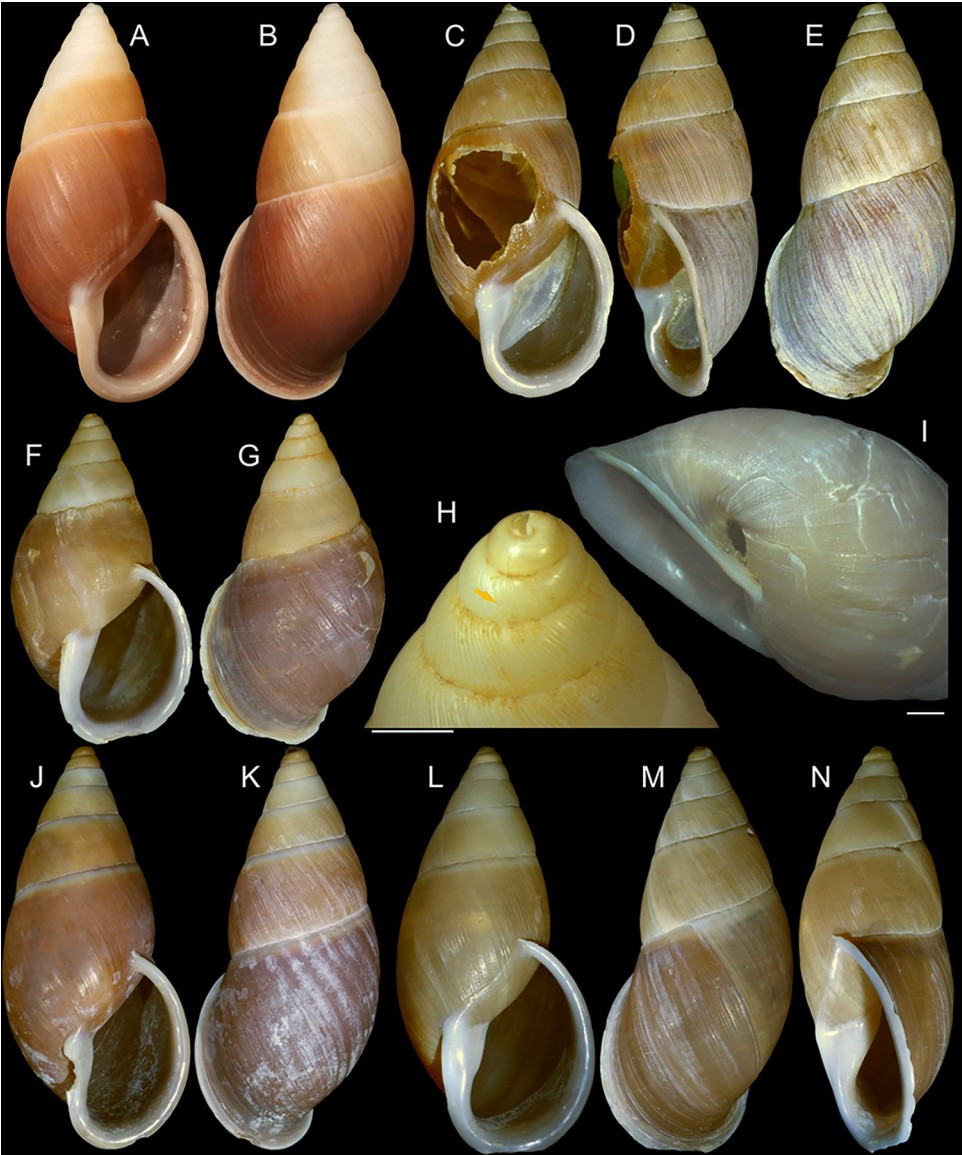

**Fig 1. *Kora corallina* shell characters.** (A–B) holotype MZSP 103910 (L 43.4 mm), frontal and dorsal views. (C–E) dissected specimen MZSP 132078, frontal, right and dorsal views, epiphragm preserved, hole artificially done for specimen extraction (L 44.8 mm). (F) MZSP 151952#1, shell slightly deformed (L 35.5 mm). (G) same, dorsal view. (H) same, detail of apex, profile-slightly apical view, arrow showing transition protoconch-teleoconch, scale = 2 mm. (I) same, detail of last whorl, left-slightly anterior view showing umbilicus, scale = 2 mm. (J–K) MZSP 151952#2, frontal and dorsal views (L 44.8 mm). (L–N) MZSP 125175, frontal, dorsal and right views (L 37.0 mm).

~half of main columellar bundle (cm) width. Both with multiple, aligned, similar-sized origins, flanking floor of haemocoel, edging at short distance pedal gland (pg) and mouth (mo); away from each other anteriorly, approaching from each other posteriorly, at ~1/3 of columellar muscle length. Anterior most branch short, as ommatophore retractor muscle (me); second small branch of right secondary muscle as genital muscle (gm); left secondary muscle (cl) with 13 branches, being posterior branch slightly broader; right secondary muscle (cr) with 6–7 branches, being posterior branch very broader. Both secondary columellar muscles attaching

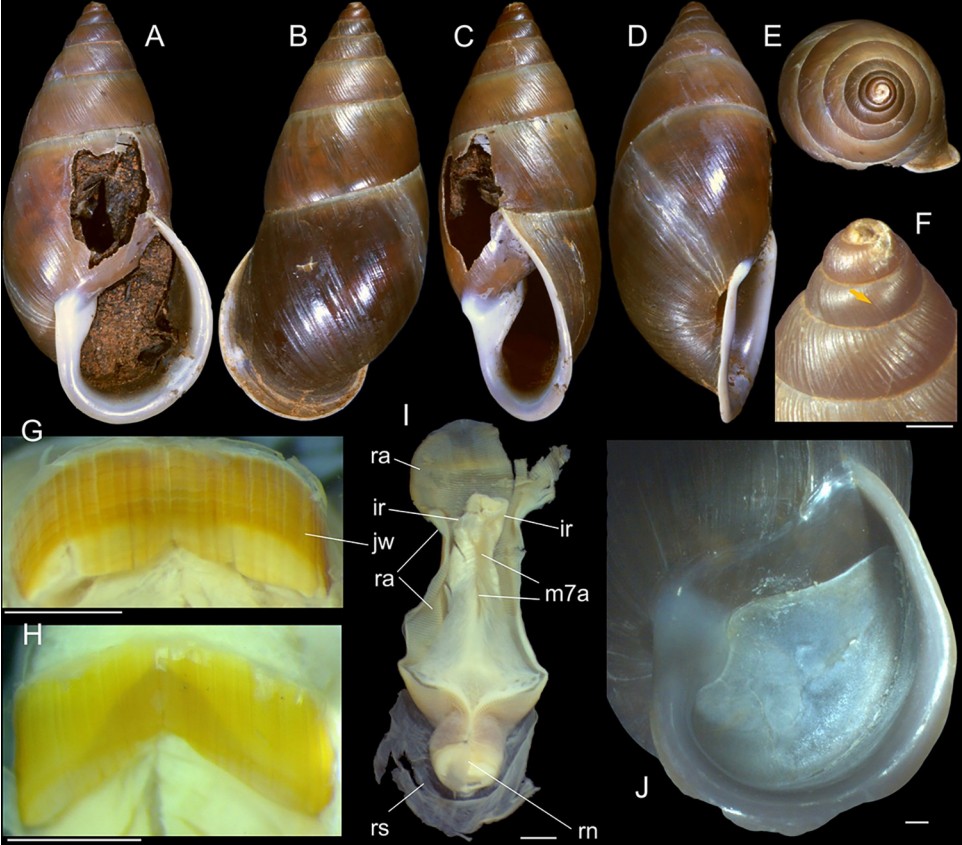

**Fig 2. *Kora corallina* shell and anatomical characters, light photos.** (A) MZSP 151952#3 shell, frontal view, naturally broken (L 40.0 mm). (B) same, dorsal view. (C) same, right view. (D) same, left-slightly anterior view showing umbilicus. (E) same, apical view. (F), same, detail of apex, profile-slightly apical view, arrow indicating transition protoconch-teleoconch. (G) jaw in situ, specimen MZSP 132078#3, ventral view, adjacent tissues removed. (H) same for specimen MZSP 132078#2, younger one. (I) radula, isolated, extended and opened longitudinally, dorsal view, specimen MZSP 132078#1. (J) detail of shell aperture of specimen MZSP 132078#1 occluded by epiphragm. Scales = 1 mm.

to main bundle only in their posterior region, jointed to radular muscle (m2) attaching externally. Pedal gland (pg) weakly protruding in posterior region of buccal area.

**Mantle organs.** (Fig 3A) Mantle border (mb) thick, lacking pigments. Pneumostome (pn) protected by simple right ventral flap (if), width ~1/5 of aperture length. Folds of mantle border weakly developed. Pneumostome (pn) ~1/8 of shell aperture length, bearing exclusively air entrance and urinary aperture (ua); flanked in internal edge by narrow urinary gutter. Anus (an) separate aperture located at right, adjacent to pneumostome. Lung of ~1.5 whorls in length, ~twice long than wide, possessing minute longitudinal muscle fibers in its wall seen by translucency; right side from pulmonary vein ~twice wider than left side. Pulmonary venation well-developed, especially in region preceding pneumostome; posterior region of pulmonary vein (cv) protruded; left 2/3 only pair of intercalated longitudinal vessels; right 1/3 mostly having perpendicular vessels rather uniformly distributed, weak posterior, becoming taller anteriorly; in region preceding pneumostome pulmonary vessel bifurcating, bearing radial arrangement of secondary vessels. Pulmonary vein (cv) running longitudinally across pallial cavity roof. Reno-pericardial area of beige color, slightly triangular, located posteriorly within pallial cavity, its posterior abutting wall of visceral cavity, occupying ~20% of cavity length and

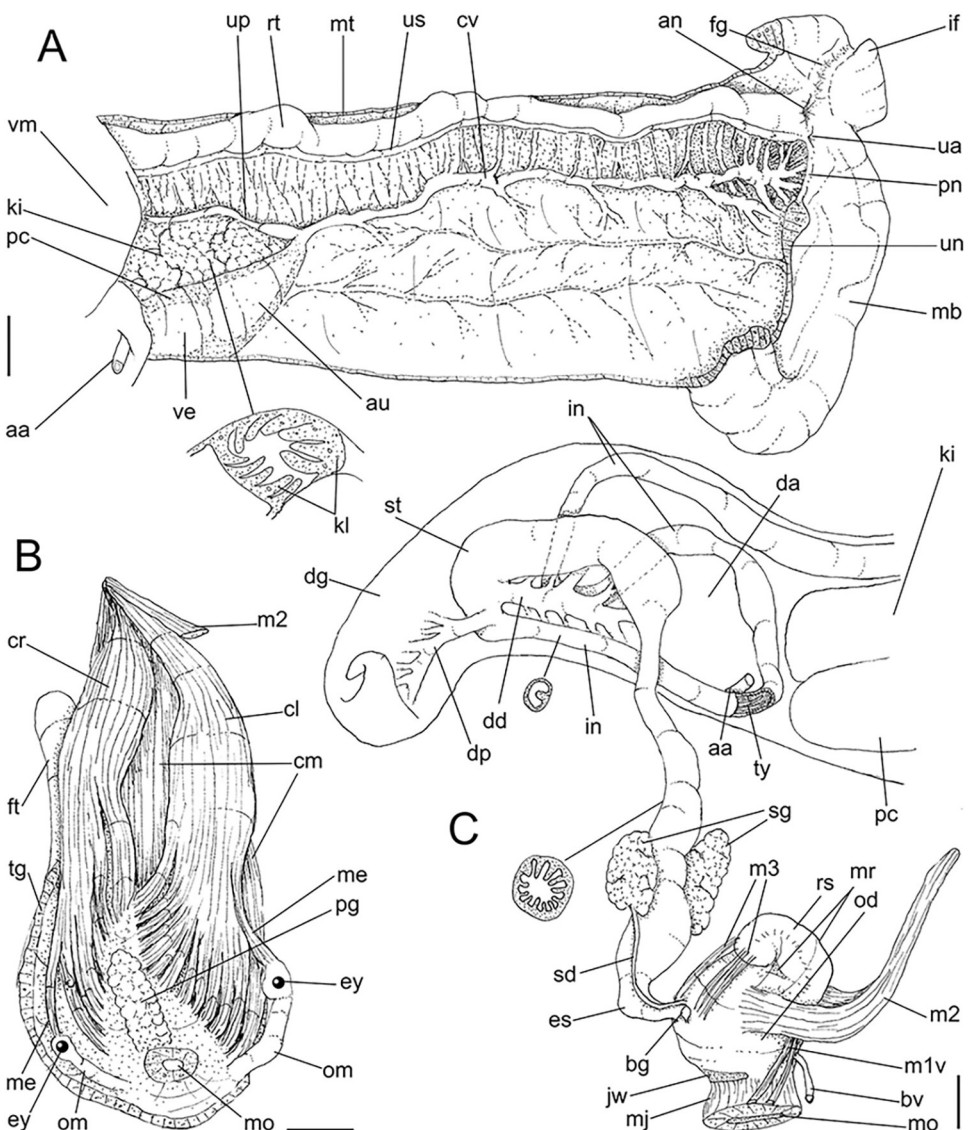

**Fig 3.** *Kora corallina* **anatomical drawings.** (A) extended pallial (pulmonary) cavity, ventral-inner view, inner edge of pneumostome sectioned and deflected upwards, transverse section of indicated region of kidney also shown. (B) head-foot, dorsal view, head, dorsal integument and internal organs removed, remaining muscles expanded. (C) foregut and midgut, mostly ventral view as in situ, topology of some adjacent structures also shown, 2 transverse sections of indicated regions of esophagus and intestine also shown, small ventral portion of intestine adjacent to pericardium removed to show inner surface. Scales = 2 mm.

~55% of its width (details below). Rectum (rt) wide. Primary (up) and secondary (us) ureters entirely closed (tubular), relatively narrow, aperture (ua) simple, directly outside at right in pneumostome region.

**Visceral mass.** (Fig 3C) ~2.5 whorls in length. Both digestive gland lobes greenish beige in color; anterior lobe (da) flattened, occupying ~1/5 of visceral volume, located just posteriorly to pallial cavity, continuous to kidney. Posterior lobe (dg) larger, extending 2 spiral whorls, occupying ~50% of visceral volume. Stomach (st) ~1/10 of visceral volume, located between both digestive gland lobes, ~3/4 whorl posterior to pallial cavity. Digestive tubes (described below) proportionally large. Gonad multi-lobed, cream color, encased between posterior lobe of digestive gland and columella, occupying ~1/3 of last whorl, ~1/15 of visceral volume.

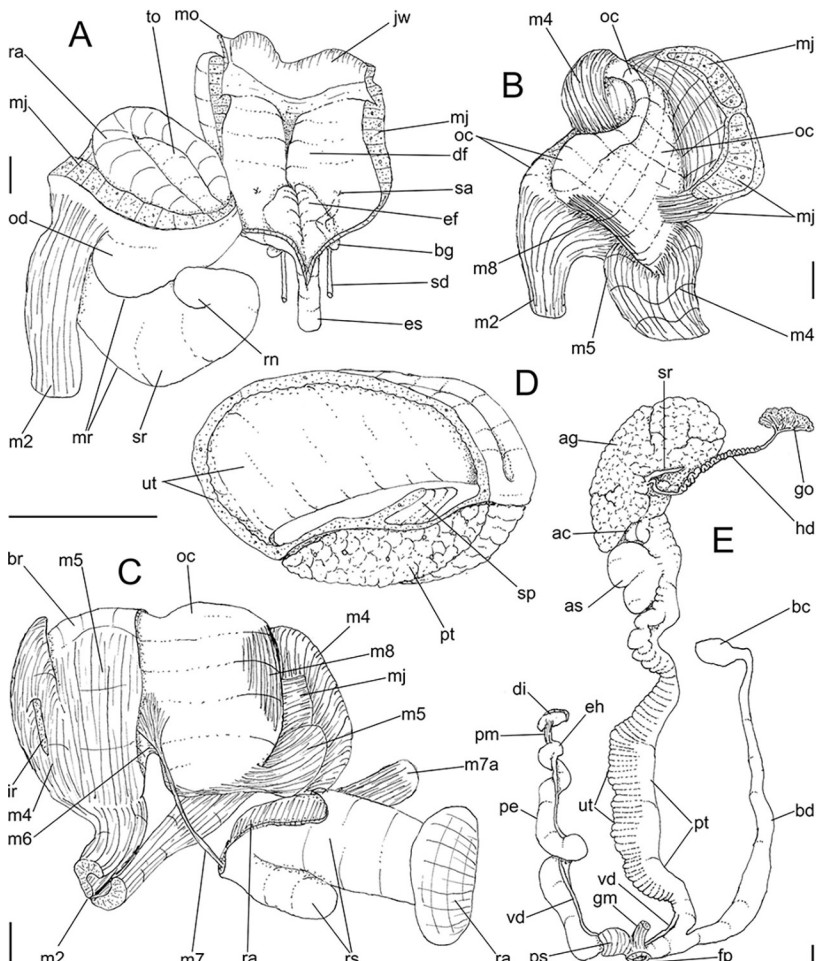

**Fig 4. *Kora corallina* anatomical drawings.** (A) buccal mass, right view, dorsal wall sectioned and deflected to right, in inner-ventral view. (B) odontophore, right-anterior view, cartilages (oc) strongly curved inwards, with right m4 and peribuccal muscles (mj) deflected. (C) same, dorsal view, radula removed and deflected downwards (m7a, located inside radular sac, slightly deflected), left muscles as in situ, right muscles deflected externally. (D) spermoviduct, middle region, short portion in transverse view. (E) genital structures, dorsal view, mostly uncoiled. Scales = 1 mm.

**Circulatory and excretory systems.** (Fig 3A) Pericardium (pc) ~twice as long as wide, located obliquely between middle and left thirds of posterior end of pallial roof, appressed against right lateral side of kidney; occupying ~5% of lung area. Auricle (au) located anteriorly, as continuation from pulmonary vein (cv); ventricle (ve) located posteriorly, larger. Kidney (ki) simple, weakly dorso-ventrally flattened; size reported above; slightly triangular, width ~1/2 of length; internally organized as successive tall glandular folds (kl), being taller in dorsal wall, and shorter in wall with visceral mass; central region hollow. Nephropore small, longitudinal slit at anterior-left corner of kidney, directed towards right, inside anterior end of primary ureter.

**Digestive system.** (Figs 3C and 4A–4C) Mouth (mo) and oral tube (mj) wide, short, thick muscular. Jaw plate (Fig 2G, 2H) thick, yellow; cutting edge convex, notched at middle or chevron-like; sculptured by successive, rather uniform, transverse, wide folds. Buccal mass elliptic, occupying~1/5 of haemocoel volume; being ~1/4 of its posterior bulged chamber, sheltering coiled radular sac (rs), covered by transparent membrane (mr). Dorsal surface of oral cavity with well-developed pair of dorsal folds (Fig 4A: df), width of each ~1/2 of dorsal wall

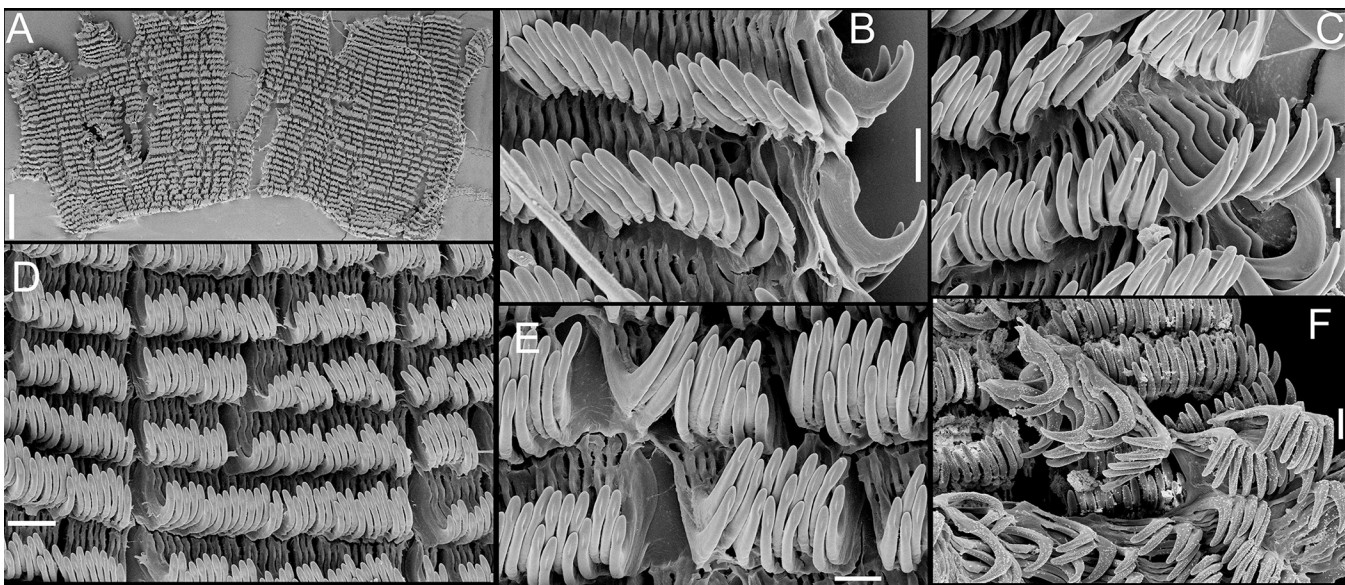

**Fig 5. *Kora corallina* radulae in SEM.** (A) whole view of a middle part, scale = 500 μm. (B) detail of marginal region, scale = 30 μm. (C) same, other level, scale = 30 μm. (D) detail of central region, scale = 50 μm. (E) detail of lateral region, scale = 30 μm. (F) detail of marginal region, another specimen, scale = 30 μm.

width; touching with each other in median line. Odontophore (Fig 3C: od) with ~50% of buccal mass volume. Odontophore muscles (Fig 4A–4C): **mj**, jaw and peribuccal muscles originating in outer-ventral surface of odontophore cartilages (Fig 4B), running towards ventral making platform covering ventral surface of cartilages (oc), afterwards splaying in dorsal wall of oral tube; **m1**, jugal muscles covering entirely haemocoelic structures, more concentrated close to mouth; **m1v**, two pairs of barrow ventral protractors jugal muscles (Fig 3C), originating in ventral surface of haemocoel close to mouth, running towards posterior, inserting in ventral-posterior region of odontophore close to m2 insertion; **m2**, radular muscle, or strong pair of retractor muscles of buccal mass, originating as single bundle in columellar muscle posterior end (Fig 3B), running anteriorly close to median line along ~60% of haemocoel length, becoming broader, inserting as two different bundles (Fig 4C) connected medially (Fig 4B) in ventro-posterior edge of odontophore, surrounding at some distance radular nucleus; **m3**, pair of thin longitudinal fibers immersed in posterior wall of odontophore (mr), between esophageal origin and radular nucleus; **m4**, main pair of dorsal tensor muscles of radula, very thick, originating in postero-medial region of odontophore cartilages, surrounding outside and medially cartilages, inserting in subradular membrane in its region correspondent to buccal cavity; **m5**, pair of auxiliary dorsal tensor muscles of radula, also thick, originating on postero-ventral region of odontophore cartilages, running towards anterior covering cartilages, inserting in subradular membrane along radular exposed (in buccal cavity) region; **m6**, horizontal muscle minute, only detectable in short portion (~10% of cartilage length) posterior to fusion between both cartilages; **m7**, narrow and slender, originated splayed in posterior region of fusion between both cartilages, run inside dorsal region od radular sac (Fig 4C); **m7a**, thick muscular bundle inside ventral stored portion of radula in radular sac, in which part of pair m4 inserts (Fig 2I: ir); **m8**, pair of wide, superficial muscles in lateral edge of cartilages, longer laterally, shorter in internal dorsal (Fig 4C) and ventral (Fig 4B) ends; **m10**, pair of narrow ventral odontophore protractor muscles, originating in ventro-anterior region of haemocoel, just ventral to mouth, running towards posterior covered by m1v, inserting in latero-posterior

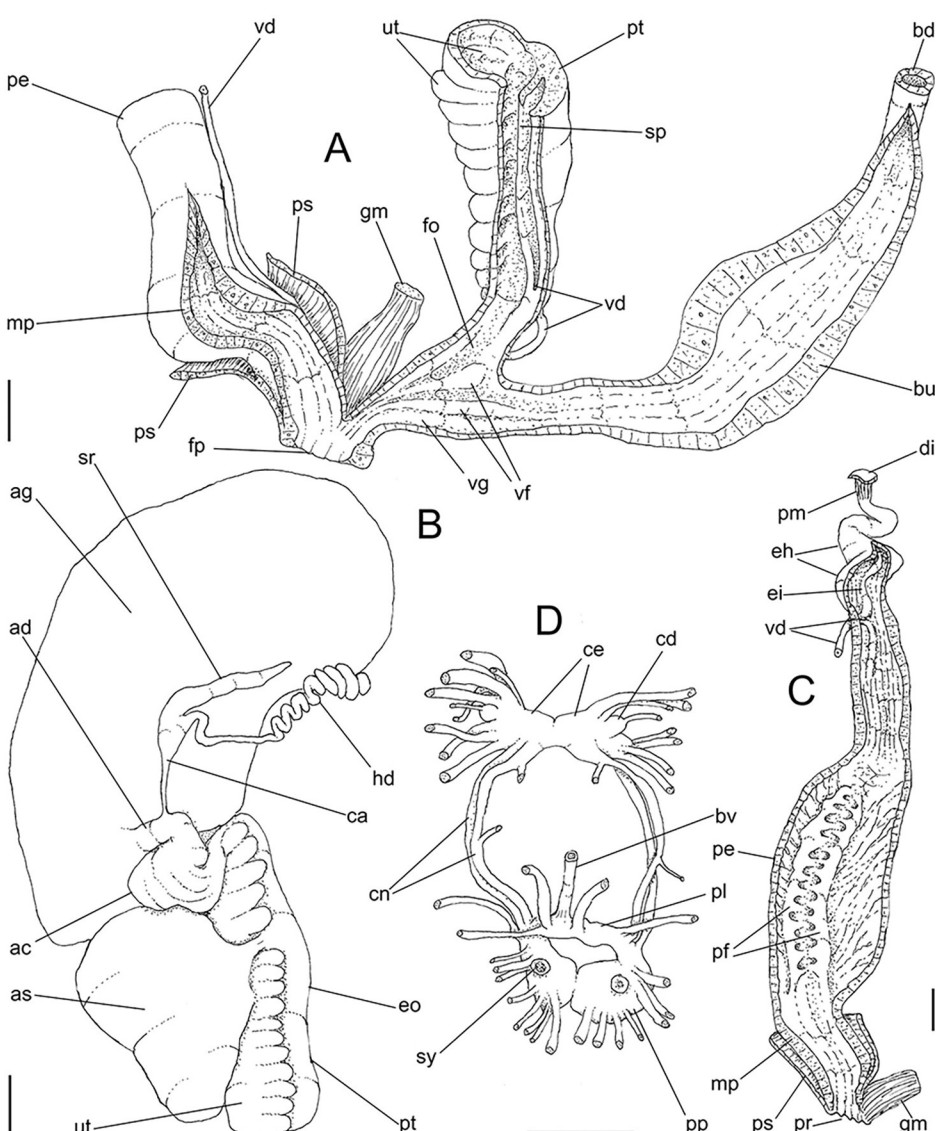

**Fig 6. *Kora corallina* anatomical drawings.** (A) genital tubes portion preceding pore, dorsal view, all of them longitudinally opened. (B) genital structures in albumen gland level if it was transparent, ventral view. (C) penis, ventral view, longitudinally opened. (D) central nervous system (nerve ring), ventral view. Scales = 1 mm.

surface of odontophore close to m2 insertions; **m11**, pair of narrow ventral tensor muscles of radula absent. Odontophore non-muscular structures: oc, pair of odontophore cartilages flattened, rather elliptical, slightly rectangular, ~1.5 times longer than wide, fused with each other along ~75% in their anterior-medial edge, posterior end roughly rounded; **sc**, subradular cartilage, with expanding region in buccal cavity protecting subradular membrane. Radular sac (rs) long, performing loop inside translucent membrane (mr) bulded posteriorly from odontophore (Fig 3C).

**Radula.** (Figs 2I and 5) ~1.5 times longer than odontophore. Composed of uniform, similar kind of tooth, with no clear differentiation among rachidian, lateral or marginal teeth, ~250 pairs of teeth per row (Fig 5A). Each tooth with elongated base, 6–7 times longer than wide, placed longitudinally, very close to each other; proximal end rounded, slightly elevated

(Fig 5B–5D); distal end bearing long, curved cusp, slightly longer than base; base reinforced by central fold gradually tapering up to ~70% of cusp length (Fig 5C); cusp tip flattened, slightly broader, rounded-barely spoon-like, possessing shallow subterminal, longitudinal, short furrow (Fig 5C and 5E). All teeth straight aligned per row, except for ~20 more marginal teeth, slightly arched aligned (Fig 5C and 5F).

Salivary glands small, covering ~1/10 of esophagus length, located between anterior and second quarter of esophageal length (Fig 3C: sg), forming two elliptic, white, thin masses. Each salivary duct differentiable in middle and anterior side of glands, with ~1/12 of esophageal width (sd). Salivary duct running in both sides of esophageal origin, penetrating buccal mass wall in region close to buccal ganglia (bg), running immersed in buccal dorsal wall along ~1/3 its length (Fig 4A). Salivary ducts opening as small pores (sa), located in middle level of posterior region of lateral edges of wide dorsal folds. Esophagus ~1-whorl long, with firm walls (Fig 3C: es); anterior region with narrow, tall longitudinal, uniform folds; posterior third weakly broader, smooth surface. Stomach (st) narrow, curved, weakly bulging; position and size described above (visceral mass); gastric walls thin, weakly muscular; inner surface mostly smooth, lacking folds. Esophageal insertion on right side, intestinal origin on left side, both close to columella. Duct to anterior lobe of digestive gland at short distance from esophagus and intestine intersection (dd) broad, running towards anterior, possessing secondary successive branches along both sides along ~1/3 whorl immersed in digestive gland. Duct to posterior lobe of digestive gland located short distance from intestinal origin, slightly posterior and at right to above-described duct, directed towards opposite side (dp), slightly narrower and possessing only branches in right side. Intestine (Fig 3C: in) initially narrower than esophageal insertion, maintaining this caliber along its entire length; performing its usual wide sigmoid loop in anterior lobe of digestive gland; its anterior region having pair of narrow typhlosoles close from each other, from stomach level, up to pericardium level (ty). Rectum and anus position described above (pallial cavity) (rt, an). Anus sessile, as slit in right end of mantle edge directly turned outside.

**Reproductive system.** (Figs 4E, 4D and 6A–6C) Gonad position described above (visceral mass), composed of 5–6 lobes with minute digitiform acini. Hermaphroditic duct (Fig 4E: hd) narrow; coiled portions occupying middle 2/3, with narrow coils; insertion preceded by straight region, and strongly curve (Fig 5B: hd). Seminal receptacle (Figs 4E and 6B: sr) small, straight, long, tapering gradually, ~10 times longer than wide, flattened. Fertilization complex or carrefour (Fig 6B: ca) simple, as wide region in receptacle base, ~1/3 of its length; totally immersed in albumen gland, tapering abruptly up to very narrow duct inserting in posterior end of spermoviduct, at side of tip of wide albumen gland duct. Albumen gland (ag) solid, white, elliptical, ~5 times larger than gonad (~1/3 whorl). Albumen gland duct subterminal, connected to distal end of spermoviduct (Fig 6B: ad), continuing as small, curved albumen chamber (Fig 6B: ac); narrowly connected to distal end of spermoviduct (eo). Spermoviduct (Figs 4E and 6B: eo) of ~1.5 whorl in length, slightly narrower than albumen gland, ~20 times longer than wide. Secondary albumen chamber (as) ~4-times larger than primary chamber (described above), connected to spermoviduct some distance anterior to it by narrow duct (Fig 6B). Prostate wide (pt), ~1/3 of spermoviduct diameter (Fig 4D); uterus with weakly glandular walls, highly, transversally, and relatively uniformly folded (Figs 4D, 4E and 5A, 5B: ut). Sperm inner longitudinal fold (Figs 4D and 6A: sp) as simple, tall, thick fold, a second small fold gradually appearing in basal third; both folds fusing with each other, originating vas deferens, slightly anterior to end of uterine level (Fig 6A: vd). Vas deferens uniformly narrow, uncoiled (Figs 4E and 6A: vd). Genital muscle in intersection vagina and penis (Figs 4E and 6A: gm). Bursa copulatrix (bc) and its duct (bd) of usual position, with ~90% of spermoviduct length (Fig 4); bursa duct with basal third with walls extraordinarily thick muscular (Fig 6A:

bu). Free oviduct (fo) and vagina (vg) simple, possessing respectively 2 and 4 wide, low, longitudinal, simple folds (Fig 6A). Penis slightly coiled, ~70% of spermoviduct length if straightened (Fig 4E: pe); epiphallus as continuation of penis, penis muscle inserted at epiphallus' tip (Figs 4E and 6C: pm), short, simple. Penis shield (Figs 4E and 6A, C: ps) with transverse muscle fibers, with ~1/7 penis length. Penis wall relatively muscular, especially in region adjacent to penis shield (mp). Epiphallus (eh) ~1/8 of penis' length, amply opened to penis; only vas deferens insertion marking its limit (Fig 6C: vd). Epiphallus inner surface with single high longitudinal fold (ei) and 8–10 secondary small, parallel folds (Fig 6C: eh). Internal penial arrangement of folds clearly with three regions (Fig 6C): (1) basal third, possessing only 3–4 longitudinal, broad, low, simple folds; (2) middle third, with strong pair of tall, longitudinal folds in a side, each one with simple external margin, and successive small secondary branches in internal margin, imbricating with its counterpart, this strong pair basal end fading, distal end rounded; mosaic of low, oblique, rather irregular folds flanking both strong folds; (3) distal third with basal region smooth, and 8–10 longitudinal folds similar to those from epiphallus gradually appearing; some of them, including one of them slightly larger, converge to vas deferens aperture (Fig 6C: vd).

**Central nervous system.**   (Fig 6D) Nerve ring located in anterior half of buccal mass. Pair of cerebral ganglia located very close with each other, commissure extremely short; each cerebral ganglion elliptic, with ~1/10 buccal area's size. Pair of cerebral nodes, or glands (cd), with ~1/8 each ganglion's size. Pair of cerebro-pedal and cerebro-pleural connectives (cn) slender, long (~2.5 times longer than each cerebral ganglion), similar-sized, running close from each other. Pair of pleural ganglia (pl) connected with each other by short commissure slightly dislocated to left, region possessing narrow blood vessel (bv); both pleural ganglia broadly connected to pedal ganglia in their outer region. Pair of pedal ganglia (pp) located close from each other, commissure extremely short; each pedal ganglion rather spheric, ~1.5 times larger than each cerebral ganglion. Pair of statocysts (sy) located in ventral surface of pedal ganglia, each one with ~1/15 of each pedal ganglion's size; internally with several rounded, crystalline statoconia.

**Distribution.**   *Kora corallina* so far has been only known in a Bahia region west from São Francisco River, more or less defined by the quadrilateral Santa Maria da Vitória–São Félix do Coribe–Serra do Ramalho–Taquarinópolis, Brazil.

**Habitat.**   Under rocks, limestone areas.

**Measurements.**   (in mm) MZSP 132078 (Fig 1C–1E): 44.8 by 21.2; MZSP 151952#1 (Fig 1F–1G) 35.5 by 19.6; MZSP 151952#2 (Fig 1J–1K): 44.8 by 20.1; MZSP 151952#3 (Fig 2A–2D): 40.0 by 19.3; MZSP 125175 (Fig 1L–1N): 37.0 by 17.1.

**Material examined.**   All types. BRAZIL (W Vailant-Mattos col.). **Bahia**. Santa Maria da Vitória (topotypes), 13˚28'S 44˚13'W, MZSP 132078, 5 spm (vi.2016); São Félix do Coribe, 13˚26'12"S 44˚12'21"W, MZSP 151952, 19 shells (4.iv.2020); Serra do Ramalho, Toca, 13˚38'13"S 43˚50'10"W, MZSP 152175, 25 shells, 152223, 1 shell (v.2019); Taquarinópolis, 13˚32'44"S 43˚50'28"S, MZSP 151808, 9 shells (v.2019).

**Taxonomic remarks.**   *Shell. Kora corallina* has an average shell size of approximately 45 mm, making it larger than *K. nigra*, *K. aetheria*, *K. vania*, and *K. curumim*, but smaller than *K. tupan* and *K. ajar*. Its shell is about 2.3 times longer than it is wide, giving it a narrower shape compared to most other congeneric species, except *K. rupestris*, from which it differs by lacking a distinctly conical outline. The shell aperture comprises about 44% of the total shell length, which is much shorter than the apertures of *K. nigra*, *K. rupestris*, *K. tupan*, *K. ajar*, *K. kremerorum*, and *K. curumim*. Additionally, unlike *K. nigra*, *K. tupan*, *K. ajar*, *K. jimenezi*, *K. kremerorum*, and *K. vania*, *K. corallina* lacks a horizontally oriented superior implantation of the outer lip.

**Anatomy.** *Kora corallina* is the only species with low folds along the mantle edge (Fig 3A). It differs from *K. nigra*, *K. aetheria*, and *K. jimenezi* by having a single intercalated pair of wide vessels to the left of the pulmonary vein (Fig 3A), as these species exhibit different vascular arrangements. The branched anterior end of the pulmonary vein (Fig 3A: cv) distinguishes *K. corallina* from *K. nigra*, *K. rupestris*, and *K. jimenezi*, which have simpler structure. Additionally, *K. corallina* displays strong venation to the right of the pulmonary vein only in the region preceding the pneumostome, a trait shared only with *K. jimenezi* among its congeners. The kidney lobe completely surrounds the kidney walls (Fig 3A: kl), differentiating *K. corallina* from *K. nigra*, *K. rupestris*, and *K. tupan*. In terms of muscle structure, *K. corallina* has 13 anterior insertions of the left accessory columellar muscle (Fig 3B: cl), more than any other congener, all of which have significantly fewer insertions. Additionally, it has 7 anterior insertions of the right accessory columellar muscle, setting it apart from *K. tupan*, *K. ajar*, *K. jimenezi*, and *K. uhlei*. These accessory columellar muscles also lack a posteriorly located medial branch, which is present in all remaining congeners except *K. jimenezi*. Furthermore, *K. corallina* lacks the odontophore muscle pair m1l, a feature present in *K. rupestris*, *K. tupan*, *K. aetheria*, *K. jimenezi*, and *K. uhlei*. The presence of two pairs of odontophore muscles m1v (Fig 3C) differentiates it from *K. tupan*, *K. ajar*, *K. jimenezi*, and *K. uhlei*, which lack these muscles, and from *K. aetheria*, which has only one pair. The connection of the odontophore muscle pair m3 to the esophagus origin distinguishes *K. corallina* from *K. rupestris*, where it connects to the m2 pair, and from *K. jimenezi*, which lacks this muscle. In having branches only on the right side of the posterior duct to the digestive gland (Fig 3C: dp), *K. corallina* differs from *K. nigra*, and its bilateral branching in the anterior duct to the digestive gland further differentiates it from all congeners, except *K. rupestris* and *K. aetheria*, which also have bilateral branches. The central notch in the jaw plate (Fig 2G, 2H) is shared only with *K. rupestris* and *K. aetheria*, distinguishing *K. corallina* from other congeners. The salivary gland aperture, located in the middle third of the buccal dorsal wall (Fig 4A: sa), differs from *K. tupan*, and the absence of a salivary papilla sets it apart from *K. jimenezi* and *K. uhlei*. The degree of fusion of the odontophore cartilages, around 75%, is similar to most congeners but greater than that of *K. nigra* (~60˚) and less than that of *K. uhlei* (~90˚). In terms of odontophore m4-m5 pairs of muscles, the m4 muscle covering m5 distinguishes *K. corallina* from *K. tupan* and *K. uhlei*, in which these muscle pairs are continuous with each other. The narrow odontophore muscle m7 (Fig 4C), with a single origin, differs from the conformations in *K. tupan*, *K. jimenezi*, and *K. uhlei*. Finally, the narrow m10 muscle in *K. corallina* differs from the broader form in *K. tupan* and *K. aetheria*, and from the filiform version in *K. jimenezi*.

**Genital system.** *Kora corallina* has a small curve at the end of the hermaphrodite duct (Fig 6B: hd), which distinguishes it from *K. nigra*. The conical shape of its carrefour (ca) is distinct from those of *K. nigra* and *K. rupestris*, and its carrefour duct is the widest among all congeneric species. The species lacks the bulged portion on the opposite side of the hermaphrodite duct at the base of the seminal receptacle (sr), differentiating it from *K. tupan*, *K. jimenezi*, and *K. uhlei*. *K. corallina* is unique in having the carrefour duct inserting directly into the albumen gland duct (Fig 6B: ad). Its albumen chamber (Fig 6B: ac) forms a simple curve, contrasting with the blind sacs found in *K. nigra*, *K. rupestris*, *K. tupan*, and *K. aetheria*. In having a single sperm fold in the spermoviduct (Fig 4D: sp), *K. corallina* differs from *K. nigra*, which has two. Additionally, it has the narrowest prostate band in the spermoviduct (~35%) among its congeners, except for *K. nigra* and *K. aetheria*, which share similar proportions. The muscular anterior portion of the bursa copulatrix duct (Fig 6A: bu) is well-defined, setting *K. corallina* apart from *K. jimenezi*. Its penis length is approximately 70% of the spermoviduct, longer than those of *K. rupestris*, *K. aetheria*, and *K. uhlei*, but shorter than those of *K. nigra*, *K. tupan*, *K. ajar*, and *K. jimenezi*. The bursa copulatrix duct is about 90% of the spermoviduct length, which is shorter than in *K. rupestris* and *K. tupan*, but longer than in *K.*

*aetheria*, *K. jimenezi*, and *K. uhlei*. The vas deferens of *K. corallina* lacks the strong curve preceding its insertion at the tip of the penis, distinguishing it from *K. nigra*, *K. tupan*, *K. ajar*, and *K. uhlei*. Its penis base has clear muscular walls (Fig 6C: mp), unlike *K. jimenezi*, which lacks them. The penis of *K. corallina* features the usual pair of inner folds, but it also has an imbricated arrangement of inner branches (Fig 6C: pf), a characteristic shared only with *K. aetheria* and *K. uhlei* among its congeners. *K. corallina* lacks the umbrella-like transverse penial fold found in most of its congeners, with the exceptions of *K. nigra* and *K. jimenezi*. The epiphallus comprises about 13% of the penial length, the shortest ratio among its congeners, though *K. rupestris* has a similar but slightly longer proportion. *K. corallina* also has a strong longitudinal fold in the epiphallus (Fig 6C: ei), a feature shared only with *K. tupan* and *K. ajar*. Its penis muscle (pm) inserts terminally in the epiphallus, unlike *K. rupestris*, *K. ajar*, and *K. uhlei*, which have more basal insertions.

**Kora nigra** Simone, 2015 **Figs 7–11.** *Kora nigra* Simone, 2015 [2]: 3 (fig 6–13, 21); Salvador & Simone, 2016 [3]: 1–6 (fig 3), 2021 [22]: 396–397 (fig 2A–B); Cavallari et al., 2016 [21]: 15 (fig 6); Salvador et al., 2022 [23]; 5 (fig 18); MolluscaBase, 2023 [11].

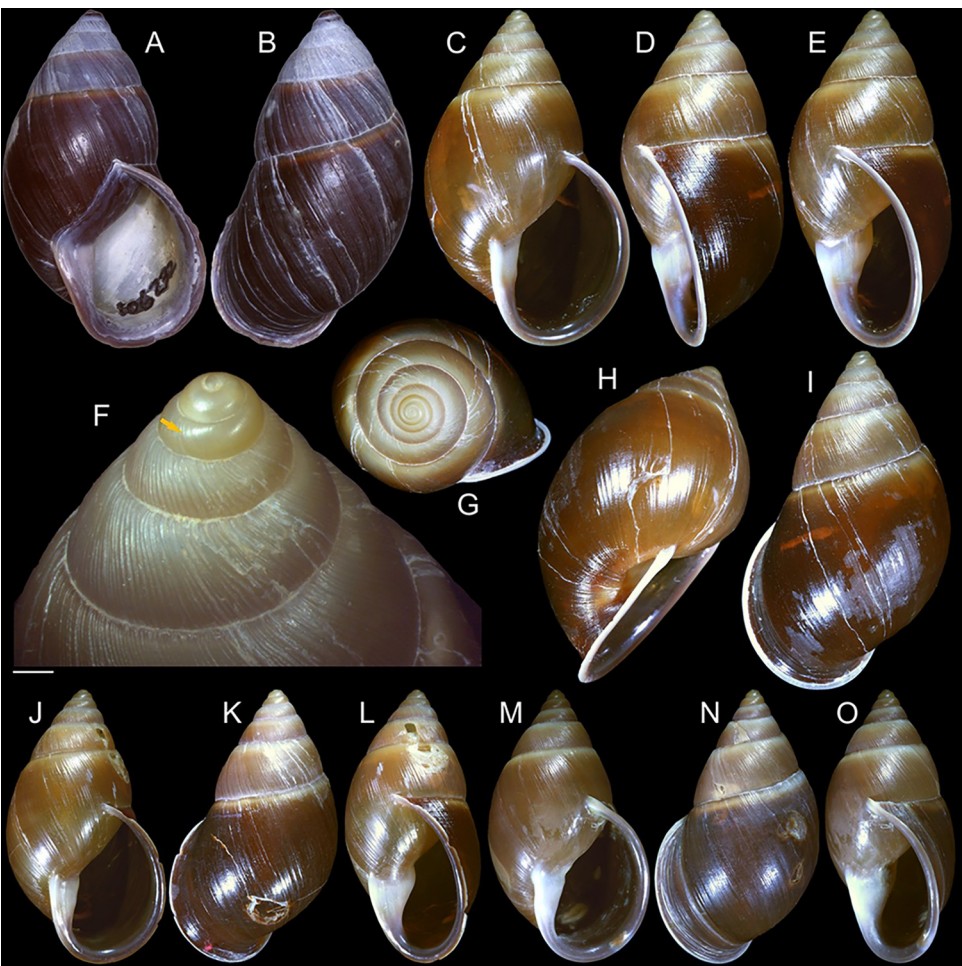

**Fig 7. *Kora nigra* shell characters.** (A–B) Holotype MZSP 106230, frontal and dorsal views (L 30.1 mm). (C) shell of dissected specimen MZSP 151828#2, frontal view (L 41.8 mm); (D) same, right view. (E) same, right-slightly ventral view. (F) same, detail of apex, profile-slightly apical view, arrow indicating transition protoconch-teleoconch, scale = 1 mm. (G) same, apical view. (H) same, anterior-slightly left view. (I) same, dorsal view. (J–L) shell of dissected specimen MZSP 151828#1, frontal, dorsal and right views (L 40.6 mm). (M–O) MZSP 151827, frontal, dorsal and right views (L 43.3 mm).

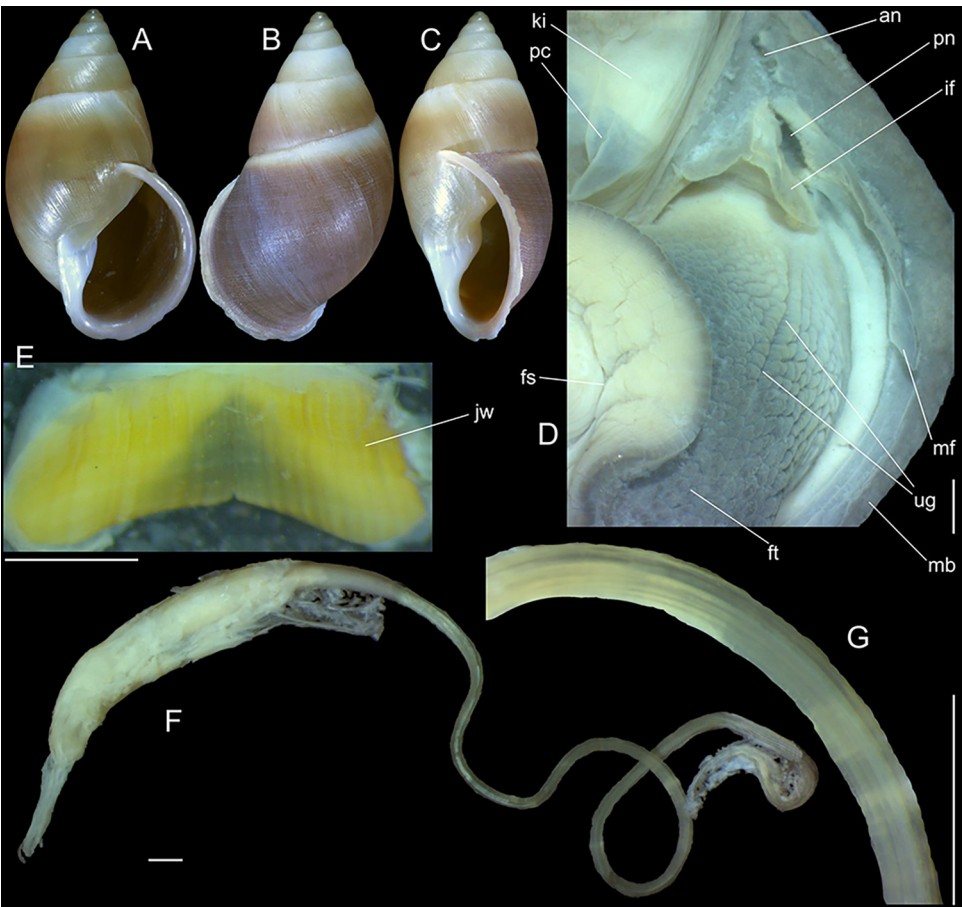

**Fig 8. *Kora nigra* shell and anatomical characters, light photos.** (A–C) shell MZSP 151830, frontal, dorsal and right views (L 39.9 mm). (D) contracted MZSP 151828#1 specimen just removed from shell, detail of mantle edge and part of head-foot in pneumostome region, frontal view. (E) jaw, ventral view, MZSP 151828#1. (F) spermatophore extracted from duct of bursa copulatrix of specimen MZSP 151828#2. (G) same, detail of its stem middle portion. Scales = 1 mm.

**Types.**　Holotype MZSP 106232; Paratypes: MZSP 106241, 1 shell; 106250, 2 shells, 104831, 5 shells; all from type locality (all examined).

**Type locality.**　BRAZIL. **Bahia**. Carinhanha, Serra do Ramalho, Gruna do Cesário, 14˚19'S 43˚47'W (Bichuette col., 12.ix.2008).

**Diagnosis.**　Size about 30 mm, ~1.6 times longer than wide; lacking dorso-ventral compression. Apex with same color as remaining shell. Subsutural lighter band present. Peristome with brown spots. Delicate spiral striae present. Aperture occupying ~48% of length and ~60% width. Implantation of outer lip slightly horizontal. Inner lip with high middle fold. Umbilicus wide. Secondary columellar muscles with 7 insertions in left and right. Two pairs of m1v. gastric posterior duct Y-shaped. Jaw rectangular. Odontophore cartilages ~60% fused. Pair m10 narrow. Carrefour duct narrow and long, inserted between albumen gland and accessory albumen chamber. Albumen chamber sac-like. Two sperm folds in spermoviduct. Penis ~100% of spermoviduct length, lacking umbrella-like fold; epiphallus ~20% of penis length; penis muscle at epiphallus tip.

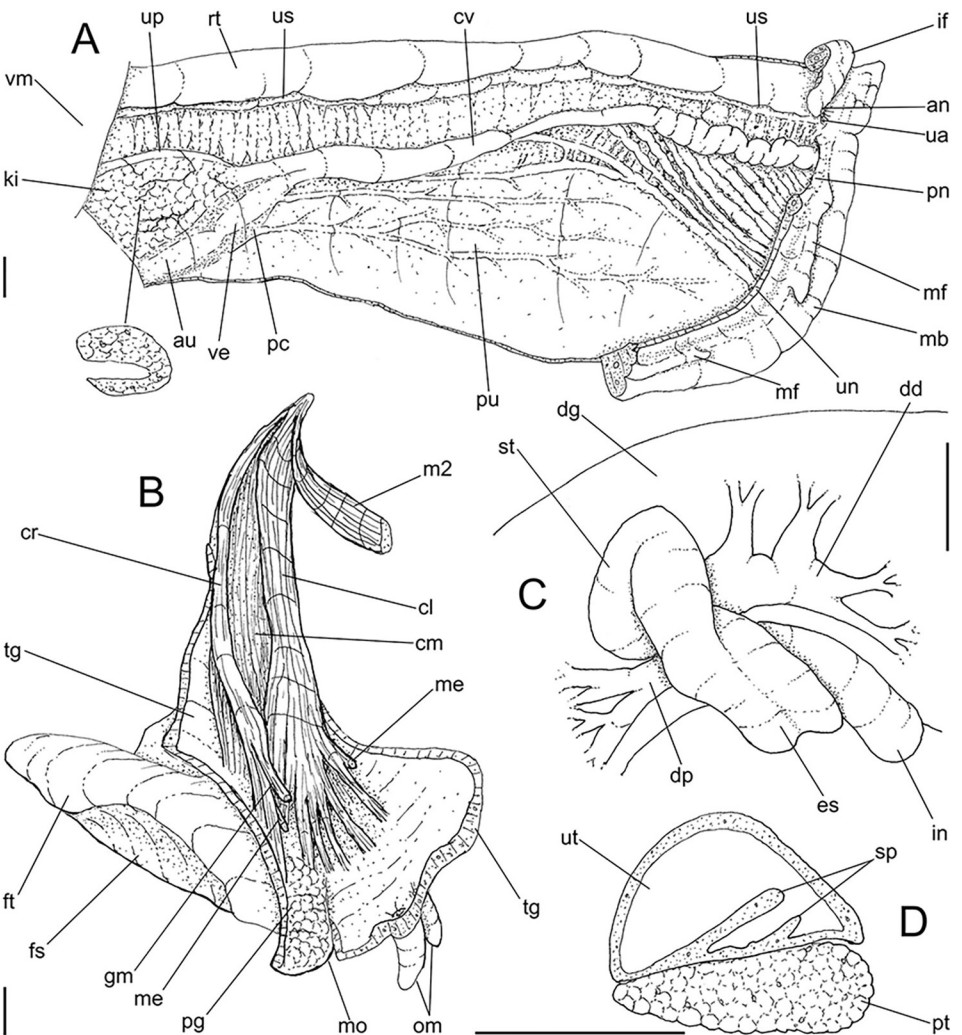

**Fig 9. *Kora nigra* anatomical drawings.** (A) extended pallial (pulmonary) cavity, ventral-inner view, inner edge of pneumostome sectioned and deflected upwards, transverse section of indicated region of kidney also shown. (B) head-foot, dorsal view, head, dorsal integument and internal organs removed, remaining muscles expanded. (C) midgut, ventral view, esophagus slightly displaced ventrally. (D) spermoviduct, transverse section of middle region. Scales = 2 mm.

## Distinctive redescription

*Shell.* (Figs 7 and 8A–8C) Proper description in [2]. Complement: maximum length up to 43 mm, vast majority ~30–35 mm; outline fusiform-obese, ~1.8x longer than wide. Color dark (Fig 7A–7O) to light brown (Fig 8A–8C), with narrow white subsutural band (Figs 7B, 7I, 7K, 7N and 8B); peristome white with some brown tinting in different degrees (high: Fig 7A, lower: Figs 7C, 7J, 7M and 8A). Apical view rounded (Fig 7G). Protoconch (Fig 7F) with ~2 whorls, wide; first whorl smooth, weak axial riblets gradually appearing in second whorl. Callus weak (Fig 7D, 7E, 7L, 7O). Aperture with inner lip bearing strong oblique, middle, slightly narrow fold (Figs 7E, 7L, 7O and 8C), almost forming stubby plica. Umbilicus broadly opened (Fig 7H), partially covered by inferior half of inner lip.

   **Head-foot.**   (Fig 8D and 9B) Similar character as preceding species, remarks and distinctions following. Integument with clear (urinary?) sulcus from pneumostome up to genital

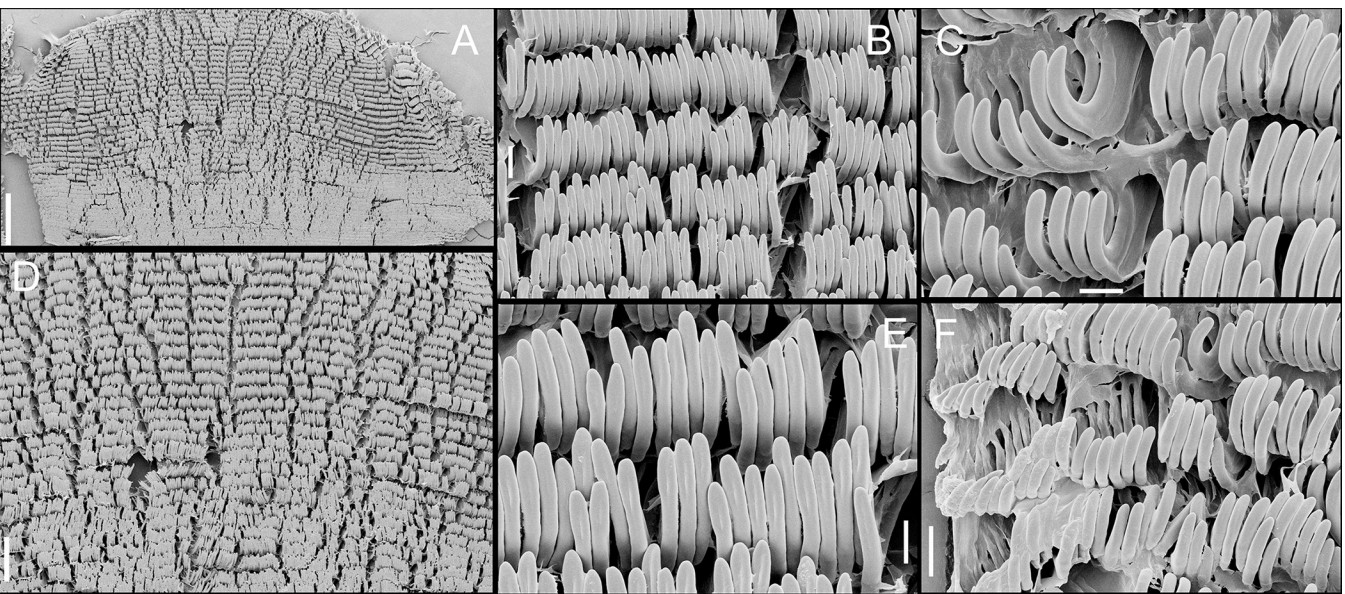

**Fig 10. *Kora nigra* radulae in SEM.** (A) whole view, scale = 500 µm. (B) detail of central region, scale = 30 µm. (C), detail of lateral region, scale = 20 µm. (D) detail of central region, scale = 200 µm. (E) detail of lateral region, scale = 20 µm. (F) detail of marginal region, scale = 30 µm.

aperture (Fig 8D: ug). Columellar muscle also divided into three bundles. Main columellar bundle (cm) slightly narrower, with ~1/2 foot width. Left secondary columellar/cephalic muscle (cl) with ~half of main columellar bundle (cm) width; with 7 transversely aligned narrow anterior branched, medial one larger, inserted more posteriorly in haemocoelic floor, remaining aligned perpendicularly from floor to left wall, being 2 dorsal ones longer than remaining, and ommatophore muscle as last one (me). Right secondary columellar/cephalic muscle (cr) similar to left one, but ~half its width; anteriorly only 5 branches, medial branch wide and more posteriorly inserted, remaining branches more anteriorly inserted on right wall, 2 more dorsal branches as ommatophore (me) and genital (gm) muscles. Pedal gland (pg) weakly protruding in posterior region of buccal area.

**Mantle organs.** (Fig 9A) Characters similar to those of preceding species, important features following. Mantle border (mb) with pair of small left and right folds (mf), with middle end pointed. Pulmonary venation well-developed, especially in region preceding pneumostome at along almost half of pulmonary cavity; left 2/3 with 3 longitudinal vessels draining to pericardial region; right 1/3 mostly having perpendicular weak vessels rather uniformly distributed, becoming taller only in close region preceding pneumostome; left region of pulmonary vein (cv) with set of oblique, successively larger vessels, located very close from each other, interspaces bearing transverse secondary vessels. Pulmonary vein (cv) broad, particularly in both ends. Reno-pericardial broadly triangular, occupying ~15% of cavity length and ~60% of its width (details below). Rectum (rt) wide. Urinary aperture (ua) simple, directly outside at right in pneumostome region, almost in continuation to anus (an).

**Visceral mass.** General features similar to preceding species.

**Circulatory and excretory systems.** (Fig 9A) Features similar to preceding species (check proportions above), except for slightly narrower heart and pericardium (pc), broader kidney (ki), and by kidney lobe being single, U-shaped glandular mass (check detail of transverse section on Fig 9A).

**Digestive system.** Most characteristics as those described for *K. corallina*, remarks and distinctions following. Jaw plate (Fig 8E) slightly thinner, yellow; cutting edge weakly convex,

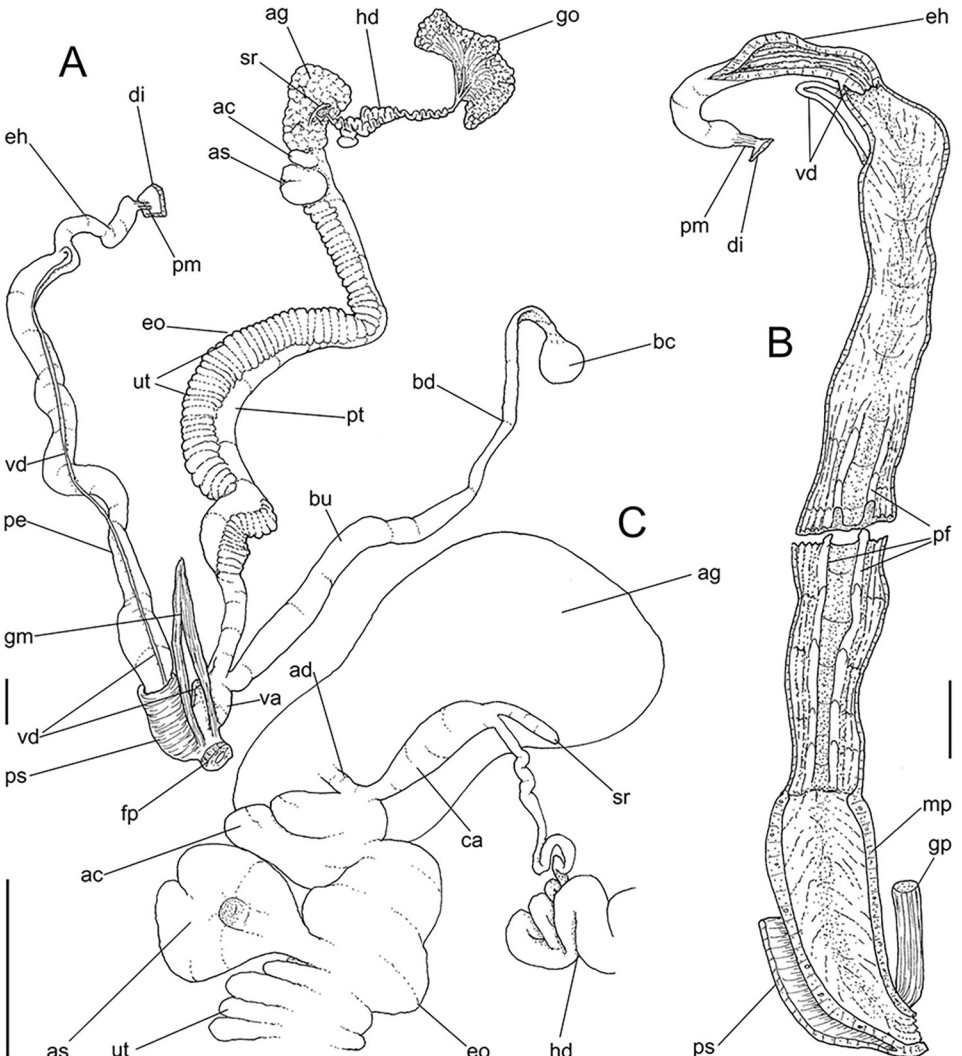

**Fig 11. *Kora nigra* anatomical drawings.** (A) genital structures, dorsal view, mostly uncoiled. (B) penis, ventral view, longitudinally opened, transverse section in its middle level also done. (C) genital structures in albumen gland level if it was transparent, ventral view. Scales = 2 mm.

with small notch at middle. Buccal mass also with similar attributes; oral cavity with dorsal folds slightly taller, especially in their posterior region; aperture of salivary glands equally positioned, but wider. Odontophore also with same characters, except for having minor fusion degree between both odontophore cartilages––~60% fused. Radular sac also sheltered inside bulged translucent sac. **Radula.** (Fig 10) ~2 times longer than odontophore. Structure and characters as those described for *K. corallina*. Except in having slightly more pairs of teeth, ~300 pairs per row (Fig 10A and 10D), and each teeth slightly more robust, being weakly broader, with shorter cusp (Fig 10B, 10C, 10E and 10F). Stomach (st) (Fig 9C) also with similar features, including position of ducts to both lobes of digestive gland; anterior duct (dd) much shorter, possessing only 4 strong branches; posterior duct (dp) also shorter, bifurcated in its base.

**Reproductive system.** (Figs 9D and 11) General organization similar to preceding species, remarks and distinctions following. Gonad proportionally larger (~1/2 whorl), composed of 8

lobes. Hermaphroditic duct (Fig 11A: hd) intensely coiled, mainly in region at some distance from its insertion, having some broader loops (Fig 11A, 11C); short portion very narrow preceding its insertion (Fig 11C). Seminal receptacle (Fig 11C: sr) very small, straight, with uniform width along its length, ~5 times longer than wide, flattened. Fertilization complex or carrefour (Fig 11C: ca) simple, wide, ~twice longer than receptacle and 3-times its width; receptacle and hermaphroditic duct inserting in its distal tip; bluntly tapering in basal end up to its insertion in beginning of albumen chamber (Fig 11C: ac). Duct of albumen gland (Fig 11C: ad) narrow, inserted in albumen chamber by side of that of carrefour. Albumen chamber relatively large, as bifid chamber (Fig 11C: ac); narrowly connected to distal end of spermoviduct (eo). Spermoviduct (Fig 11A, C: eo) ~25 times longer than wide. Secondary albumen chamber (as) weakly larger than primary chamber (described above), connected to spermoviduct some distance anterior to it by narrow duct (Fig 11C). Prostate wide (pt), ~1/3 of spermoviduct diameter (Fig 9D); uterus with weakly glandular walls, highly, transversally, and uniformly folded (Fig 11A: ut). Sperm inner longitudinal fold (Fig 9D: sp) with simple, tall, thick primary fold, and second small fold all along spermoviduct length. Remaining features of basal region of genital structures similar to preceding species, except for uterus having more uniform and narrow folds, and by duct of bursa copulatrix having taller and more irregular folds in its base. Penis proportionally longer, almost as long as spermoviduct if straightened (Fig 11A: pe). Penis shield (ps) with ~1/10 penis length. Penis with especial thick muscular region in its basal third. Epiphallus (eh) ~1/4 of penis' length, amply opened to penis; vas deferens insertion and abrupt narrowing marking its limit (Fig 11A: vd). Epiphallus inner surface with ~8 uniform, narrow, parallel folds (Fig 11B: eh); its basal end bulging inside penis distal end, papilla-like, partially covering vas deferens aperture (Fig 11B: vd). Internal penial arrangement of folds also clearly with three regions (Fig 11B): (1) basal third, corresponding to strongly muscular region, with smooth surface; (2) middle third, with strong pair of tall, simple, longitudinal folds in a side, smooth area between both; small, narrow, longitudinal secondary folds surrounding both main folds; these folds suddenly appearing just after basal muscular portion, fading in region preceding distal third; (3) distal third region almost smooth, only divergent/coalescent, weak, narrow wrinkles.

**Spermatophore.** (Fig 8F–8G) Found inside duct of bursa copulatrix. Chitinous walls, color greenish-yellow. Having 2 portions: (1) basal portion, ~70% of its length, slender hollow stem; walls longitudinally folded, translucent, relatively hard; basal aperture irregular; superior region gradually broadening, with smooth walls; (2) distal portion, ~30% of its length, bulged, elongated sperm chamber; tip flaccid, sharp pointed, irregular; base connected to stem in a side, with irregular, disform aperture in other side; walls relatively firm, weakly flattened.

**Central nervous system.** Same characters as preceding species.

**Distribution.** *Kora nigra* has so far been known both, in Bahia and Minas Gerais, but in regions close from each other, close to their border and very close to São Francisco River, in the region of Serra do Ramalho, Bahia, and in region of Itacarambi, Minas Gerais, Brazil.

**Habitat.** Under rocks, limestone areas.

**Measurements** (in mm): MZSP 106230#1 (Fig 6J–6K): 40.6 by 23.2; 106230#2 (Fig 6C–6E): 41.8 by 22.4; MZSP 151827 (Fig 6M–6O): 43.3 by 24.5; MZSP 151830 (Fig 6A–6C): 39.9 by 22.5.

**Material examined.** All types. BRAZIL (W Vailant-Mattos col.). **Minas Gerais**. Itacarambi, Serra de Itacarambi, 15˚01'42"S 44˚13'15"W, MZSP 151830, 10 shells (i.2020), Fazenda ICIL, 15˚00'39"S 44˚03'24"W, MZSP 151827, 4 shells, MZSP 151828, 4 spm (i.2020).

**Taxonomic remarks. Taxonomic remarks.** *Shell. Kora nigra* has an average shell size of approximately 30 mm, making it in the smallest category in the genus, a size range only shared with *K. aetheria*, *K. vania*, and *K. curumim*. Its shell is about 1.6 times longer than it is wide,

giving it an obese shape compared to most other congeneric species, a feature also shared with *K. ajar*, *K. curumim*, the remaining congeners are considerably slender. The shell aperture comprises about 48% of the total shell length, which is a medium measure, it is slightly shorter than the apertures of *K. rupestris*, *K. tupan*, *K. ajar*, *K. uhlei*, *K. kremerorum*, and *K. curumim*; but considerably longer than that of *K. corallina*, *K. aetheria*, *K. jimenezi* and *K. vania*. Additionally, like *K. tupan*, *K. ajar*, *K. jimenezi*, *K. kremerorum*, and *K. vania*, *K. nigra* has a horizontally oriented superior implantation of the outer lip. Also, the shell usually bears a very dark-brown color and dark spots in the peristome.

**Anatomy.** *Kora nigra* has in its mantle edge narrow folds with pointed end, a character only shared with *K. rupestris* (Fig 9A: mf). It differs from all congeners in only having large vessels draining to posterior region to the left of the pulmonary vein (Fig 9A). The simple anterior end of the pulmonary vein (Fig 9A: cv) distinguishes *K. nigra* from *K. corallina*, *K. rupestris* and *K. jimenezi*, which have branched structure. Additionally, *K. nigra* displays strong venation to the right of the pulmonary vein up to middle level of pulmonary cavity, a trait shared with *K. tupan*, *K. ajar*, *K. aetheria* and *K. uhlei*. The kidney lobe has a single solid lobe surrounding anterior, dorsal and ventral surfaces (Fig 9A: ki), a feature shared only with *K. tupan*. In terms of muscle structure, *K. nigra* has 7 anterior insertions of the left accessory columellar muscle (Fig 9B: cl), a condition only shared with *K. rupestris*. Additionally, it also has 7 anterior insertions of the right accessory columellar muscle (cr), similarly only to *K. corallina*, *K. rupestris* and *K. aetheria*. These accessory columellar muscles have a posteriorly located medial branch, which is present in all remaining congeners except *K. jimenezi* and *K. corallina*. Furthermore, *K. nigra* lacks the odontophore muscle pair m1l, a feature present in *K. rupestris*, *K. tupan*, *K. aetheria*, *K. jimenezi*, and *K. uhlei*. The presence of two pairs of odontophore muscles m1v differentiates it from *K. tupan*, *K. ajar*, *K. jimenezi*, and *K. uhlei*, which lack these muscles, and from *K. aetheria*, which has only one pair. The connection of the odontophore muscle pair m3 to the esophagus origin distinguishes *K. corallina* from *K. rupestris*, where it connects to the m2 pair, and from *K. jimenezi*, which lacks this muscle. In having a Y-shaped posterior duct to the digestive gland (Fig 9C: dp), *K. nigra* differs from all congeners, and in having branches only in the right side in the anterior duct to the digestive gland differentiated it from *K. corallina*, but its short shape, with broad, stubby branches is exclusive. The rectangular shape of the jaw plate (Fig 8E) is shared only with *K. tupan*, *K. ajar* and *K. uhlei*. The salivary gland aperture, located in the middle third of the buccal dorsal wall, differs from *K. tupan*, and the absence of a salivary papilla sets it apart from *K. jimenezi* and *K. uhlei*. The degree of fusion of the odontophore cartilages, around 60%, puts *K. nigra* as the species with the shorted fusion degree amongst the congeners. In terms of odontophore m4-m5 pairs of muscles, the m4 muscle covering m5 distinguishes *K. nigra* from *K. tupan* and *K. uhlei*, in which these muscle pairs are continuous with each other. The narrow odontophore muscle m7, with a single origin, differs from the conformations in *K. tupan*, *K. jimenezi*, and *K. uhlei*. Finally, the narrow m10 muscle in *K. nigra* differs from the broader form in *K. tupan* and *K. aetheria*, and from the filiform version in *K. jimenezi*.

**Genital system.** *Kora nigra* is the only species that lacks a small curve at the end of the hermaphrodite duct (Fig 11C: hd), which distinguishes its congeners. The elongated shape of its carrefour (Fig 11C: ca) is distinct from all remaining species, connecting directly to spermoviduct practically lacking duct. The species lacks the bulged portion on the opposite side of the hermaphrodite duct at the base of the seminal receptacle (sr), differentiating it from *K. tupan*, *K. jimenezi*, and *K. uhlei*. *K. nigra* has the carrefour inserting between the albumen gland duct (ad) and the beginning of the albumen chamber (ac) (Fig 11C), a condition only shared with *K. tupan*, *K. aetheria* and *K. uhlei*. Its albumen chamber (Fig 11C: ac) is sac-like, a similar condition shared with *K. rupestris*, *K. tupan*, and *K. aetheria*. *K. nigra* is the single species in having

two sperm folds along the spermoviduct (Fig 9D: sp). Additionally, it has the narrowest prostate band in the spermoviduct (~35%) among its congeners, except for *K. corallina* and *K. aetheria*, which share similar proportions. The muscular anterior portion of the bursa copulatrix duct (Fig 6A: bu) is well-defined, setting *K. nigra* apart from *K. jimenezi*. Its penis length is as long as the spermoviduct, a condition only shared with *K. jimenezi*. The bursa copulatrix duct is about 90% of the spermoviduct length, which is shorter than in *K. rupestris* and *K. tupan*, but longer than in *K. aetheria*, *K. jimenezi*, and *K. uhlei*. The vas deferens of *K. nigra* has the strong curve preceding its insertion at the tip of the penis, approaching it from *K. corallina*, *K. tupan*, *K. ajar*, and *K. uhlei*. Its penis base has clear muscular walls (Fig 11B: mp), unlike *K. jimenezi*, which lacks them. The penis of *K. nigra* features the usual pair of inner folds, but it is simple, straight lacking branches (Fig 11B: pf), a characteristic shared only with *K. rupestris*, *K. ajar* and *K. jimenezi* among its congeners. *K. nigra* lacks the umbrella-like transverse penial fold found in most of its congeners, with the exceptions of *K. corallina* and *K. jimenezi*. The epiphallus comprises about 25% of the penial length, being longer than *K. corallina*, *K. rupestris* and *K. tupan*, but shorter than *K. aetheria*. *K. nigra* lacks strong longitudinal fold in the epiphallus (Fig 11B), setting it apart from *K. corallina*, *K. tupan* and *K. ajar*. Its penis muscle (pm) inserts terminally in the epiphallus, unlike *K. rupestris*, *K. ajar*, and *K. uhlei*, which have more basal insertions.

***Kora rupestris* Salvador & Simone, 2016 Figs 12–15.** *Kora rupestris* Salvador & Simone, 2016 [3]: 1–6 (fig 4–11, 15–17), 2021 [21]: 396–397 (fig 2A–B); Simone, 2022 [24]: 44 (fig 7); MolluscaBase, 2023 [11].

*Kora* sp: Cavallari et al., 2016 [21]: 15 (fig 7).

**Types.** Holotype MZSP 121416; Paratypes: MZSP 121441, 2 shells from type locality (all examined).

**Type locality.** BRAZIL. Bahia; Carinhanha, Canabrava Hill, ~14˚18'18"S 43˚45'54"W (A. Bianchi col., viii/2012).

**Diagnosis.** Size about 45 mm, ~2.3 times longer than wide; lacking dorso-ventral compression. Apex with light color. Subsutural lighter band present. Peristome white. Delicate spiral striae present. Aperture occupying ~50% of length and ~66% width. Implantation of outer lip slightly vertical. Inner lip with high middle fold. Umbilicus wide. Secondary columellar muscles with 7 insertions in left and in right. Two pairs of m1v; additional pair m3 close to m2 insertion. Jaw with central notch. Odontophore cartilages ~75% fused. Pair m10 narrow. Carrefour duct narrow and long, inserted in tip of spermoviduct. Albumen chamber sac-like. Penis ~60% of spermoviduct length, umbrella-like fold present, with 5 rods; epiphallus ~15% of penis length; penis muscle at epiphallus base.

**Distinctive redescription.** *Shell.* (Fig 12A–12K) Complement to original description [3]. Complement: length up to 46.5 mm, outline rather conic, elongated, ~2.2x longer than wide. Transverse section circular (Fig 12G). Color orange (Fig 12A, 12B) to brown (Fig 12C–12H), subsutural pale band (Fig 12C, 12F, 12I–12K); peristome completely white (Fig 12A, 12B) or with outer lip brown (Fig 12C, 12F, 12I, 12K). last whorl having only narrow axial undulations and growth lines (Fig 12B) or associating minute spiral striae more (Fig 12D, 12E) or less (Fig 12J) dense. Callus low, weak (Fig 12A, 12C, 12F, 12I, 12K). Aperture with inner lip bearing strong oblique, relatively tall, narrow fold (Fig 12F, 12K), almost forming stubby plica. Outer lip implantation more vertical shaped (Fig 12A), or more horizontal implanted (Fig 12I), with intermediaries (Fig 12C). Umbilicus narrowly opened (Fig 2H), partially covered by inferior half of inner lip.

**Head-foot.** With similar features as *K. nigra*. Except for left secondary columellar muscle slightly broader; and by pedal gland deeper and more developed.

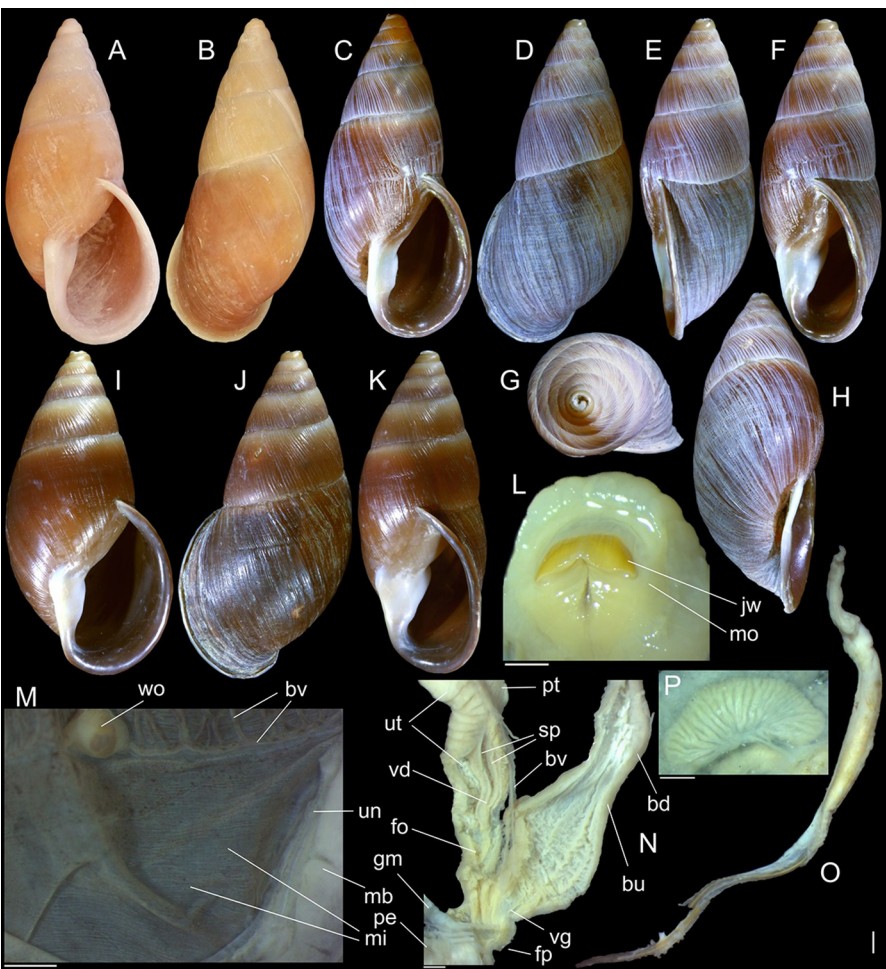

**Fig 12. *Kora rupestris* shell and anatomical characters, light photos.** (A–B) Holotype shell MZSP 121416, frontal and dorsal views (L 37.7 mm). (C) shell of dissected specimen MZSP 151890#3, frontal view (L 46.1 mm). (D) same, dorsal view. (E) same, right view. (F) same, right-slightly ventral view. (G) same, apical view. (H) same, left-slightly anterior view, showing umbilicus. (I–K) shell MZSP 151891, frontal, dorsal and right views (L 44.1 mm). (L) mouth, ventral view, MZSP 151890#2 with jaw exposed. (M) pallial (pulmonary) cavity, detail of anterior-left region, ventral view, MZSP 151890#1. (N) basal region of genital system, dorsal view, both left tubes (right in Fig) opened longitudinally with inner surface exposed, only penis base shown, MZSP 151890#1. (O) spermatophore found inside duct of bursa copulatrix of MZSP 151890#1 (some mucus still covering it) (P) gonad in situ, ventral view, #2. Scales = 1 mm.

**Mantle organs.** (Figs 12M, 13A) Most features similar to *K. corallina*, distinctions and remarks following. Mantle border (mb) with pair of secondary folds (mf) small and pointed, located in opposed sides. Pulmonary venation weakly poorly developed, including in region preceding pneumostome, possessing only transverse, low vessels. Pulmonary vein (cv) running longitudinally across pallial cavity roof. Pallial wall in entire pulmonary cavity with uniform cover of minute longitudinal muscle fibers (Fig 12M: mi). Primary (up) and secondary (us) ureters entirely slightly broader.

**Visceral mass.** With same characters as *K. corallina*.

**Circulatory and excretory systems.** (Fig 13A) General characteristics as those described for *K. corallina*, except for pericardium (pc) slightly narrower if compared to kidney; and for arrangement of kidney lobe, as 4 successive, transverse, tall folds.

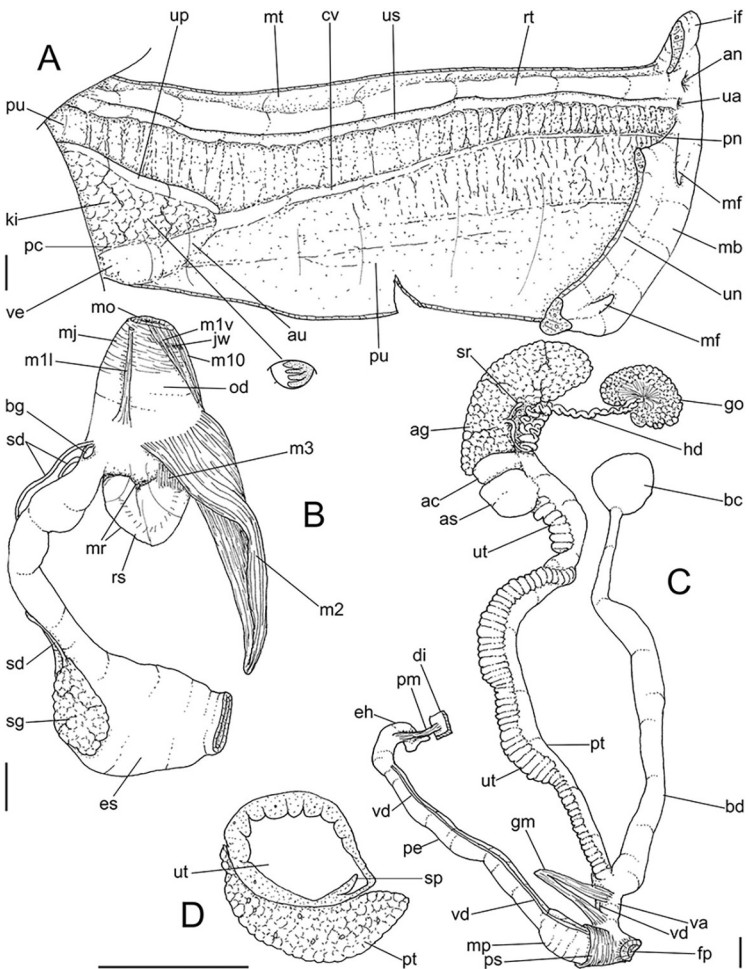

**Fig 13.** *Kora rupestris* **anatomical drawings.** (A) extended pallial (pulmonary) cavity, ventral-inner view, inner edge of pneumostome sectioned and deflected upwards, transverse section of indicated region of kidney also shown. (B) foregut, right view. (C) genital structures, dorsal view, mostly uncoiled. (D) spermoviduct, transverse section of middle region. Scales = 2 mm.

**Digestive system.** (Fig 13B) Overall morphology similar to that of *K. corallina*. Differences and remarks following. Peribuccal circular muscles (mj) longer and thicker. Jaw plate (Fig 12L) strongly notched at middle; sculptured by successive, rather uniform, transverse, wide folds. Buccal cavity with esophageal folds (ef) slightly taller and dark pigmented, having salivary apertures (sa) in their outer-anterior edges. Odontophore intrinsic, extrinsic muscles, and other structures similar to *K. corallina*, except for **m1v**, pair single and wider; **m1l**, narrow pair of jugal lateral protractor muscles, originating in lateral region of mouth, running towards posterior, inserting in lateral region of buccal mass posterior end; **m3**, pair short and wide, originating in inner insertion of m2, shortly inserting along membrane surrounding radular sac; **m10**, broader and more visible. **Radula** (Fig 14) with same attributes as *K. corallina*, except for tip of teeth slightly more rounded, with subterminal furrow shorter (Fig 14B, 14D); and marginal teeth slightly more arched (Fig 14E). Stomach with anterior duct to digestive gland slightly narrower.

**Reproductive system.** (Figs 12N–12O and 13C, 13D and 15) General structures similar to preceding species, remarks and distinctions following. Gonad (go) proportionally larger, not

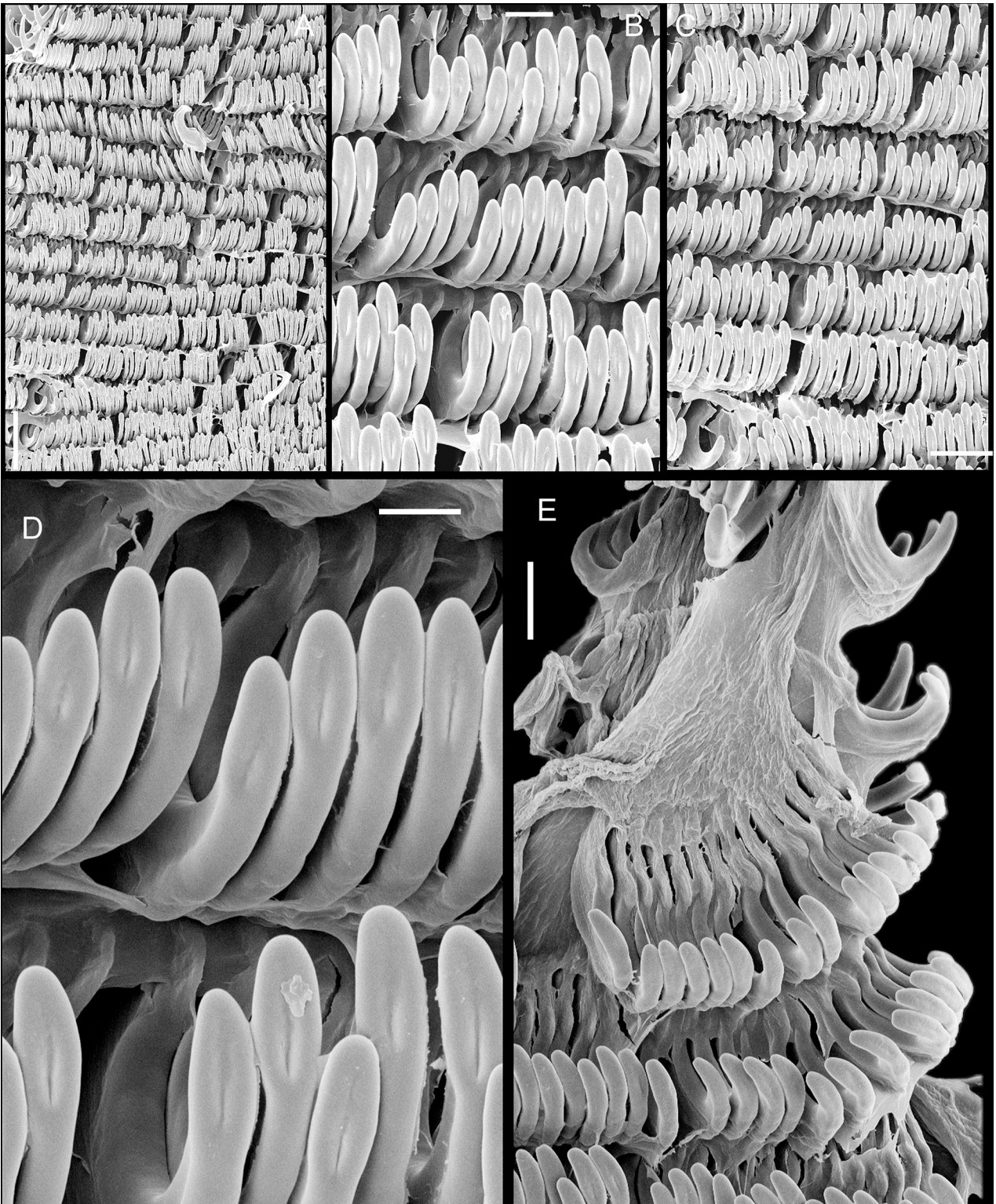

**Fig 14. *Kora rupestris* radulae in SEM.** (A) central view, scale = 100 μm. (B) same, detail, scale = 20 μm. (C) lateral region, scale = 50 μm. (D) same, detail, scale = 10 μm. (E) marginal region, scale = 30 μm.

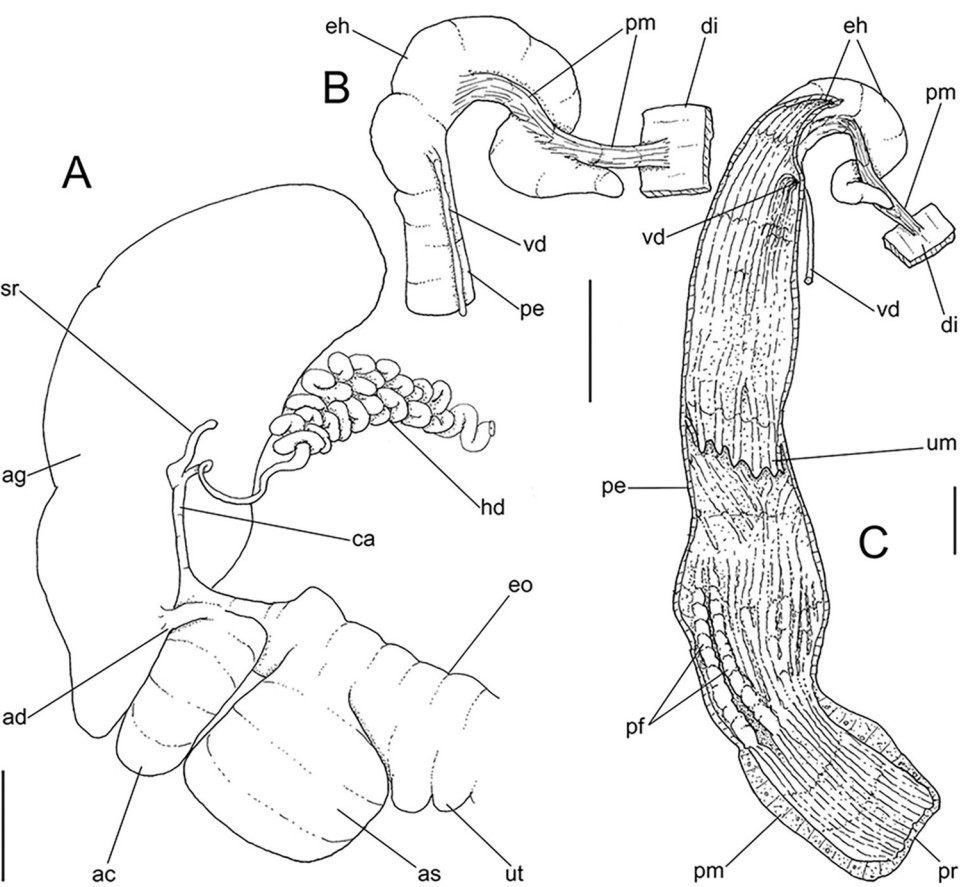

**Fig 15. *Kora rupestris* anatomical drawings.** (A) genital structures in albumen gland level if it was transparent, ventral view. (B) penis, uncoiled distal end. (C) penis, ventral view, longitudinally opened. Scales = 2 mm.

clearly divided in lobes (Fig 12P). Hermaphroditic duct (Figs 13C and 15A: hd) intensely coiled, particularly in its half close to its insertion; except for short narrow region preceding its insertion (Fig 15A). Seminal receptacle minuscule (Fig 15A: sr), straight, long, tapering gradually, ~8 times longer than wide, tip rounded. Fertilization complex or carrefour (Fig 15A: ca) as simple, long, relatively wide duct, lacking bulged regions; inserting directly in spermoviduct beginning. Albumen gland (ag) ~5 twice larger than gonad. Albumen gland duct (Fig 15A: ad) subterminal, connected to spermoviduct as narrow, separated duct. Albumen chamber (ac) as blind sac, ~twice longer than wide, connected to spermoviduct beginning by narrow region, jointed to duct of albumen gland. Secondary albumen chamber (as) ~3-times larger than primary chamber, with its same length, broad, abruptly tapering in its insertion at short distance from albumen chamber insertion. Spermoviduct (eo) slightly narrower, slender ~30 times longer than wide. Prostate wide (pt), ~1/2 of spermoviduct diameter (Fig 12D); uterus with glandular walls, highly, transversally, and relatively uniformly folded (Fig 13C: ut). Sperm inner longitudinal fold (Fig 13D: sp) simple, low, thick fold, a second small fold gradually appearing in basal third; both folds fusing with each other, originating vas deferens, slightly anterior to end of uterine level (Fig 12N: vd). Genital muscle in two bundles attached to vaginal outer wall (Fig 13C: gm). Bursa copulatrix (bc) and its duct (bd) almost as long as spermoviduct length (Fig 13C); bursa duct with basal 2/3 with walls extraordinarily thick muscular (Figs 12N: bu, 13C). Basal region of duct of bursa copulatrix with special arrangement of irregular inner folds

as shown in Fig 12B: bd. Penis slightly mostly straight, ~60% of spermoviduct length if straightened (Fig 12C: pe). Penis muscle inserted along lateral walls of epiphallus, leaving tip free (Fig 15B, 15C: pm), long, simple. Penis walls weakly muscular, except for region adjacent to penis shield (Figs 13C, 15C:mp). Epiphallus (eh) ~1/7 of penis' length, amply opened to penis; only vas deferens insertion marking its limit (Fig 17C: vd) and weak sudden change of folds. Epiphallus inner surface only with 8–10 narrow, small, parallel folds (Fig 15C: eh). Internal penial arrangement of folds clearly with 4 regions (Fig 15C): (1) basal 1/4, highly muscular region (pm), possessing mosaic of low, uniform, longitudinal folds; (2) sub-basal 1/4, with strong pair of tall, longitudinal folds in a side, located close from each other, each one with simple margins; mosaic of low, longitudinal, rather irregular folds flanking both strong folds; (3) sub-distal 1/4, with basal region almost smooth, only having 4–5 oblique, low folds, separated from each other; **umbrella-like fold** (Fig 15B: um) located distally to this region, tall and narrow, possessing 5 rods exceeding fold's edge, septum-like, inserted transversally in inner penis wall; (4) distal region, possessing only longitudinal folds, those more basal as continuation of umbrella-like fold rods, gradually fading posteriorly, becoming lower and more numerous; some of them converging to aperture of vas deferens (Fig 15B: vd), other continuous with epiphallus inner folds.

**Central nervous system.**  Same characters as *K. corallina*, the single detected difference is slightly shorter connectives.

**Distribution.**  *Kora rupestris* is the only species with a wide range. It is known from Serra do Ramalho, Bahia (~13.5˚S) up to Januária, Minas Gerais (~15˚S), a range of ~250 km. It is also, the single species that occurs in both sides of the São Fransisco River, with a sample collected in Iuiú, a city in east side from São Francisco River (the others are in the west side). As the species' shell is very characteristic, and very distinct from the other species, and the access to complete specimens was very scanty (Table 1) (only samples from Januária, Minas Gerais were complete), it was not possible to check if all these populations are really conspecific. A conservative approach was so far adopted, based on the conchology only.

**Habitat.**  Under rocks, limestone areas.

**Measurements.**  (in mm) MZSP 121416#3 (Fig 9C–9H): 46.1 by 20.2; MZSP 151891 (Fig 9I–9K): 44.1 by 21.8.

**Material examined.**  All types. BRAZIL (W Vailant-Mattos col.). **Bahia**. Cocos, Itaguari, 14˚37'06"S 45˚31'13"W, altitude 730 m, MZSP 151920, 8 shells (ii.2020); Iuiu, 14˚25'50"S 43˚33'50"W, MZSP 151765, 5 shells (i.2022); Serra do Ramalho, Pedreira, 13˚32'44"S 43˚50'28"S, MZSP 151956, 26 shells (v.2019). **Minas Gerais**. Januária, border of National Park Paruaçu, 14˚57'55"S 44˚04'21"W, MZSP 151890, 5 spm, MZSP 151891, 9 shells (i.2020), Barreiros, 15˚28'47"S 44˚22'03"W, MZSP 152132, 5 shells (iv.2020); Cônego Marinho, 15˚19'15"S 44˚27'52"W, MZSP 152163, 4 shells (iv.2020), Vale do Gala, 15˚19'15"S 44˚27'52"W, MZSP 152062, 5 shells (iv.2020)

**Taxonomic remarks.**  *Shell. Kora rupestris* has an average shell size of approximately 45 mm, making it larger than *K. nigra*, *K. aetheria*, *K. vania*, and *K. curumim*, but smaller than *K. tupan* and *K. ajar*. Its shell is about 2.3 times longer than it is wide, giving it a narrower shape compared to most other congeneric species, except *K. corallina*, from which it differs by a more rounded outline. The shell aperture comprises about 50% of the total shell length, which is shorter than the apertures of *K. tupan* and *K. ajar*, but longer than *K. corallina*, *K. nigra*, *K. aetheria*, *K. jimenezi* and *K. vania*. Additionally, unlike *K. nigra*, *K. tupan*, *K. ajar*, *K. jimenezi*, *K. kremerorum*, and *K. vania*, *K. corallina* lacks a horizontally oriented superior implantation of the outer lip. *K. rupestris* is distinct in having the more conic, pointed shell shape amongst its congeners,

**Anatomy.** *Kora rupestris* has in its mantle edge narrow folds with pointed end, a character only shared with *K. nigra* (Fig 13A: mf). It differs from *K. nigra*, *K. aetheria*, and *K. jimenezi* by having a single intercalated pair of wide vessels to the left of the pulmonary vein (Fig 13A), as these species exhibit different vascular arrangements. The simple anterior end of the pulmonary vein (Fig 13A: cv) distinguishes *K. rupestris* from *K. corallina*, *K. nigra* and *K. jimenezi*, which have branched structure. Additionally, *K. rupestris* displays almost absent venation to the right of the pulmonary vein being a unique condition among its congeners. The kidney lobe has four tall anterior folds (Fig 13A), differentiating from all remaining species. In terms of muscle structure, *K. rupestris* has 7 anterior insertions of the left accessory columellar muscle, a condition only shared with *K. nigra*. Additionally, it has 7 anterior insertions of the right accessory columellar muscle, setting it apart from *K. tupan*, *K. ajar*, *K. jimenezi*, and *K. uhlei*. These accessory columellar muscles have a posteriorly located medial branch, present in all remaining congeners except *K. corallina* and *K. jimenezi*. Furthermore, *K. rupestris* has the odontophore muscle pair m1l, approaching it from *K. tupan*, *K. aetheria*, *K. jimenezi*, and *K. uhlei*. The presence of two pairs of odontophore muscles m1v (Fig 13B) differentiates it from *K. tupan*, *K. ajar*, *K. jimenezi*, and *K. uhlei*, which lack these muscles, and from *K. aetheria*, which has only one pair. The connection of the odontophore muscle pair m3 to the insertion of the pair m2 is exclusive of *N. rupestris*. In having branches only on the right side of the posterior duct to the digestive gland, *K. rupestris* differs from *K. nigra*, and its bilateral branching in the anterior duct to the digestive gland further differentiates it from all congeners, except *K. corallina* and *K. aetheria*, which also have bilateral branches. The central notch in the jaw plate (Fig 12L) is shared only with *K. corallina* and *K. aetheria*, distinguishing *K. rupestris* from other congeners. The salivary gland aperture, located in the middle third of the buccal dorsal wall (Fig 4A: sa), differs from *K. tupan*, and the absence of a salivary papilla sets it apart from *K. jimenezi* and *K. uhlei*. The degree of fusion of the odontophore cartilages, around 75%, is similar to most congeners but greater than that of *K. nigra* (~60˚) and less than that of *K. uhlei* (~90˚). In terms of odontophore m4-m5 pairs of muscles, the m4 muscle covering m5 distinguishes *K. rupestris* from *K. tupan* and *K. uhlei*, in which these muscle pairs are continuous with each other. The narrow odontophore muscle m7, with a single origin, differs from the conformations in *K. tupan*, *K. jimenezi*, and *K. uhlei*. Finally, the narrow m10 muscle in *K. rupestris* differs from the broader form in *K. tupan* and *K. aetheria*, and from the filiform version in *K. jimenezi*.

**Genital system.** *Kora rupestris* has a small curve at the end of the hermaphrodite duct (Fig 15A: hd), which distinguishes it from *K. nigra*. The shape of its carrefour (Fig 15A: ca) is entirely narrow, which is distinct from all its congeners. The species lacks the bulged portion on the opposite side of the hermaphrodite duct at the base of the seminal receptacle (sr), differentiating it from *K. tupan*, *K. jimenezi*, and *K. uhlei*. *K. rupestris* is unique in having the carrefour duct inserting directly in the tip of the spermoviduct (Fig 15A: ca). Its albumen chamber (Fig 15A: ac) is sac-like, a similar condition shared with *K. nigra*, *K. tupan*, and *K. aetheria*. In having a single sperm fold in the spermoviduct (Fig 13D: sp), *K. rupestris* differs from *K. nigra*, which has two. Additionally, it has the broadest prostate band in the spermoviduct (~50%) among its congeners, a condition shared with *K. jimenezi*. The muscular anterior portion of the bursa copulatrix duct (Fig 12N: bu) is well-defined, setting *K. rupestris* apart from *K. jimenezi*. Its penis length is approximately 60% of the spermoviduct, being the shorter proportion among its congeners, a condition shared with *K. uhlei*. The bursa copulatrix duct is as long as the spermoviduct, a condition only shared with *K. tupan*. The vas deferens of *K. rupestris* lacks the strong curve preceding its insertion at the tip of the penis, distinguishing it from *K. nigra*, *K. tupan*, *K. ajar*, and *K. uhlei*. Its penis base has clear muscular walls (Fig 15C: mp), unlike *K. jimenezi*, which lacks them. The penis of *K. corallina* features the usual pair of inner folds, but

it simple, relatively short, lacking branches (Fig 15C: pf), a characteristic shared with *K. nigra*, *K. ajar* and *K. jimenezi*; but these folds are distinct in being thick and apparently glandular. *K. rupestris* has the umbrella-like transverse penial fold, with 5 rods (Fig 15C: um) found in most of its congeners, with the exceptions of *K. corallina*, *K. nigra* and *K. jimenezi*. The epiphallus comprises about 15% of the penial length, a short ratio among its congeners, though *K. corallina* has a similar but still further shorter proportion. *K. rupestris* lacks a strong longitudinal fold in the epiphallus (Fig 15C), distinguishing it from *K. tupan* and *K. ajar*. Its penis muscle (pm) inserts in the base of the epiphallus, in the similar model of *K. rupestris*, *K. ajar*, and *K. uhlei*. The spermatophore of *K. rupestris* has the shortest basal rod if compared to those found in its congeners (Fig 12O).

*Kora tupan* new species Figs 16–20. ZooBank. urn:lsid:zoobank.org:act: A8261BE4-E22E-41FF-B845-2C21DCCB273B.

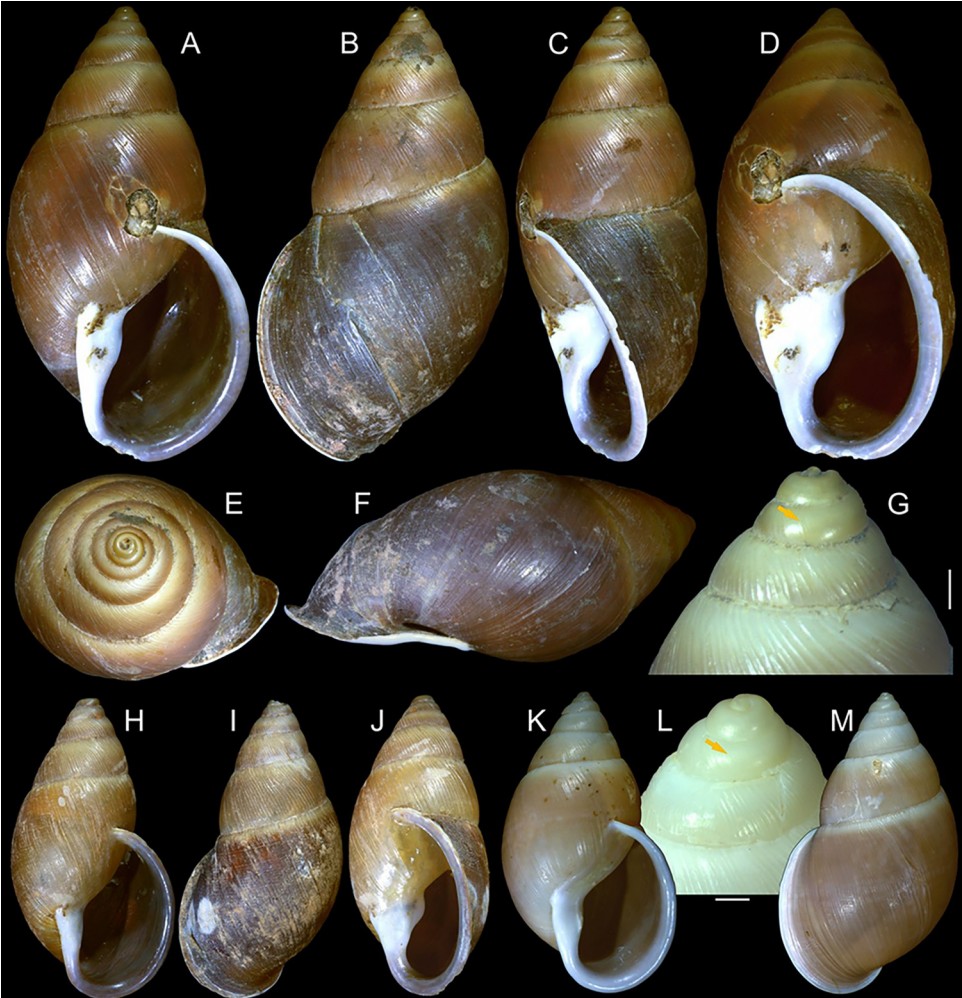

**Fig 16. *Kora tupan* shell characters.** (A) holotype MZSP 161200, frontal view (L 55.8 mm). (B) same, dorsal view. (C) same, right view. (D) same, right-slightly ventral and anterior view. (E) same, apical view. (F) same, left-slightly anterior view showing umbilicus. (G) apex, profile, arrow indicating transition protoconch-teleoconch. (H–J) paratype MZSP 151817, frontal, dorsal and right views (L 47.0 mm). (K) paratype MZSP 151823, frontal view (L 53.1 mm). (L) same, detail of apex, profile-slightly apical view, arrow indicating transition protoconch-teleoconch. (M) same, dorsal view. Scales = 1 mm.

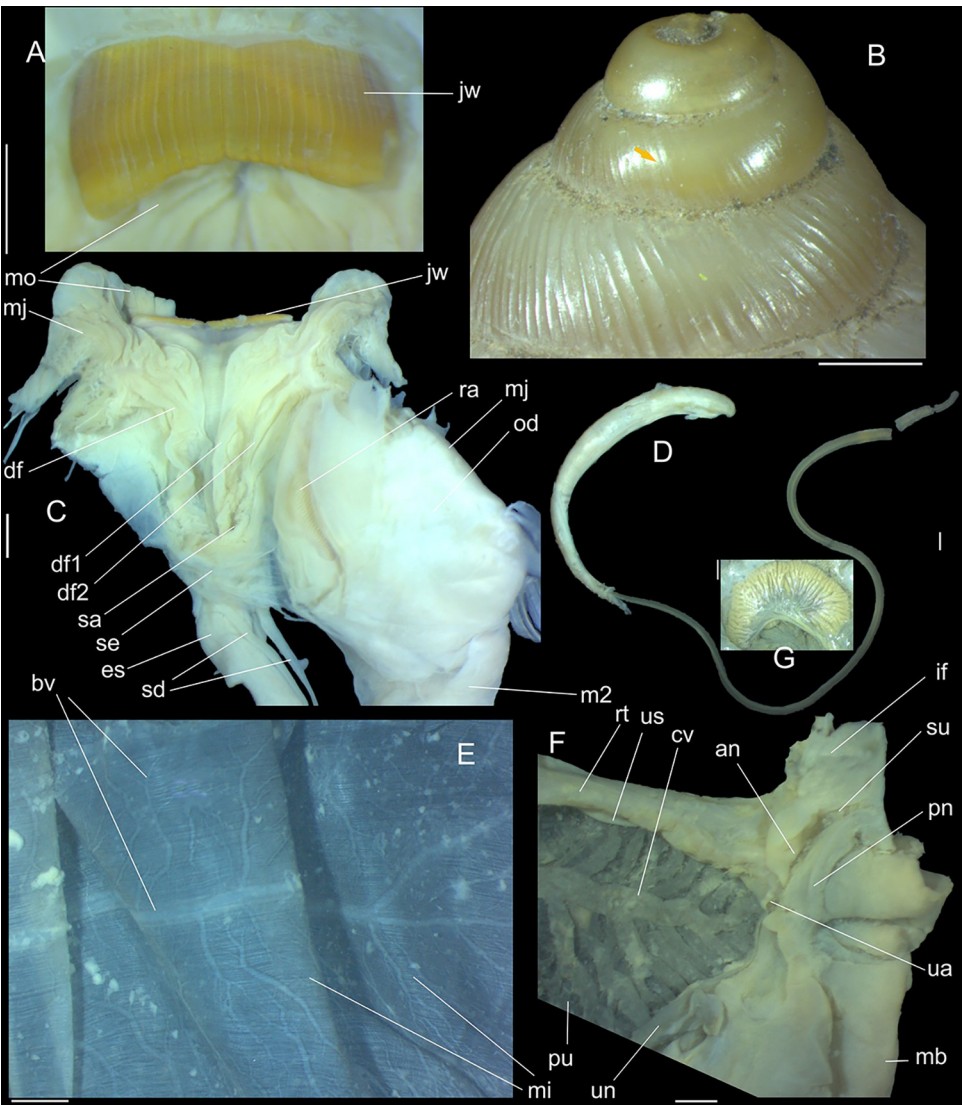

**Fig 17. *Kora tupan* shell and anatomical characters, light photos.** (A) jaw in situ, ventral view, MZSP 151817#1. (2) protoconch of holotype (MZSP 161200) and first teleoconch whorl, profile, arrow indicating transition protoconch-teleoconch. (C) buccal mass, ventral view, odontophore sectioned along left edge and deflected to right, inner dorsal surface exposed, MZSP 151817#1. (D) spermatophore found inside duct of bursa copulatrix of MZSP 151817#1 (broken at end). (E) pallial hoof (lung), ventral view, detail of antero-left region showing venation and pallial micro musculature, MZSP 151817#2. (F) region of pneumostome, ventral view, its inner edge sectioned and deflected upwards to show inner structures, MZSP 151817#1. (G) gonad in situ, ventral view, MZSP 151817#2. Scales = 1 mm.

**Types.** Holotype MZSP 161200, complete spm; paratypes: MZSP 151817, 3 spm, MZSP 151823, 1 shell, all from type locality.

**Type locality.** BRAZIL. **Minas Gerais**; São João da Ponte, Gruta do Índio, 15˚50'44"S 44˚00'03"W (W. Vailant-Mattos col., i.2020).

**Diagnosis.** Size about 55 mm, ~1.9 times longer than wide; dorso-ventrally slightly compressed. Apex with same color as remaining shell. Subsutural lighter band present. Peristome white. Delicate spiral striae present. Aperture occupying ~54% of length and ~70% width. Implantation of outer lip slightly horizontal. Inner lip with high middle fold. Umbilicus wide. Secondary columellar muscles with 8 insertions in left and 6 in right. pairs of m1v lacking. Jaw

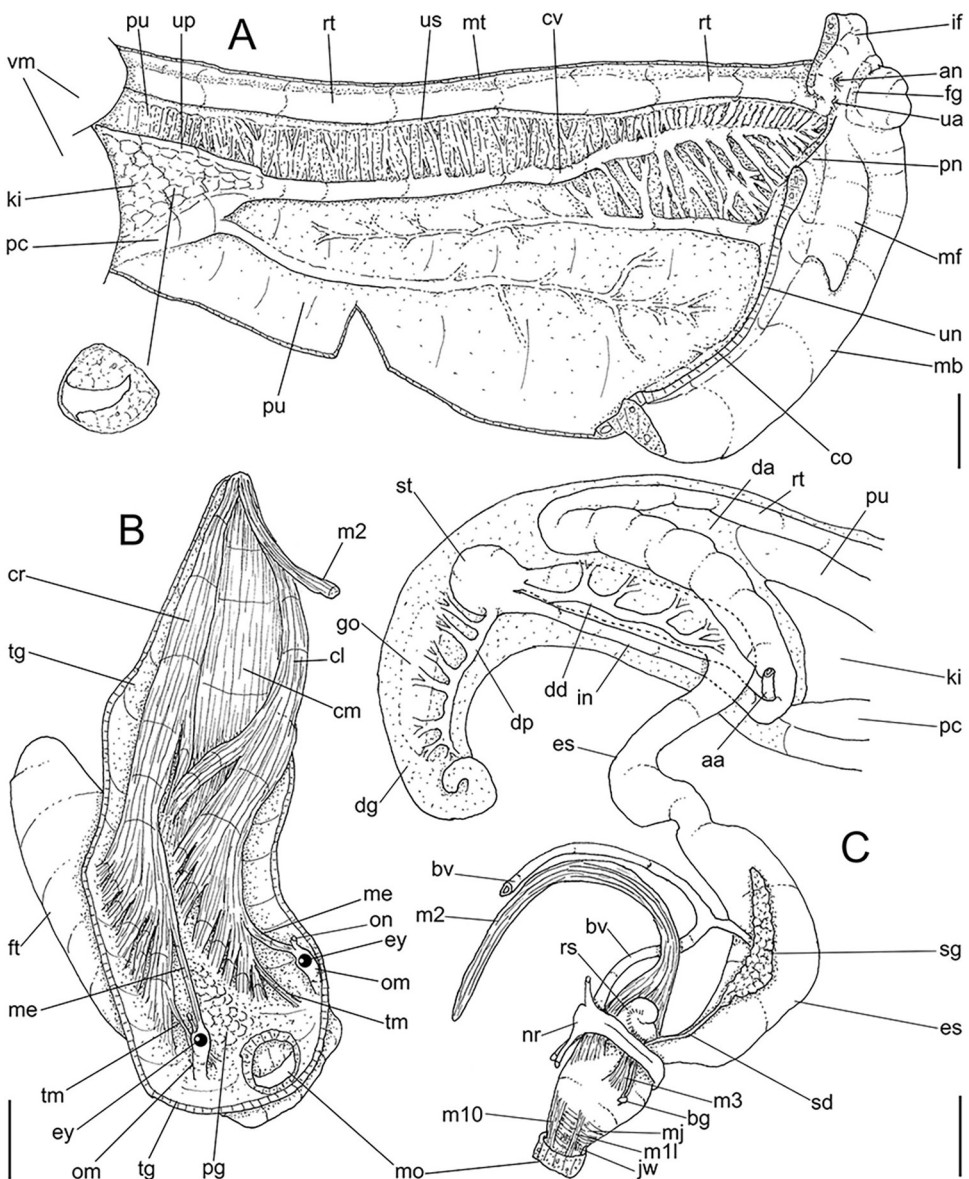

**Fig 18.** *Kora tupan* **anatomical drawings.** (A) extended pallial (pulmonary) cavity, ventral-inner view, inner edge of pneumostome sectioned and deflected upwards, transverse section of indicated region of kidney also shown. (B) head-foot, dorsal view, head, dorsal integument and internal organs removed, remaining muscles expanded. C) foregut and midgut, mostly ventral view as in situ, topology of some adjacent structures (gonad included) also shown, distal esophageal region shown if transparent. Scales = 5 mm.

rectangular. Odontophore cartilages ~75% fused. Pair m10 broad, m7 as 3 bundles. Carrefour bearing bulged portion, duct narrow and long, inserted between duct of albumen gland and albumen chamber. Albumen chamber sac-like. Penis ~85% of spermoviduct length, umbrella-like fold present, with 3 rods; epiphallus ~20% of penis length; penis muscle at epiphallus tip.

**Description (distinctive in anatomy).** *Shell.* Length up to 56 mm, outline fusiform-globose, ~1.9 longer than wide. Color pale (Fig 16K, 16M) to middle brown (Fig 16A–16D, 16H–16J) in spire, gradually becoming dark brown in last whorl (Fig 16B, 16F, 16I, 16M); subsutural pale band in all whorls well-developed (Fig 16A, 16C, 16I, 16M). Protoconch (Figs 16G, 16L,

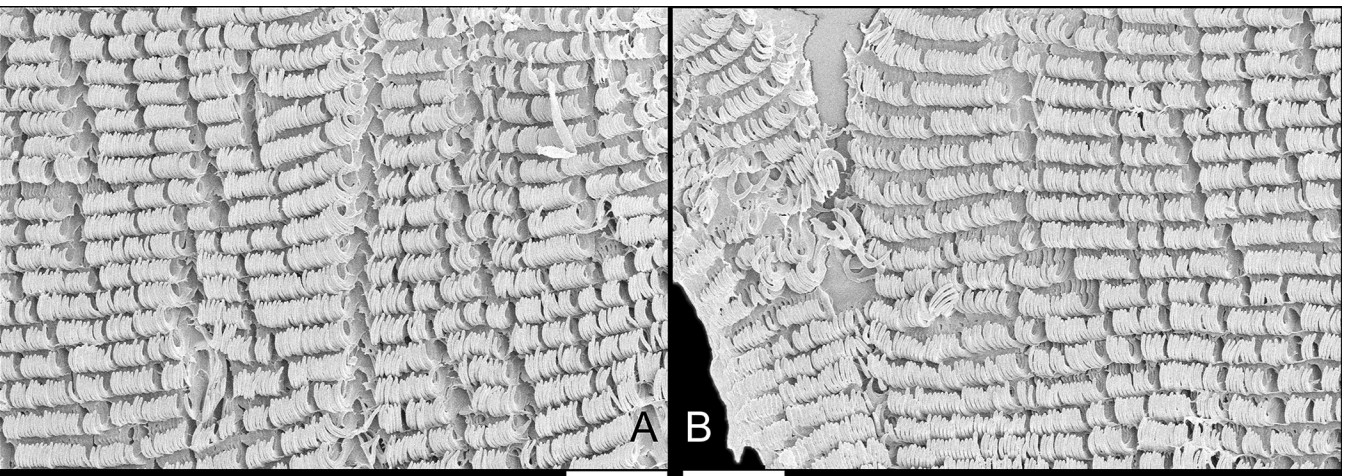

**Fig 19. *Kora tupan* radulae in SEM.** (A) detail of central region. (B) detail of lateral-marginal regions. Scales = 200 μm.

17B) with 2 whorls, bluntly pointed; length ~5% of shell length, and ~11% of shell width; mostly smooth, barely sculptured by axial riblets in last whorl. Limit between protoconch and teleoconch weakly visible, weakly prosocline. Teleoconch of ~4.5 whorls successively and uniformly increasing; whorls weakly concave; suture weakly deep; sculpture absent, except for growth lines and delicate axial, uniform undulations, ~60 in penultimate whorl. Dorso-ventrally very softly flattened (Fig 16E). Peristome weakly dislocated to right; deflected, except for region of callus. Callus weak (Fig 16A, 16D, 16H, 16J, 16K). Aperture wide, somewhat dislocated from spire longitudinal axis; length ~54% of shell length, ~70% of shell width. Outer lip inserted distantly from adjacent suture, simple, arched. Inner lip concave, superior half weakly convex, mostly showing outer surface of last whorl; inferior half weakly convex, concave only inferiorly; bearing oblique fold in limit with superior half, having weak elevation preceding its end in inner lip (Fig 16C, 16D, 16J); tooth length ~35% of peristome length. Umbilicus opened, narrow, partially covered by inferior half of inner lip (Fig 16F).

**Head-foot.** (Fig 18B) With similar features as *K. corallina*. Except for both secondary columellar muscles slightly narrower, and with fewer basal insertions, 6 in right (cr), 8 in left (cl); additionally pair of central-medial longer insertions slightly symmetrical sized.

**Mantle organs.** (Figs 17E, 17F and 18A) Most features similar to *K. corallina*, distinctions and remarks following. Mantle border (mb) with large secondary fold (mf) at left from pneumostome (pn) with pointed left end. Pulmonary venation strongly developed, especially in region preceding pneumostome, where vessels touch each other. Pulmonary vein (cv) entirely broad; its anterior end with ramifications; those at right from it basically straight, more developed anteriorly and in region adjacent to kidney; those at left from it longer and more complex, with anastomosis with next main longitudinal secondary vessel at left from it (Fig 18A). Region at left from pulmonary vein (cv) with single pair of long, longitudinal, intercalated vessels; collar vessel (co) also broad. Pallial wall in entire pulmonary cavity with uniform cover of minute longitudinal muscle fibers (Fig 17E: mi).

**Visceral mass.** (Fig 18C) With same characters as *K. corallina*. Except for digestive gland, with both lobes (anterior and posterior) connected with each other, and portion posterior to stomach slightly longer.

**Circulatory and excretory systems.** (Fig 18A) General characteristics as those described for *K. corallina*, except for pericardium (pc) with about half its width; and for arrangement of kidney lobe, as only 2 broad, tall folds.

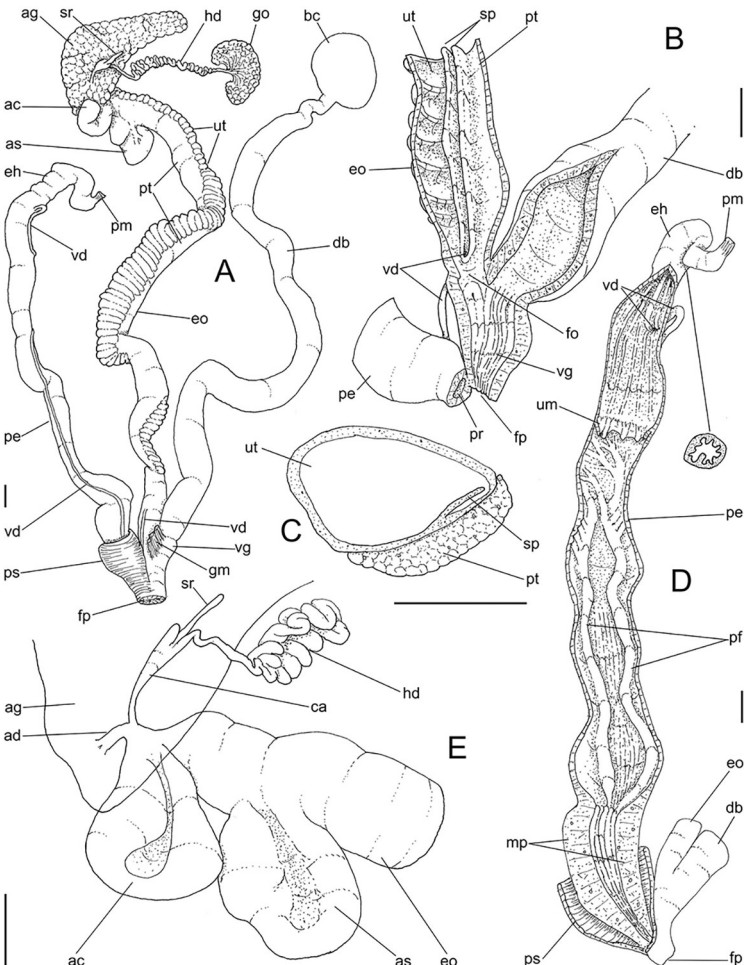

**Fig 20.** *Kora tupan* **anatomical drawings.** (A) genital structures, dorsal view, mostly uncoiled. (B) genital tubes portion preceding pore, dorsal view, 2 of them longitudinally opened. (C) spermoviduct, middle region, short portion in transverse view. (D) Penis longitudinally opened, transverse section of indicated region of epiphallus also shown, short portion of adjacent regions of genital tubes also shown. (E) genital structures in albumen gland level if it was transparent, ventral view. Scales = 2 mm.

**Digestive system.** (Fig 18C) overall morphology similar to that of *K. corallina*. Differences and remarks following. Peribuccal circular muscles (mj) longer and thicker. Jaw plate (Fig 17A, 17C) thick, rather rectangular, cutting edge only weakly concave; sculptured by successive, rather uniform, transverse, narrow folds. Buccal cavity (Fig 17C) with esophageal folds (ef) taller and subfolded, having salivary apertures (sa) immersed in posterior end of deep furrow. Odontophore intrinsic, extrinsic muscles, and other structures similar to *K. corallina*, except for **m1v**, absent; **m1l**, narrow pair of jugal lateral protractor muscles, originating in lateral region of mouth, running towards posterior, inserting in lateral region of buccal mass posterior end; **m3**, similar, but slightly wider; **m7**, originated in same region in cartilages, but divided into 3 distinct and equidistant bundles; **m4-m5**, of difficult separation, being continuation from each other; **m10**, broader and more visible. **Radula.** (Fig 19) with same characters as *K. corallina*, except in having more pairs per row (~350); and in having proportionally smaller and slightly more arched teeth. Esophagus slightly broader, mainly in its anterior region. Salivary glands narrow (Fig 18A: sg). Stomach (Fig 18C: st) bulbed, with anterior (dd) and posterior (dp) ducts to digestive gland more elongated; anterior duct with only right

branches. Middle portion of intestine, in its region after crossing aorta (aa) much wider, running obliquely, direct to right-posterior region of visceral mass (in).

**Reproductive system.** (Fig 20) General structures similar to preceding species, remarks and distinctions following. Gonad (go) proportionally longer, ~1.5 whorl, not clearly divided in lobes (Fig 17G), with well-developed digital acini. Hermaphroditic duct (Fig 20A, E: hd) gradually more intensely coiled towards anterior; except for short narrow region preceding its insertion (Fig 20E). Seminal receptacle small (Fig 20E: sr), straight, long, cylindric, ~10 times longer than wide, tip rounded; inserted between two bulged terminal regions, being one of them insertion of hermaphroditic duct (Fig 20E). Fertilization complex or carrefour (Fig 20E: ca) elongated, triangular, tapering gradually up to becoming narrow duct inserted in end of albumen gland duct. Albumen gland (Fig 20A: ag) ~twice larger than gonad. Albumen gland duct (Fig 20E: ad) subterminal, connected to tip of spermoviduct as narrow, separated duct. Albumen chamber (Fig 20A, 20C: ac) as flattened, rather triangular blind sac, ~as long as wide, connected to beginning of spermoviduct by narrow region, jointed to duct of albumen gland. Secondary albumen chamber (as) ~with same size and form of primary chamber, insertion at some distance from albumen chamber insertion. Spermoviduct (Fig 20A: eo) slightly narrower, slender, ~30 times longer than wide. Prostate wide (Fig 20C: pt), almost 1/2 of spermoviduct diameter; uterus lacking glandular walls, highly, transversally, and relatively uniformly folded (Fig 20A–20C: ut). Sperm inner longitudinal fold (Fig 20C: sp) simple, tall, narrow fold, second small fold gradually appearing in basal third; both folds fusing with each other, originating vas deferens, slightly anterior to end of uterine level (Fig 20B: vd). Genital muscle wide, attached to vaginal outer wall (Fig 20A: gm). Bursa copulatrix (bc) and its duct (bd) as long as spermoviduct length (Fig 20A); bursa duct with basal 3/5 with walls thick muscular (Fig 20B). Basal region of duct of bursa copulatrix with smooth inner surface (Fig 20B). Free oviduct and vagina with thick muscular walls (Fig 20B: fo, vg). Penis slightly mostly straight, weakly coiled, ~85% of spermoviduct length if straightened (Fig 20A: pe). Penis muscle inserted terminally in epiphallus tip (Fig 20A, 20D: pm), very short, simple. Penis walls weakly muscular, except for region adjacent to penis shield (Fig 20D: mp), region with very thick muscular walls. Epiphallus (eh) ~1/5 of penis' length, amply opened to penis; only vas deferens insertion marking its limit (Fig 20D: vd). Epiphallus inner surface only with ~8 narrow, small, parallel folds, being one of them larger, converging in vas deferens aperture (Fig 20D: eh). Internal penial arrangement of folds clearly with 3 regions (Fig 20D): (1) basal 1/5, highly muscular region (pm), lumen narrow, possessing only 4–5 low, uniform, longitudinal folds; (2) following basal 3/5, with strong pair of wide, rounded in section, longitudinal folds in a side, located close from each other, each one with simple margins in their basal 2/3, apical 1/3 possessing successive oblique branches in side turned to its counterpart, each main fold, after this, becoming oblique and successively branching; mosaic of low, longitudinal, regular folds flanking both strong folds; smooth space between both folds; **umbrella-like fold** (Fig 20D: um) located distally to this region, tall and narrow, possessing 3 rods exceeding fold's edge, septum-like, inserted transversally in inner penis wall; (3) distal region, possessing only 5–6 separated longitudinal, uniform folds; some of them converging to aperture of vas deferens (Fig 20D: vd), other continuous with epiphallus inner folds.

**Central nervous system.** (Fig 18C: nr) Same characters as *K. corallina*.

## Distribution

Known only from the for region of the type locality.

**Habitat.** Under rocks, limestone areas.

**Etymology.** The specific epithet is in apposition, and refers to the native Tupi-Guarani godhead Tupã, the Thunder-God, creator of the earth, heaven, and seas. This is an allusion to the species being the largest of the genus.

**Measurements.** (in mm) MZSP 161200 (holotype, Fig 11A–11F): 55.8 by 29.2; MZSP 151817 (paratype, Fig 11H–11J): 47.0 by 24.4; MZSP 151823 (paratype, Fig 11K–11M): 53.1 by 29.6.

**Material examined.** The types.

**Taxonomic remarks.** *Shell. Kora tupan* is the largest species, with some shells reaching 60 mm; only *K. ajar* has similar portions, but slightly shorter. Its shell is about 1.9 times longer than it is wide, being narrower than *K. nigra*, *K. ajar* and *K. curumim*, but more obese than *K. corallina* and *K. rupestris*. It is weak dorso-ventral flattened (Fig 16E), a condition only shared with *K. aetheria* and *K. kremerorum*. The shell aperture comprises about 54% of the total shell length, which is the amplest apertural proportion among its congeners, a condition only shared with *K. ajar*. Additionally, like *K. nigra*, *K. tupan*, *K. ajar*, *K. jimenezi*, *K. kremerorum*, and *K. vania*, *K. tupan* has a horizontally oriented superior implantation of the outer lip.

**Anatomy.** *Kora tupan* has tall folds in the mantle edge (Fig 18A), with pointed tip, a condition only shared with *K. ajar*. It differs from *K. nigra*, *K. aetheria*, and *K. jimenezi* by having a single intercalated pair of wide vessels to the left of the pulmonary vein (Figs 17E, 18A), as these species exhibit different vascular arrangements. The branched anterior end of the pulmonary vein (Fig 18A: cv) distinguishes *K. tupan* from *K. nigra*, *K. rupestris*, and *K. jimenezi*, which have simpler structure. Additionally, *K. tupan* displays strong venation to the right of the pulmonary vein up to half of pulmonary cavity length, a trait shared with *K. nigra*, *K. ajar*, *K. aetheria* and *K. uhlei*. The kidney lobe is a pair of solid masses in the dorsal and ventral sides, slightly similar only to *K. nigra* (Fig 18A: ki). In terms of muscle structure, *K. tupan* has 8 anterior insertions of the left accessory columellar muscle (Fig 18B: cl), an exclusive number. Additionally, it has 6 anterior insertions of the right accessory columellar muscle, also setting it apart from all remaining congeners. These accessory columellar muscles have strong posteriorly located medial branches, present in all remaining congeners except *K. corallina* and *K. jimenezi*. Furthermore, *K. tupan* has the odontophore muscle pair m1l, a feature also present in *K. rupestris*, *K. tupan*, *K. aetheria*, *K. jimenezi*, and *K. uhlei*. The absence of the pairs of odontophore muscles m1v (Fig 18C) approaches it from *K. ajar*, *K. jimenezi*, and *K. uhlei*. The connection of the odontophore muscle pair m3 to the esophagus origin distinguishes *K. tupan* from *K. rupestris*, where it connects to the m2 pair, and from *K. jimenezi*, which lacks this muscle. In having branches only on the right side of the posterior duct to the digestive gland (Fig 18C: dp), *K. tupan* differs from *K. nigra*, and its only right branching in the anterior duct to the digestive gland further differentiates it from *K. corallina*, *K. rupestris* and *K. aetheria*, which have bilateral branches. The rectangular outline of the jaw plate (Fig 17A) is shared with *K. nigra*, *K. ajar and K. uhlei*. The salivary gland aperture, located in the posterior third of the buccal dorsal wall (Fig 17C: sa), differs *K. tupan* from most congeners, being shares only with *K. jimenezi*. The degree of fusion of the odontophore cartilages, around 75%, is similar to most congeners but greater than that of *K. nigra* (~60˚) and less than that of *K. uhlei* (~90˚). In terms of odontophore m4-m5 pairs of muscles, the m4 displays as continuation of m5, a character only shared with *K. uhlei*. The odontophore muscle m7 divided into 3 bundles is exclusive of this species. Finally, the broad m10 muscle in *K. tupan* is only shared with *K. aetheria*, differing from the filiform version in *K. jimenezi*, and from narrow form of remaining species.

**Genital system.** *Kora tupan* has a small curve at the end of the hermaphrodite duct (Fig 20E: hd), which distinguishes it from *K. nigra*. The conical shape of its carrefour (ca) is distinct from those of *K. nigra* and *K. rupestris*, and its carrefour duct is narrow and long, similarly to most congener species, except *K. corallina*, *K. ajar* and *K. jimenezi*, which have other

arrangements. The species has the bulged portion on the opposite side of the hermaphrodite duct at the base of the seminal receptacle (Fig 20E: sr), approaching it from *K. jimenezi* and *K. uhlei*. *K. tupan* has the carrefour duct inserting between albumen gland duct and the albumen chamber duct (Fig 20E: ad), a character shared with *K. nigra*, *K. aetheria* and *K. uhlei*. Its albumen chamber (Fig 20E: ac) is sac-like, similarly to those of *K. nigra*, *K. rupestris* and *K. aetheria*. In having a single sperm fold in the spermoviduct (Fig 20C: sp), *K. tupan* differs from *K. nigra*, which has two. Additionally, it has a wide prostate band in the spermoviduct (~45%), being much wider than *K, corallina*, *K. nigra* and *K. aetheria*. The muscular anterior portion of the bursa copulatrix duct (Fig 20B) is well-defined, setting *K. tupan* apart from *K. jimenezi*. Its penis length is approximately 85% of the spermoviduct, longer than those of *K. corallina*, *K. rupestris*, *K. aetheria*, and *K. uhlei*, but shorter than those of *K. nigra*, *K. ajar*, and *K. jimenezi*. The bursa copulatrix duct is about as long as the spermoviduct, a condition only shared with *K. rupestris*. The vas deferens of *K. tupan* has the strong curve preceding its insertion at the tip of the penis, aporoaching it from *K. nigra*, *K. ajar*, and *K. uhlei*. Its penis base has clear muscular walls (Fig 20D: mp), being particularly thick, unlike *K. jimenezi*, which lacks them. The penis of *K. tupan* features the usual pair of inner folds, but it also has a terminal arrangement of outer, wing-like branches (Fig 20D: pf), an exclusive characteristic. *K. tupan* has the umbrella-like transverse penial fold, bearing 3 rods; it is found in most of its congeners, with the exceptions of *K. nigra* and *K. jimenezi*. The epiphallus comprises about 20% of the penial length, being shorter than those of *K. corallina* and *K. rupestris*; but longer than remaining species. *K. tupan* also has a strong longitudinal fold in the epiphallus (Fig 20D), a feature shared only with *K. corallina* and *K. ajar*. Its penis muscle (pm) inserts terminally in the epiphallus, unlike *K. rupestris*, *K. ajar*, and *K. uhlei*, which have more basal insertions.

***Kora ajar* new species Figs 21–23.** **ZooBank.** urn:lsid:zoobank.org:act: D72F5CEE-AF7B-4B6A-BE94-C8D48DD53DA2.

**Types.** Holotype MZSP 163700, 1 complete spm; paratypes: MZSP 151867, 7 spm, MZSP 151866, 39 shells, USNM, 2 shells, MNRJ, 2 shells, all from type locality. BRAZIL. **Minas Gerais**; Itacarambi, Serra de Itacarambi, 15˚01'42"S 44˚13'15"W, 740 m altitude, MZSP 151829, 7 spm (W. Vailant-Mattos col., i.2020).

**Type locality.** BRAZIL. **Minas Gerais**; Itacarambi, Vargem Grande, 466 m altitude, 15˚00'50"S 44˚04'34"W (W. Vailant-Mattos col., 7.ii.2020).

**Diagnosis.** Size about 55 mm, ~1.7 times longer than wide; lacking dorso-ventral compression. Apex with same color as remaining shell. Subsutural lighter band present. Delicate spiral striae present. Peristome with brown spots. Aperture occupying ~54% of length and ~71% width. Implantation of outer lip slightly horizontal. Inner lip with high middle fold. Umbilicus wide. Radula with narrow teeth, tip of cusps pointed, lacking expansion and subterminal furrow. Secondary columellar muscles with 5 insertions in left and 4 in right. Lacking m1v. Jaw rectangular. Odontophore cartilages ~75% fused. Pair m10 narrow, pair m2a present. Carrefour duct narrow and short, inserted in albumen chamber. Albumen chamber in curve. Penis ~90% of spermoviduct length, umbrella-like fold present, with 5 rods; epiphallus ~25% of penis length; penis muscle at epiphallus base.

**Description (distinctive in anatomy).** *Shell*. Length up to 50 mm, outline fusiform-globose, ~1.7x longer than wide. Color brown (Fig 21), slightly lighter in tip, subsutural pale band in all whorls well-developed (Fig 21B, 21C, 21I, 21J, 21L). Protoconch (Figs 21G, 22A) with 2 whorls, bluntly pointed; length ~4% of shell length, and ~9.5% of shell width; mostly smooth, barely sculptured by axial riblets in last whorl. Limit between protoconch and teleoconch weakly visible, weakly prosocline. Teleoconch of ~4.2 whorls successively and uniformly increasing; whorls weakly concave; suture weakly deep; sculpture absent, except for growth lines and delicate axial, uniform undulations, ~60 in penultimate whorl; weak spiral striae

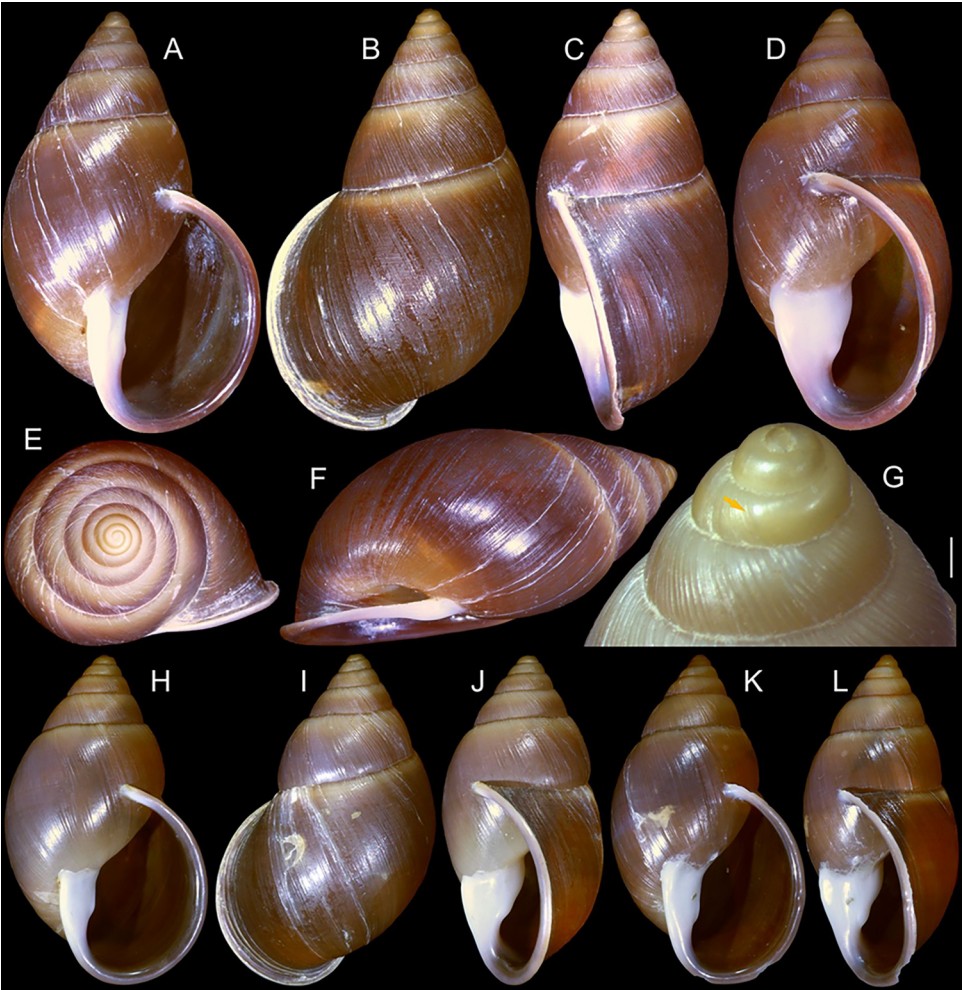

**Fig 21.** *Kora ajar* **shell characters.** (A) holotype MZSP 163700, frontal view (L 47.4 mm). (B) same, dorsal view. (C) same, right view. (D) same, right-slightly ventral view. (E) same, apical view. (F) same, left view showing umbilicus. (G) same, detail of apex, arrow showing transition protoconch-teleoconch, scale = 1 mm. (H–J) paratype MZSP 151866#1, frontal, dorsal and right views (L 43.0 mm). (K–L) paratype MZSP 151866#2, frontal and right views (L 49.1 mm).

gradually appearing in last whorls, more visible in last whorl (Fig 21B, 21C, 21F, 21I, 21L), distribution relatively uniform from suture up to inferior region of last whorl. Transverse section rounded (Fig 21E). Peristome weakly dislocated to right; deflected, except for region of callus. Callus weak (Fig 21A, 21D, 21H, 21J, 21K, 21L). Aperture wide, weakly dislocated from spire longitudinal axis; length ~54% of shell length, ~71% of shell width. Outer lip inserted distantly from adjacent suture, simple, arched. Inner lip concave, superior half weakly convex, mostly showing outer surface of last whorl; inferior half weakly convex, concave only inferiorly; bearing oblique, broad fold in limit with superior half, having weak elevation preceding its end in inner lip (Fig 21D, 21J, 21L); tooth length ~30% of peristome length. Umbilicus opened, relatively wide, partially covered by inferior half of inner lip (Fig 21F).

**Head-foot.** (Fig 23A) With similar features as *K. corallina*. Except for both secondary columellar muscles slightly narrower, and with fewer basal insertions, 4 in right (cr), 5 in left (cl); additionally pair of central-medial longer insertions very asymmetric, left one almost as small as remaining basal branches; right one very big, making right secondary columellar

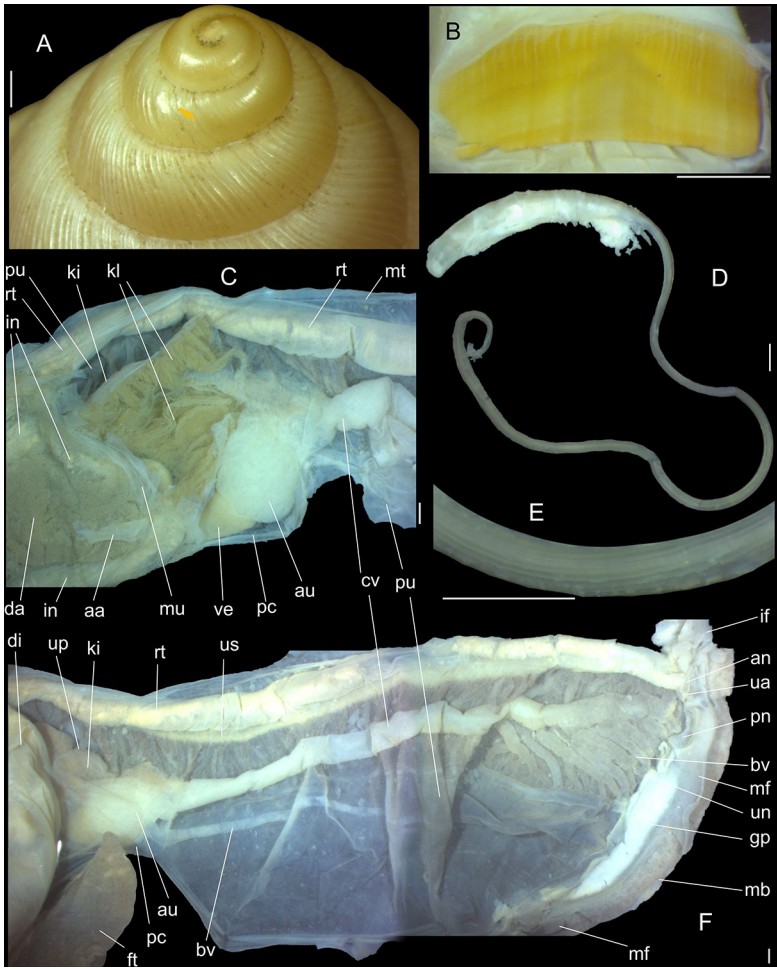

**Fig 22. _Kora ajar_ shell and anatomical characters, light photos.** (A) apex of MZSP 151866#1, profile-slightly apical view, arrow showing transition protoconch-teleoconch. (B) jaw in situ, ventral view, MZSP 151829#1. (C) reno-pericardial area, ventral view, kidney ventral wall opened along left edge and deflected upwards, ventral pericardial wall and head-foot removed, MZSP 151829#1. (D) spermatophore found inside duct of bursa copulatrix of MZSP 151829#2, stem digitally restored (it is broken in 3 levels). (E) same, detail of stem middle region. (F) extended pallial (pulmonary) cavity, ventral-inner view, inner edge of pneumostome sectioned and deflected upwards. Scales = 1 mm.

muscle (cr) almost bifid in its insertion. Also, insertion of secondary columellar muscles more medially positioned.

**Mantle organs.** (Fig 22C, 22F) Most features similar to _K. corallina_, distinctions and remarks following. Mantle border (mb) with large secondary fold (Fig 22F: superior mf) at left from pneumostome (pn) with tall pointed left end. Pallial edge gland in left-dorsal region of mantle border (Fig 22F: gp), white, claviform. Pulmonary venation strongly developed, especially in region preceding pneumostome up to almost half whorl posterior to it, with vessels touching each other. Pulmonary vein (cv) entirely broad; its anterior end bifid close to pneumostome (pn); those at right from it basically straight, more developed anteriorly and in region adjacent to kidney; those at left from it longer and more complex, obliquely crowding in region preceding pneumostome, with anastomosis with next main longitudinal secondary vessel at left from it (Fig 22F). Region at left from pulmonary vein (cv) with single pair of long, longitudinal, intercalated vessels (bv).

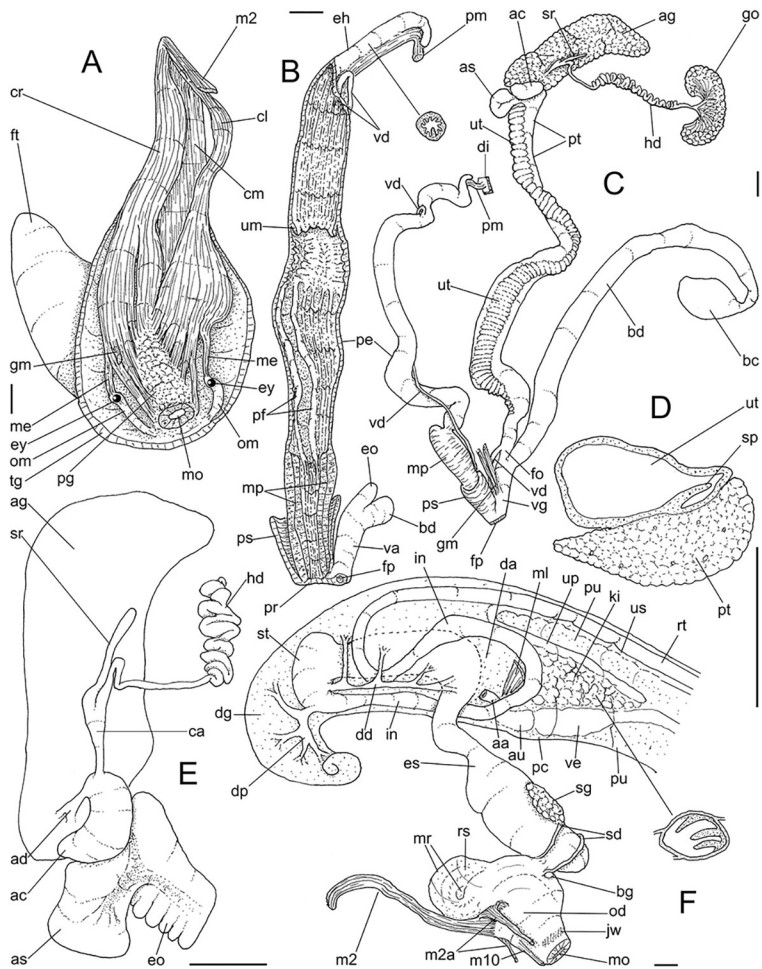

**Fig 23.** *Kora ajar* **anatomical drawings.** (A) head-foot, dorsal view, head, dorsal integument and internal organs removed, remaining muscles expanded. (B) penis, ventral view, longitudinally opened, transverse section of indicated region of epiphallus also shown. (C) genital structures, dorsal view, mostly uncoiled. (D) spermoviduct, transverse section of middle region. (E) genital structures in albumen gland level if it was transparent, ventral view. (F) Posterior end of pallial cavity and foregut-midgut, mostly ventral view as in situ, topology of some adjacent structures also shown, distal esophageal region shown if transparent, transverse section of indicated region of kidney also shown. Scales = 2 mm.

**Visceral mass.** (Fig 23F) With same characters as *K. corallina*. Except for stomach positioned more posteriorly, and by presence of pallial pre-rectal muscle (ml), originating in columellar region of shell in level of pericardium, running slightly flattened, fan-like, towards ventral, through digestive gland, very close to anterior aorta and adjacent portion of intestine, inserting splaying in pallial floor posterior end.

**Circulatory and excretory systems.** (Figs 22C, 22F and 23F) General characteristics as those described for *K. corallina*, except for pericardium (pc) with 2/3 its width; and for arrangement of kidney lobe, as 3 main narrow, tall folds (Fig 22C: kl and 23F: ki).

**Digestive system.** (Fig 23F) Overall morphology similar to that of *K. corallina*. Differences and remarks following. Peribuccal muscles and oral tube relatively narrow. Jaw plate (Figs 22B and 23F: jw) thick, rather rectangular, cutting edge straight; sculptured by successive, uniform, transverse, narrow folds. Odontophore intrinsic, extrinsic muscles, and other structures similar to *K. corallina*, except for **m1v**, **m1l,** both absent; **m2a**, narrow, working as pair of ventral

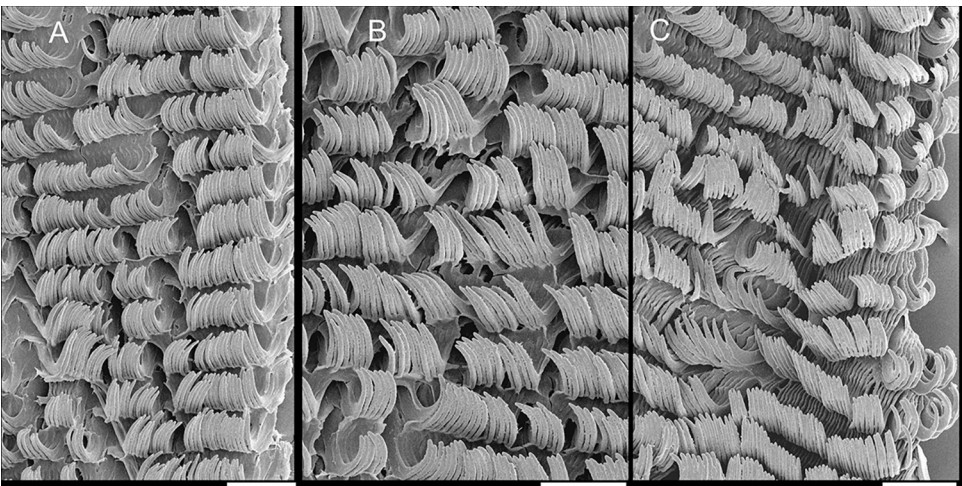

**Fig 24.** *Kora ajar* **radulae in SEM.** (A) detail of central region. (B) detail or lateral region. (C) detail of marginal region. Scales = 100 μm.

protractors of buccal mass, originating in lateral region of mouth, running towards anterior, inserting in edge of m2, in its both lateral insertions (replacing m1v); **m3**, similar, but as single wider, weak bundle covering dorsal end of membrane surrounding radular sac (mr); **m7**, with single, fan-like origin; **m10**, pair narrow. **Radula** (Fig 24) most attributes similar to those of *K. corallina*, except in having teeth much slenderer, from base to cusp; tip of cusp sharp pointed, lacking expansion and subterminal furrow; marginal teeth slightly more arched (Fig 24C). Esophagus (es) slightly broader, mainly in its anterior region. Salivary glands (sg) with shorter ducts (sd). Stomach (st) bulbed, with anterior (dd) and posterior (dp) ducts to digestive gland narrower and more elongated; anterior duct with only right branches, each one very long.

**Reproductive system.** (Fig 23B–23E) General structures similar to preceding species, remarks and distinctions following. Gonad (Fig 23C: go) proportionally longer, ~1.4 whorl, not clearly divided in lobes, with well-developed, aligned digital acini. Hermaphroditic duct (hd) coiled only in its middle region; narrow and straight in both ends (Fig 23C), always with tight curve preceding its insertion in carrefour (Fig 23E: bd). Seminal receptacle elongated (Fig 23C, 23E: sr), straight, weakly flattened, ~15 times longer than wide, tip rounded, base slightly broader by side insertion of hermaphroditic duct (Fig 23E). Fertilization complex or carrefour (Fig 23E: ca) elongated, narrowly triangular, tapering gradually up to narrow duct inserted in albumen gland chamber-spermoviduct limit. Albumen gland (Fig 23C: ag) ~twice larger than gonad. Albumen gland duct (Fig 23E: ad) subterminal, narrow, connected to tip of albumen chamber. Albumen chamber (Fig 23C, 23E: ac) flattened, curved blind sac, ~twice long than wide, connected to beginning of spermoviduct by narrow region. Secondary albumen chamber (as) ~1.5 time larger than primary chamber, slightly triangular, insertion at some distance from albumen chamber insertion. Spermoviduct (Fig 23C: eo) slightly narrower, slender, ~30 times longer than wide. Prostate wide (Fig 23C, 23D: pt), ~1/2 of spermoviduct diameter; uterus lacking glandular walls, highly, transversally, and relatively uniformly folded (Fig 20C–20E: ut). Sperm inner longitudinal fold (Fig 23D: sp). Anterior end of spermoviduct, entire free oviduct, vagina and ¾ of duct of bursa copulatrix with walls very thick muscular; their inner lumen tightly narrow, with thin longitudinal folds. Bursa copulatrix (bc) and its duct (bd) slightly shorter than spermoviduct length (Fig 23C); bursa duct with basal 3/4 with walls thick muscular. Penis slightly mostly straight, weakly coiled, ~90% of spermoviduct length if straightened (Fig 23C: pe). Penis muscle inserted subterminally in epiphallus, with fibers

inserted since its base, in region of vas deferens insertion, running attached to epiphallus side (Fig 23B, 23C: pm). Penis walls weakly muscular, except for basal ¼, region adjacent to penis shield (Fig 23C: mp), region with very thick muscular walls twice longer than penis shield. Epiphallus (eh) ~1/4 of penis' length, amply opened to penis; only vas deferens insertion marking its limit (Fig 23B: vd). Epiphallus inner surface only with ~8 narrow, small, parallel folds, being one of them larger, converging in vas deferens aperture. Internal penial arrangement of folds clearly with 3 regions (Fig 23B): (1) basal 1/4, highly muscular region (pm), lumen narrow, possessing only 4–5 low, uniform, longitudinal folds; (2) following basal 1/2, with strong pair of narrow, rounded in section, longitudinal folds in a side, located close from each other, each one with simple margins in their basal 2/3, apical 1/3 composed of successive small nodes, 6–7 in number, positioned turned to each other; mosaic of 4 low, longitudinal, regular folds flanking both strong folds; smooth space between both folds; distal third of area between both main folds, secondary folds abruptly disappearing, with only irregular surface produced by nodes of main folds; **umbrella-like fold** (Fig 23B: um) located distally to this region, between middle and distal peins thirds, low and narrow, possessing 5–6 small rods exceeding fold's edge, septum-like, inserted transversally in inner penis wall; (3) distal ¼ region, possessing only 7–8 separated, longitudinal, uniform folds; some of them converging to aperture of vas deferens (Fig 23B: vd), other continuous with epiphallus inner folds.

**Spermatophore.**    (Fig 22D, 22E) Found inside duct of bursa copulatrix. Virtually similar to that described for *K. nigra*, except for stem slightly shorter; and distal, bulged portion having blunt (instead in having sharp) tip.

**Central nervous system.**    Same characters as *K. corallina*.

**Distribution.**    Known only from the for region of the type locality.

**Habitat.**    Cerrado biome, altitude of 466 to 740 m.

**Etymology.**    The specific epithet is from the Latin word *ajar*, also applied in English, correspondent to the umbilicus aperture.

**Measurements.**    (in mm). holotype MZSP 163700 (Fig 13A–13G): 47.4 by 28.8; MZSP 151866#1 (Fig 13H–13J): 43.0 by 26.4; #2 (Fig 13K–13L): 49.1 by 27.9.

**Material examined.**    The types.

**Taxonomic remarks.**    *Shell. Kora ajar* is one of the largest species, being only supplanted by *K, tupan*, which is slightly larger. Its shell is about 1.7 times longer than it is wide, giving it a obese, only with *K. curumim* has a similar rank, from which it differs by in having a much ampler peristome and in being much larger. The shell aperture comprises about 54% of the total shell length, being the amplest peristome among the congeners, with *K. tupan* with te same rank. Additionally, like *K. nigra*, *K. tupan*, *K. jimenezi*, *K. kremerorum*, and *K. vania*, *K. ajar* lacks a horizontally oriented superior implantation of the outer lip.

**Anatomy.**    *Kora ajar* has wide folds along the mantle edge (Fig 22F: mf), with pointed ends. It differs from *K. nigra*, *K. aetheria*, and *K. jimenezi* by having a single intercalated pair of wide vessels to the left of the pulmonary vein (Fig 22F), as these species exhibit different vascular arrangements. The branched anterior end of the pulmonary vein (Fig 22F: cv) distinguishes *K. ajar* from *K. nigra*, *K. rupestris*, and *K. jimenezi*, which have simpler structure. Additionally, *K. ajar* displays strong venation to the right of the pulmonary vein along half of pulmonary length, a character shared with *K. nigra*, *K. tupan*, *K. aetheria* and *K. uhlei*. The kidney lobe has 3 tall anterior folds (Fig 23F), being only comparable with *K. rupestris*, which has 4. In terms of muscle structure, *K. ajar* has 5 anterior insertions of the left accessory columellar muscle (Fig 23A: cl), a character shared only with *K. aetheria*. Additionally, it has 4 anterior insertions of the right accessory columellar muscle, an exclusive number. These accessory columellar muscles also has a posteriorly located medial branch, present in all remaining congeners except *K. jimenezi*, but this pair is asymmetric, which is exclusive. Furthermore, *K. ajar*

lacks the odontophore muscle pair m1l, a feature present in *K. rupestris*, *K. tupan*, *K. aetheria*, *K. jimenezi*, and *K. uhlei*. The absence of the odontophore muscles m1v (Fig 23F) approaches it from *K. tupan*, *K. jimenezi*, and *K. uhlei*. The connection of the odontophore muscle pair m3 to the esophagus origin distinguishes *K. ajar* from *K. rupestris*, where it connects to the m2 pair, and from *K. jimenezi*, which lacks this muscle. In having branches only on the right side of the posterior duct to the digestive gland (Fig 23F: dp), *K. ajar* differs from *K. nigra*, and its only right branches in the anterior duct to the digestive gland further approaches it from all congeners, except *K. corallina*, *K. rupestris* and *K. aetheria*, which have bilateral branches. The rectangular shape of the jaw plate (Fig 22B) is shared only with *K. nigra*, *K. tupan* and *K. uhlei*. The salivary gland aperture, located in the middle third of the buccal dorsal wall, differs from *K. tupan*, and the absence of a salivary papilla sets it apart from *K. jimenezi* and *K. uhlei*. The degree of fusion of the odontophore cartilages, around 75%, is similar to most congeners but greater than that of *K. nigra* (~60°) and less than that of *K. uhlei* (~90°). In terms of odontophore m4-m5 pairs of muscles, the m4 muscle covering m5 distinguishes *K. ajar* from *K. tupan* and *K. uhlei*, in which these muscle pairs are continuous with each other. The narrow odontophore muscle m7, with a single origin, differs from the conformations in *K. tupan*, *K. jimenezi* and *K. uhlei*. Finally, the narrow m10 muscle in *K. ajar* differs from the broader form in *K. tupan* and *K. aetheria*, and from the filiform version in *K. jimenezi*.

**Genital system.** *Kora ajar* has a small curve at the end of the hermaphrodite duct (Fig 23E: hd), which distinguishes it from *K. nigra*. The conical shape of its carrefour (ca) is distinct from those of *K. nigra* and *K. rupestris*, and its carrefour duct is narrow and short, being similar only to *K. jimenezi*. The species lacks the bulged portion on the opposite side of the hermaphrodite duct at the base of the seminal receptacle (sr), differentiating it from *K. tupan*, *K. jimenezi*, and *K. uhlei*. *K. ajar* is unique in having the carrefour duct inserting directly into the albumen chamber (Fig 23E: ac). Its albumen chamber (Fig 23E: ac) forms a simple curve, contrasting with the blind sacs found in *K. nigra*, *K. rupestris*, *K. tupan*, and *K. aetheria*. In having a single sperm fold in the spermoviduct (Fig 23D: sp), *K. ajar* differs from *K. nigra*, which has two. Additionally, it has a wide prostate band in the spermoviduct (~45%) among its congeners, being narrower than *K. corallina*, *K. nigra* and *K. aetheria*. The muscular anterior portion of the bursa copulatrix duct is well-defined, setting *K. ajar* apart from *K. jimenezi*. Its penis length is approximately 90% of the spermoviduct, only *K. nigra* and *K. jimenezi* have proportionally longer penis. The bursa copulatrix duct is about 90% of the spermoviduct length, which is shorter than in *K. rupestris* and *K. tupan*, but longer than in *K. aetheria*, *K. jimenezi*, and *K. uhlei*. The vas deferens of *K. ajar* has the strong curve preceding its insertion at the tip of the penis, approaching it from *K. nigra*, *K. tupan* and *K. uhlei*. Its penis base has clear muscular walls (Fig 23B: mp), unlike *K. jimenezi*, which lacks them. The penis of *K. ajar* features the usual pair of inner folds, with simple shape, lacking branches (Fig 23B: pf), distinguishing it from *K. corallina*, *K. aetheria* and *K. uhlei* among its congeners, which have branched folds. *K. ajar* has the umbrella-like transverse penial fold, with 5 rods, found in most of its congeners, with the exceptions of *K. corallina*, *K. nigra* and *K. jimenezi*. The epiphallus comprises about 25% of the penial length, as most of its congeners, with *K. corallina*, *K. rupestris* and *K. tupan* with shorter proportions, and *K. aetheria* with longer epiphallus. *K. ajar* also has a strong longitudinal fold in the epiphallus (Fig 23B), a feature shared only with *K. tupan* and *K. corallina*. Its penis muscle (pm) inserts in the base of the epiphallus, a character shared with *K. rupestris*, and *K. uhlei*.

***Kora aetheria* new species Figs 25–27.**

**ZooBank.**   urn:lsid:zoobank.org:act:D19B73B4-F62D-4753-8595-54E92445E5E2.

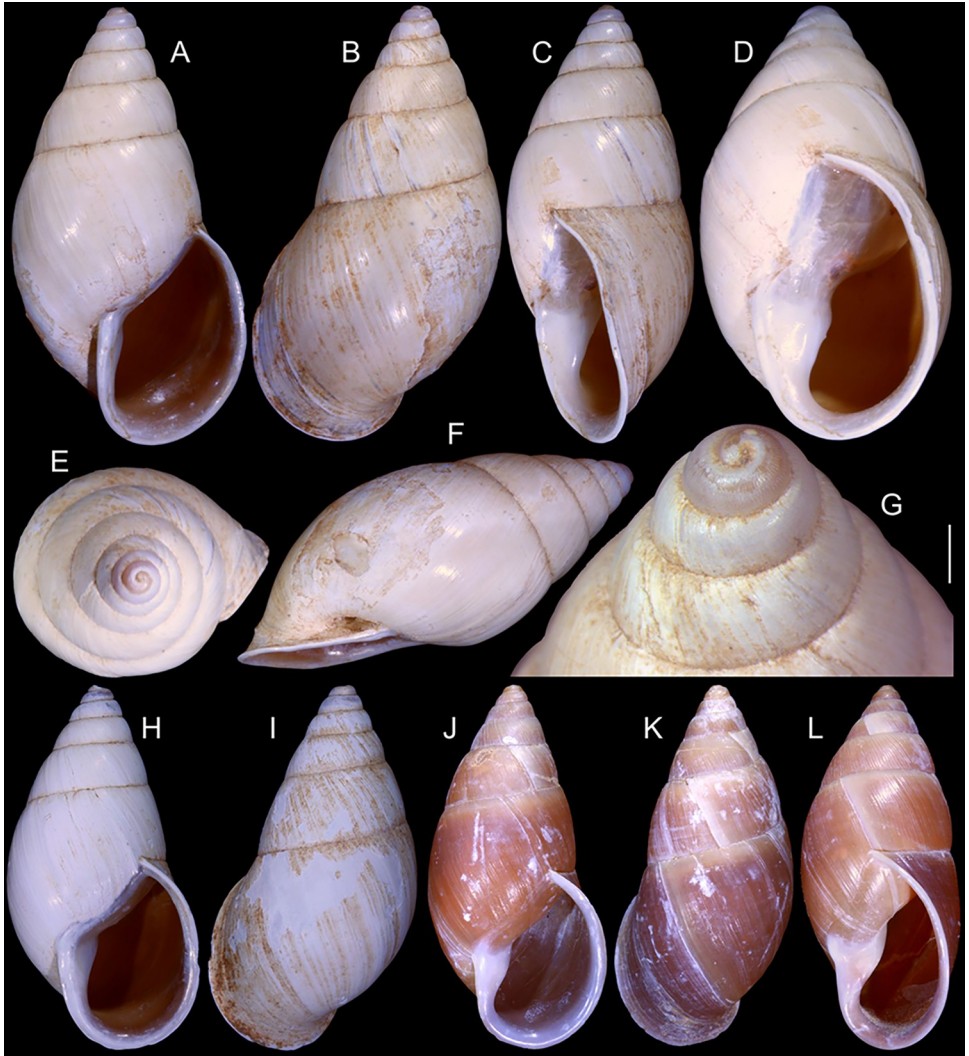

**Fig 25. *Kora aetheria* shell characters.** (A) holotype MZSP 163400, frontal view (L 29.4 mm). (B) same, dorsal view. (C), same, right view. (D) same, right-slightly antero-ventral view. (E) same, apical view. (F) same, left view showing umbilicus. (G) apex, profile-slightly apical view, scale = 1 mm. (H–I) paratype MZSP 153856, frontal and dorsal views (L 30.3 mm). (J–L) paratype 152249, frontal, dorsal and right views (L 32.6 mm).

**Types.** Holotype MZSP 163400, 1 complete spm; paratypes: MZSP 153856, 4 spm, from type locality. BRAZIL. **Bahia**; Serra do Ramalho, Pedreira, 13˚26'40"S 43˚49'05"W, MZSP 152249, 23 shells, USNM, 1 shell, MNRJ, 1 shell (W. Vailant-Mattos col., v.2019).

**Type locality.** BRAZIL. **Bahia**; Serra do Ramalho, Toca, 13˚38'13"S 43˚50'10"W (W. Vailant-Mattos col., v.2019).

**Diagnosis.** Size about 30 mm, ~2 times longer than wide; dorso-ventrally weakly compressed. Apex with same color as remaining shell. Subsutural lighter band absent. Peristome white. Delicate spiral striae absent. Aperture occupying ~45% of length and ~60% width. Implantation of outer lip slightly vertical. Inner lip lacking high middle fold. Umbilicus narrow. Secondary columellar muscles with 5 insertions in left and 7 in right. One pair of m1v. Jaw with central notch. Odontophore cartilages ~75% fused. Pair m10 broad. Carrefour duct narrow and long, inserted between duct of albumen gland and spermoviduct. Albumen

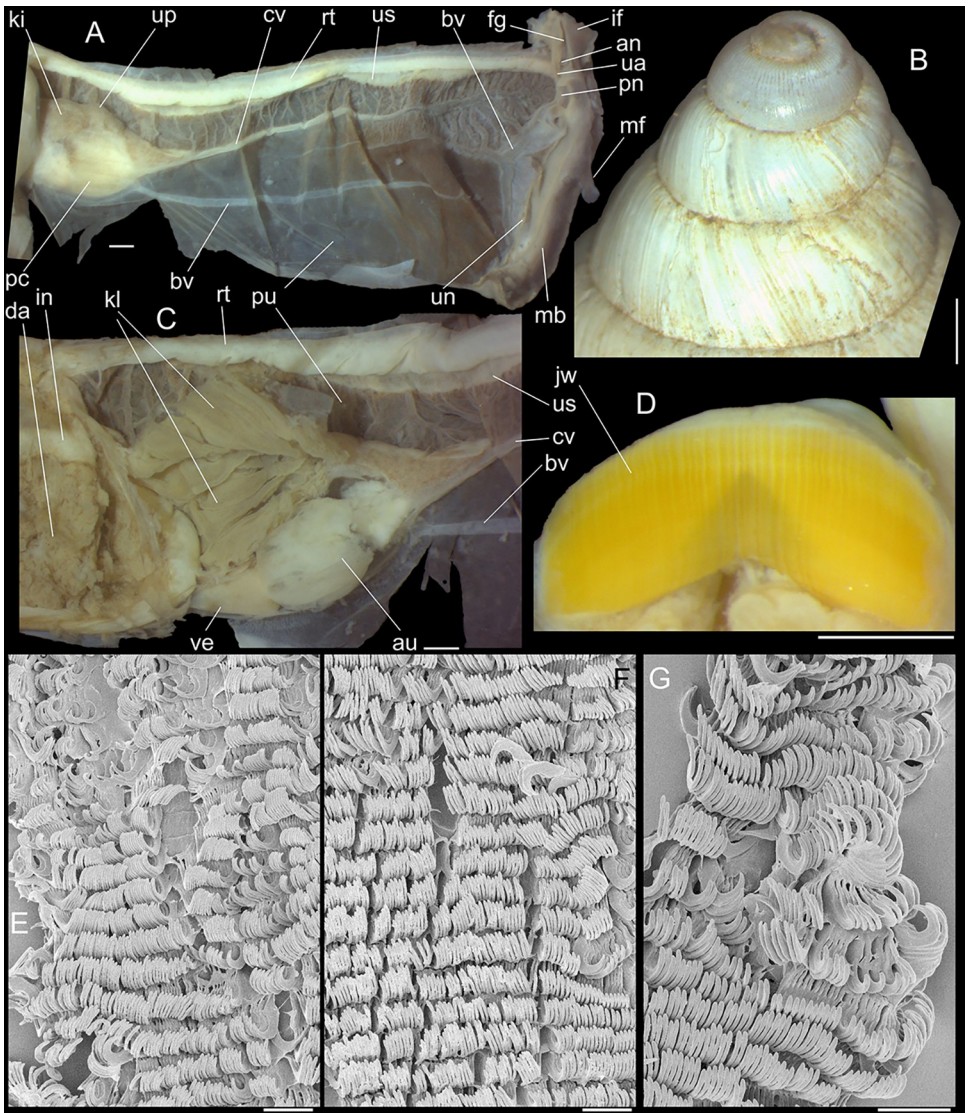

**Fig 26.** *Kora aetheria* **shell and anatomical characters, light photos, SEM of radula.** (A) extended pallial (pulmonary) cavity, ventral-inner view, inner edge of pneumostome sectioned and deflected upwards, MZSP 153856#1. (B) shell apex, profile-slightly apical view, MZSP 153856#2. (C) reno-pericardial area, ventral view, kidney ventral wall opened along left edge and deflected upwards, ventral pericardial wall removed, MZSP 153856#1. (D) jaw in situ, ventral view, MZSP 153856#1. Scales = 1 mm. (E–G) Radulae in SEM, scales = 100 μm. (E) detail of central region. (F) detail of lateral region. (G) detail of marginal region.

chamber sac-like. Penis ~65% of spermoviduct length, umbrella-like fold present, with 6 rods; epiphallus ~33% of penis length; penis muscle at epiphallus tip.

**Description (distinctive in anatomy).** *Shell.* (Figs 25, 26B) length up to 33 mm, outline fusiform-elongated, ~1.9–2.0x longer than wide. Color uniform beige (Fig 25A–25I) to light brown (Fig 25J–25L) with paler apex, gradually becoming dark brown in last whorl (Fig 25K); subsutural pale band in all whorls well-developed in darker specimens (Fig 25J–25L). Protoconch (Figs 25G and 26B) with 2.2 whorls, bluntly pointed; length ~4% of shell length, and ~14% of shell width (Fig 25E); almost entirely sculptured by uniform, axial riblets. Limit between protoconch and teleoconch weakly visible, weakly prosocline. Teleoconch of ~4.5 whorls successively and uniformly increasing; whorls weakly concave; suture weakly deep;

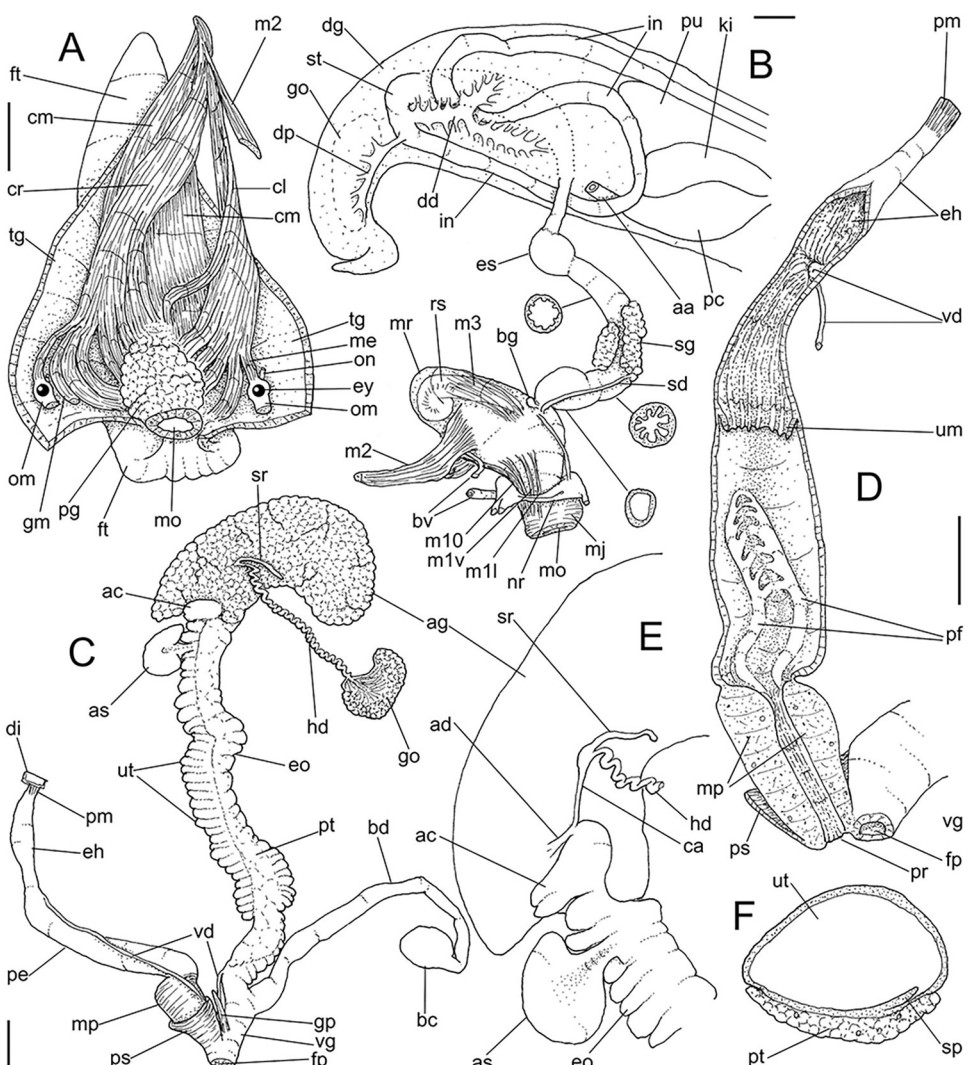

**Fig 27. *Kora aetheria* anatomical drawings.** (A) head-foot, dorsal view, head, dorsal integument and internal organs removed, remaining muscles expanded. (B) foregut and midgut, mostly ventral view as in situ, topology of some adjacent structures also shown (gonad included), 3 transverse sections of indicated regions of esophagus also shown, distal portion of esophagus represented if it was transparent. (C) genital structures, dorsal view, mostly uncoiled. (D) penis, ventral view, longitudinally opened. (E) genital structures in albumen gland level if it was transparent, ventral view. (F) spermoviduct, transverse section of middle region. Scales = 2 mm.

sculpture absent, except for growth lines and delicate axial, uniform undulations, ~50 in penultimate whorl. Dorso-ventrally very softly flattened (Fig 25E). Peristome very weakly dislocated to right; deflected, except for region of callus. Callus weak (Fig 25J, 25L) to relatively thick (Fig 25A, 25C, 25D, 25H). Aperture wide; length ~44–47% of shell length, ~60% of shell width. Outer lip inserted distantly from adjacent suture, simple, arched. Inner lip concave, superior half weakly convex, mostly showing callus or outer surface of last whorl; inferior half straight or weakly convex, concave only inferiorly; bearing low, oblique, uniform fold in limit with superior half (Fig 25C, 25D, 25L); tooth length ~35% of peristome length. Umbilicus opened, narrow, partially covered by inferior half of inner lip (Fig 25F).

**Head-foot.** (Fig 27A) with similar features as *K. corallina*. Except for details of both secondary columellar muscles, with fewer basal insertions, 7 in right (cr), 5 in left (cl);

additionally, pair of central-medial longer insertions slightly symmetrical sized and similar to remaining basal insertions. Pedal gland (pg) wider and shorter.

**Mantle organs.** (Fig 26A, 26C) Most features similar to *K. corallina*, distinctions and remarks following. Mantle border (mb) with large secondary fold (mf) at left from pneumostome (pn) with rounded, projected left end. Pulmonary venation strongly developed, especially in region preceding pneumostome, vessels not touching each other, showing secondary transverse venation in interspaces. Pulmonary vein (cv) relatively narrow; its anterior end with ramifications, bifid at end; those at right from it basically straight, more developed anteriorly; those at left from it longer and more complex, with anastomosis with next main longitudinal secondary vessel at left from it (Fig 26A: bv). Region at left from pulmonary vein (cv) with single pair of long, longitudinal, intercalated vessels; right vessel Y-shaped, with strong bifurcation anterior, covering collar vessel (Fig 26A: bv).

**Visceral mass.** (Fig 27B) With similar characters as *K. corallina*.

**Circulatory and excretory systems.** (Fig 26C) General characteristics as those described for *K. corallina*. Arrangement of kidney lobe (kl), as 4 successive larger, tall folds, with ventral lobe well-developed.

**Digestive system.** (Fig 27B) Overall morphology similar to that of *K. corallina*. Differences and remarks following. Peribuccal circular muscles (mj) longer and thicker. Jaw plate (Fig 26D) thin, curved, cutting edge only concave, notched at middle; sculptured by successive, rather uniform, transverse, narrow folds. Buccal cavity also similar, except for pair of salivary ducts having subterminal bulging. Odontophore intrinsic, extrinsic muscles, and other structures similar to *K. corallina*, **m1l**, present; **m2**, slightly narrower, inserted in buccal mass in 3 separated, isometric bundles; **m3**, similar, but slightly wider; **m10**, broader and more visible. **Radula** (Fig 26E–26G) with similar characteristics as *K. corallina*, except for teeth slightly more arched, mainly marginal teeth (Fig 26G). Esophagus (es) narrow, walls thicker, with internal 7–8 longitudinal, separated folds. Salivary glands narrow (sg). Stomach (st) narrow, as simple curve, with anterior (dd) and posterior (dp) ducts to digestive gland slender; anterior duct strongly bifurcated.

**Reproductive system.** (Fig 27C–27F) General structures similar to preceding species, remarks and distinctions following. Hermaphroditic duct (Fig 27C, 27E: hd) entirely, uniformly and delicately coiled. Seminal receptacle small (sr), very elongated, weakly flattened, ~15 times longer than wide, tip rounded; inserted by side of insertion of hermaphroditic duct (Fig 27C). Fertilization complex or carrefour (Fig 27E: ca) elongated, initially (1/4) bulged, abruptly tapering, becoming narrow, long duct, equivalent to ¾ its length; inserting at tip of spermoviduct. Albumen gland (Fig 27C: ag) ~4x larger than gonad. Albumen gland duct (Fig 27E: ad) subterminal, very narrow, connected to tip of spermoviduct in same region of carrefour duct. Albumen chamber (Fig 27C, 27E: ac) bulged blind sac, ~twice longer than wide, widely connected to beginning of spermoviduct. Secondary albumen chamber (as) ~double of primary chamber, balloon-like, duct narrow, insertion at some distance from albumen chamber insertion. Spermoviduct (Fig 27C: eo) slightly shorter, ~10 times longer than wide. Prostate wide but low (Fig 27F: pt), ~30% of spermoviduct diameter; uterus lacking glandular walls, highly, transversally, and relatively uniformly folded (Fig 27C, 27F: ut). Sperm inner longitudinal fold (Fig 27F: sp) simple, tall, narrow fold. Anterior end of spermoviduct, entire free oviduct, vagina and ¾ of duct of bursa copulatrix with walls very thick muscular; their inner lumen tightly narrow, with thin longitudinal folds. Genital muscle very narrow, attached to vaginal outer wall (Fig 26C: gm). Bursa copulatrix (bc) and its duct (bd) ~80% of spermoviduct length (Fig 27C); bursa duct with basal 1/3 with walls thick muscular. Basal region of duct of bursa copulatrix with 4–5 longitudinal, tall folds. Free oviduct and vagina with thick muscular walls; inner lumen of both narrow, bearing only 4–5 longitudinal, tall folds. Penis slightly

straight, ~65% of spermoviduct length (Fig 27C: pe). Penis muscle inserted terminally in epiphallus tip (Fig 27C, 27D: pm), very short, simple. Penis walls weakly muscular, except for region adjacent to penis shield (Fig 27D: mp), region with very thick muscular walls equivalent to ~1/3 of its length. Epiphallus (eh) ~1/3 of penis' length, amply opened to penis; only vas deferens insertion marking its limit (Fig 27D: vd). Epiphallus inner surface only with ~8–10 narrow, low, parallel folds, being one of them slightly larger (Fig 27D: eh). Internal penial arrangement of folds clearly with 3 regions (Fig 27D): (1) basal ~1/3, highly muscular region (pm), lumen very narrow, possessing only 4–5 low, uniform, longitudinal folds; (2) following basal ~1/3, with strong pair of wide, rounded in section, longitudinal-slightly curved folds in a side, located close from each other, each one with simple margins in their basal half; apical half possessing 6–7 successive, uniform, oblique branches in side turned to its counterpart, each secondary branch almost touching its pair of other fold; smooth space between both folds in both sides, including wide distance between them and umbrella-like fold; **umbrella-like fold** (Fig 27D: um) located distally to this region, low and narrow, possessing 6 rods exceeding fold's edge, septum-like, inserted transversally in inner penis wall; (3) distal region, possessing only 56 separated longitudinal, uniform folds; some of them converging to aperture of vas deferens (Fig 27D: vd), other continuous with epiphallus inner folds.

**Central nervous system.**   (Fig 27B: nr) Same characters as *K. corallina*.

**Distribution.**   Known only from the for region of Serra do Ramalho, Bahia.

**Habitat.**   Under rocks, limestone areas.

**Etymology.**   The specific epithet is a Latin word in feminine genitive meaning ethereal, heavenly, divine, celestial, an allusion to the simplicity of the shell shape.

**Measurements.**   (in mm) MZSP 163400 (holotype, Fig 25A–25G): 29.4 by 15.3; MZSP 153856 (Fig 25H–25I): 30.3 by 15.1; MZSP 152249 (Fig 25J–25L): 32.6 by 16.6.

**Material examined.**   The types.

**Taxonomic remarks.**   *Shell. Kora aetheria* has an average shell size of approximately 30 mm, making it in the smaller rank among its congeners, a rank shared with *K. nigra*, *K, vania* and *K. curumim*. Its shell is about 2.0 times longer than it is wide, giving it ae elongated shape compared to most other congeneric species, it is, however, more obese than *K. corallina* and *K. rupestris*; and more elongated than *K. nigra*, *K. ajar* and *K. curumim*. It has a relatively shining surface, with a pale, light color, associated in being dorso-ventral flattened (Fig 25E); the first is an exclusive feature, while the last is shared with *K. tupan* and *K. kremerorum*. The shell aperture comprises about 45% of the total shell length, which is much shorter than the apertures of *K. nigra*, *K. rupestris*, *K. tupan*, *K. ajar*, *K. kremerorum*, and *K. curumim*. Additionally, unlike *K. nigra*, *K. tupan*, *K. ajar*, *K. jimenezi*, *K. kremerorum*, and *K. vania*, *K. aetheria* lacks a horizontally oriented superior implantation of the outer lip.

**Anatomy.**   *Kora aetheria* is the only species with tall folds along the mantle edge and rounded, projected ends (Fig 26A: mf). It differs from *K. nigra*, *K. aetheria*, and *K. jimenezi* by having a single intercalated pair of wide vessels to the left of the pulmonary vein (Fig 3A), as these species exhibit different vascular arrangements; but, additionally, the right vessel has an Y-shaped anterior end (Fig 26A: bv). The branched anterior end of the pulmonary vein (Fig 26A: cv) distinguishes *K. aetheria* from *K. nigra*, *K. rupestris*, and *K. jimenezi*, which have simpler structure. Additionally, *K. aetheria* displays strong venation up to the middle level of pulmonary cavity, as most species, except *K. corallina*, *K, rupestris* and *K. jimenezi*. The kidney lobe has several tall folds along anterior kidney walls (Fig 26C: kl), being similar to *K. corallina*, *K. jimenezi* and *K. uhlei*. In terms of muscle structure, *K. aetheria* has 5 anterior insertions of the left accessory columellar muscle (Fig 27A: cl), the same number as *K. ajar*. Additionally, it has 7 anterior insertions of the right accessory columellar muscle, setting it apart from *K. tupan*, *K. ajar*, *K. jimenezi*, and *K. uhlei*. These accessory columellar muscles have a posteriorly

located medial branch, which is present in all remaining congeners except *K. jimenezi*. Furthermore, *K. aetheria* has the odontophore muscle pair m1l, a feature also present in *K. rupestris*, *K. tupan*, *K. jimenezi*, and *K. uhlei*. The presence of a pair of odontophore muscles m1v (Fig 27B) differentiates it from *K. tupan*, *K. ajar*, *K. jimenezi*, and *K. uhlei*, which lack these muscles, and from *K. corallina*, *K. nigra and K. rupestris* that have 2 pairs. The connection of the odontophore muscle pair m3 to the esophagus origin distinguishes *K. corallina* from *K. rupestris*, where it connects to the m2 pair, and from *K. jimenezi*, which lacks this muscle. In having branches only on the right side of the posterior duct to the digestive gland (Fig 27B: dp), *K. aetheria* differs from *K. nigra*, and its bilateral branching in the anterior duct to the digestive gland further differentiates it from all congeners, except *K. rupestris* and *K. aetheria*, which also have bilateral branches, but further differs in having a Y-shape (Fig 27B: dd). The central notch in the jaw plate (Fig 26D) is shared only with *K. rupestris* and *K. corallina*, distinguishing *K. aetheria* from other congeners. The salivary gland aperture, located in the middle third of the buccal dorsal wall, differs from *K. tupan*, and the absence of a salivary papilla sets it apart from *K. jimenezi* and *K. uhlei*. The degree of fusion of the odontophore cartilages, around 75%, is similar to most congeners but greater than that of *K. nigra* (~60˚) and less than that of *K. uhlei* (~90˚). In terms of odontophore m4-m5 pairs of muscles, the m4 muscle covering m5 distinguishes *K. aetheria* from *K. tupan* and *K. uhlei*, in which these muscle pairs are continuous with each other. The narrow odontophore muscle m7, with a single origin, differs from the conformations in *K. tupan*, *K. jimenezi*, and *K. uhlei*. Finally, the broad m10 muscle in *K. aetheria* similar to those of *K. tupan* and *K. aetheria*.

**Genital system.** *Kora aetheria* has a small curve at the end of the hermaphrodite duct (Fig 27E: hd), which distinguishes it from *K. nigra*. The conical shape of its carrefour (ca) is distinct from those of *K. nigra* and *K. rupestris*, and its carrefour duct is narrow and long, differing from *K. corallina*, *K. nigra*, *K. ajar* and *K. jimenezi*. The species lacks the bulged portion on the opposite side of the hermaphrodite duct at the base of the seminal receptacle (sr), differentiating it from *K. tupan*, *K. jimenezi*, and *K. uhlei*. *K. aetheria* is unique in having the carrefour duct inserting between the albumen gland duct (ad) and the beginning of the spermoviduct (eo) (Fig 27E). Its albumen chamber (Fig 27E: ac) is sac-like, similarly to *K. nigra*, *K. rupestris* and *K. tupan*; but it is unique in having a bifid tip. In having a single sperm fold in the spermoviduct (Fig 27F: sp), *K. aetheria* differs from *K. nigra*, which has two. Additionally, it has the narrowest prostate band in the spermoviduct (~35%) among its congeners, except for *K. nigra* and *K. corallina*, which share similar proportions. The muscular anterior portion of the bursa copulatrix duct is well-defined, setting *K. aetheria* apart from *K. jimenezi*. Its penis length is approximately 65% of the spermoviduct, longer than those of *K. rupestris* and *K. uhlei*, but shorter than those of *K. corallina*, *K. nigra*, *K. tupan*, *K. ajar*, and *K. jimenezi*. The bursa copulatrix duct is about 80% of the spermoviduct length, which is the shortest amongst its congeners, a condition shared with *K. jimenezi* and *K. uhlei*. The vas deferens of *K. aetheria* lacks the strong curve preceding its insertion at the tip of the penis, distinguishing it from *K. nigra*, *K. tupan*, *K. ajar*, and *K. uhlei*. Its penis base has clear muscular walls (Fig 27D: mp), unlike *K. jimenezi*, which lacks them; but it is particularly thick. The penis of *K. aetheria* features the usual pair of inner folds, but it has an imbricated arrangement of inner branches in its distal half (Fig 27D: pf), a characteristic shared only with *K. corallina* and *K. uhlei* among its congeners. *K. aetheria* has the umbrella-like transverse penial fold, with 6 hods, a feature found in most of its congeners, with the exceptions of *K. corallina*, *K. nigra* and *K. jimenezi*. The epiphallus comprises about 33% of the penial length, the longest ratio among its congeners. *K. aetheria* lacks strong longitudinal fold in the epiphallus (Fig 27D), distinguishing it from *K. corallina*, *K. tupan* and *K. ajar*. Its penis muscle (pm) inserts terminally in the epiphallus, unlike *K. rupestris*, *K. ajar*, and *K. uhlei*, which have more basal insertions.

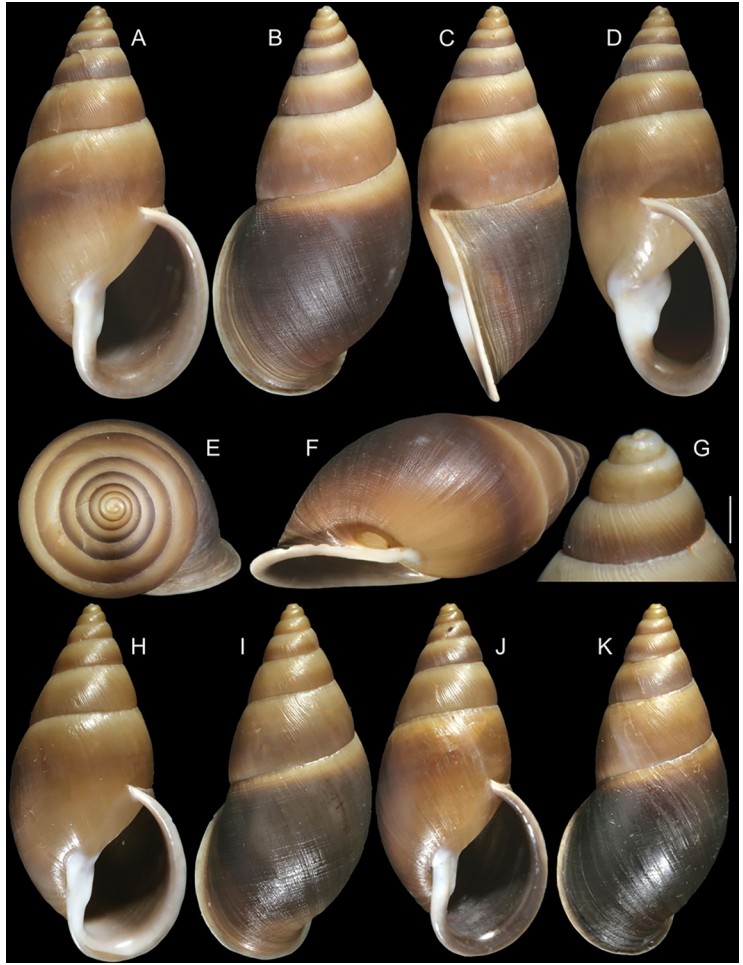

**Fig 28. *Kora jimenezi* shell characters.** (A) holotype MZSP 151907, frontal view (L 43.8 mm). (B) same, dorsal view. (C) same, right view. (D) same, right-slightly ventral view. (E) same, apical view. (F) same, left-slightly anterior view, showing umbilicus. (G) same, detail of apex, profile, scale = 1 mm. (H–I) paratype MZSP 151906#1, frontal and dorsal views (L 41.3 mm). (J–K) paratype MZSP 151906#2, frontal and dorsal views (L 37.6 mm).

*Kora jimenezi* new species Figs 28–33.     ZooBank.   urn:lsid:zoobank.org:
act:105DB150-42A9-4569-952A-2C130DB634FB.

**Types.**   Holotype MZSP 151907; paratypes: MZSP 151906, 58 shells, MNRJ, 2 shells, USNM, 2 shells, all from type locality. BRAZIL. **Minas Gerais**; Itacarambi (W. Vailant-Mattos col.), 15˚10'30"S 44˚13'24"W, altitude, 530 m, MZSP 152191, 28 shells (ii.2019), border of National Park of Peruaçu, 15˚06'35"S 44˚10'13"W, MZSP 152190, 18 shells (i.2020); Between Januária and Barreiros, 15˚28'47"S 44˚22'03"W, MZSP 152162, 4 shells, 152139, 2 spm (iv.2020).

**Type locality.**   BRAZIL. **Minas Gerais**; Itacarambi, west downtown, 15˚00'38"S 44˚07'10"W, altitude 540–550 m (W. Vailant-Mattos col., ii.2019).

**Diagnosis.**   Size about 45 mm, ~2 times longer than wide; lacking dorso-ventral compression. Apex with same color as remaining shell. Subsutural lighter band present. Delicate spiral striae present. Peristome white. Delicate spiral striae present. Aperture occupying ~45% of length and ~70% width. Implantation of outer lip slightly horizontal. Inner lip with high middle fold. Umbilicus narrow. Kidney elongated, narrow. Secondary columellar muscles with 3

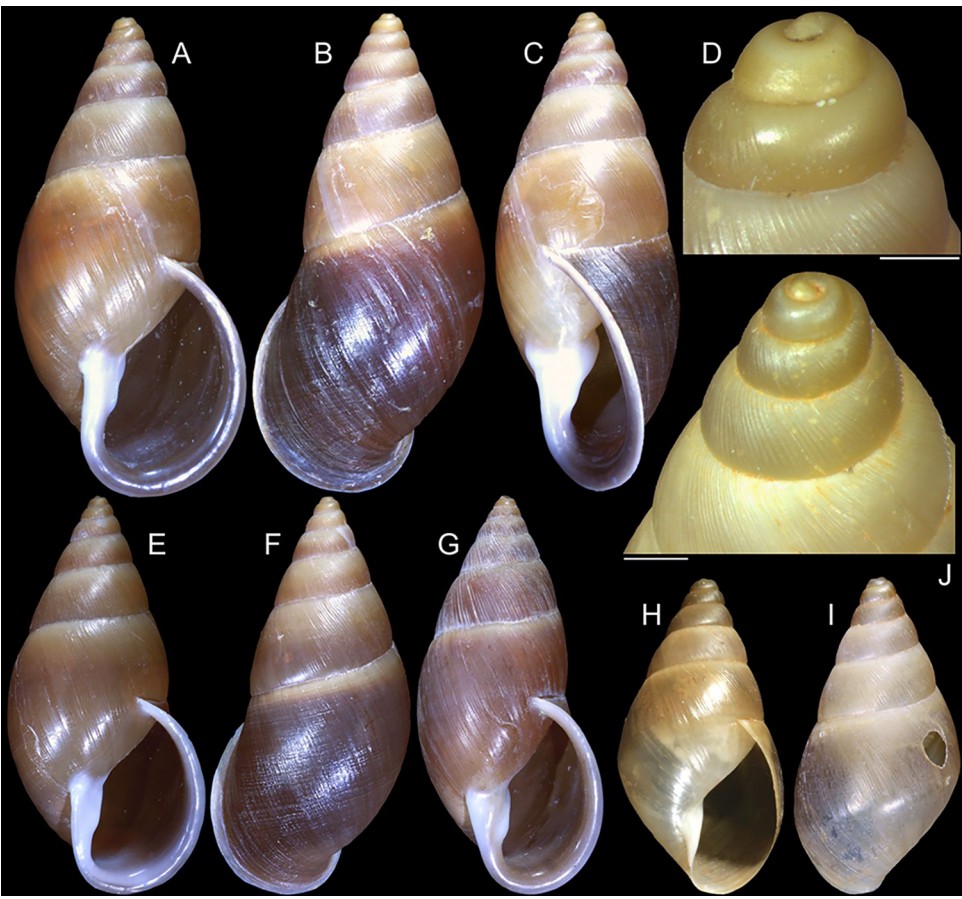

**Fig 29. *Kora jimenezi* shell characters.** (A–C) paratype MZSP 151906#3 (L 38.6 mm), frontal, dorsal, and right views. (D) same, detail of apex in profile, scale = 1 mm. (E–F) paratype MZSP 151906#4 (L 41.4 mm), frontal and dorsal views. (G) paratype MZSP 151906#5 (L 42.5 mm), frontal view. (H–I) paratype MZSP 151906#6 (L 23.8 mm), young specimen, frontal and dorsal views. (J) same, detail of apex, profile-slightly apical view, scale = 1 mm.

insertions in left and in right. Lacking pairs m1v. Jaw arched. Odontophore cartilages ~75% fused. Pair m10 filiform, pair m2a present. Carrefour duct narrow and short, inserted in duct of albumen chamber; bulged region present. Albumen chamber in curve. Penis ~100% of spermoviduct length, lacking umbrella-like fold; epiphallus ~25% of penis length; penis muscle at epiphallus tip.

### Description (distinctive in anatomy)

*Shell.* Length up to 44 mm, outline fusiform-elongated, 1.9–2.2x longer than wide. Color brown, lighter in spire, gradually becoming dark brown in last whorl (Figs 28B, 28F, 28I, 28K and 29B, 29F); subsutural pale band in all whorls well-developed (Figs 28D, 28F and 29B, 29F, 29G). Protoconch (Figs 28G and 29D, 29J, 29I) with 2 whorls, bluntly pointed; length ~10% of shell length, and ~13% of shell width (Fig 28E); mostly smooth, barely sculptured by axial riblets in last whorl. Limit between protoconch and teleoconch weakly visible, weakly prosocline. Teleoconch of ~4.2 whorls successively and uniformly increasing; whorls weakly concave; suture weakly deep. Sculpture growth lines and delicate axial, uniform undulations, ~60 in penultimate whorl; also weak, uniform spiral striae, clearer in last whorl, 55–60 in last whorl, each stria composed of minute aligned pits, separated from each other by equivalent distance

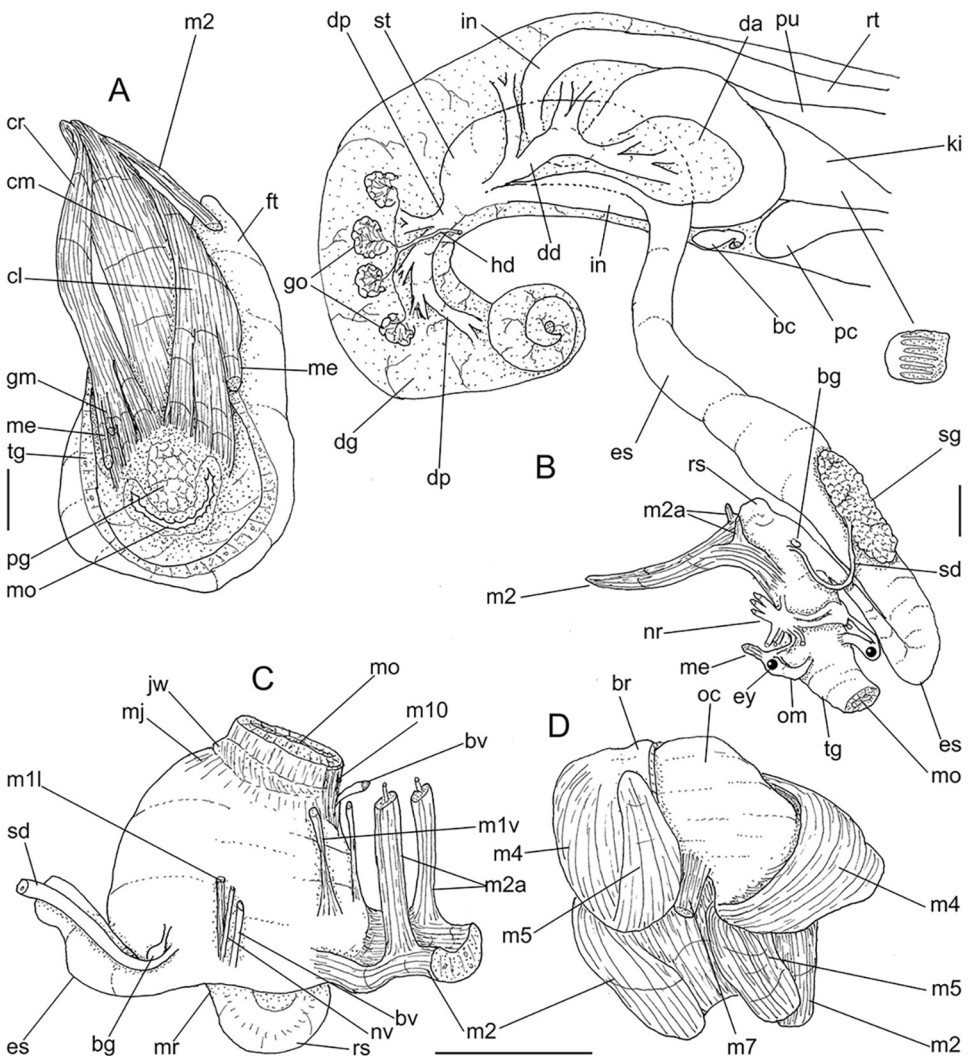

**Fig 30. *Kora jimenezi* anatomical drawings.** (A) head-foot, dorsal view, head, dorsal integument and internal organs removed, remaining muscles expanded. (B) foregut and midgut, mostly ventral view as in situ, topology of some adjacent structures also shown (gonad included), transverse section of indicated region of kidney also shown. (C) buccal mass, right view, esophagus and m2 only partially shown. (D) odontophore, dorsal view, superficial layer of membranes and muscles removed, both cartilages deflected, left muscles as in situ, right muscled expanded. Scales = 2 mm.

of their width. Transverse section rounded (Fig 28E). Peristome weakly dislocated to right; deflected, except for region of callus. Callus weak (Figs 29A, 29H, 29J and 30A, 30E, 30G, 30H). Aperture wide, somewhat dislocated from spire longitudinal axis; length ~45% of shell length, ~70% of shell width. Outer lip inserted distantly from adjacent suture, simple, arched. Inner lip concave, superior half weakly convex, mostly showing outer surface of last whorl; inferior half weakly convex, concave only inferiorly; bearing oblique fold in limit with superior half, having weak elevation preceding its end in inner lip (Figs 28D and 29C); tooth length ~31% of peristome length. Umbilicus opened, relatively wide, partially covered by inferior half of inner lip (Fig 28F).

**Head-foot.** (Fig 30A) With similar features as *K. corallina*. Except for details of both secondary columellar muscles, with fewer and broader basal insertions, 3 in each side (cr, cl);

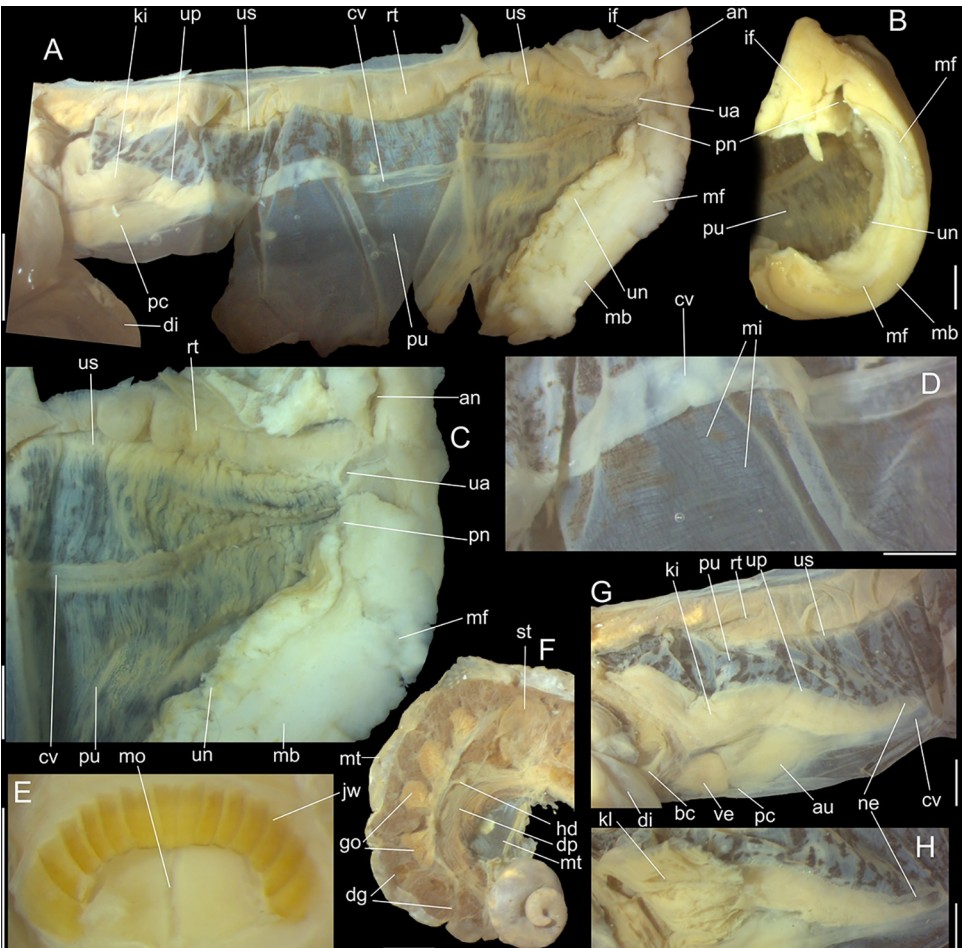

**Fig 31.** *Kora jimenezi* **anatomical photos.** (A) extended pallial (pulmonary) cavity, ventral-inner view, inner edge of pneumostome sectioned and deflected upwards, scale = 5 mm. (B) mantle edge, frontal view as in situ, inner region not shown, scale = 2 mm. (C) same as Fig A, detail of pneumostome region, scale = 2 mm. (D) same, detail of middle region showing longitudinal micro muscular fibers (mi), scale = 1 mm. (E) jaw in situ, ventral view, scale = 1 mm. (F) visceral mass, detail of posterior region partially uncoiled, right view, mantle opened longitudinally and deflected, scale = 2 mm. (G) reno-pericardial region, ventral view, scale = 2 mm. (H) same, posterior region of kidney opened longitudinally in its left edge and deflected upwards.

additionally, pair of central-medial longer insertions slightly as medial part of remaining basal insertions. Pedal gland (pg) much shorter.

**Mantle organs.**   (Fig 31A–31D) Most features similar to *K. corallina*, distinctions and remarks following. Mantle border (mb) with large secondary fold (mf) at left from pneumostome (pn) with rounded, projected left end (Fig 31A, 31B). Pulmonary venation not so developed, composed mainly of pulmonary vein (cv) and perpendicular set of narrow secondary vessels in both sides (Fig 31A, 31C) especially in region preceding pneumostome. Pulmonary vein (cv) relatively wide; its anterior end lacking ramifications, ending in edge of pneumostome (Fig 31C: cv). Region at left from pulmonary vein (cv) lacking visible large vessels (Fig 31A). Mante with well-developed longitudinal muscular micro-fibers (Fig 31D: mi).

**Visceral mass.**   (Figs 30B and 31F) With similar characters as *K. corallina*. Gonad (go) located more anterior, ~2.5 whorls anterior to posterior end.

Circulatory and excretory systems.

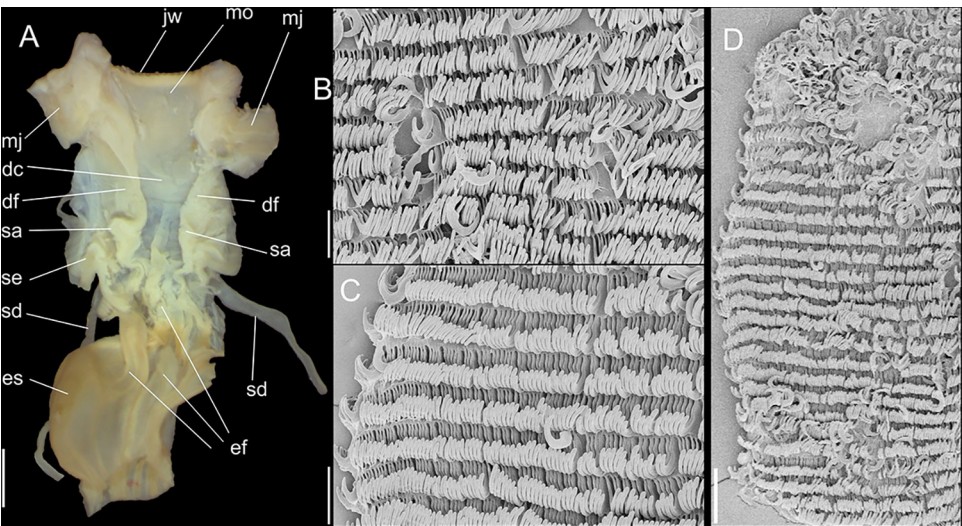

**Fig 32. *Kora jimenezi* anatomical photos and radula in SEM.** (A) foregut, ventral view, odontophore removed, esophagus opened longitudinally, inner surface exposed, scale = 1 mm. (B) Radula in SEM, detail of central region, scale = 100 μm. (C) same, detail of lateral and marginal regions, scale = 100 μm. (D) same, wide view of marginal region, scale = 200 μm.

(Fig 31A, 31G, 31H) general characteristics as those described for *K. corallina*; except for kidney shape, very antero-posteriorly elongated (ki); kidney ~8-times longer than wide; nephropore (ne) in anterior end, at beginning of primary ureter (up). Arrangement of kidney lobe (Figs 30B and 31H: kl), as 6–7 tall folds, fulfilling internal space almost completely.

**Digestive system.** (Fig 30B–30D) Overall morphology similar to that of *K. corallina*. Differences and remarks following. Jaw plate (Fig 31E) thin, narrow, arched; sculptured by successive, rather uniform, transverse, broad folds. Buccal cavity also similar, except for pair of dorsal folds relatively narrow, producing wide dorsal chamber (dc); salivary ducts aperture in tip of papilla (sa), located in medial-posterior side of dorsal folds medial edge. Odontophore intrinsic, extrinsic muscles, and other structures similar to those of *K. corallina*, except for: **m1l**, present, very narrow (Fig 30C); **m2**, very thick, with single bundle; **m2a**, present, and broad, possessing small nerve inside (Fig 30C); **m3**, not visible; **m5**, part originated in m4; **m7**, thick, as single bundle (Fig 30D); **m8** and **m10**, both small. **Radula** (Fig 32B–32D) with same features of *K corallina*, except in having slightly more erected cusp, keeping base more exposed. Esophagus (Figs 30 and 32A: es) initially narrow, becoming broad, being like that up to gastric insertion; internally well-developed longitudinal folds (Fig 32A: ef). Salivary (sg) glands as 2 separated, narrow masses. Stomach (Fig 30B: st) narrow, as simple curve, with anterior (dd) and posterior (dp) ducts to digestive gland relatively broad; anterior duct strongly bifurcated at base, having branches only in its right side.

## Reproductive system

(Fig 31) General structures similar to preceding species, remarks and distinctions following. Gonad as 4–5 well-separated, rounded masses (Figs 30B and 31F: go), united by narrow branches of gonoducts. Hermaphroditic duct (hd), initially narrow and uncoiled (Figs 30B and 31F), gradually becoming intensely coiled, with thick and irregular coils (Fig 33B, 33E). Seminal receptacle small (Fig 33B, 33E: sr), elongated, weakly flattened, ~6 times longer than wide, tip performing 1 whorl (Fig 33E); inserted between curved insertion of hermaphroditic

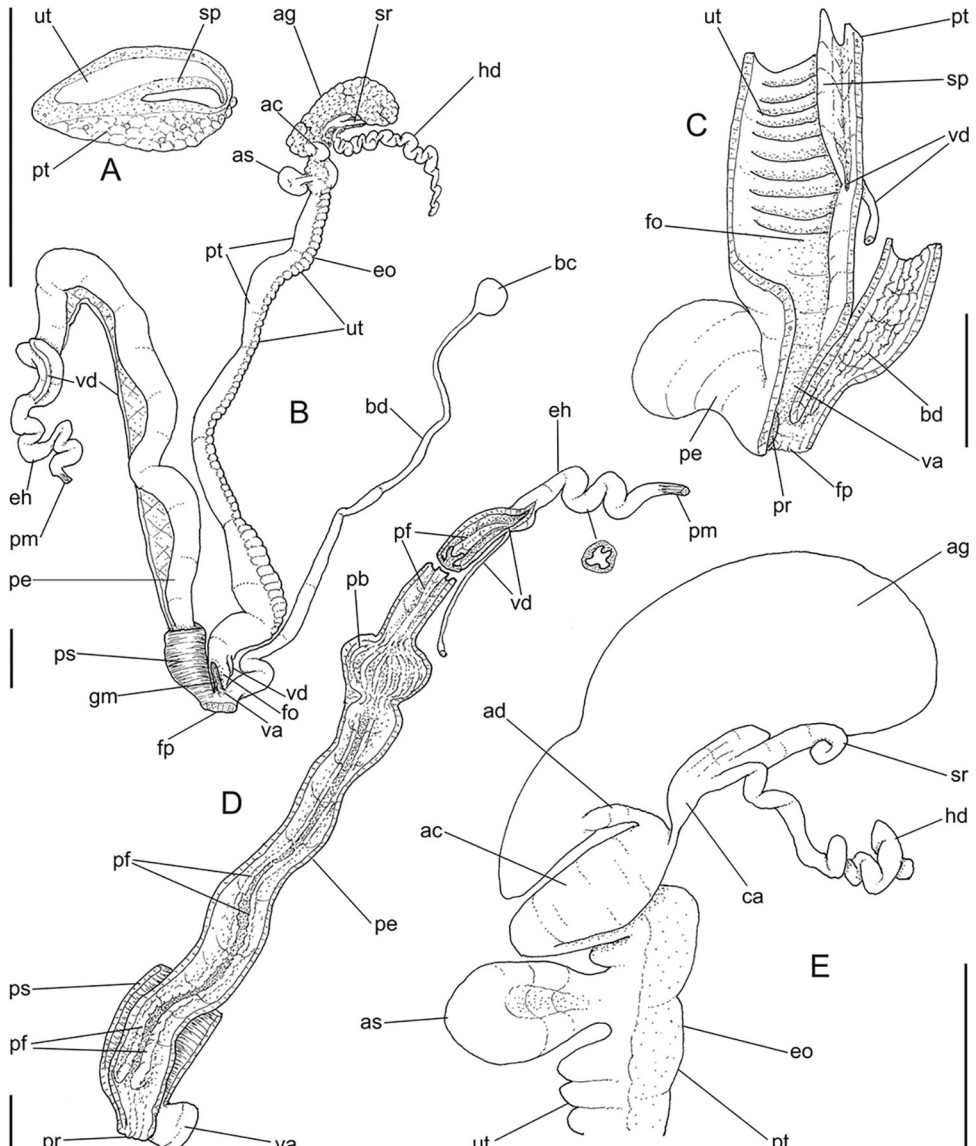

**Fig 33. *Kora jimenezi* anatomical drawings.** (A) spermoviduct, transverse section in its middle region. (B) genital structures, dorsal view, gonad extracted (see Figs 30B, 31F). (C) same, detail of anterior region, dorsal view, female portions opened longitudinally. (D) penis, opened longitudinally, with transverse section of indicated region of epiphallus also shown. (E) genital structures in albumen gland level if it was transparent, ventral view. Scales = 2 mm.

duct and wide projection of carrefour (ca), about as wide as receptacle, ~1/3 its length. Fertilization complex or carrefour (Fig 33E: ca) (except posterior projection) conic, abruptly tapering, becoming narrow, short duct; inserting at tip of spermoviduct, in intersection of albumen gland duct (ad) and its chamber (ac). Albumen gland (ag) of ~1/3 whorl in length. Albumen gland duct (Fig 33E: ad) subterminal, narrow, connected to tip of spermoviduct by side of carrefour duct. Albumen chamber (Fig 33B, 33E: ac) bulged curve, ~3-times longer than wide, widely connected with its proximal end continuous with spermoviduct (eo). Secondary albumen chamber (as) slightly larger than of primary chamber, balloon-like, duct broad, insertion at some distance from albumen chamber insertion. Spermoviduct (Fig 33B: eo) long and narrow, ~25 times longer than wide. Prostate wide but low (Fig 33A: pt), almost 1/2 of

spermoviduct diameter; uterus lacking glandular walls, highly, transversally, and relatively uniformly folded (Fig 33A–33C: ut). Sperm inner longitudinal fold (Fig 33A, 33C: sp) simple, tall, narrow fold, with distal edge sharp. Anterior regions of genital tubes with walls weakly muscular (Fig 33C); free oviduct (fo) and vagina (va) with inner surface smooth. Bursa copulatrix (bc) and its duct (bd) ~80% of spermoviduct length (Fig 33B); bursa duct with basal half slightly thicker than distal half. Basal region of duct of bursa copulatrix with 4–5 longitudinal, long, rather irregular folds (Fig 33C: bd). Genital muscle (Fig 33B: gm) small, narrow. Penis relatively broad, ~as long as spermoviduct length (Fig 33B: pe). Penis muscle inserted terminally in epiphallus tip (Fig 33B, 33D: pm), short, simple. Penis walls entirely weakly muscular (Fig 33D). Epiphallus (eh) ~1/4 of penis' length, amply opened to penis; only vas deferens insertion and end of special inner fold (pf) marking its limit (Fig 33D: vd). Epiphallus inner surface only with ~4 narrow, low, parallel folds (Fig 33D: eh). Internal penial arrangement of folds clearly with 3 regions (Fig 33D): (1) basal ~2/3, after short basal region having only longitudinal wrinkles, pair of strong folds abruptly appearing (pf), with rounded end, running close from each other, interspace smooth, equivalent to their width, all along this region, narrowing gradually; surrounding both folds smooth surface; (2) following ~1/6 as bulged region, preceded by sphincter-like constriction, internally having only 7–8 longitudinal, uniform, separated, narrow folds; (3) distal ~1/3 as narrow region, internally having only single, tall fold (pf), fold basal half only planar, gradually increasing, becoming Y-shaped in section, its distal end tapering and converging to aperture of vas deferens (vd).

**Central nervous system.**   (Fig 30B: nr) Same characters as *K. corallina*.

**Distribution.**   Known only from the for region of Itacarambi, Minas Gerais.

**Habitat.**   Under rocks or in rock crevices.

**Etymology.**   The specific epithet is in honor of David Jimenez, an expedition sponsor and remarkable contributor to the malacology.

**Measurements.**   (in mm): MZSP 151907 (holotype, Fig 28A–28G): 43.8 by 24.2; MZSP 151906#1 (Fig 28H–28I): 41.3 by 20.1; #3 (Fig 29A–29D): 38.6 by 18.7; #4 (Fig 29E, 29F): 41.4 by 19.6; paratype MZSP 151906#5 (Fig 29G): 42.5 by 18.9.

**Material examined.**   The types.

**Taxonomic remarks.**   *Shell. Kora jimenezi* has an average shell size of approximately 45 mm, making it larger than *K. nigra*, *K. aetheria*, *K. vania*, and *K. curumim*, but smaller than *K. tupan* and *K. ajar*. Its shell is about 2.0 times longer than it is wide, giving it an elongated shape compared to most other congeneric species, except for *K. corallina* and *K. rupestris*, which are further more elongated, but it is more elongated than *K. nigra*, *K. ajar* and *K. curumim*. The shell aperture comprises about 45% of the total shell length, which is much shorter than the apertures of *K. nigra*, *K. rupestris*, *K. tupan*, *K. ajar*, *K. kremerorum*, and *K. curumim*. Additionally, like *K. nigra*, *K. tupan*, *K. ajar*, *K. kremerorum*, and *K. vania*, *K. jimenezi* has a horizontally oriented superior implantation of the outer lip.

**Anatomy.**   *Kora jimenezi* has wide folds along the mantle edge (Fig 31A, 31C: mf), with rounded ends, a model shared only with *K. uhlei*. It differs from all congener species in lacking well-developed wide vessels to the left of the pulmonary vein (Fig 31A), as the remaining species exhibit well-developed vessels in this lung region. The simple anterior end of the pulmonary vein (Fig 31A: cv) approaches *K. jimenezi* from *K. nigra*, *K. rupestris*, and *K. jimenezi*, while the remaining species have a branched anterior end. Additionally, *K. jimenezi* displays strong venation to the right of the pulmonary vein only in the region preceding the pneumostome, a trait shared only with *K. corallina* among its congeners. The kidney lobe completely surrounds the kidney walls (Fig 31H: kl), differentiating *K. jimenezi* from *K. nigra*, *K. rupestris* and *K. tupan*. In terms of muscle structure, *K. corallina* has only 3 anterior insertions of the left accessory columellar muscle (Fig 30A: cl), being the fewer from all its congeners, except for

*K. uhlei*. Additionally, it also has 3 anterior insertions of the right accessory columellar muscle, being the fewer from all its congeners, except for *K. uhlei*. These accessory columellar muscles also lack a posteriorly located medial branch, which is present in all remaining congeners except *K. corallina*. Furthermore, *K. jimenezi* has the odontophore muscle pair m1l, similarly to *K. rupestris*, *K. tupan*, *K. aetheria* and *K. uhlei*. The absence of the pairs of odontophore muscles m1v (Fig 30C) approached it from *K. tupan*, *K. ajar* and *K. uhlei*, which also lack these muscles, while the remaining species have 1–2 pairs of m1v. The species is the only one lacking a clear m3 in the odontophore among its congeners. In having branches only on the right side of the posterior duct to the digestive gland (Fig 30B: dp), *K. jimenezi* differs from *K. nigra*, and its only right branching in the anterior duct to the digestive gland further differentiates it from *K. corallina*, *K. rupestris* and *K. aetheria*, which have bilateral branches. The narrow and arched shape of the jaw plate (Fig 31E) is exclusive, differing from all other congeners. The salivary gland aperture, located in the posterior third of the buccal dorsal wall (Fig 32A: sa), is only similar to *K. tupan* condition, and the presence of of a salivary papilla is a shared feature with *K. uhlei*. The degree of fusion of the odontophore cartilages, around 75%, is similar to most congeners but greater than that of *K. nigra* (~60˚) and less than that of *K. uhlei* (~90˚). In terms of odontophore m4-m5 pairs of muscles, the m4 muscle covering m5 distinguishes *K. jimenezi* from *K. tupan* and *K. uhlei*, in which these muscle pairs are continuous with each other. The odontophore muscle m7 of *K. jimenezi* (Fig 30D) is the only in being stubby, broad, thick. Also, this species is singular in having the pair m8 narrow, while all other congeners these muscles are wide. Finally, the filiform m10 muscle in *K. jimenezi* differs from all other congeners.

**Genital system.** *Kora jimenezi* has a small curve at the end of the hermaphrodite duct (Fig 33E: hd), which distinguishes it from *K. nigra*. The conical shape of its carrefour (ca) is distinct from those of *K. nigra* and *K. rupestris*, and its carrefour duct being narrow and short is similar only to *K. ajar* and *K. uhlei*. The species has a bulged portion on the opposite side of the hermaphrodite duct at the base of the seminal receptacle (sr), approaching it from *K. tupan* and *K. uhlei*; besides, this bulger region is very wide in *K. jimenezi* (Fig 33E). *K. jimenezi* is unique in having the tip of the seminal receptacle (sr) coiled, and the carrefour duct inserting directly into the albumen chamber (Fig 33E: ac). Its albumen chamber (Fig 33E: ac) forms a simple curve, contrasting with the blind sacs found in *K. nigra*, *K. rupestris*, *K. tupan*, and *K. aetheria*. In having a single sperm fold in the spermoviduct (Fig 33A: sp), *K. jimenezi* differs from *K. nigra*, which has two. Additionally, it has the broadest prostate band in the spermoviduct (~50%) among its congeners, only *K. rupestris* share similar condition. *K, jimenezi* is the single species lacking muscular anterior portion of the bursa copulatrix duct (Fig 33C) which is present in all remaining congener species. Its penis length is approximately as long as the spermoviduct, only *K. nigra* has this same condition. The bursa copulatrix duct is about 80% of the spermoviduct length, which is the shortest condition amongst its congeners, only *K. aetheria* and *K. uhlei* have this same condition. The vas deferens of *K. jimenezi* lacks the strong curve preceding its insertion at the tip of the penis, distinguishing it from *K. nigra*, *K. tupan*, *K. ajar*, and *K. uhlei*. Its penis base lacks clear muscular walls (Fig 33D), being it the single species with this attribute. The penis of *K. jimenezi* features the usual pair of inner folds, but it simple, lacking branches (Fig 6C: pf), distinguishing it from *K. corallina*, *K. aetheria* and *K. uhlei* among its congeners; additionally, this pair of folds is unique in being narrow and very long. *K. jimenezi* lacks the umbrella-like transverse penial fold found in most of its congeners, with the exceptions of *K. nigra* and *K. corallina*. The epiphallus comprises about 25% of the penial length, the longest ratio among its congeners, though similar proportions are present in *K. nigra*, *K. ajar* and *K. uhlei*. *K. jimenezi* lacks strong longitudinal fold in the epiphallus (Fig 33D), distinguishing if from *K. corallina*, *K. tupan* and *K. ajar*; however, a strong fold is in

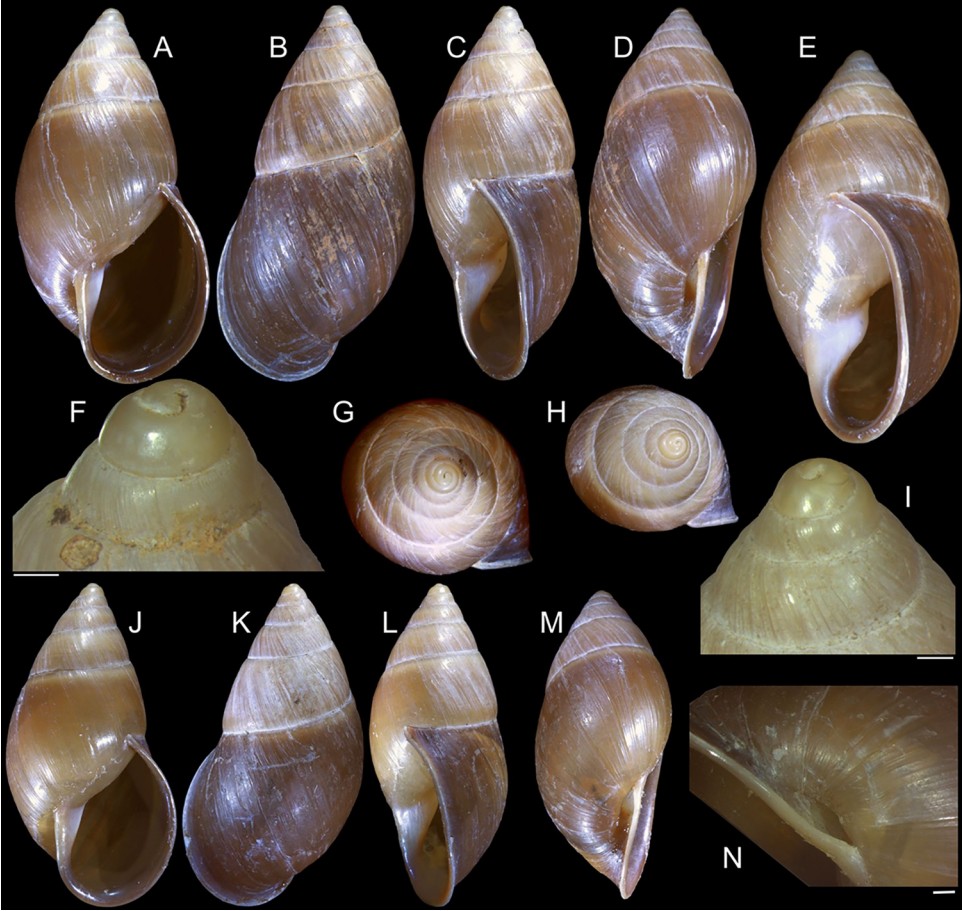

**Fig 34. *Kora uhlei* shell of types.** (A–G) holotype MZSP 165720 (L 41.4 mm). (A) frontal view. (B) dorsal view. (C) right view. (D) left-slightly anterior view. (E) right-slightly anterior view; (F) detail of apex in profile. (G) apical view. (H–N) paratype MZSP 164889#2 (L 40 mm). (H) apical view. (I) detail of apex in profile. (J) frontal view, (K) dorsal view. (L) right view. (M) left-slightly anterior view. (N) detail of umbilicus, left-slightly anterior view. Scales = 1 mm.

continuation from epiphallus aperture, inside posterior penis region, Y-in section, which is exclusive. Its penis muscle (pm) inserts terminally in the epiphallus, unlike *K. rupestris*, *K. ajar*, and *K. uhlei*, which have more basal insertions. The species has a distinct model of gonad (Fig 31F), with separated, aligned lobes.

***Kora uhlei* new species Figs 34–38.** **ZooBank.** urn:lsid:zoobank.org:act: B8283016-B72B-41A6-A1B3-A9CADB1D757F.

**Types.** Holotype MZSP 165720, spm; paratypes: MZSP 164889, 7 spm, MZSP 164888, 15 shells, all from type locality.

**Type locality.** BRAZIL. **Minas Gerais**; Matias Cardoso, near Lagoa do Cajueiro State Park, near São Francisco River, 14˚53'10"S 43˚54'51"W (Wesley Vailant-Mattos col., vi.2023).

**Diagnosis.** Size about 45 mm, ~2 times longer than wide; lacking dorso-ventral compression. Apex with same color as remaining shell. Subsutural lighter band present. Peristome with brown spots. Delicate spiral striae absent. Aperture occupying ~50% of length and ~65% width. Implantation of outer lip slightly vertical. Inner lip with high middle fold. Umbilicus wide. Secondary columellar muscles with 3 insertions in left and in right. Absence of pairs of m1v. Jaw rectangular. Odontophore cartilages ~90% fused. Pair m10 narrow, pair m7 filiform. Carrefour duct narrow and long, inserted between duct of albumen gland and albumen

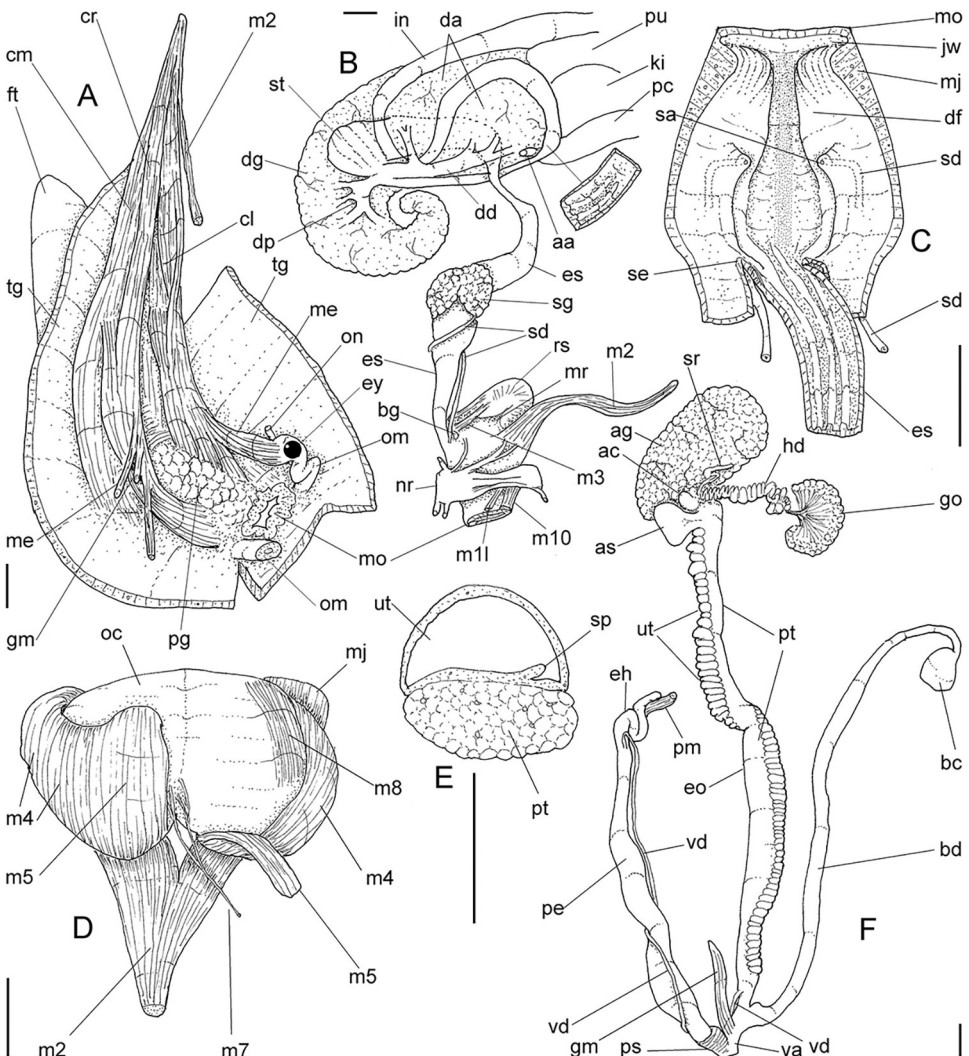

**Fig 35.** *Kora uhlei* **anatomical drawings.** (A) head-foot, dorsal view, head, dorsal integument and internal organs removed, remaining muscles expanded, left ommatophore only partially shown. (B) foregut and midgut, mostly ventral view as in situ, topology of some adjacent structures also shown, portion of indicated portion of intestine shown opened. (C) buccal mass, ventral view, odontophore removed. Esophagus opened longitudinally. (D) odontophore, dorsal view, superficial layer of membranes and muscles removed, both cartilages deflected, left muscles as in situ, right muscled expanded. (E) spermoviduct, transverse section in its middle region. (F) genital structures, dorsal view. Scales = 2 mm.

chamber; bulged region present. Albumen chamber in curve. Penis ~60% of spermoviduct length, umbrella-like fold present, with 3 rods; epiphallus ~25% of penis length; penis muscle at epiphallus base.

## Description (distinctive in anatomy)

*Shell.* (Fig 34) Length up to 42 mm, outline fusiform-globose, ~2x longer than wide. Color brown, lighter in spire, gradually becoming dark brown in last whorl (Fig 34B, 34K); subsutural pale band developed (Fig 34B, 34D, 34J, 34K). Protoconch (Fig 34F, 34I) with 2 whorls, bluntly pointed; length ~5.5% of shell length, ~17% of shell width (Fig 34G, 34H); mostly smooth, barely sculptured by axial riblets in last half whorl. Limit between protoconch and

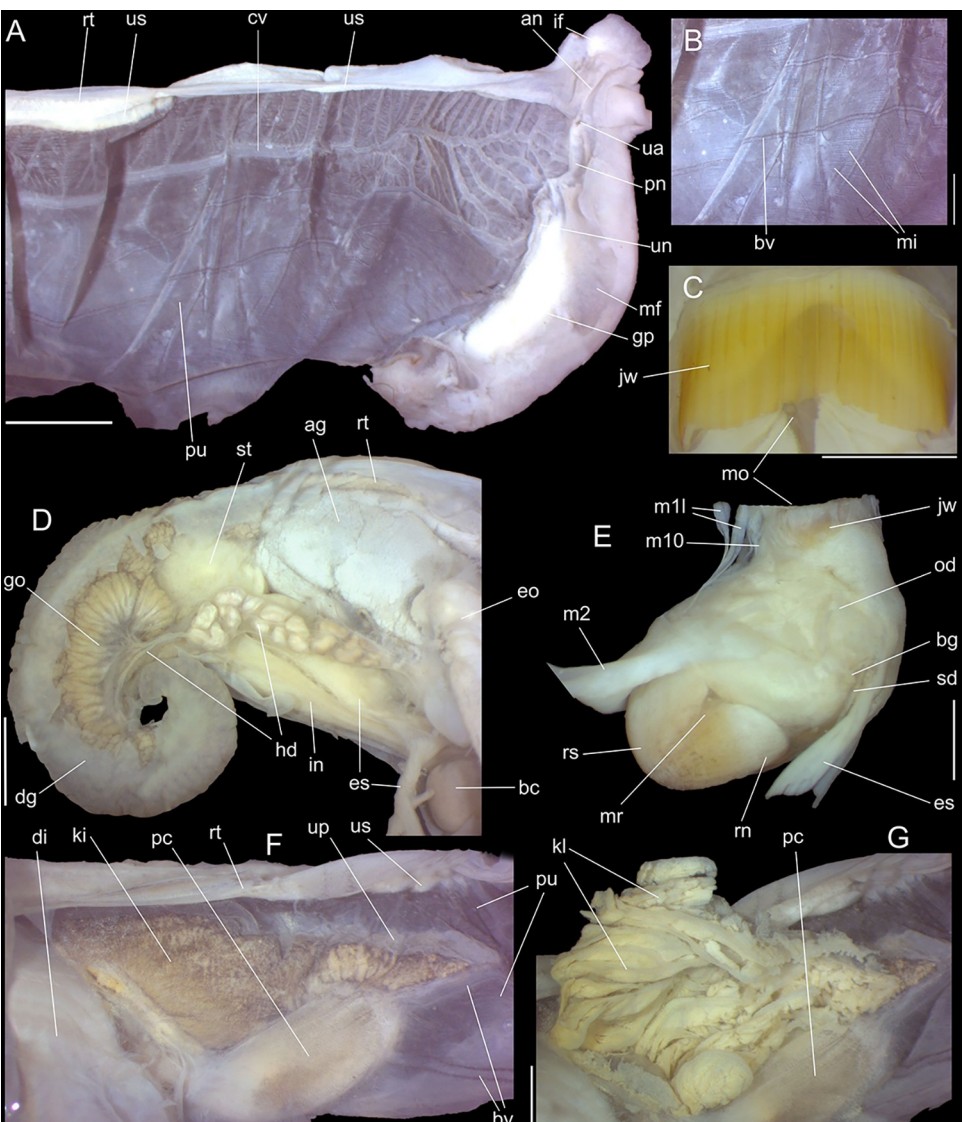

**Fig 36.** ***Kora uhlei*** **anatomical fotos of holotype MZSP 165720.** (A) extended pallial (pulmonary) cavity, ventral-inner view, inner edge of pneumostome sectioned and deflected upwards, scale = 5 mm. (B) same, detail of its mid region, showing some vessels and longitudinal micro-muscular fibers, scale = 2 mm. (C) jaw plate in situ, ventral view, scale = 1 mm. (D) posterior end of visceral mass, uncoiled and with part of superficial mantle removed, scale = 3 mm. (E) Buccal mass, left view, scale = 2 mm. (F) reno-pericardial region od pallial roof, ventral view, scale = 2 mm. (G) same, kidney ventral wall opened along its left edge and deflected upwards, exposing renal lobe, scale = 2 mm.

teleoconch visible, weakly prosocline. Teleoconch of ~4.2 whorls successively and uniformly increasing; whorls weakly concave, profile almost straight; suture shallow. Sculpture growth lines and delicate axial, uniform undulations, ~60 in penultimate whorl; spiral striae absent. Transverse section rounded (Fig 34G, 34H). Peristome not dislocated; deflected, except for region of callus. Callus weak (Fig 34A, 34C, 34E, 34J, 34L). Aperture wide, not dislocated from spire longitudinal axis; length ~50% of shell length, ~65% of shell width. Outer lip inserted distantly from adjacent suture, simple, arched. Inner lip concave, superior half weakly convex, mostly showing outer surface of last whorl covered by thin callus; inferior half almost straight, concave only inferiorly; bearing oblique fold in limit with superior half (Fig 34C, 34E, 34L);

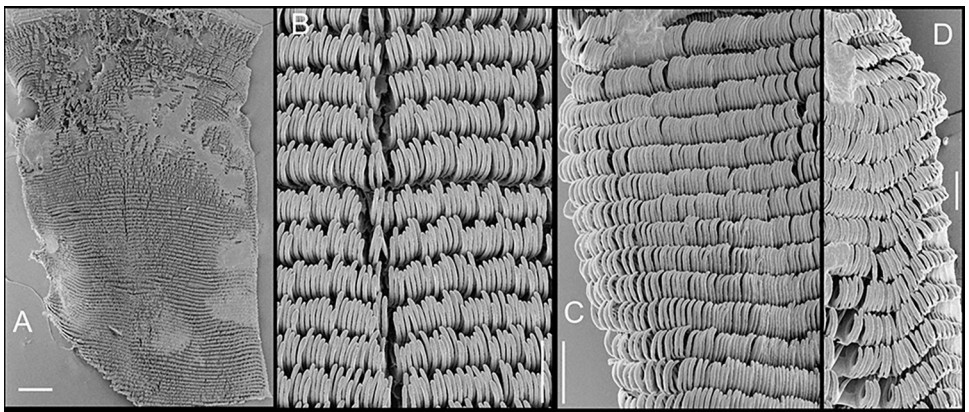

**Fig 37. *Kora uhlei* radulae in SEM.** (A) panoramic view, scale = 50 μm. (B) detail of central region, scale = 100 μm. (C) detail of lateral and marginal region, scale = 100 μm. (D) detail of marginal region, scale = 50 μm.

tooth length ~35% of peristome length. Umbilicus opened, relatively wide, partially covered by inferior half of inner lip (Fig 34D, 34M, 34N).

**Head-foot.** (Fig 35A) With similar features as *K. corallina*. Except for details of both secondary columellar muscles, with fewer and broader basal insertions, 3 in left side (cl), and

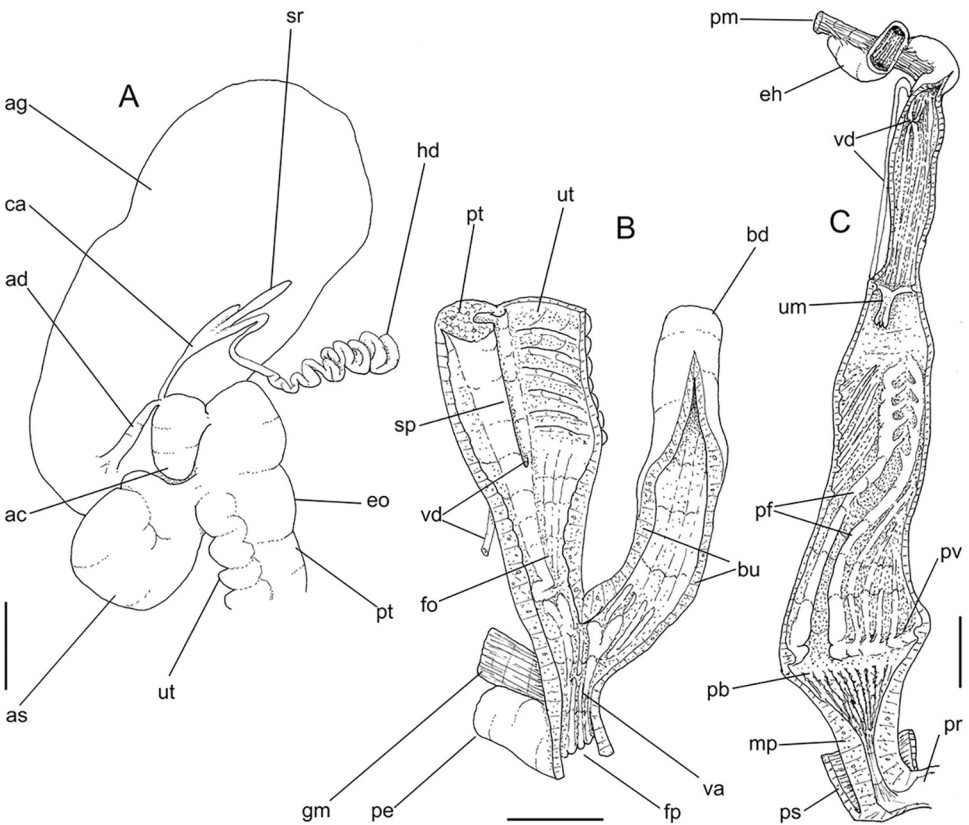

**Fig 38. *Kora uhlei* anatomical drawings.** (A) genital structures in albumen gland level if it was transparent, ventral view. (B) genital structures, detail of anterior region, dorsal view, female portions opened longitudinally. (C) penis, opened longitudinally, epiphallus with only small window-like orifice. Scales = 2 mm.

single, wide bundle in right side (cr), having a single additional narrow anterior branch adjacent to genital (gm) and ommatophore (me) muscles; additionally, pair of central-medial longer insertions well-developed, but turned backwards (instead of anteriorly).

**Mantle organs.** (Fig 36A, 36B) Most features similar to *K. corallina*, distinctions and remarks following. Mantle border (mb) with large secondary fold (mf) at left from pneumostome (pn) with blunt, projected left end (Fig 36A: mf). Mantle edge gland well-developed, white (gp). Pulmonary venation strong in region of pneumostome, very branched in anterior end of pulmonary vein (cv); those at left of pulmonary vein intercalated with its branches and branches from collar vessel. Vessels at right from pulmonary vein, perpendicular positioned, well-developed all along pulmonary length, also intercalated and slightly more developed in region preceding pneumostome. Pulmonary vein (cv) relatively narrow. Region at left from pulmonary vein (cv) with visible large, intercalated vessels (Fig 36A). Mante with well-developed longitudinal muscular micro-fibers (Fig 36B: mi).

**Visceral mass.** (Figs 35B and 36D) With similar characters as *K. corallina*. Gonad (go) located more ~1 whorl anterior to posterior end.

**Circulatory and excretory systems.** (Fig 36F, 36G) General characteristics as those described for *K. corallina*; except for kidney shape, relatively antero-posteriorly elongated (ki); kidney ~3-times longer than wide, triangular. Reno-pericardial area occupying ~1/15 of pulmonary roof. Arrangement of kidney lobe (Fig 36G: kl), as 6–7 tall folds, fulfilling internal space almost completely.

**Digestive system.** (Fig 35B–35D) Overall morphology similar to that of *K. corallina*. Differences and remarks following. Jaw plate (Fig 36C) thin, wide (~2.5 times wider than long), almost straight; sculptured by successive, rather uniform, transverse, narrow folds; cutting edge chevron-like. Buccal cavity also similar, except for pair of dorsal folds (Fig 35C: df) relatively narrow, producing wide dorsal chamber between them; salivary ducts aperture in tip of papilla (sa), located in medial-middle side of dorsal folds medial edge. Odontophore intrinsic, extrinsic muscles, and other structures similar to those of *K. corallina*, differences and notes following (Figs 35B, 35D and 36E) **m1l**, present, narrow and long; **m2**, narrow, inserted as 2 bundles; **m3**, longitudinal pair, in esophageal insertion area; **m5**, mostly by side of m4, originated from cartilages; **m7**, extremely narrow, each one originated in 2 different points (Fig 35D); **m8**, broad. Pair of cartilages fused with each other in higher degree, ~90%. **Radula** (Fig 37) with same attributes as *K corallina*, except in having more pairs of teeth per row (~300) (Fig 37A); cusp more elongated, touching adjacent row (Fig 37B, 37C), with tip slightly more arched; 7–8 more marginal teeth with great diminishment of size (Fig 37D). Salivary (sg) glands as 2 rounded masses, partially united in medial region (Fig 35B: sg). Stomach (Figs 35B and 36D: st) bulged; internally with 4–5 transverse folds, with anterior (dd) and posterior (dp) ducts to digestive gland relatively broad; anterior duct large bifurcation at base, having branches only in its right side. Posterior duct relatively short, also having only branches in right side. Intestine (in) with pair of low parallel folds initiating in stomach, ending in region adjacent to pericardium (Fig 35B).

**Reproductive system.** (Figs 35F and 38) General structures similar to preceding species, remarks and distinctions following. Gonad as single block of digitiform acini (Figs 35F and 36D: go). Hermaphroditic duct (hd), initially narrow and uncoiled (Fig 36D), abruptly becoming intensely coiled, with very thick, irregular coils (Figs 35F and 36D: hd). Seminal receptacle small (Figs 35F and 38A: sr), elongated, weakly flattened, ~6 times longer than wide, tip bluntly pointed (Fig 38A); inserted between curved insertion of hermaphroditic duct and narrow projection of carrefour (ca), about as wide as receptacle, ~1/3 its length. Fertilization complex or carrefour (Fig 38A: ca) (except posterior projection) conic, tapering, becoming narrow, long duct, as long as carrefour itself; inserting in intersection of albumen gland duct (ad) and its

chamber (ac). Albumen gland (ag) of ~1/3 whorl in length. Albumen gland duct (Fig 38A: ad) subterminal, narrow, connected to superior side of albumen chamber (ac), by side of carrefour duct. Albumen chamber (Figs 35F and 38A: ac) as small blind-sac, ~twice longer than wide, connected to tip of spermoviduct by narrow, short duct located also in its posterior region, opposite to insertion of carrefour. Secondary albumen chamber (as) ~3x larger than of primary chamber, bulged, balloon-like, duct narrow, insertion at some distance from albumen chamber insertion. Spermoviduct (Fig 35F: eo) long and narrow, ~20 times longer than wide. Prostate wide, tick (Fig 35E: pt), almost 1/2 of spermoviduct diameter and thickness; uterus lacking glandular walls, highly, transversally, and relatively uniformly folded (Figs 35F and 38B: ut). Sperm inner longitudinal fold (Figs 35E, 38B: sp) simple, posteriorly short, gradually becoming taller, broad, fulfilled by prostate gland in anterior region (Fig 38B). Anterior regions of genital tubes with muscular walls (Fig 35B); free oviduct (fo) and vagina (va) with inner surface possessing longitudinal, wide, low folds. Bursa copulatrix (bc) and its duct (bd) ~80% of spermoviduct length (Fig 35F); bursa duct with basal third with walls thick muscular (Fig 38B: bu). Basal region of duct of bursa copulatrix with 4–5 longitudinal, long, wide folds (Fig 38B: bd). Genital muscle (Figs 35F and 38B: gm) large, single. Penis relatively broad, ~60% of spermoviduct length (Fig 35F: pe). Penis shield (ps) short, ~10% of penis length. Penis muscle inserted subterminally, along epiphallus lateral side (Figs 35F and 38C: pm), long, with epiphallus zigzagging it. Penis walls entirely weakly muscular, except for its base, with ~1/5 thick muscular (Fig 38C: mp). Epiphallus (eh) ~1/4 of penis' length if straightened, amply opened to penis; only vas deferens insertion marking its limit (Fig 38C: vd). Epiphallus inner surface only with ~8–10 narrow, low, parallel folds (Fig 38C: eh). Internal penial arrangement of folds clearly with 3 regions (Fig 38C): (1) basal ~1/4, thick muscular region, with inner lumen with 10–12 longitudinal, simple, uniform folds; (2) previous region suddenly opening to wide, bulged area possessing wide transverse fold, encircling almost entire local circumference, except for short region, in which edges turn perpendicularly, originating pair of strong, longitudinal folds running along ~1/2 of penis length; circling these pair of main folds 4–5 secondary longitudinal folds, running parallel to main folds, with interspaces equivalent to their width; pair of main folds with smooth interspace, equivalent to their width, their distal third possessing 5–6 short, uniform, intercalated branches, turned to its pair, up to sudden folds end, smooth ~1/4 of this region's length preceding umbrella-like transverse fold; umbrella-like transverse fold (um) very narrow, possessing only 3 rods; (3) distal ~1/3, slightly narrow, from umbrella-like fold, continuing to epiphallus, internally with 6–7 longitudinal, narrow, simple, separated, uniform folds, 2 of them converging in aperture of vas deferens (vd).

**Central nervous system.** (Fig 35B: nr) Same characters as *K. corallina*.

**Distribution.** Known only from the for region of the type locality.

**Habitat.** Under rocks or rock crevices.

**Etymology.** The specific epithet is in honor of Mauricio Uhle, São Paulo, shell collector, sponsor of expeditions, and enthusiastic contributor to the malacology.

**Measurements.** (in mm): holotype (Fig 34A–34G): 41.4 by 20.6; paratype MZSP 164889# (Fig 34H–34G): 40.0 by 20.1.

**Material examined.** The types.

**Taxonomic remarks.** *Shell. Kora uhlei* has an average shell size of approximately 45 mm, making it larger than *K. nigra*, *K. aetheria*, *K. vania*, and *K. curumim*, but smaller than *K. tupan* and *K. ajar*. Its shell is about 2.0 times longer than it is wide, giving it a narrower shape compared to *K. nigra*, *K. ajar* and *K. curumim*, but more globose than *K. corallina* and *K. rupestris*. The shell aperture comprises about 50% of the total shell length, which is an ample peristome, only *K. tupan* and *K. ajar* have wider apertures; and *K. corallina*, *K aetheria*, *K. jimenezi* and *K. vania* have shorter apertures. Additionally, unlike *K. nigra*, *K. tupan*, *K. ajar*,

*K. jimenezi*, *K. kremerorum*, and *K. vania*, *K. corallina* lacks a horizontally oriented superior implantation of the outer lip.

**Anatomy.** *Kora uhlei* has wide folds along the mantle edge (Fig 36A), with rounded tip, a condition distinct from all its congeners, but only shared with *K. jimenezi*. It differs from *K. nigra*, *K. aetheria*, and *K. jimenezi* by having a single intercalated pair of wide vessels to the left of the pulmonary vein (Fig 36A, 36B), as these species exhibit different vascular arrangements. The branched anterior end of the pulmonary vein (Fig 36A: cv) distinguishes *K. uhlei* from *K. nigra*, *K. rupestris*, and *K. jimenezi*, which have simpler structure; but the intercalated kind of branching is exclusive feature. Additionally, *K. uhlei* displays strong venation up to middle level of the pulmonary cavity, distinguishing it from other arrangements of *K. corallina*, *K. rupestris* and *K. jimenezi*. The kidney lobe completely surrounds the kidney walls (Fig 36G: kl), differentiating *K. uhlei* from *K. nigra*, *K. rupestris*, and *K. tupan*. In terms of muscle structure, *K. uhlei* has 3 anterior insertions both, of the left and right accessory columellar muscles (Fig 35A: cl, cr), the fewer condition amongst its congeners, only shared with *K. jimenezi*. These accessory columellar muscles have a posteriorly located medial branch, similarly to all remaining congeners except *K. jimenezi*. Furthermore, *K. uhlei* has the odontophore muscle pair m1l, also present in *K. rupestris*, *K. tupan*, *K. aetheria*, *K. jimenezi*, and *K. uhlei*. It lacks the odontophore muscles m1v (Fig 35D), similarly to *K. tupan*, *K. ajar*, *K. jimenezi*, and *K. uhlei*, which also lack these muscles, and differs from remaining species that have them. The connection of the odontophore muscle pair m3 to the esophagus origin distinguishes *K. uhlei* from *K. rupestris*, where it connects to the m2 pair, and from *K. jimenezi*, which lacks this muscle. In having branches only on the right side of the posterior duct to the digestive gland (Fig 35B: dp), *K. uhlei* differs from *K. nigra*, and in having only right branches in the anterior duct to the digestive gland (dd) further approaches it from all congeners, except *K, corallina*, *K. rupestris* and *K. aetheria*, which have bilateral branches; additionally, as an idiosyncrasy, both ducts are broad and relatively short. The rectangular shape of the jaw plate (Fig 36C) is different from *K. corallina*, *K. rupestris*, *K. aetheria* and *K jimenezi*, which have other models. The salivary gland aperture, located in the middle third of the buccal dorsal wall (Fig 35C: sa), differs from *K. tupan* and *K. jimenezi*, and the presence of a salivary papilla is a shared feature with *K. jimenezi*. Its degree of fusion of the odontophore cartilages, around 90% (Fig 35D: oc), is the greatest degree of fusion among its congeners. In terms of odontophore m4-m5 pairs of muscles, the m4 muscle is in continuation to m5, a condition only founding in *K. tupan*, and distinct from all other congener species. *K. uhlei* is the only species in having a filiform m7 pair (Fig 35D), being an exclusivity. Finally, the narrow m10 muscle in *K. uhlei* differs from the broader form in *K. tupan* and *K. aetheria*, and from the filiform version in *K. jimenezi*.

**Genital system.** *Kora uhlei* has a small curve at the end of the hermaphrodite duct (Fig 38A: hd), which distinguishes it from *K. nigra*. The conical shape of its carrefour (ca) is distinct from those of *K. nigra* and *K. rupestris*, and its carrefour duct is narrow and long, being distinct from those of *K. corallina*, *K. ajar* and *K. jimenezi*, which possess other chapes. The species has the bulged portion on the opposite side of the hermaphrodite duct at the base of the seminal receptacle (sr), approaching it from *K. tupan* and *K. jimenezi*; but this budging region is narrow and long, a distinction. *K. uhlei* has the carrefour duct inserting between the albumen duct (ad) and the albumen chamber (ac) (Fig 38A), a condition only shared with *K. nigra* and *K. tupan*. Its albumen chamber (Fig 38A: ac) forms a wide curve, contrasting with the blind sacs found in *K. nigra*, *K. rupestris*, *K. tupan*, and *K. aetheria*. In having a single sperm fold in the spermoviduct (Fig 35E: sp), *K. uhlei* differs from *K. nigra*, which has two. Additionally, it has the relatively wide prostate band in the spermoviduct (~45%), only *K. rupestris* and *K. jimenezi* have wider prostates, while *K. corallina*, *K. nigra* and *K. aetheria* have narrower prostates. The muscular anterior portion of the bursa copulatrix duct (Fig 38B: bu) is well-defined,

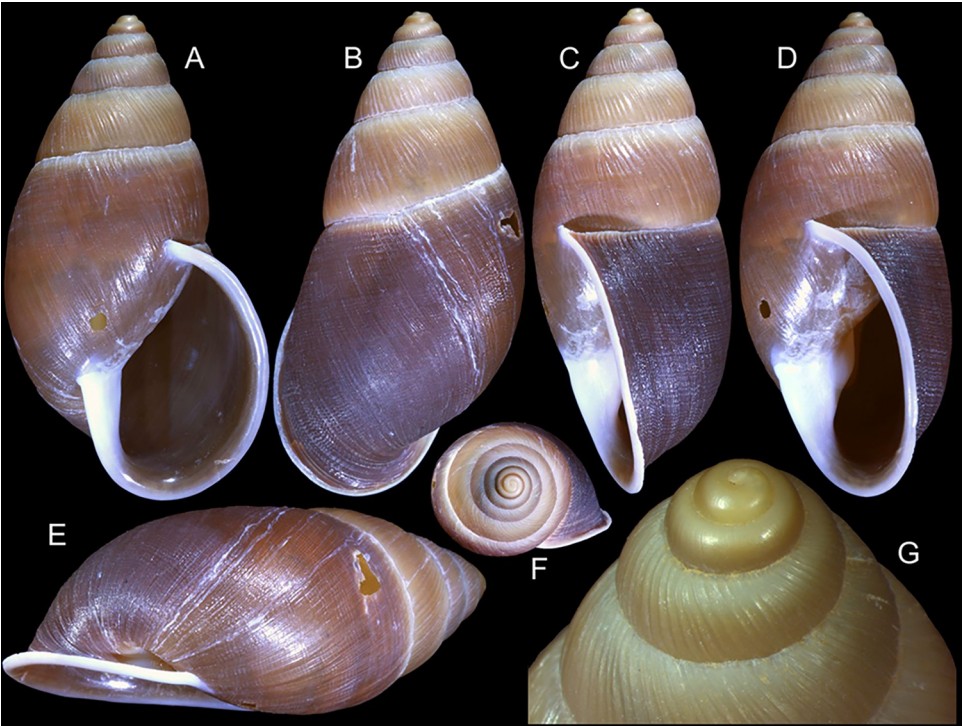

**Fig 39. *Kora kremerorum* holotype MZSP 151809 shell characters.** (A) frontal view (L 40.8 mm). (B) dorsal view. (C) right view. (D) right-slightly ventral view. (E) left-slightly anterior view showing umbilicus. (F) apical view. (G) detail of apex, profile-slightly apical view, scale = 1 mm.

setting *K. uhlei* apart from *K. jimenezi*. Its penis length is approximately 60% of the spermoviduct (Fig 35F: pe), being the shorter proportion among its congeners, a condition only shared with *K. rupestris*. The bursa copulatrix duct is about 80% of the spermoviduct length, which is the shortest condition, also shared with *K. aetheria and K. jimenezi*. The vas deferens of *K. uhlei* has the strong curve preceding its insertion at the tip of the penis, apporaching it from *K. nigra*, *K. tupan* and *K. ajar*. Its penis base has clear muscular walls (Fig 38C: mp), unlike *K. jimenezi*, which lacks them. The penis of *K. uhlei* features the usual pair of inner folds, but it also has an imbricated arrangement of inner branches (Fig 38C: pf), a characteristic shared only with *K. aetheria* and *K. corallina* among its congeners; but the imbrication is only in the distal portion, as an idiosyncrasy. *K. uhlei* has the umbrella-like transverse penial fold, bearing 3 rods, which is found in most of its congeners, with the exceptions of *K. corallina*, *K. nigra* and *K. jimenezi*. The epiphallus comprises about 25% of the penial length, being longer than *K. corallina*, *K. rupestris* and *K. tupan*, but shorter than that of *K. aetheria*. *K. uhlei* lacks a strong longitudinal fold in the epiphallus (Fig 38C), a distinguishing it from *K. corallina*, *K. tupan* and *K. ajar*. Its penis muscle (pm) inserts in the base of the epiphallus, similarly to *K. rupestris*, *K. ajar*, and *K. uhlei*, and distinct from remaining congeners, which have apical insertions.

*Kora kremerorum* new species Figs 39, 40

**ZooBank.**   urn:lsid:zoobank.org:act:D9F0CAAC-68E9-40AD-8440-7C62765AEF82.

**Types.**   Holotype MZSP 151809; paratypes: MZSP 151810, 19 shells, MNRJ, 1 shell, USNM, 1 shell, all from type locality.

**Type locality.**   BRAZIL. **Minas Gerais**; São João da Ponte, near Olímpio Campos, 15˚50'44"S 44˚00'03"W, altitude 759 m (W. Vailant-Mattos col., i.2020).

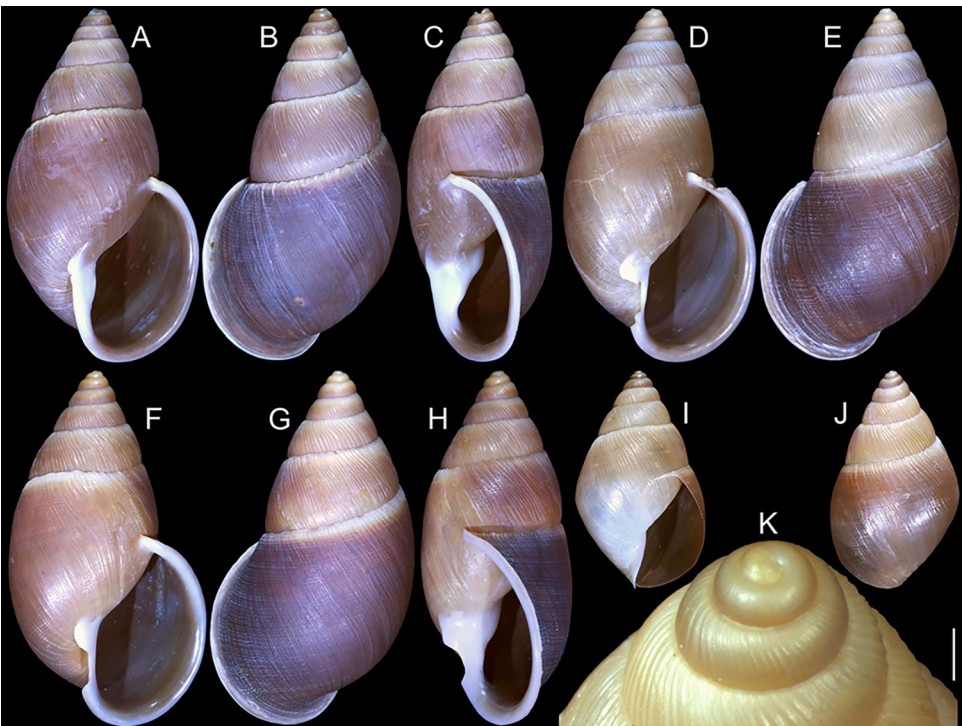

**Fig 40. *Kora kremerorum* paratypes MZSP 151810 shell characters.** (A–C) #1 (L 48.6 mm), frontal, dorsal and right views. (D–E) #2 (L 41.9 mm), frontal and dorsal views. (F–H) #3 (L 43.5 mm). (I–J) #4 young specimen (L 30.9 mm), frontal and dorsal views. (K) same, detail of apex, profile-slightly apical view, scale = 1 mm.

**Diagnosis.** Size about 45 mm, ~1.9 times longer than wide; dorso-ventral weakly compressed. Apex with same color as remaining shell. Subsutural lighter band present. Peristome white. Delicate spiral striae present. Aperture occupying ~49% of length and ~70% width. Implantation of outer lip slightly horizontal. Inner lip with high middle fold. Umbilicus wide.

**Description.** *Shell.* Length up to 44 mm, outline fusiform-globose, 1.9x longer than wide. Color brown, lighter in spire, gradually becoming dark brown in last whorl (Figs 39B and 40B, 40E, 40G); subsutural pale band in all whorls well-developed (Figs 39C, 39D and 40A, 40E, 40G, 40J). Protoconch (Figs 39G and 40I, 40K) with 2.2 whorls, bluntly pointed; length ~4% of shell length, and ~13% of shell width (Fig 39F); first whorl smooth, second whorl sculptured by spaced axial riblets. Limit between protoconch and teleoconch weakly visible, weakly prosocline. Teleoconch of ~4.2 whorls successively and uniformly increasing; whorls weakly concave; suture weakly deep. Sculpture well-developed, uniform, delicate axial undulations, forming axial riblets, ~60 in penultimate whorl; also weak, uniform spiral striae, clearer in last whorl, 45–50 in last whorl, each stria composed of minute aligned pits, separated from each other by equivalent distance of their width. Dorso-ventrally slightly flattened (Fig 39E). Peristome weakly dislocated to right; deflected, except for region of callus. Callus weak (Figs 39A, 39D, 40A, 40C, 40D, 40F, 40H). Aperture wide, somewhat dislocated from spire longitudinal axis; length ~49% of shell length, ~70% of shell width. Outer lip inserted distantly from adjacent suture, simple, arched. Inner lip concave, superior half weakly convex, mostly showing outer surface of last whorl; inferior half almost straight (Figs 39A and 40D, 40F) to weakly convex, concave only inferiorly (Fig 40A); bearing oblique fold in limit with superior half, having weak elevation preceding its end in inner lip (Figs 39D and 40C, 40H); tooth length ~31% of

peristome length. Umbilicus opened, narrow, partially covered by inferior half of inner lip (Fig 39E).

**Distribution.**   Known only from the for region of the type locality.

**Habitat.**   Under rocks or rock crevices.

**Etymology.**   The specific epithet is in honor of Lee and Jan Kremer, shell collectors, expedition sponsors and enthusiastic of the malacology.

**Measurements.**   (in mm) MZSP 151809 (holotype, Fig 39A–39E): 40.8 by 21.8; paratypes MZSP 151810 #1 (Fig 40A–40C): 48.6 by 26.7; #2 (Fig 40D, 40E): 41.9 by 21.2; #3 (Fig 40F–40H): 43.5 by 23.6.

**Material examined.**   The types.

**Taxonomic remarks.**   *Shell. Kora kremerorum* has an average shell size of approximately 45 mm, making it larger than *K. nigra*, *K. aetheria*, *K. vania*, and *K. curumim*, but smaller than *K. tupan* and *K. ajar*. Its shell is about 1.9 times longer than it is wide, giving it an elongated shape compared to most other congeneric species; it is, however, wider than *K. corallina and K. rupestris*, but narrower than *K. nigra*, *K. ajar* and *K. curumim*. The shell aperture comprises about 49% of the total shell length, which is much wider than those of *K. corallina*, *k. aetheria*, *K. jimenezi* and *K. vania*, but narrower than those of *K. tupan and K. ajar*. Additionally, like *K. nigra*, *K. tupan*, *K. ajar*, *K. jimenezi*, and *K. vania*, *K. kremerorum* has a horizontally oriented superior implantation of the outer lip. Additionally, the species can be easily distinguished by the well-marked axial sculpture in the spire, which is the more developed among the species studied in this paper. This strong axial sculpture is only comparable to that of *K. arnaldoi*, from which it distinguishes by more inflated shell and the deeper suture. The shell is also dorso-ventrally slightly flattened (Fig 39F), a condition only shared with *K. tupan* and *K. aetheria*, and absent in remaining congeners.

***Kora vania* new species Fig 41.**

**ZooBank.**   urn:lsid:zoobank.org:act:F49EC154-BBD7-4FEF-A84F-7CED97C2E780.

**Types.**   Holotype MZSP 165500. Paratypes: MZSP 163776, 2 shells, MZSP 164899, 8 shells, from type locality.

**Type locality.**   BRAZIL. **Minas Gerais**; Montalvânia, E of, 14°25'25"S 44°21'43"W (W. Vailant-Mattos col., vi.2023).

**Diagnosis.**   Size about 30 mm, ~1.9 times longer than wide; lacking dorso-ventral compression. Apex with same color as remaining shell. Subsutural lighter band present. Peristome white. Delicate spiral striae scanty. Aperture occupying ~46% of length and ~65% width. Implantation of outer lip slightly horizontal. Inner lip with high middle fold. Umbilicus narrow.

**Description.**   *Shell*. Length ~37 mm, outline fusiform-elongate, ~1.9x longer than wide. Color light brown, with darker region half whorl preceding aperture (Fig 41D, 41C); light subsutural band present (Fig 41C, 41E). Protoconch (Fig 41G, 41H) width of 2.25 mm, of ~2 convex whorls, first whorl smooth, axial narrow ribs gradually appearing in second whorl; transition with teleoconch unclear, weakly prosocline; occupying 4.6% of shell length, 15.5% of shell width in holotype. Teleoconch of ~5 whorls successively and uniformly increasing; whorls slightly concave; suture well-marked; spire angle ~45°. Sculpture absent, except for growth lines; surface slightly glossy; spiral striae very weak and sparse, 6–7 in last whorl (Fig 41B, 41J, 41M). Transverse section circular (Fig 41F). Peristome slightly dislocated to right (Fig 41A, 41I, 41L); slightly prosocline, ~15° in relation to longitudinal shell axis (Fig 41C, 41K). Callus thin (Fig 41C, 41D, 41K). Aperture wide; length ~46% of shell length, ~65% of shell width. Outer lip inserted very distantly from adjacent suture, in inferior slope; simple, arched. Inner lip concave, superior half weakly convex, constituted by weak callus; inferior half also convex due to strong middle fold (Fig 41A, 41C, 41D, 41I, 41L); bearing oblique, low

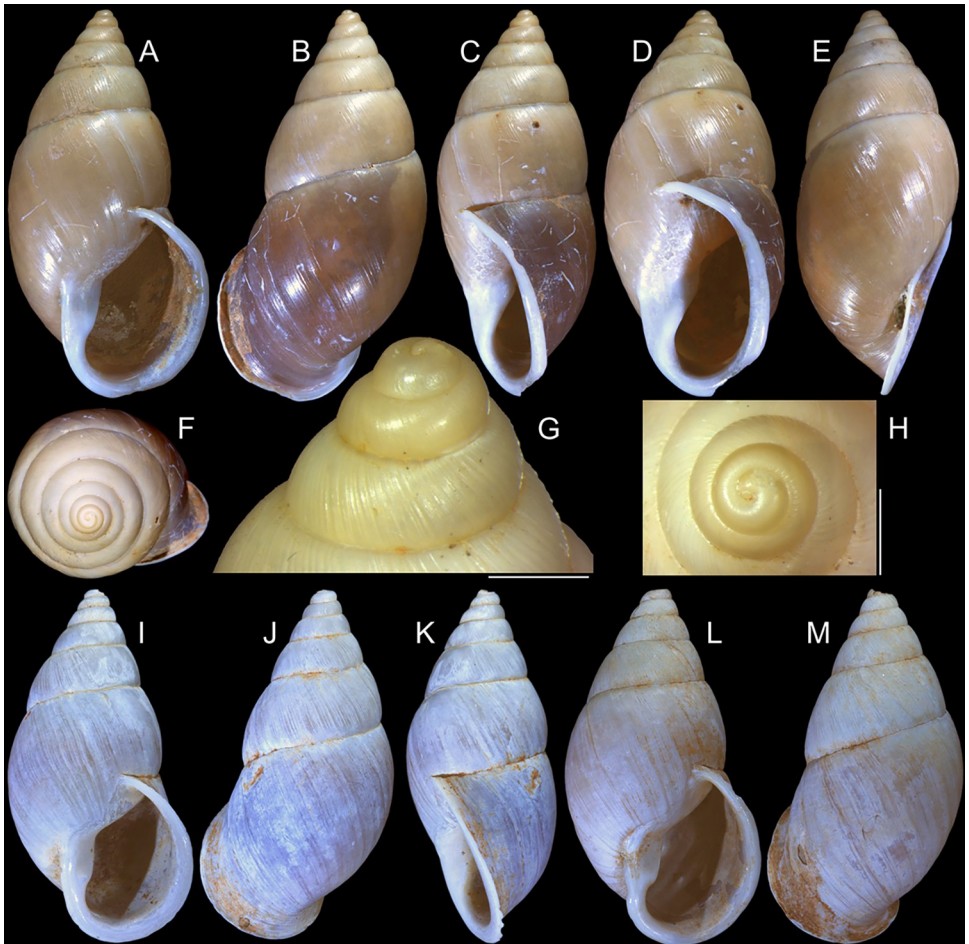

**Fig 41. *Kora vania* types.** (A–H) holotype MZSP 165500 (L 38.0 mm). (A), frontal view. (B) dorsal view. (C) right view. (D) right-slightly ventral view. (E) left-slightly anterior view. (F) apical view. (G) apex in profile, scale = 2 mm. (H) same, apical view. (I–K) paratype MZSP 163776#1 (L 34.7 mm), frontal, dorsal and right views. (L–M) paratype MZSP 163776#2 (L 38.8 mm), frontal and dorsal views.

fold in limit with superior half (Fig 41D); tooth length ~40% of peristome length. Umbilicus opened, covered by inferior half of inner lip (Fig 41E).

## Distribution

Known only from the for region of the type locality.

**Habitat.** Cerrado region.

**Etymology.** The specific epithet is in apposition, and is a Latinization of the final part of the locality of occurrence, the city of Montalvânia.

**Measurements.** Holotype MZSP 165500 (Fig 41A–41H): 38.0 by 20.3; paratype MZSP 163776#1 (Fig 41I–41K): 34.7 by 17.8; #2 (Fig 41L–41M): 38.8 by 20.8.

**Material examined.** The types.

**Taxonomic remarks.** *Shell. Kora vania* has an average shell size of approximately 30 mm, in the smaller category for the genus; this condition is only shared with *K. nigra*, *K. aetheria* and *K. curumim*. Its shell is about 1.9 times longer than it is wide, giving it an elongated shape compared to most other congeneric species, it is, however, wider than *K. corallina* and *K.*

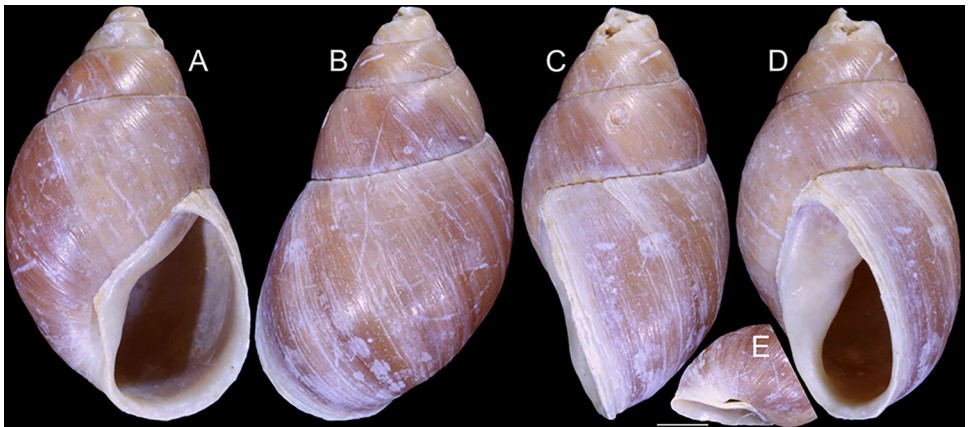

**Fig 42. *Kora curumim* holotype MZSP 152077 shell characters.** (A) frontal view (L 27.7 mm). (B) dorsal view. (C) right view. (D) right-slightly ventral view. (E) detail of anterior region, left-slightly ventral view, scale = 5 mm.

*rupestris*, but narrower than *K. nigra*, *K. ajar* and *K. curumim*. The shell aperture comprises about 46% of the total shell length, which is much shorter than the apertures of *K. nigra*, *K. rupestris*, *K. tupan*, *K. ajar*, *K. uhlei*, *K. kremerorum*, and *K. curumim*; but it is wider than that of *K. corallina*. Additionally, like *K. nigra*, *K. tupan*, *K. ajar*, *K. jimenezi*, and *K. kremerorum*, *K. vania* has a horizontally oriented superior implantation of the outer lip. Additionally, *K. vania* has the proportional widest last whorl, making the shell rather elongated, with a proportionally small spire; also, its surface is relatively shining in freshly collected specimens (Fig 41A–41F); and its first whorls have deeper suture (Fig 41G). These are distinctions that help in the individualization of the species.

*Kora curumim* new species Fig 42.

**ZooBank.** urn:lsid:zoobank.org:act:CD86562F-7B7D-4149-AE23-E420378975EB.

**Types.** Holotype MZSP 152077.

**Type locality.** BRAZIL. **Minas Gerais**; Inaí, Pedra da Fartura, 16˚31'37"S 46˚49'18"W (W. Vailant-Mattos col., iv.2020).

**Diagnosis.** Size about 30 mm, ~1.7 times longer than wide; lacking dorso-ventral compression. Apex with same color as remaining shell. Subsutural lighter band absent. Peristome white. Delicate spiral striae absent. Aperture occupying ~50% of length and ~65% width. Implantation of outer lip slightly vertical. Inner lip lacking high middle fold. Umbilicus almost closed. Umbilicus wide.

## Description

*Shell.* Length ~28 mm, outline fusiform-globose, 1.7x longer than wide. Color light brown, intercalating spiral wide darker and lighter bands along whorls (Fig 42A–42C). Protoconch not seen. Teleoconch of ~4.5 whorls successively and uniformly increasing; whorls weakly concave; suture weakly deep, canaliculated. Sculpture absent, except for growth lines; surface slightly glossy. Transverse section circular. Peristome not dislocated. Callus relatively thick (Fig 42A, 42D). Aperture wide, not dislocated from spire longitudinal axis; length ~50% of shell length, ~65% of shell width. Outer lip inserted distantly from adjacent suture, simple, arched. Inner lip concave, superior half weakly convex, constituted by callus; inferior half almost straight (Fig 42A, 42D); bearing oblique, low fold in limit with superior half (Fig 42D); tooth length ~31% of peristome length. Umbilicus tightly opened, very narrow, covered by inferior half of inner lip (Fig 42E).

**Distribution.** Known only from the for region of the type locality.

**Habitat.** Cerrado region.

**Etymology.** The specific epithet is in apposition, and is derived from Tupy word *curumim*, meaning child, small, an allusion to the small size of the specimen.

**Measurements.** (in mm) MZSP 152077 (holotype, Fig 42): 27.7 by 16.1.

**Material examined.** The type.

**Taxonomic remarks.** *Shell. Kora curumim* has a shell length smaller than 28 mm, which makes it the smaller species in the genus. Its shell is about 1.7 times longer than it is wide, giving it an elongated shape compared to most other congeneric species, it is, however, wider than *K. corallina*, *K. rupestris*, *K. tupan*, *K. aetheria*, *K. jimenezi*, *K. uhlei*, *K kremerorum* and *K. vania*; only *K nigra* is wider that it. The shell aperture comprises about 50% of the total shell length, which is shorter than the apertures of *K. tupan* and *K. ajar*; but it is wider than that of *K. corallina*, *K. nigra*, *K. aetheria*, *K. jimenezi* and *K. vania*. Additionally, inlike *K. nigra*, *K. tupan*, *K. ajar*, *K. jimenezi*, and *K. kremerorum*, *K. curumim* lacks a horizontally oriented superior implantation of the outer lip. Additionally, *K. curumim* has the narrower umbilicus, being it almost closed, which makes easy to recognize the species.

## Genus *Koltrora* new genus

**ZooBank.** urn:lsid:zoobank.org:act:ABA4B542-3282-4916-B2B0-132ADEBDE415.

**Diagnosis.** Shell thin, translucent. Protoconch with 2 whorls, smooth, with weak axial riblets in last whorl. Teleoconch sculpture only axial, uniform undulations. Umbilicus open. Peristome deflected, wide, weakly dislocated. Ureter totally closed (tubular). Odontophore pair m8 absent; ventral tensor muscle of radula lost. Odontophore cartilages totally fused with each other. Two ducts to anterior lobe of digestive gland. Accessory albumen chamber absent. Uterus with glandular walls. Epiphallus widely opened to penis, with penis muscle subterminal. Penis with transverse, simple inner fold at middle. Calcified epiphragm present.

**List of included taxa.** Monotypic so far, only *K. pyrostoma* n. sp. the type species.

**Etymology.** The genus name is in apposition, a contraction of Coltro–in honor to the Coltro brothers (José and Marcus, who have contributed considerably to Brazilian Malacology, by collecting and donating material) and *Kora*, the genus in which the new one has some similarity.

**Gender.** Feminine.

**Taxonomic discussion.** See below.

***Koltrora pyrostoma* new species Figs 43–48.**

**ZooBank.** urn:lsid:zoobank.org:act:E6EC25FB-9685-45F4-BA76-560201653CC5.

**Types.** Holotype MZSP 163500, 1 complete spm; paratypes: MZSP 151789, 10 spm, MZSP 151791, 2 shells, MZSP 151790, 19 shells, MNRJ, 1 shell, USNM, 1 shell, all from type locality. BRAZIL. **Bahia**; Bom Jesus da Lapa (W. Vailant-Mattos col.), near Pirâmide Luxor Hotel, 13˚15'35"S 43˚25'12"W, altitude 500–520 m, MZSP 152017, 21 shells (17.iv.2019)

**Type locality.** BRAZIL. **Bahia**; Bom Jesus da Lapa, way to Morrão, 13˚09'45"S 43˚18'33"W, altitude 440–460 m (W. Vailant-Mattos col., 17.iv.2019).

**Diagnosis.** Size about 30 mm, ~1.8 times longer than wide; dorso-ventrally weakly compressed. Apex with same color as remaining shell. Subsutural lighter band absent. Peristome white. Delicate spiral striae absent. Aperture occupying ~50% of length and ~57% width. Implantation of outer lip slightly vertical. Inner lip lacking high middle fold. Umbilicus narrow. Secondary columellar muscles with 4 insertions in left and 2 in right. Absence of pairs of m1v. Jaw rectangular. Odontophore cartilages ~100% fused. Pair m10 broad, absence of m8. Double anterior duct to digestive gland. Carrefour duct narrow and long, inserted in tip of

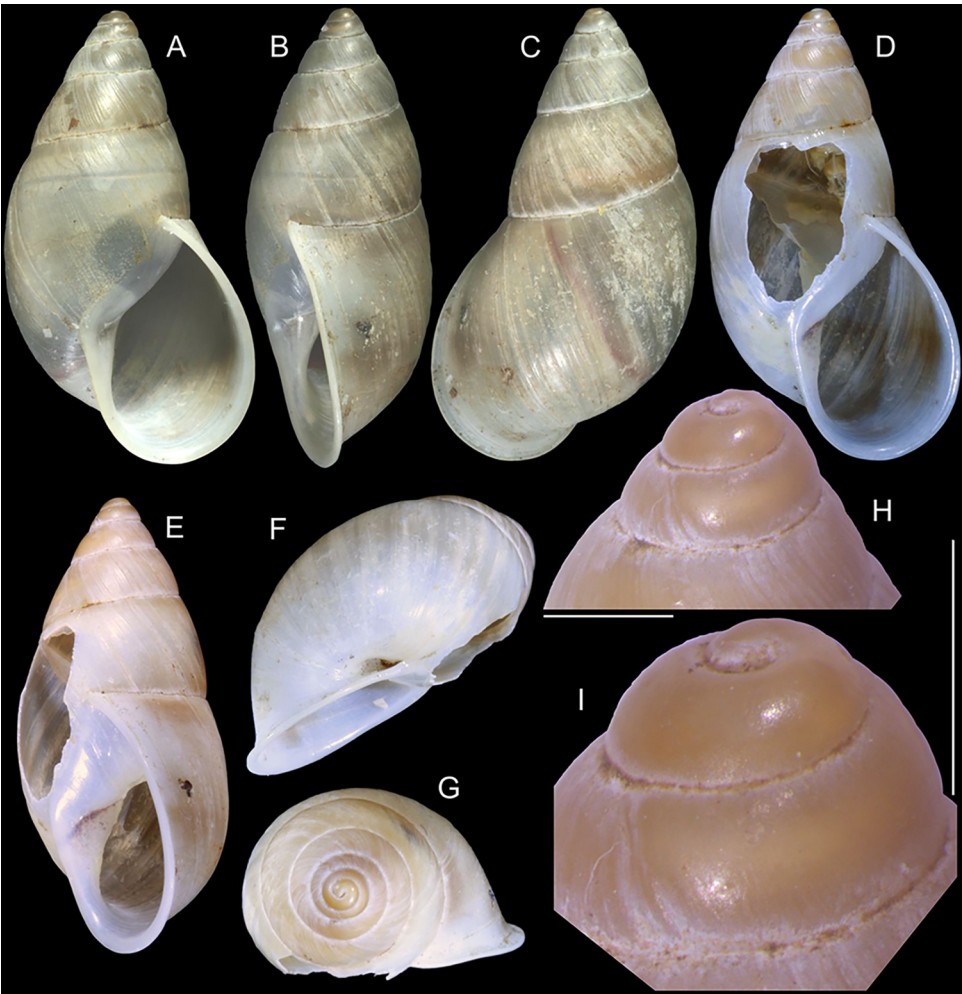

**Fig 43. *Koltrora pyrostoma* holotype MZSP 163500 shell characters.** (A) frontal view (L 29.1 mm), specimen alive retracted inside. (B) same, right view. (C) same, dorsal view. (D) present state after extraction of specimen, frontal view. (E) right-slightly ventral view. (F) anterior-left view, showing umbilicus. (G) apical view. (H) apex, profile. (I) protoconch, profile. Scales = 1 mm.

spermoviduct. Albumen chamber sac-like. Penis ~50% of spermoviduct length, lacking umbrella-like fold; epiphallus ~25% of penis length; penis muscle subterminal in epiphallus.

**Description (distinctive in anatomy).** *Shell.* (Figs 43 and 44A–44E and 45A–45D, 45G, 45H) Length up to 32 mm, outline fusiform-elongate, ~1.8 longer than wide. Color uniform pale beige to cream, walls translucent. Protoconch (Figs 43H, 43I and 44D, 44E and 45C) with 2 whorls, bluntly pointed; length ~7% of shell length, and ~6.5% of shell width (Fig 43G); mostly smooth, barely sculptured by axial riblets in last whorl. Limit between protoconch and teleoconch weakly visible, weakly prosocline. Teleoconch of ~4 whorls successively and uniformly increasing; whorls weakly concave; suture weakly deep; sculpture absent, except for growth lines and delicate axial, uniform undulations, ~60 in penultimate whorl. Dorso-ventrally softly flattened (Fig 43F, 43G). Peristome weakly dislocated to right, slightly oblique; deflected. Callus plane, relatively well-developed in adult (Figs 43A, 43D, 43E and 42A), weak in young (Figs 44D and 45A, 45D). Aperture wide, somewhat dislocated from spire longitudinal axis; length ~50% of shell length, ~57% of shell width. Outer lip inserted distantly from

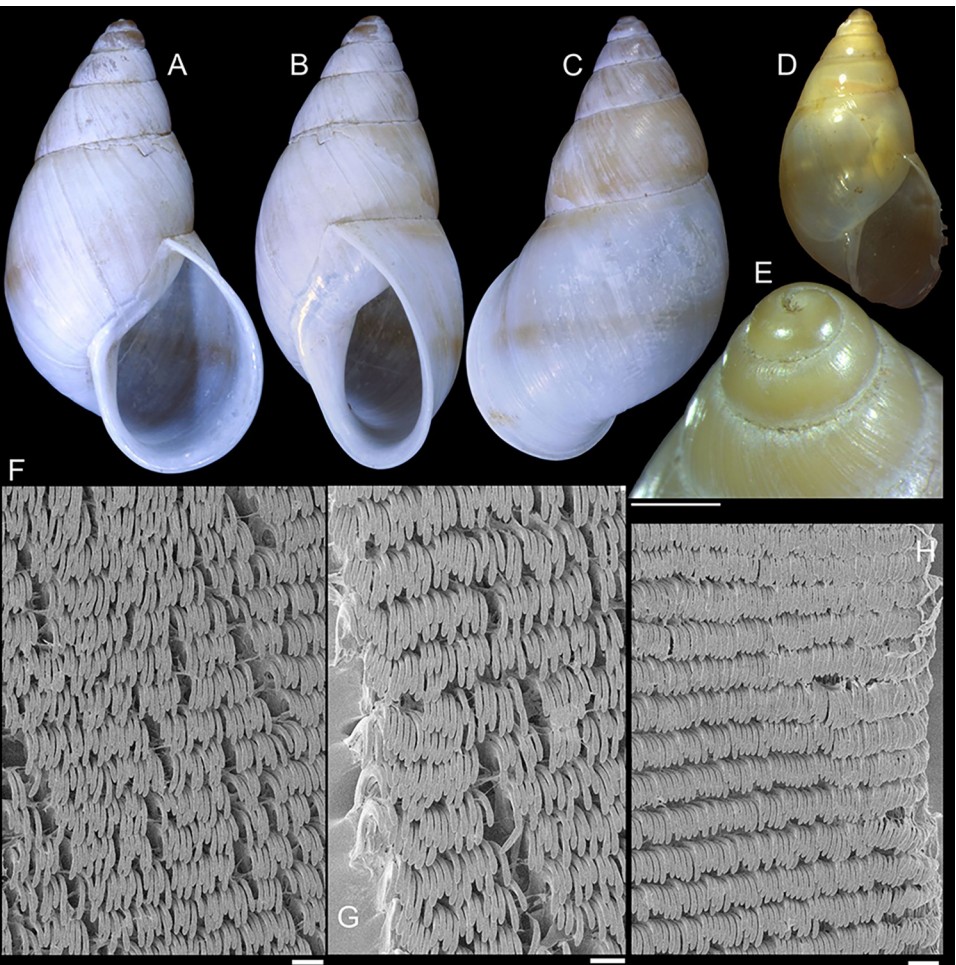

**Fig 44.** *Koltrora pyrostoma* **paratypes shells and radula.** (A–C) MZSP 191791, frontal, right and dorsal views (L 31.9 mm). (D) MZSP 151789, frontal view (L 27.0 mm). (E) same, detail of apex, profile-slightly apical view, scale = 1 mm. (F–G) radula in SEM, central region, scales = 50 μm. (H) same, lateral region, scale = 50 μm.

adjacent suture, simple, arched. Inner lip concave, superior half weakly convex, mostly composed of plane callus; inferior half weakly straight to weakly convex; bearing very low oblique fold in limit with superior half (Figs 43E and 44B). Umbilicus opened, narrow, partially covered by inferior half of inner lip (Fig 43F).

**Epiphragm.** Present in few specimens (Fig 45D) calcified, thin, occluding entire aperture; dislocated posteriorly from peristome.

**Head-foot.** (Figs 44B and 45J) Of normal shape, resembling *K corallina*. Color bluish beige with some red pigment in furrows of mosaic of dorsal foot integument (Fig 45A, 45G: ft, H), becoming uniformly bluish beige in preserved specimens (Fig 46J). Clear oblique furrow (Fig 45J: fe) running at right in integument from pneumostome (pn) up to very anteriorized genital pore (Fig 45E: fp). Columellar muscle thick, 1.2 whorls in length. Main columellar bundle (cm) ~3/4 of foot width. Each secondary columellar/cephalic muscles with ~1/3 of main columellar bundle (cm) width. Right cephalic muscle (Fig 46B: cr) with 2 broad insertions, being medial insertion broader, with also broad tentacular, ommatophore muscles and small genital muscles as more lateral branches; left cephalic muscle (cl) similarly organized, but with

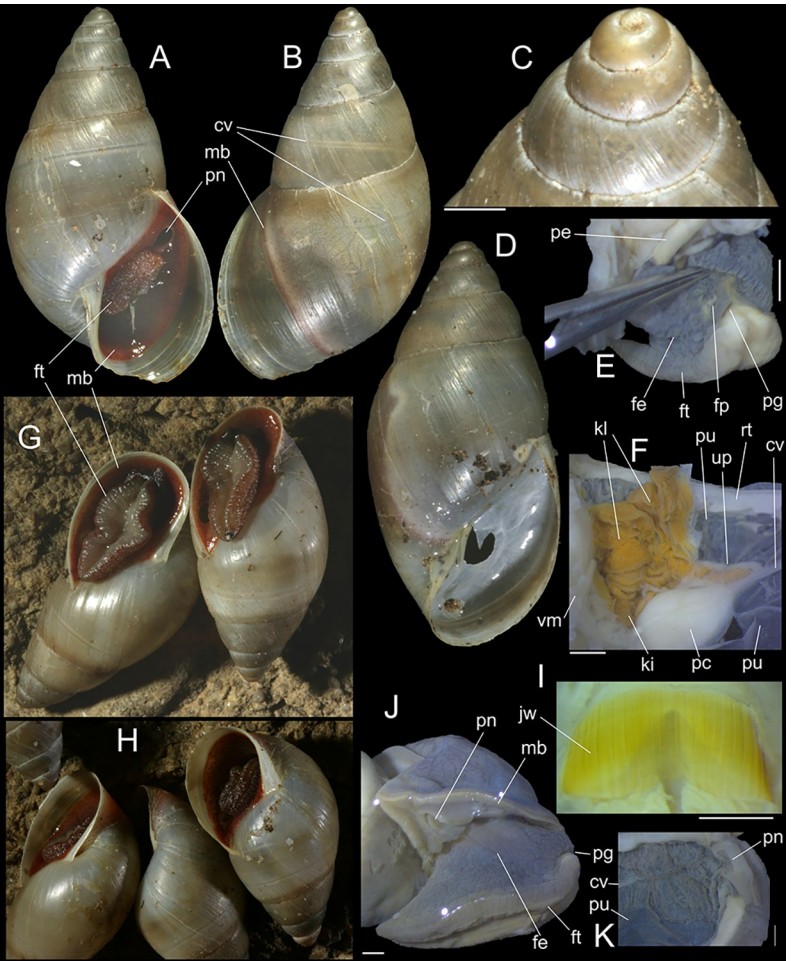

**Fig 45.** *Koltrora pyrostoma* **anatomical features in light photos of paratypes MZSP 151789.** (A–B) alive retracted specimen #7, frontal and dorsal views (L 26.3 mm). (C) same, detail of apex, profile-slightly apical view. (D) specimen #6, frontal view, with epiphragm (L 24.2 mm). (E) head-foot anterior view, forceps deflecting integument upwards to show genital aperture (fp). (F) reno-pericardial region, ventral view, ventral wall of kidney cut along left edge and deflected upwards. (G–H) alive semi-retracted specimens showing red mantle edge (average L 36 mm). (J) extracted specimen, anterior region, right view. (I) jaw in situ, ventral view. (K) pulmonary cavity, anterior-right region, ventral edge of pneumostome sectioned and deflected upwards. Scales = 1 mm.

4 slightly narrow aligned insertions. Pedal gland (pg) short, weakly protruding in posterior region of buccal area (mo).

## Mantle organs

(Figs 45J, 45K and 46A) With similarities to *K. corallina*, distinctions and remarks following. Mantle edge (mb) thick, strongly red pigmented (Fig 45A, 45G: mb, 45H); seen through shell translucency (Figs 43C and 45B, 45D); red pigment disappearing in preserved specimens (Fig 45E, 45J, 45K). Right pallial fold of mantle edge well-developed (Fig 46A: mf), with projected, bluntly pointed left end. Pneumostome (Fig 46A: pn) bearing exclusively air entrance and urinary aperture (ua); anus (an) as separate aperture located at right, adjacent to pneumostome. Lung of ~1.5 whorls in length, ~twice long than wide; scarcely possessing minute longitudinal muscle fibers in its wall seen by translucency, more concentrated in region near pulmonary vein. Pulmonary venation well-developed, especially in region preceding pneumostome;

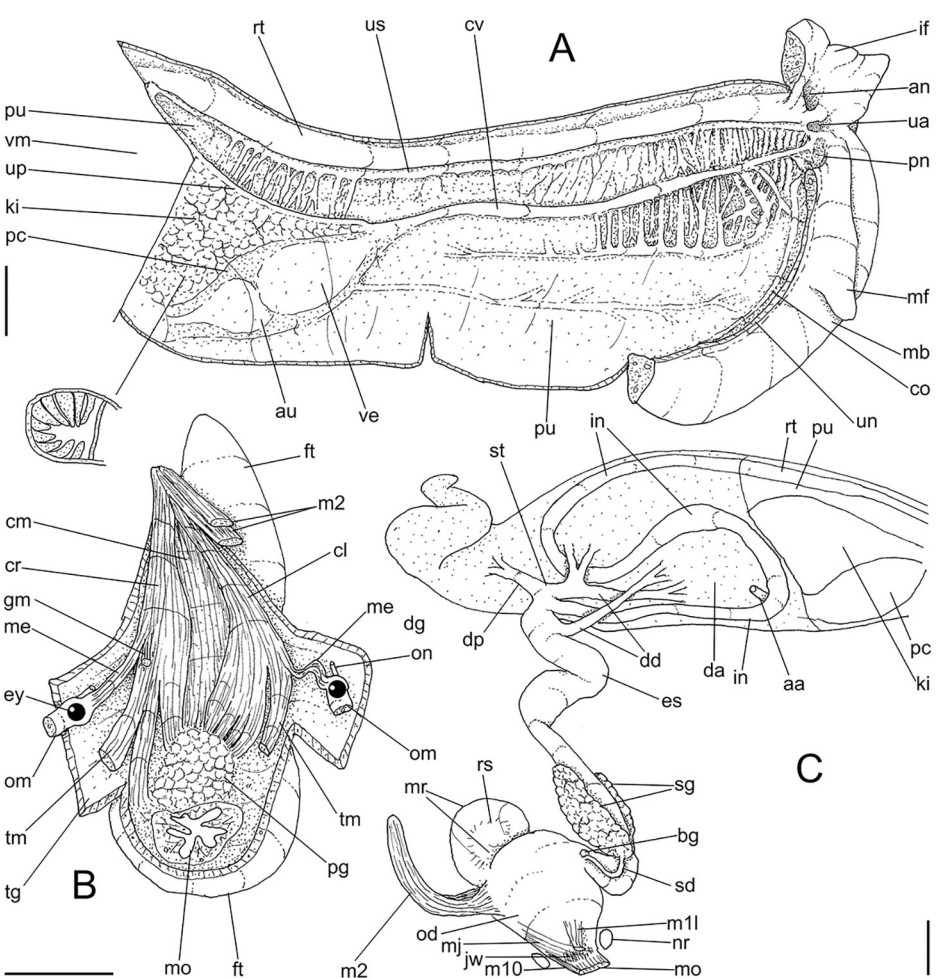

**Fig 46.** *Koltrora pyrostoma* **anatomical drawings.** (A) extended pallial (pulmonary) cavity, ventral-inner view, inner edge of pneumostome sectioned and deflected upwards, transverse section of indicated region of kidney also shown. (B) head-foot, dorsal view, head, dorsal integument and internal organs removed, remaining muscles expanded. (C) foregut and midgut, mostly ventral view as in situ, topology of some adjacent structures also shown. Scales = 2 mm.

posterior region of pulmonary vein (cv) protruded, relatively straight; left 2/3 with only pair of narrow intercalated longitudinal vessels; right 1/3 mostly having perpendicular vessels rather uniformly distributed, weak in middle, becoming taller in region adjacent to kidney and anteriorly; pulmonary vessel bifurcating very close to pneumostome (Figs 45K and, 46A: cv), with other subterminal branches overlapping right vessels producing strange anastomoses (Figs 45K and 46A). Collar vessel (co) also well-developed, running slightly away from mantle edge. Reno-pericardial area of light brown color, slightly triangular with slender anterior prolongation (Figs 45F and 46A: ki), occupying ~35% of cavity length and ~65% of its width (details below). Rectum (rt) wide. Primary (Figs 45F and 46A: up) and secondary (us) ureters entirely closed (tubular), relatively narrow, aperture (ua) simple, directly outside at right in pneumostome region.

## Visceral mass

(Fig 46C) With ~3 whorls in length, with similar attributes as *K corallina*. Except for stomach smaller and slightly more anterior localized.

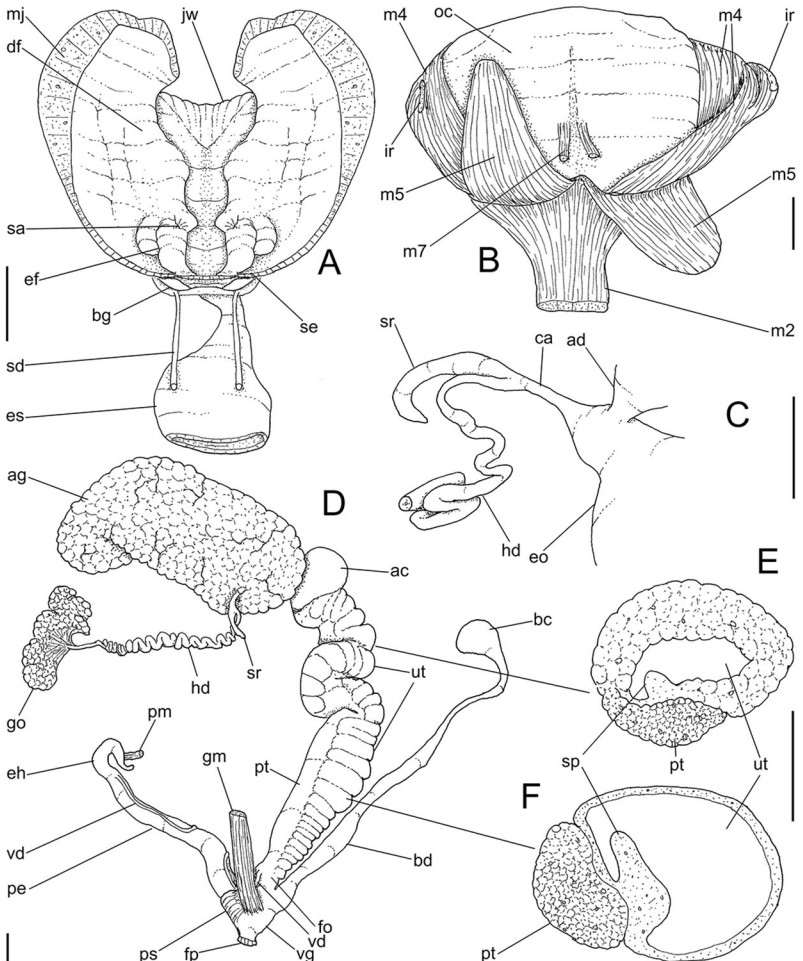

**Fig 47.** *Koltrora pyrostoma* **anatomical drawings.** (A) buccal mass, isolated dorsal wall, ventral view, with adjacent portion of esophagus. (B) odontophore, dorsal view, superficial layer of muscles and membranes removed, both cartilages deflected, left muscles as in situ, right muscles deflected outside. (C) genital structures in albumen gland level if it was transparent, ventral view. (D) genital structures, dorsal view, mostly uncoiled. (E–F) spermoviduct, transverse sections of indicated regions. Scales = 1 mm.

## Circulatory and excretory systems

(Figs 45F and 46A) General Bauplan similar to *K. corallina*, with following remarks. Pericardium (pc) slightly broader. Kidney (ki) size reported above; slightly triangular, as long as wide, with anterior slender sharp projection. Nephrostome at anterior tip of this projection, inside primary ureter beginning. Internally organized as successive tall glandular folds (Fig 45F: kl), of relative similar height.

## Digestive system

(Fig 44C) General organization resembling that of *K. corallina*, distinctions and remarks following. Jaw plate (Fig 45I) thin, yellow, translucent, ~1.5x broader than long; cutting edge softly convex, slightly notched at middle; sculptured by successive, rather uniform, transverse, wide folds. Buccal mass with radular sac large, bulging ~1/3 of its posterior side, sheltering coiled radular sac (rs), covered by transparent membrane (mr). Dorsal surface of oral cavity with broad and low pair of dorsal folds (Fig 47A: df), width of each almost 1/2 of dorsal wall

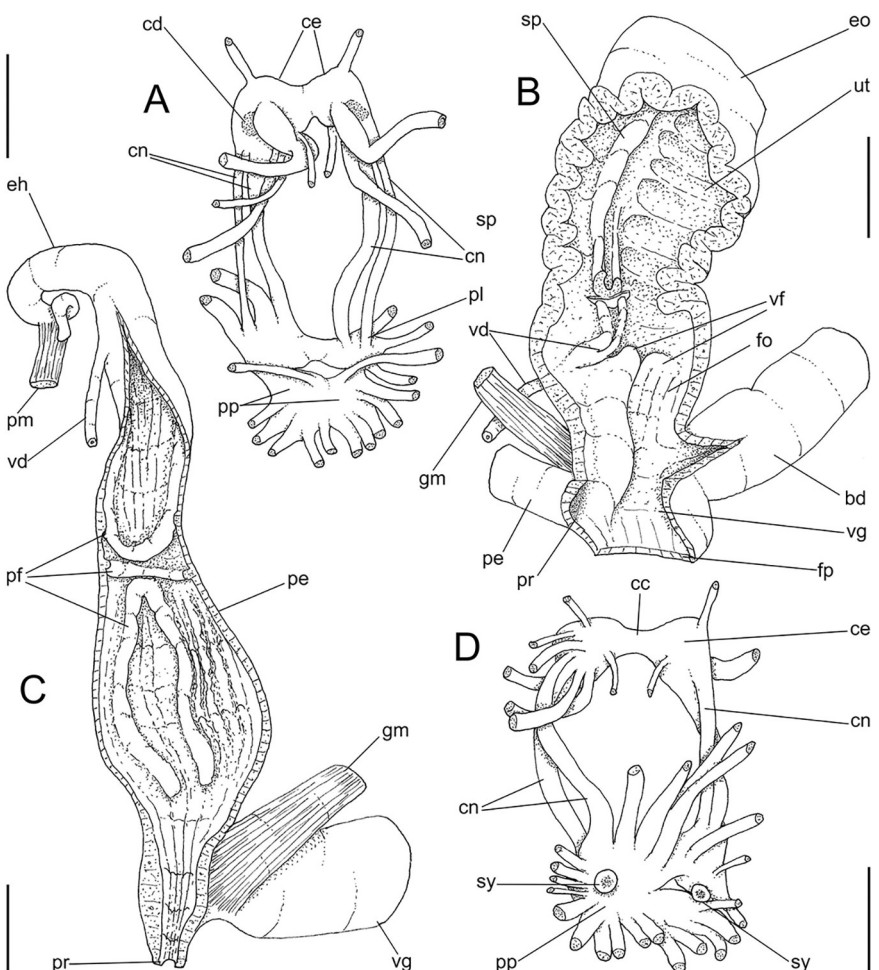

**Fig 48. *Koltrora pyrostoma* anatomical drawings.** (A) central nervous system (nerve ring), dorsal view. (B) basal end of spermoviduct, dorsal view, mostly opened longitudinally, transverse section in subterminal region of sperm inner longitudinal fold (sp) artificially done, some adjacent structures also shown. (C) penis, ventral view, longitudinally opened, some adjacent structures also shown. (D) nerve ring, ventral view. Scales = 1 mm.

width; not touching with each other in median line, keeping shallow dorsal chamber. Odontophore (Fig 46C: od) with ~50% of buccal mass volume. Odontophore muscles (Fig 47B) with overall features of *K., corallina*, with following remarks: **m1v**, absent; **m1l**, relatively wide pair of dorso-lateral protractor muscles, originating in lateral region of mouth, running short distance towards anterior, inserting in latero-dorsal region of buccal mass surface (Fig 46C); **m3**, not detected; **m4**, with separated branch connected to radular sac and m7a (Fig 47B: ir); **m5**, originating only median portion on postero-ventral region of odontophore cartilages, ~80% on m4; **m6**, absent; **m7**, narrow and slender, originated as 2 branches in posterior region of fusion between both cartilages, run inside dorsal region od radular sac; **m8**, absent; **m10**, broad. Pair of odontophore cartilages almost entirely fused with each other in their anterior-medial edge (Fig 47B: oc). Radular sac (Fig 46C: rs) long, performing loop inside translucent membrane (mr) bulded posteriorly from odontophore. **Radula.** (Fig 44F–44H) ~2.5 times longer than odontophore. Composed of uniform, similar kind of tooth, with no clear differentiation among rachidian, lateral or marginal teeth, ~100 pairs of teeth per row. Each tooth with elongated base, 6–7 times longer than wide, placed longitudinally, very close to each other;

proximal end rounded; distal end bearing long, curved cusp, slightly longer than base; cusp slightly flattened, base reinforced by central fold gradually tapering up to ~70% of cusp length; cusp tip flattened, slightly broader, rounded-barely spoon-like, possessing shallow subterminal, longitudinal, short furrow. All teeth straight aligned per row, except for 8–10 more marginal teeth, slightly arched aligned (Fig 44G, 44H).

Salivary glands covering ~1/20 of esophagus length, located between anterior and second quarter of esophageal length (Fig 46C: sg), forming two elliptic, white, thin masses. Each salivary duct differentiable in anterior side of glands (sd). Salivary duct running in both sides of esophageal origin, penetrating buccal mass wall in region close to buccal ganglia (Fig 47A: bg), running immersed in buccal dorsal wall along ~1/6 its length (Fig 47A). Salivary ducts opening as small pores (sa), located medially in posterior region of oral cavity, at anterior tip of pair of short longitudinal, broad folds (ef). Esophagus (Fig 46C: es) entirely narrow, simple; inner surface with narrow, separated longitudinal folds. Esophageal duct to digestive gland (Fig 46C: dd inferior) long, located at short distance from stomach, connected to anterior lobe of digestive gland. Stomach (st) small, narrow, curved, not bulging; position and size described above (visceral mass); gastric walls thin, not muscular; inner surface mostly smooth, lacking folds. Duct to anterior lobe of digestive gland at short distance from intestine intersection (dd superior) narrow; strongly bifid, right branch running perpendicularly to right, left branch long, running towards anterior covered ventrally by esophageal duct. Duct to posterior lobe of digestive gland located short distance from intestinal origin, directed towards opposite side (dp), as wide as other ducts, bifurcating only after long distance. Intestine (in) entirely narrow, performing its usual wide sigmoid loop in anterior lobe of digestive gland. Rectum and anus position described above (pallial cavity) (rt, an). Anus sessile, as slit in right end of mantle edge directly turned outside (Fig 46A: an).

**Reproductive system.** (Fig 47C–47F and 48B, C) General structures similar to preceding species, remarks and distinctions following. Gonad composed of 5–6 lobes with minute digitiform acini. Hermaphroditic duct (Fig 47D: hd) narrow; coiled portions occupying middle 2/3, with narrow coils; insertion preceded by straight region, and strongly curve (Fig 47C: hd). Seminal receptacle (Fig 47C, 47D: sr) small, curved, long, tip pointed, ~6 times longer than wide, flattened. Fertilization complex or carrefour (Fig 47C: ca) simple, as narrow, duct-like region in receptacle base, ~1/2 of its length; totally immersed in albumen gland, inserting in posterior end of spermoviduct, at side of tip of wide albumen gland duct. Albumen gland (Fig 47D: ag) ~6 times larger than gonad (~1/3 whorl). Albumen gland duct subterminal, conic, connected laterally to distal end of spermoviduct (Fig 47C: ad). Albumen chamber (Fig 47D: ac) as wide sac in spermoviduct initial portion, widely connected to distal end of spermoviduct (eo). Spermoviduct (Fig 47D and 48B: eo) of ~1 whorl in length, slightly narrower than albumen gland, ~10 times longer than wide. Secondary albumen chamber absent. Prostate narrow (pt), ~1/4 of spermoviduct diameter (Fig 47E, 47F); uterus with thick glandular walls posteriorly, gradually becoming thinner glandular walled anteriorly; broadly, transversally, relatively uniformly folded (Figs 47D, 48B: ut). Sperm inner longitudinal fold as simple, low, very thick fold (Fig 47E, 47F: sp); a second small fold gradually appearing only in basal end; both folds fusing with each other, originating vas deferens, slightly anterior to end of uterine level (Fig 48B: vd). Vas deferens uniformly narrow, uncoiled (Figs 47D and 48B: vd). Genital muscle in intersection vagina and penis (Figs 47D and 48B: gm), wide and long. Bursa copulatrix (bc) and its duct (bd) of usual position, with ~70% of spermoviduct length (Fig 47D); bursa duct weakly muscular (Fig 48B: bd). Free oviduct (fo) and vagina (vg) simple, possessing 2 very wide, low, longitudinal, simple folds (Fig 48B). Penis almost straight, ~50% of spermoviduct length (Fig 47D: pe); epiphallus as continuation of penis, penis muscle inserted subterminally in epiphallus (Figs 47D, 48C: pm), short, simple. Penis shield (Fig 47D: ps) with transverse

muscle fibers, with ~1/10 penis length. Penis wall weakly muscular, except for thick muscular walls in region adjacent to penis shield (Fig 48C). Epiphallus (eh) ~1/4 of penis' length, amply opened to penis; only vas deferens insertion marking its limit (Fig 48C: vd). Epiphallus inner surface with 8–10 small, low, parallel folds. Internal penial arrangement of folds clearly with three regions (Fig 48C): (1) basal 1/4, possessing only 3–4 longitudinal, broad, low, simple folds, correspondent to penial muscular portion; (2–3) remaining ¾ portion, divided at middle by transverse fold, of rounded profile, surrounding entirely penis wall; inferior half having pair of longitudinal folds united with each other in distal end, close to transverse fold, in opposed end both folds finishing abruptly, between both folds almost smooth space, surrounding them 5 secondary longitudinal folds parallel to them; distal half with similar arrangement than inferior half, but with fusion between both folds more ample, and both folds running longitudinally very close from each other, surrounding them 6–7 low, wide, longitudinal secondary folds, all them continuing to epiphallus, with some converging to vas deferens aperture.

**Central nervous system.** (Fig 48A, 48D) Characters of ganglia and statocysts virtually similar to those described for *K. corallina*. Except for cerebral (cc) and pedal commissures slightly longer; and by pleural ganglia (pl) slightly proportionally larger.

**Distribution.** Known only from the for region of Bom Jesus da Lapa, Bahia, Brazil.

**Habitat.** Under rocks, in limestone areas.

**Measurements** (in mm): MZSP 163500 (holotype, Fig 43A–43E): 29.1 by 16.3; MZSP 191791 (Fig 43A): 31.9 by 17.2.

**Material examined.** All types.

**Etymology.** The specific epithet is a junction of the Greek words *pyro*, meaning fire, and Greek *stoma*, meaning mouth, in allusion to the red color of the aperture when the animal is alive (Fig 43A, 43G, 43H), red color easily seen through translucent shell (Fig 45B: mb).

**Taxonomic discussion.** Despite similarities in the radula and shell shape, this taxon cannot be considered part of the genus *Kora*, primarily due to the following distinguishing features. It is relatively small, averaging 30 mm, which makes it smaller than the smallest *Kora* species. Its shell walls are thin and fragile, contrasting with the relatively thick shells of all *Kora* species. Additionally, the shell is completely translucent and colorless, lacking the characteristic brown pigmentation in varying shades typically observed in *Kora*. Its protoconch is entirely smooth (Figs 43H, 43I and 44E), unlike the weak axial ribs commonly found in *Kora* protoconchs' last whorl. Furthermore, because it lacks the wrinkles or reticulated sculpturing seen respectively in the protoconchs of *Bulimulus* (Lea, 1814) and *Rhinus* (Martens, 1860), genera with which it bears superficial resemblance, this taxon cannot be classified under either.

Regarding its anatomical features, *Koltrora* lacks several key traits fundamental for classification as *Kora*. These include the absence of the m3 buccal mass muscles, absence of successive branching in the ducts to the digestive gland, and the absence of m8 pair of odontophore muscles. Additionally, the carrefour lacks connection to the albumen duct or albumen chamber, instead connecting directly to the spermoviduct, following the typical pattern among orthalicoideans. The accessory albumen chamber and the muscular portion at the base of the bursa copulatrix duct are also absent.

Moreover, *Koltrora* exhibits unique features, including a duplicated anterior duct to the digestive gland (Fig 46C: dd), complete fusion of the odontophore cartilages (Fig 47B: oc), the m7 pair of odontophore muscles as two separate strips (Fig 47B), and a distinct arrangement of the inner penis fold, with a transverse fold at the middle level.

Further arguments for its generic separation are presented in the Discussion and Phylogenetic Analysis sections.

.

## Genus *Neopetraeus* Martens, 1885

**Complemented diagnosis.** Shell elongated to obese; with mosaic of spots and bands on light basal color; higher variation of sculptures. Protoconch of ~2 smooth whorls, variating from smooth up to reticulated, from rounded to carinated. Umbilicus usually well-developed, resulted of columellar hollow area, producing inner lip usually with middle region bulged. Ureter 50–30% opened (as groove). Both pairs of dorsal tensor muscles of radula (m4 and m5) as indistinct single mass; odontophore pair m8 absent; ventral tensor muscle of radula present. Accessory albumen chamber present.

*Neopetraeus lobbii* **(Reeve, 1849) Figs 49–53.**
*Bulimus lobbii* Reeve, 1849 [25]: pl 71 (fig 516).

*Neopetraeus lobbii*: Pilsbry, 1897 [26]: 177–178 (pl. 29 fig 24–26) (+ ancient synonymy); Breure & Araujo, 2017 [27]: 82–83 (fig 31F); MolluscaBase, 2023 [6] (fig).

**Types.** Lectotype NHMUK 1975431, 1 shell (Fig 49J–49M) (examined)

**Type locality.** Balsas, Banks of the Amazon near Balsas, Peru [25].

**Distinctive description.** *Shell.* (Fig 49A–49G, 49J–49M) Length up to 44 mm, outline fusiform-elongate, ~2.2 longer than wide. Color uniform white to very light beige as base, plus axial bands randomly variating from dark, middle and light brown, from uniformly (Fig 49J–49M) to randomly (Fig 49A–49D) distributed, bands from suture to suture, in last whorl entering in columellar area and umbilicus (Fig 49A, 49D, 49J), concentrated in region preceding peristome (Fig 49C, 49D, 49K). Protoconch described as having 2.25 whorls with delicate reticulate [26] (only axial riblets detected in damaged specimen–Fig 49F, 49G). Teleoconch of ~5 whorls successively and uniformly increasing; whorls weakly concave; suture weakly deep; sculpture absent, almost smooth and shining, except for growth lines and delicate axial, uniform undulations in region preceding aperture. Transverse section circular (Fig 49F, 49M). Peristome weakly dislocated to right, ample; deflected. Callus weak, almost absent (Fig 49A, 49D, 49J). Aperture wide, somewhat dislocated from spire longitudinal axis; length 41–50% of shell length, 57–70% of shell width. Outer lip inserted distantly from adjacent suture, simply arched. Inner lip concave, superior half weakly convex, mostly composed of exposed last whorl; inferior half weakly straight; bearing very low oblique fold in limit with superior half (Fig 49C, 49D). Umbilicus opened, narrow, partially covered by inferior half of inner lip (Fig 49F). More details in [25, 26].

**Head-foot.** (Fig 50B) Of normal shape, resembling *K corallina*. Distinctions and remarks following. Columellar muscle (cm) and its secondary units (cl, cr) narrower. Right cephalic muscle (Fig 50B: cr) particularly narrow, with 4 slim, aligned, isometric insertions, plus ommatophore and small genital muscles as more lateral branches; left cephalic muscle (cl) similarly organized, but ~3-times wider, with only 3 insertions, 2 of them broad insertions, being middle insertion larger, plus ommatophore muscle as more lateral branch, medial-most insertion very narrow and more posteriorly inserted. Pedal gland (pg) long and narrow, weakly protruding in posterior region of buccal area (mo).

**Mantle organs.** (Figs 50A and 51A) With similarities to *K. corallina*, distinctions and remarks following. Mantle edge (mb) very thick, lacking secondary folds or glands. Pneumostome (pn) bearing exclusively air entrance and urinary groove (ur); anus (an) as separate aperture located at right, adjacent to pneumostome. Lung of ~1.5 whorls in length, ~3x longer than wide; lacking longitudinal muscle fibers. Pulmonary venation well-developed only in region preceding pneumostome; entire pulmonary vein (cv) protruded, relatively broad; left 2/3 lacking longitudinal vessels, bearing only sparse transverse minute vessels; right 1/3 mostly having perpendicular vessels rather uniformly distributed, weak in middle and posterior regions, becoming taller anteriorly; pulmonary vessel bifurcating very close to pneumostome, anterior quarter of pulmonary vessel with relatively symmetric set of oblique, wide, bifurcating vessels.

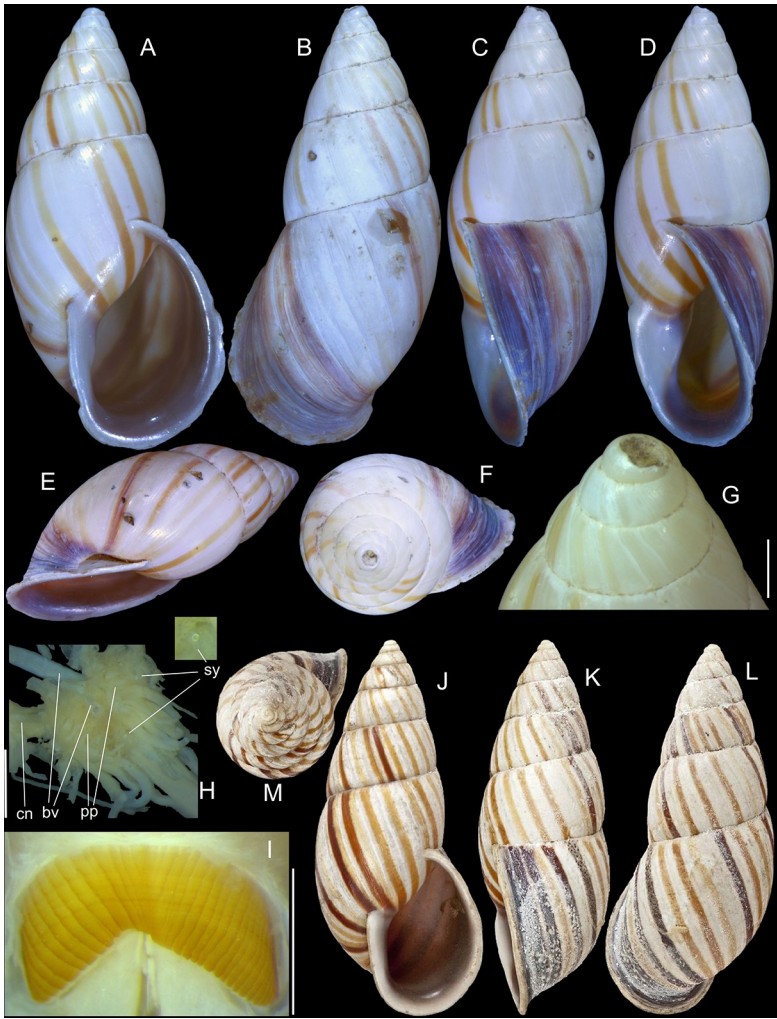

**Fig 49. *Neopetraeus lobbi* shell and anatomical characters, specimen MZSP 158045.** (A) frontal view (L 40.2 mm). (B) dorsal view. (C) right view. (D) right-slightly ventral view. (E) left-anterior view. (F) apical view. (G) apex, profile. (H) nerve ring, detail of pair of pedal ganglia, ventral view, with detail of statocyst enlarged. (I) jaw in situ, ventral view. Scales = 1 mm. (J–M) Lectotype NHMUK 1975431, frontal, right, dorsal and apical views (L 44.4 mm) (copyright © The Trustees of the Natural History Museum, London, published under permission).

Reno-pericardial area of light brown color, slightly triangular (ki), occupying ~20% of cavity length and ~50% of its width (details below). Rectum (rt) wide. Primary (Fig 50A: up) and most of secondary (us) ureters entirely closed (tubular), relatively narrow; ~20% of anterior region of ureter opened (as groove) (ur), running like this up to pneumostome, in urinary furrow flanking perpendicularly inner edge of pneumostome (ur).

**Visceral mass.** (Fig 51A) Of ~3 whorls in length, with similar attributes as *K corallina*. Except for stomach smaller and slightly more anterior localized.

**Circulatory and excretory systems.** (Figs 50A and 51A) General Bauplan similar to *K. corallina*, with following remarks. Pericardium (pc) slightly narrower. Kidney (ki) size reported above; triangular, 1.5x longer than wide. Internally organized as 5 successive tall glandular folds (Fig 50A: ki), of relative similar height.

**Digestive system.** (Figs 50C, 50D, 51A–51C) General organization resembling that of *K. corallina*, distinctions and remarks following. Jaw plate (Fig 49I) thick, yellow, curved, ~1.5x

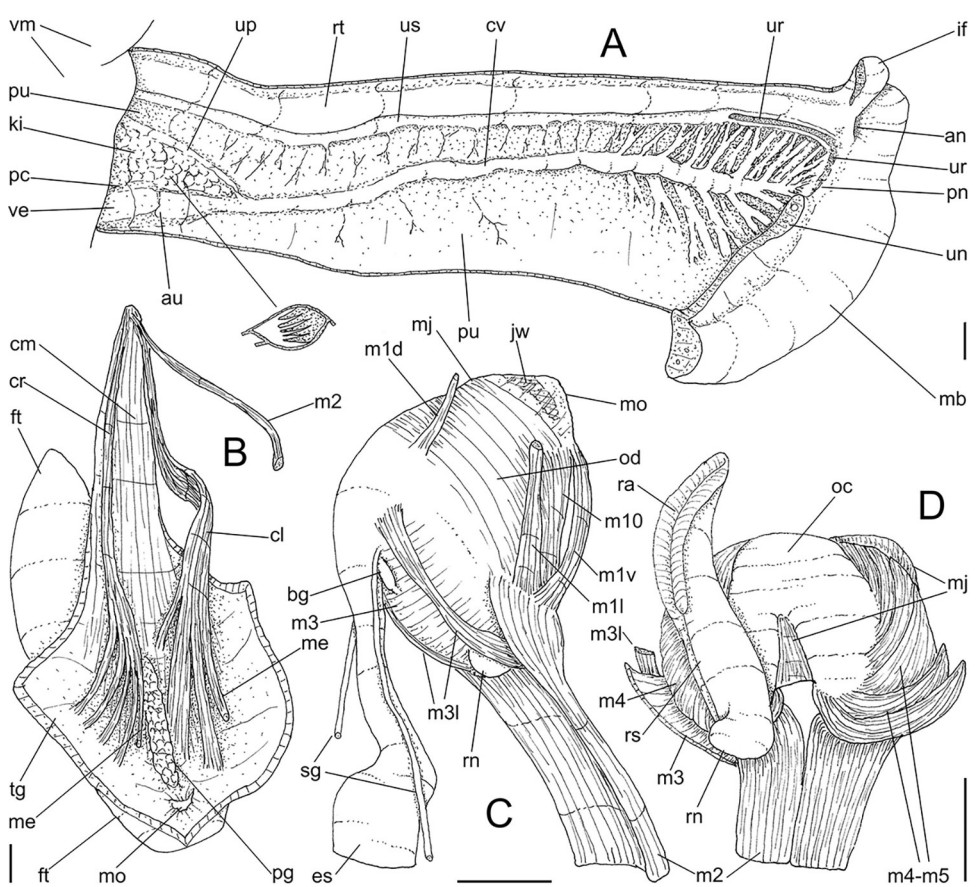

**Fig 50. *Neopetraeus lobbi* anatomical drawings.** (A) extended pallial (pulmonary) cavity, ventral-inner view, inner edge of pneumostome sectioned and deflected upwards, transverse section of indicated region of kidney also shown. (B) head-foot, dorsal view, head, dorsal integument and internal organs removed, remaining muscles expanded. (C) foregut, right view. (D) odontophore, dorsal view, cartilages deflected, radula removed and deflected to left, right muscles expanded, left muscles as in situ. Scales = 2 mm.

broader than long; cutting edge chevron-like; sculptured by successive, rather uniform, oblique, wide folds, convergent at middle. Buccal mass with radular sac small, weakly bulging beyond buccal mass (Fig 50C: rn). Dorsal surface of oral cavity (Fig 51B) with broad and low pair of dorsal folds (df), width of each ~1/3 of dorsal wall width; shallow dorsal chamber between them. Odontophore (Fig 50C: od) ~70% of buccal mass volume. Odontophore muscles (Figs 50C, 50D and 51B, 51C) with overall features of *K. corallina*, with following remarks: **m1v**, broad and long; **m1l**, relatively wide pair of dorso-lateral protractor muscles, originating in lateral region of mouth, running short distance towards anterior, inserting in latero-dorsal region of buccal mass surface (Fig 50C); **m1d**, small pair of dorsal protractor muscles of buccal mass, originating in dorsal region of mouth, running towards posterior covering dorsal surface of buccal mass, inserting in 2/3 of buccal mass length lateral surface (Fig 50C); **m3**, as cover of transverse fibers covering entire posterior surface of odontophore; **m3l**, pair of dorso-ventral muscles located each one in each side of esophageal insertion, covering m3, running up to m2 insertion; **m4**, very wide, composed of several separated V-shaped layers (Figs 50D and 51C); **m5**, incorporated to m4, not individualized; **m6**, absent; **m7**, absent; **m8**, absent; **m10**, broad; **m11**, pair of ventral tensor muscles of radula, narrow, originating in posterior edge of ventral surface of cartilages, running between cartilages and mj, inserting splaying in subradular

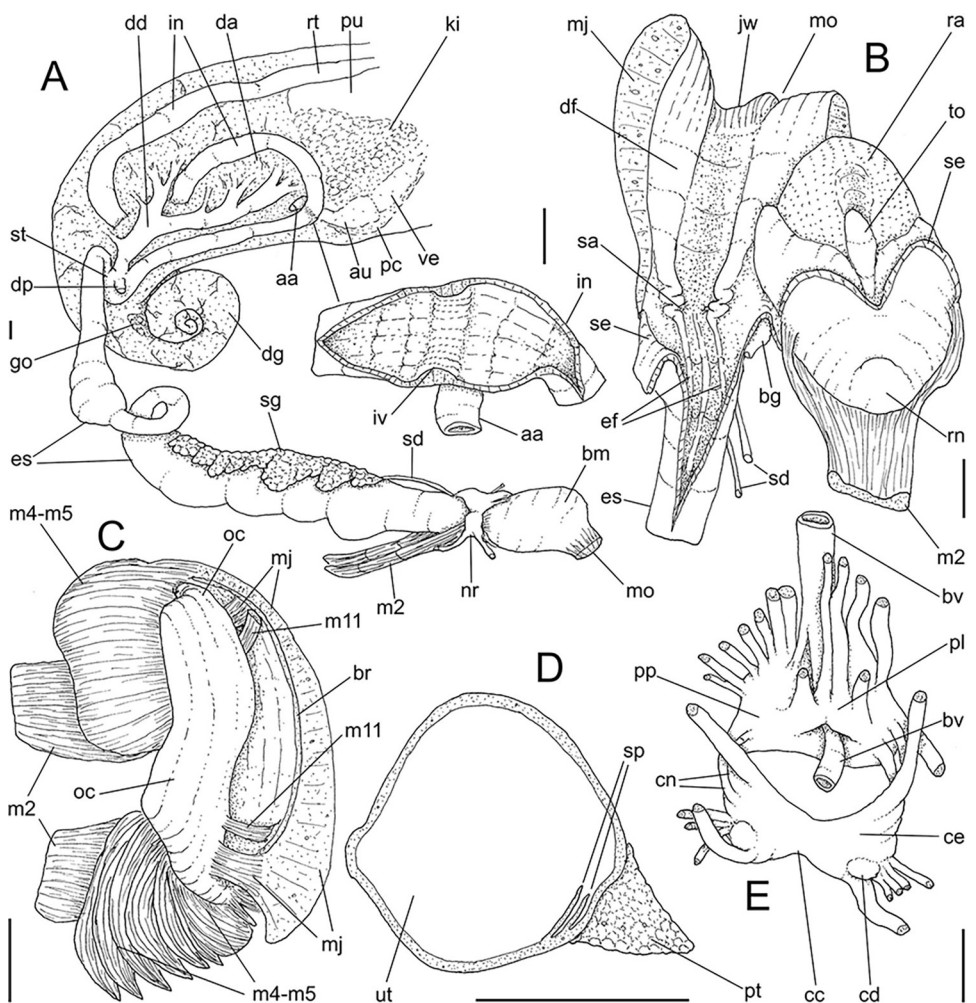

**Fig 51.** *Neopetraeus lobbi* **anatomical drawings.** (A) foregut and midgut, mostly ventral view as in situ, topology of some adjacent structures also shown, with detail of indicated stretch of intestine longitudinally opened. (B) buccal mass, ventral view, odontophore sectioned along its right and posterior edges and deflected to right, esophagus partially opened longitudinally. (C) odontophore, anterior view, cartilages (oc) curved ventrally, peribuccal muscles (mj) sectioned transversally and its anterior half removed, both (mj-oc) deflected from each other, right muscles (inferior in Fig) expanded, left muscles as in situ. (D) spermoviduct, transverse section of middle region. (E) central nervous system (nerve ring), dorsal view. Scales = 1 mm.

membrane in its region adjacent of tip of cartilages; **mj**, inserted in latero-ventral edged of cartilages, forming ventral platform (Fig 51C). Pair of odontophore cartilages ~50% fused with each other in their anterior-medial edge (Fig 50D: **oc**). Radular sac (Fig 50D: rs) short, slightly longer than odontophore. **Radula** (Fig 52) Composed of uniform, similar kind of tooth, with weak differentiation among rachidian, lateral or marginal teeth, ~50 pairs of teeth per row; each row strongly arched in margins, widely V-shaped in center (Fig 52A, 52B). Rachidian with rectangular base, length ~twice width; cusp stubby, triangular, located in distal end of base, with ~half of base length; situated perpendicularly to base, slightly arched inwards; tip bluntly pointed. First lateral teeth similar to rachidian, but with ~double cusp, with rounded tip, and appearance of minuscule subterminal inner secondary cusp (Fig 52D); remaining lateral teeth with cusp as large as base, and secondary cusp gradually increasing towards external up to becoming ~half main cusp's size (Fig 52C). No clear border between lateral and marginal

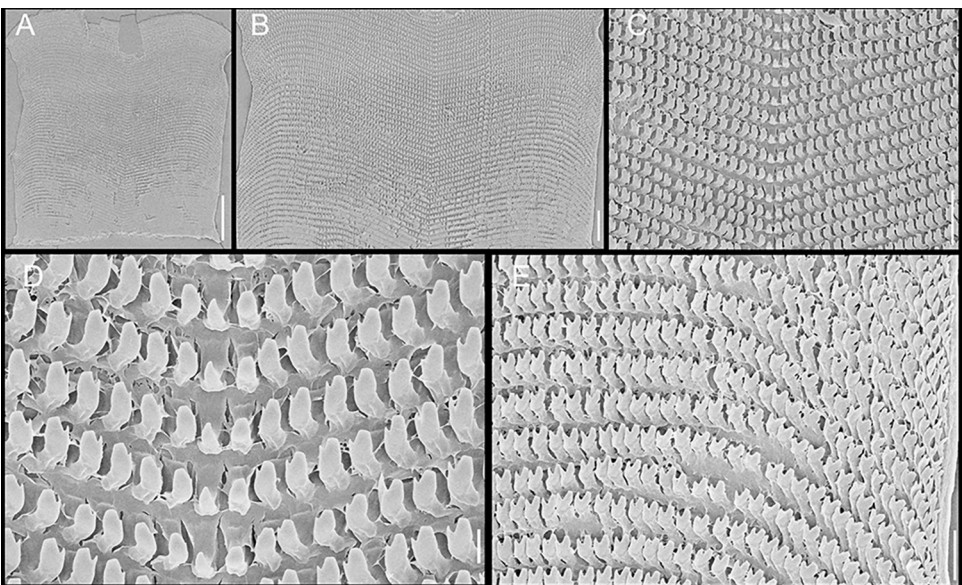

**Fig 52. *Neopetraeus lobbi* radulae in SEM.** (A) panoramic view, scale = 500 μm. (B) wide view, scale = 200 μm. (C) detail of central region, scale = 100 μm. (D) same, higher magnification, scale = 20 μm. (E) detail of lateral and marginal regions, scale = 50 μm.

teeth. Marginal teeth similar to lateral teeth, narrowing gradually towards margins, where row arching; more marginal teeth very narrow (Fig 52E). Salivary glands covering ~1/2 of esophagus length, located between anterior and second quarter of esophageal length (Fig 51A: sg), forming single multilobed, flattened, yellow mass. Each salivary duct differentiable in anterior side of glands (sd). Salivary duct running in both sides of esophageal origin, penetrating buccal mass wall in region close to buccal ganglia (Fig 51A: sd), running immersed in buccal dorsal wall along ~1/6 its length (Fig 51B). Salivary ducts opening as small pores (Fig 51B: sa), located medially in zigzag portion of posterior region of dorsal folds. Esophagus (Fig 51A: es) with irregular width along its length; inner surface with narrow, separated longitudinal folds. Stomach (Fig 51A: st) small, narrow, curved, weakly bulging; position and size described above (visceral mass); gastric walls thin, not muscular; inner surface mostly smooth, lacking folds. Duct to anterior lobe of digestive gland at short distance from intestine and esophageal intersections (dd) broad (as broad as intestine); bifid after some distance, with branches to both regions of anterior lobe of digestive gland (da). Duct to posterior lobe of digestive gland small, located in middle region of gastric ventral wall (dp). Intestine (Fig 51A: in) relatively narrow, performing its usual wide sigmoid loop in anterior lobe of digestive gland. Rectum and anus position described above (pallial cavity) (rt, an). Anus sessile, as slit in right end of mantle edge directly turned outside (Fig 51A: an).

**Reproductive system.** (Figs 51D and 53) General structures similar to preceding species, remarks and distinctions following. Gonad not clearly divided in lobes, acini not digitiform. Hermaphroditic duct (Fig 53B: hd) broad; dark pigmented, strongly coiled, mainly its middle region; insertion preceded by rather straight region (Fig 53D: hd). Seminal receptacle (Fig 53D: sr) small, straight, tip curved, ~4 times longer than wide, cylindric. Fertilization complex or carrefour (Fig 53D: ca) simple, tapering region in receptacle base, slightly longer than it, still possessing bulging region opposed to hermaphrodite duct insertion; totally immersed in albumen gland; insertion very narrow, in posterior end of spermoviduct, at tip of narrow albumen gland duct (ad) and intersection of albumen chamber (ac). Albumen gland (Fig 53B: ag) ~as

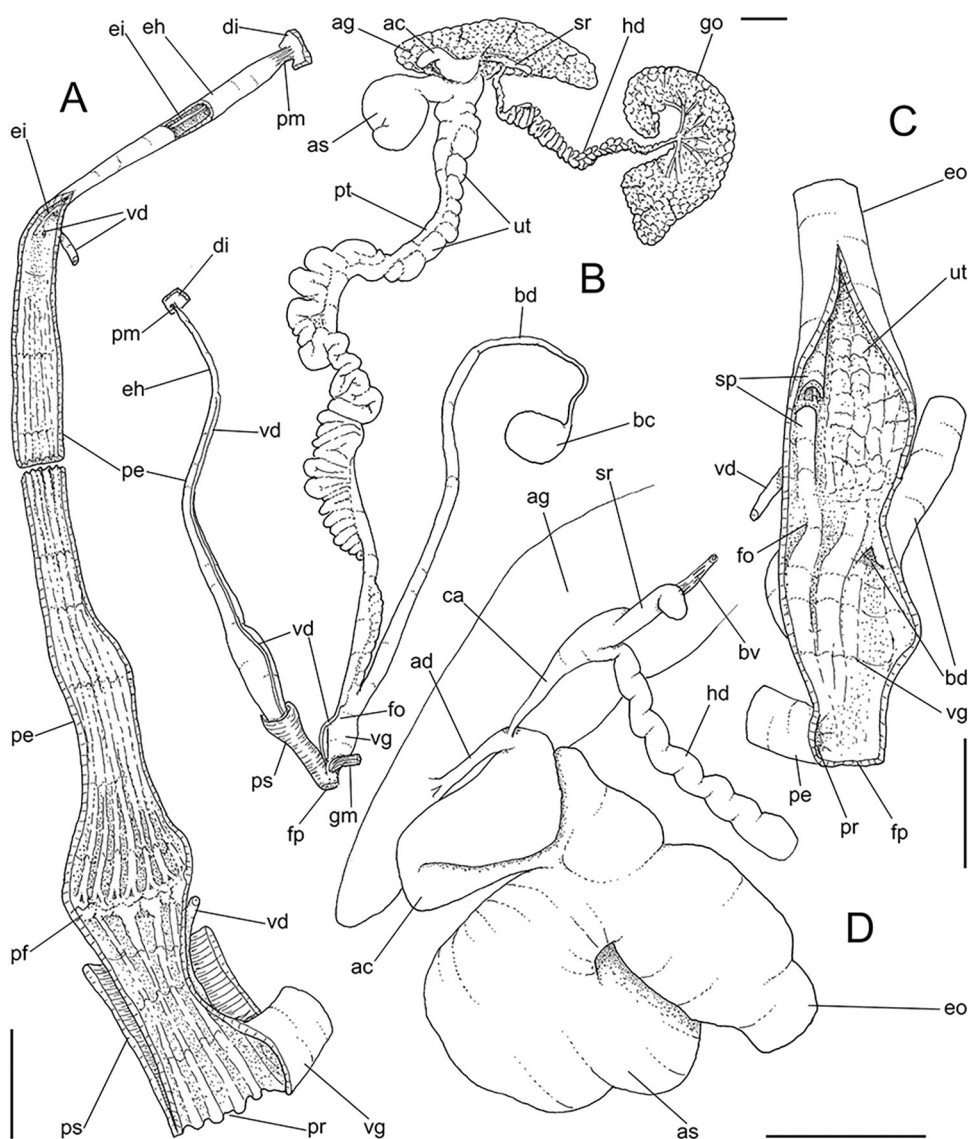

**Fig 53.** *Neopetraeus lobbi* **anatomical drawings.** (A) penis, ventral view, longitudinally opened, transverse section artificially done in its middle level, short portion of middle region of epiphallus (eh) also done, some adjacent structures also shown. (B) genital structures, dorsal view, mostly uncoiled. (C) spermoviduct, basal region longitudinally mostly opened, some adjacent structures also shown. (D) genital structures in albumen gland level if it was transparent, ventral view. Scales = 2 mm.

large as gonad (~1/4 whorl). Albumen gland duct subterminal, narrow (Fig 53D: ad), connected laterally to distal end of spermoviduct/albumen chamber. Albumen chamber (Fig 53D: ac) as simple, bulged curve connected to distal end of spermoviduct (eo). Spermoviduct (Fig 53B–53D: eo) of ~1.5 whorl in length, slightly narrower than albumen gland, ~20 times longer than wide. Secondary albumen chamber large (~half of albumen gland), balloon-like, with wide chamber, narrow duct connected just anterior to primary albumen gland chamber (Fig 53B, 53D: as). Prostate narrow (pt), ~1/6 of spermoviduct diameter (Figs 51D, 53B); uterus with walls not glandular (Figs 51D, 53C: ut); folds performing irregular zigzag (Fig 53B: ut). Sperm inner longitudinal fold double, tall, narrow posteriorly (Fig 51D: sp); one of them becoming still taller, curving covering other, becoming duct like (Fig 53C: sp), covering vas

deferens origin, slightly anterior to end of uterine level (Fig 53C: vd). Vas deferens uniformly narrow, uncoiled (Fig 53B: vd). Genital muscle small, in vaginal base (Fig 53B: gm). Bursa copulatrix (bc) and its duct (bd) of usual position, with ~70% of spermoviduct length (Fig 53B); bursa duct weakly muscular (Fig 53C: bd). Free oviduct (fo) and vagina (vg) simple, possessing 2 very wide, low, longitudinal, simple folds (Fig 53C). Penis almost straight, ~50% of spermoviduct length (Fig 53D: pe), base wider, gradually tapering up to pointed epiphallus; epiphallus as continuation of penis, penis muscle inserted terminally at epiphallus tip (Fig 53A, 53B: pm), very short, simple. Penis shield (ps) with ~1/4 penis length. Penis wall weakly muscular, all along its length (Fig 53A). Epiphallus (eh) ~1/3 of penis' length, amply opened to penis; only vas deferens insertion marking its limit (Fig 53A: vd). Epiphallus (eh) smooth, except for single, narrow, longitudinal fold (Fig 53A: ei), ending by side of vas deferens aperture. Internal penial arrangement of folds clearly with 2 regions (Fig 53A): (1) basal 1/3, possessing only 4 longitudinal, narrow, low, simple folds, with broad interspaces; (2) separated by transverse, low fold (pf) in which longitudinal folds end, occupying remaining 2/3 of penis length, bearing only 6–7 longitudinal, narrow folds, initially more evident and separated from each other, gradually becoming close from each other and weaker towards posterior, up to disappearing at some distance from epiphallus insertion.

**Central nervous system.** (Fig 51E) Characters of ganglia and statocysts virtually similar to those described for *K. corallina*. Except in being apparently more concentrated, and with shorter connectives. Pair of pleural ganglia (pl) located closer to pedal ganglia. Nerve ring located posteriorly to buccal mass (Fig 51A: nr).

**Distribution.** Known only from the for region of Amazon, Peru.

**Habitat.** Branches of *Jatropha* sp [25], Pata trees and cacti [26].

**Measurements** (in mm): MZSP 158045 (Fig 49A–49E): 40.2 by 18.1; Lectotype NHMUK 1975431 (Fig 49J–49L): 44.4 by 20.0.

**Material examined.** PERU. **Amazonas**; Balsas, Chacanto, 6˚50'36.45"S 78˚01'43.92"W, 880 m altitude, MZSP 158045, 1 spm (Valentin Mogollón col., ii.2016; Frederico Gutierres leg.).

**Taxonomic remarks.** See Discussion.

*Neopetraeus tessellatus* **(Shuttleworth, 1852) Figs 54–57.** *Bulimus tessellatus* Shuttleworth, 1852 [28]: 200.

*Neopetraeus tessellatus*: Breure, 1978 [29]: 217 (plus additional synonymy); Breure & Araujo, 217 [27]: 83 (Fig 31G) (plus additional synonymy).

Type locality: Peruvia.

**Distinctive description.** *Shell*. (Fig 54) Length ~40 mm, outline fusiform-globose, ~1.5 longer than wide. Color uniform white to light beige as base, plus mosaic of dark to light brown spots, more evident in 2 last whorls, randomly organized, but successively repeated, modifying gradually along whorl; sometimes spirally organized ([27]: Fig 31G) sometimes axially (Fig 54H). Protoconch with 2.5 whorls, slightly shouldered; sculptured by strong axial, uniform riblets, with interspaces with twice their width, fulfilled by narrow axial lines (Fig 54J, 54K), these axial lines very close from each other, from suture to suture (Fig 54K); limit with teleoconch clear, slightly prosocline (Fig 54J). Teleoconch of ~5 whorls successively and uniformly increasing; whorls weakly concave; suture weakly deep; sculpture absent, almost smooth, opaque, except for growth lines and delicate axial, uniform undulations in region preceding aperture. Transverse section circular (Fig 54F, 54G). Peristome weakly dislocated to right, ample; deflected. Callus weak, almost absent (Fig 54A, 54B, 54E, 54H). Aperture relatively narrow, somewhat dislocated from spire longitudinal axis; length ~57% of shell length, ~65% of shell width. Outer lip inserted not so distant from adjacent suture, simply arched. Inner lip concave, superior half weakly convex, mostly composed of exposed last whorl and

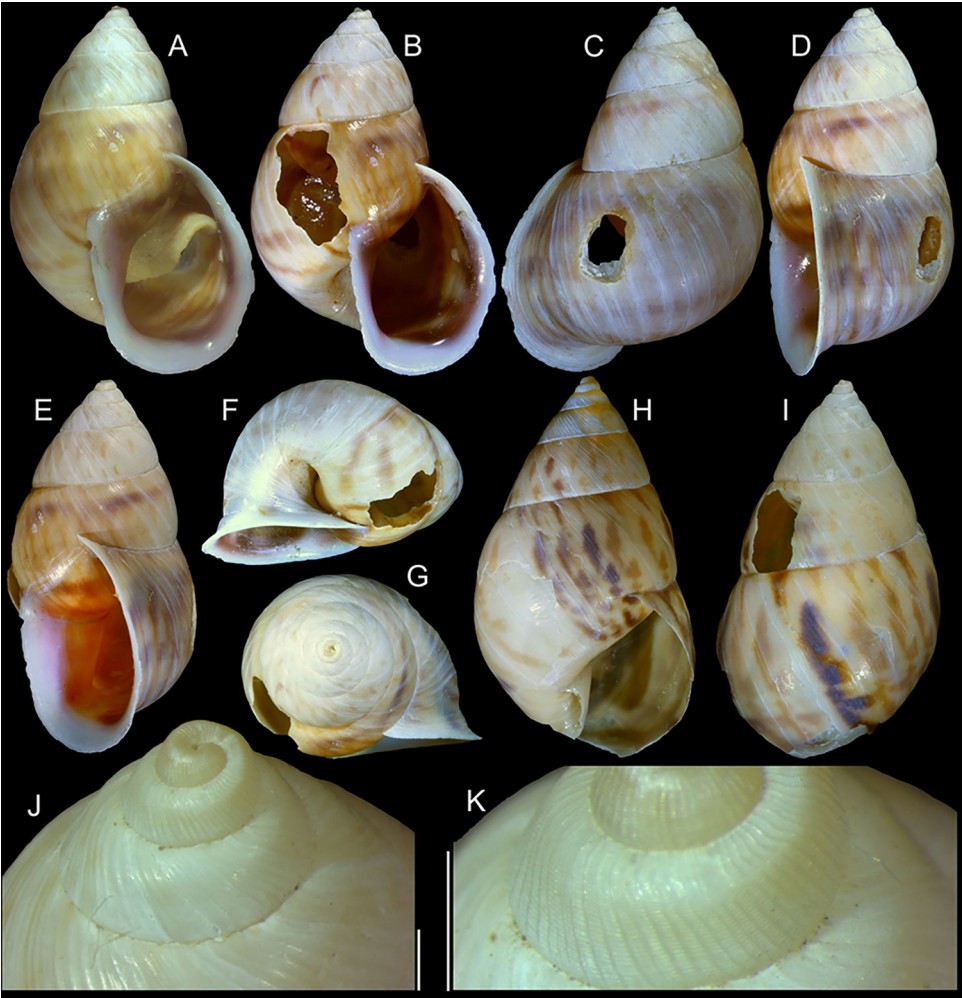

**Fig 54.** *Neopetraeus tessellatus* **shell characters (MZSP 158044).** (A) complete specimen #1 still inside shell, frontal view (L 36.9 mm). (B) same, specimen extracted by means of a hole done in beginning of last whorl. (C), same, dorsal view. (D) same, right view. (E) same, right-slightly ventral view. (F) same, anterior view, showing umbilicus. (G) same, apical view. (H–I) specimen #2, frontal and dorsal views (L 33.1 mm). (J) same, detail of apex, profile-slightly apical view. (K) same, detail of second whorl of protoconch showing sculpture. Scales = 1 mm.

thin callus; inferior half weakly straight; fold formed by umbilicus occupying entire inferior half (Fig 54E, 54F). Umbilicus widely opened, narrow, ventrally covered by inner half of inner lip (Fig 54F). More details in [28, 29].

**Head-foot.** Same characters of *N. lobbii*, except for having a columellar muscle ~25% shorter, and left secondary columellar muscle ~20% broader.

**Mantle organs.** (Figs 55D, 56A) With similarities to *N. lobbii*, distinctions and remarks following. Mantle edge (mb) also very thick and lacking secondary folds or glands. Lung of only ~1 whorl in length, almost as long as wide; lacking longitudinal muscle fibers. Pulmonary venation much less-developed, more visible in region preceding pneumostome; entire pulmonary vein (cv) protruded, relatively broad; left 2/3 lacking longitudinal vessels or any visible vessels, except for some few more developed in region of pneumostome; right 1/3 mostly having perpendicular, narrow vessels rather uniformly distributed, almost absent posteriorly, weak in middle and anterior regions; pulmonary vessel trifurcating very close to pneumostome. Reno-pericardial area of pale beige color, slightly triangular (ki), occupying ~40% of

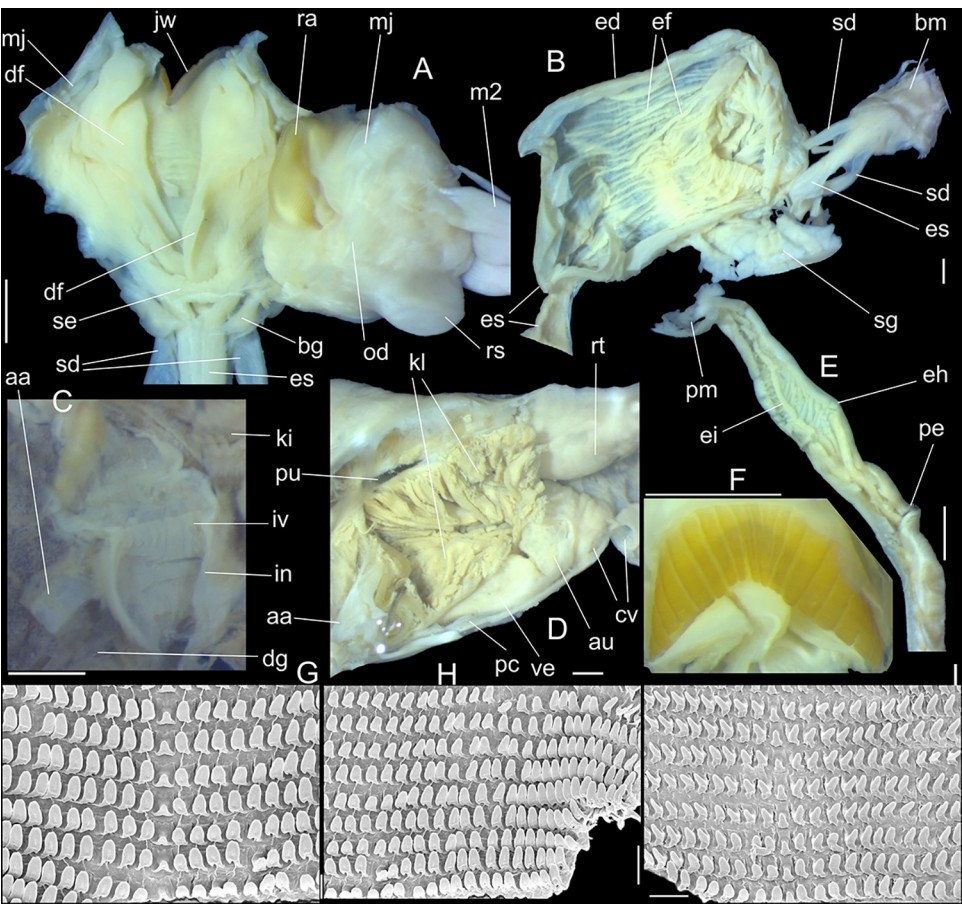

**Fig 55.** *Neopetraeus tessellatus* **anatomical characters (MZSP 158044) light photos.** (A) buccal mass, ventral view, odontophore sectioned along right edge (left in Fig) and deflected to right, showing dorsal inner surface of oral cavity. (B) foregut, dorsal view, dilated esophageal region opened longitudinally showing its inner surface. (C) intestine, ventral view, region close to pericardial region, opened longitudinally, adjacent structures also shown. (D) reno-pericardial region of pallial roof, ventral view, renal wall sectioned along its left edge and deflected upwards. (E) distal end of penis and epiphallus, opened longitudinally. (F) Jaw in situ, ventral view. Scales = 1 mm. (G–I) radulae in SEM. (G) detail of central region, scale = 30 μm. (H) detail of lateral and marginal regions, scale = 60 μm. (I) detail of central region, scale = 60 μm.

cavity length and ~50% of its width (details below). Rectum (rt) wide. Primary (up) and ~40% of secondary (us) ureters closed (tubular), relatively narrow; ~60% of anterior region of ureter opened, as widely opened groove (ur), with elevated edges, smooth inside, running like this up to pneumostome, in urinary furrow flanking anus (an) left edge.

**Visceral mass.**　With similar attributes as *N. lobbii.*

**Circulatory and excretory systems.**　(Figs 55D and 56A) General Bauplan similar to *N. lobbii*, except in being proportionally larger, and in having kidney lobe almost entirely solid (not organized in folds), with single, central, tight lumen (Fig 55D).

**Digestive system.**　(Fig 56B, 56C) General organization resembling that of *N. lobbii*, distinctions and remarks following. Jaw plate (Fig 55F) thin, yellow, strongly curved, ~2x broader than long; cutting edge deeply concave; sculptured by successive, fewer, wide, oblique, wide folds, convergent at middle. Buccal mass with radular sac small, weakly bulging beyond buccal mass (Figs 55B and 56B, 56C: rn). Dorsal surface of oral cavity (Fig 55A) similar, but lacking zigzag posterior portion, with dorsal folds (df) broad up to their posterior region, overlapping

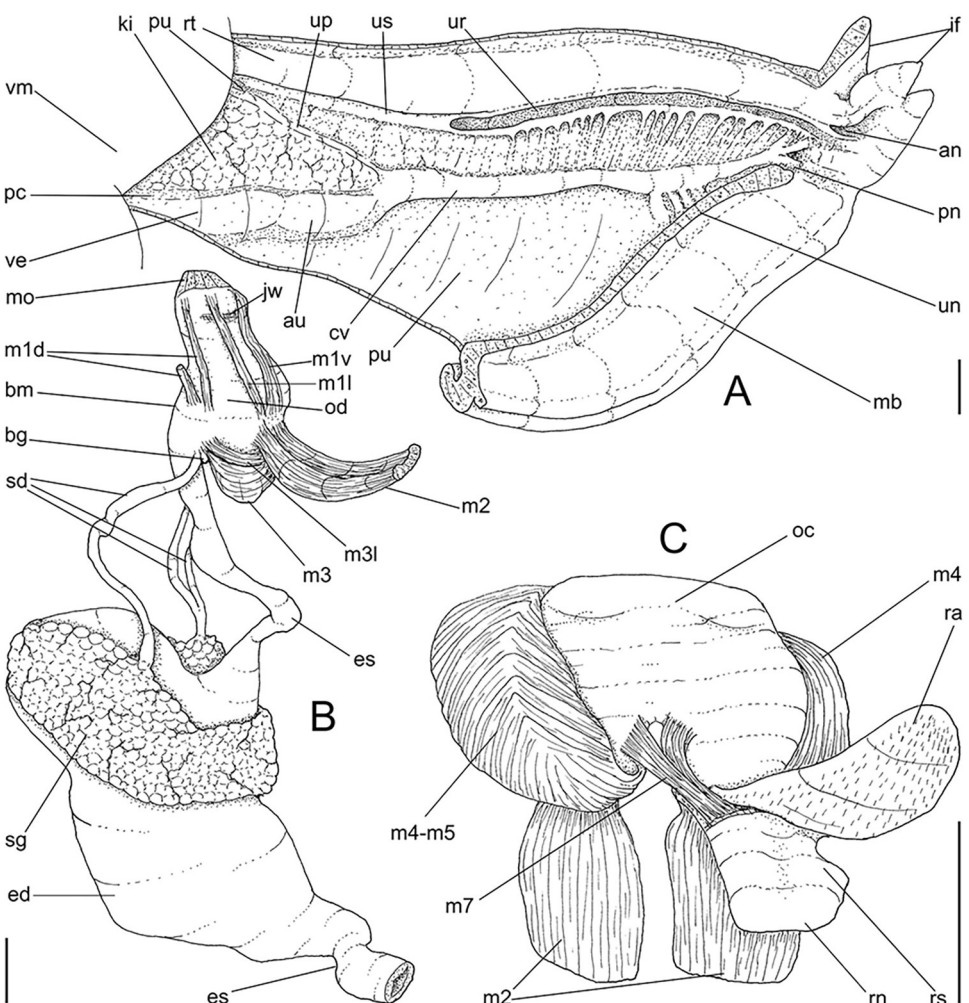

**Fig 56.** *Neopetraeus tessellatus* **anatomical drawings.** (A) extended pallial (pulmonary) cavity, ventral-inner view, inner edge of pneumostome sectioned and deflected upwards. (B) foregut, right extended view. (C) odontophore, dorsal view, superficial layer of muscles and membranes removed, radula removed and deflected downwards, right muscles also deflected. Scales = extended pallial (pulmonary) cavity, ventral-inner view, inner edge of pneumostome sectioned and deflected upwards. Scales = 2 mm.

posteriorly. Odontophore (Fig 56B: od) ~50% of buccal mass volume. Odontophore muscles (Fig 56B, 56C) with overall features of *N. lobbii*, with following remarks: **m1v** and **m1l**, similar, elongated; **m1d**, double; **m3** and **m3l**, similarly organized; **m4**-**m5**, also composed of several separated V-shaped layers; **m7**, pair present and broad, originated close to fusion of both cartilages (Fig 56C). Pair of odontophore cartilages ~70% fused with each other in their anterior-medial edge (Fig 56C: **oc**). **Radula** (Fig 55G–55I) composed of rachidian plus ~40 pairs per row, with similar features as *N. lobbii*, except for: rachidian ~30% wider (Fig 55G), with concave posterior edge; lateral and marginal teeth lacking secondary cusps and more inclined externally (Fig 55H, 55I). Salivary glands covering ~1/3 of esophagus length, located at middle of esophageal length (Figs 55B, 56B: sg), thick, forming single multilobed, yellowish-cream mass. Each salivary duct very broad, differentiable in anterior edge of glands (sd); left duct bifid. Esophagus (Figs 55B and 56B: es) initially narrow; posterior half with large crop, as enormous dilatation (Figs 55B and 56B: ed) bulged anteriorly, tapering posteriorly; internally bearing longitudinal, relatively regular folds (ef). Remaining gastric and intestinal features similar

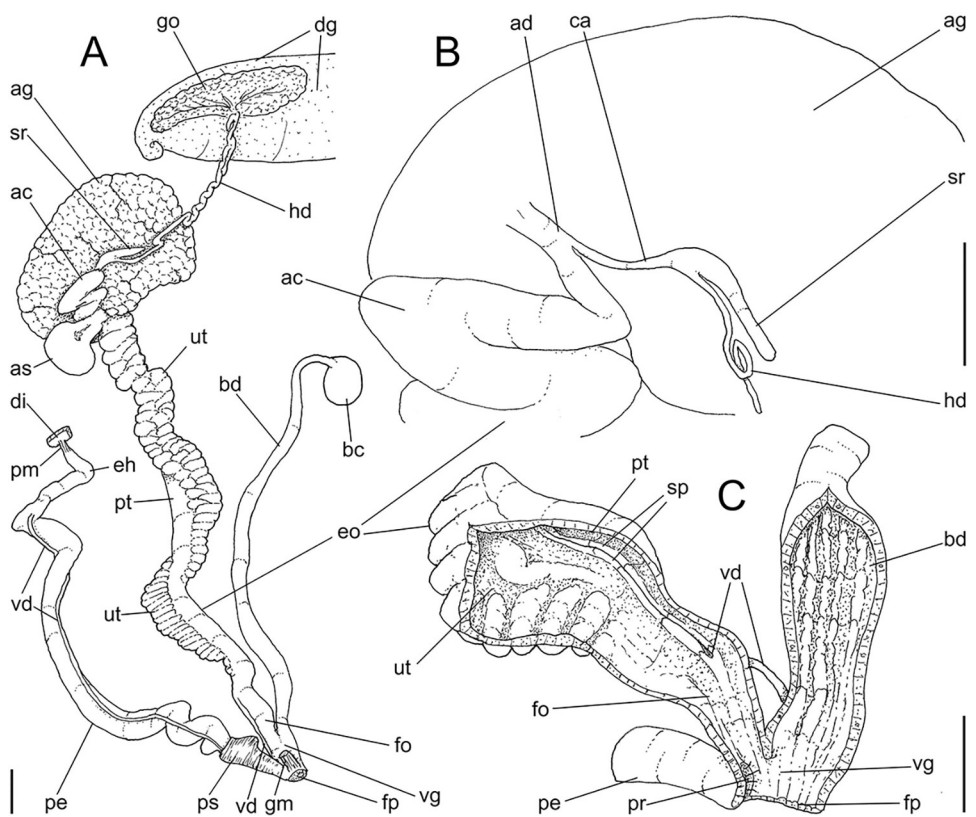

**Fig 57.** *Neopetraeus tessellatus* **anatomical drawings of genital structures.** (A) whole dorsal view, uncoiled, terminal end of visceral mass also shown. (B) genital structures in albumen gland level if it was transparent, as in situ, ventral view. (C) genital tubes portion preceding pore, dorsal view, 2 of them longitudinally opened. Scales = 2 mm.

to those of *N. lobbii*, except for intestine having clear transverse wide fold in region adjacent to pericardium (Fig 55C: iv) and to anterior aorta (aa).

**Reproductive system.**   (Figs 55E and 57) General structures similar to *N. lobbii*, remarks and distinctions following. Gonad not clearly divided in lobes, acini not digitiform; located close to end of visceral whorls (Fig 57A: go). Hermaphroditic duct (Fig 57A: hd) narrow, slightly coiled along almost its entire length; insertion preceded by rather straight region (Fig 57B: hd). Seminal receptacle (Fig 57A, B: sr) small, straight, tip rounded, ~6 times longer than wide, cylindric. Fertilization complex or carrefour (Fig 57B: ca) simple, small conic region in receptacle base, succeeded by elongated, narrow duct as long as receptacle slightly; insertion very narrow, by side of posterior end of spermoviduct, at tip of narrow albumen gland duct (ad). Albumen gland (Fig 57A: ag) ~2x larger than gonad (~1/3 whorl). Albumen gland duct subterminal, narrow (Fig 57B: ad), connected frontally to distal end of spermoviduct and carrefour insertion. Albumen chamber (Fig 57A, 57B: ac) as simple, bulged curve connected to distal end of spermoviduct (eo). Spermoviduct (Fig 57A, 57C: eo) of ~1.5 whorl in length, ~3x narrower than albumen gland, ~15 times longer than wide. Secondary albumen chamber present, ~15% of albumen gland size, balloon-like, with wide chamber, narrow duct connected just anterior to primary albumen gland chamber (Fig 57A: as). Basic composition of spermoviduct and genital muscle similar to those of *N. lobbii*, except in lacking closure of sperm groove (sp). Bursa copulatrix (bc) and its duct (bd) with ~90% of spermoviduct length (Fig 57A); bursa duct mostly weakly muscular, except for its thicker walls of anterior region (Fig 57C: bd).

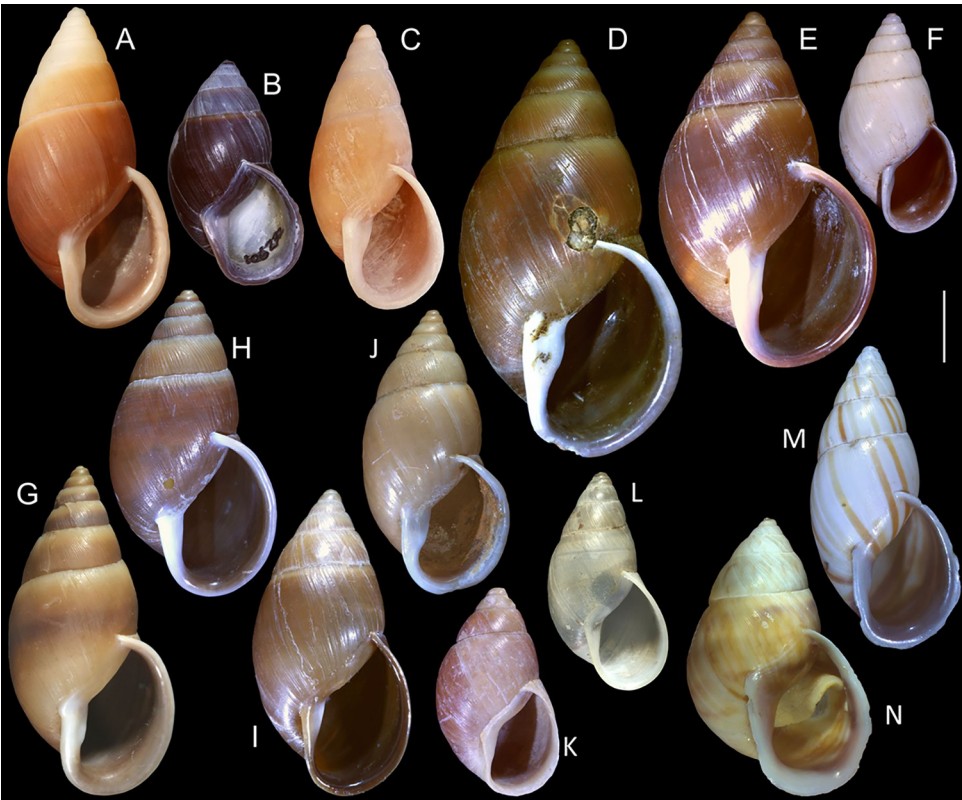

**Fig 58. Gallery of studied shells in same scale, frontal views.** (A) *Kora corallina* holotype MZSP 103910. (B) *K. nigra* holotype MZSP 106230. (C) *K. rupestris* holotype MZSP 121416. (D) *K. tupan* holotype MZSP 161200. (E) *K. ajar* holotype MZSP 163700. (F) *K. aetheria* holotype MZSP 163400. (G) *K. jimenezi* holotype MZSP 151907. (H) *K. kremerorum* holotype MZSP 151809. (I) *K. uhlei* holotype MZSP 165720. (J) *K. vania* holotype MZSP 165500. (K) *K. curumim* holotype MZSP 152077. (L) *Koltrora pyrostoma* holotype MZSP 163500. (M) *Neopetraeus lobbii* MZSP 158045. (N) *Neopetraeus tessellatus* MZSP 158044. Scale = 10 mm.

Penis general attributes similar to those of *N. lobbii*, except in being longer (~90% of spermoviduct length); in having uniform width along its length (not tapering posteriorly); epiphallus (eh) proportionally shorter and broader, internally having strong longitudinal fold (Fig 55E: ei), flanked by secondary, small oblique folds; wall uniformly more muscular; and in lacking bulged basal region with transverse fold (transverse fold absent).

**Central nervous system.** Same characteristics of *N. lobbii*.

**Distribution.** Known only from the for region of Lima, Peru.

**Habitat.** No data.

**Measurements** (in mm): MZSP 158044: #1 (Fig 54A–54E): 36.9 by 23.7.

**Material examined.** PERU. **Lima**; Cañón del Pato, Kiman Aullu, 8˚46'25.85"S 77˚52'26.28"W, 840–870 m altitude, MZSP 158044, 2 spm (Valentin Mogollón col., 18.vi.2010; Frederico Gutierres leg.).

**Taxonomic remarks.** See Discussion. Confront of shells in a same scale of all specimens studied in this paper is in Fig 58.

## Genus *Anctus* Martens, 1860

*Anctus angiostomus* (Wagner, 1827) **Fig 59.** The taxonomic treatment and anatomical description by Simone (1998) [18] are particularly complete. Some missing data of that, necessary to this species be included in the phylogeny of the present paper, are here completed.

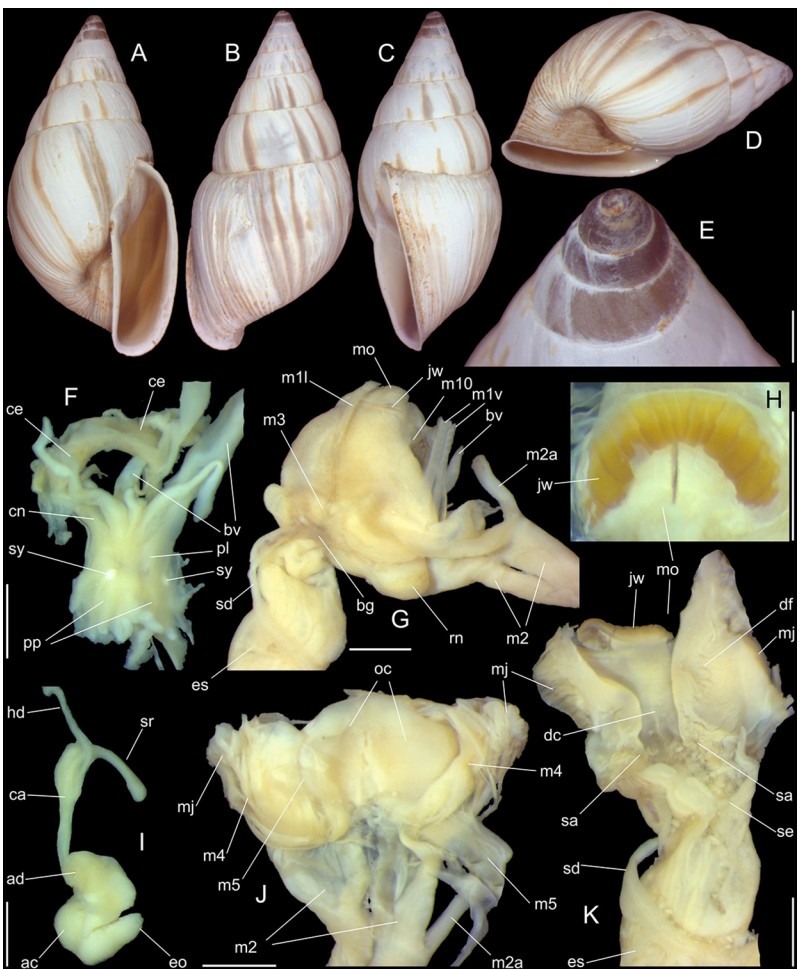

**Fig 59.** ***Anctus angiostomus*** **anatomical features, MZSP 27937, 4 spm.** (A) shell (L 28.1 mm), frontal view. (B) same, dorsal view; (C) same, right view; (D) same left view; (E) detail of apex, profile-slightly apical view. (F) nerve ring, ventral view. (G) buccal mass, right view. (H) jaw in situ, ventral view. (I) region of carrefour of reproductive system, removed from albumen gland, ventral view. (J) odontophore, dorsal view, both cartilages deflected, right muscles expanded, left muscles as in situ. (K) dorsal wall of buccal mass, ventral-inner view, odontophore removed. Scales = 1 mm.

**Shell.** (Fig 59A–59E) With very elongated peristome (Fig 59A, 59C), wide umbilicus (Fig 59A, 59D), delicate undulations if sculpture (Fig 59B). Protoconch dark brown, smooth, of 2 whorls (Fig 59E).

**Head-foot.** Secondary columellar muscles with 2 wide insertions in right component, and 4 wide insertions in left component.

**Digestive system.** (Fig 59G, 59H, 59J, 59K) Jaw plate wide horseshoe-shaped, with broad transverse folds (Fig 59H). Buccal mass with well-developed pair m1l and 2 pairs of m1v (Fig 59G), also pair m2a and pair m3 dorso-ventral in esophageal-odontophore limit. Odontophore cartilages (oc) almost 100% fused with each other (Fig 59J), with pair m5 covering medial region of m4; m7 wanting. Salivary ducts (Fig 59K: sd) broad; aperture of salivary glands (sa) very posteriorized on dorsal folds (df).

**Reproductive system.** (Fig 59I) Presence of bulged region by side of base of seminal receptacle (sr). Carrefour (ca) narrow and elongated duct inserted in albumen gland duct (ad). Albumen chamber (ac) as curve preceding spermoviduct.

**Central nervous system.** (Fig 59F) Similar to preceding species, cerebral commissure relatively wide. Pleural ganglia (pl) close to pedal ganglia (pp), bearing commissure.

**Measuremens** (in mm): MZSP 27937: 28.1 by 14.0.

## Phylogenetic analysis

As informed above, a phylogenetic analysis was performed with the single objective of comparing the species studied herein in a more orthodox scenario, as well as to take advantage of already available data in the literature on other orthalicoideans to analyze the taxonomic/phylogenetic position of them in a wider context, with some biogeographic conclusions (Fig 60) (more details below). Also, a secondary objective is to compare the present result based on a morphologic-based dataset with others based on molecular ones.

The matrix of characters (S2 Appendix), which includes ingroup and outgroups, was built based on the 94 characters (251 states) listed in the S1 Appendix. The processing of this dataset as reported above resulted in a single cladogram (Fig 61), which is analyzed in the Discussion item. Its statistics are length: 391; consistency index: 54, retention index: 76. For the obtention of a single cladogram, the method by [31] was applied.

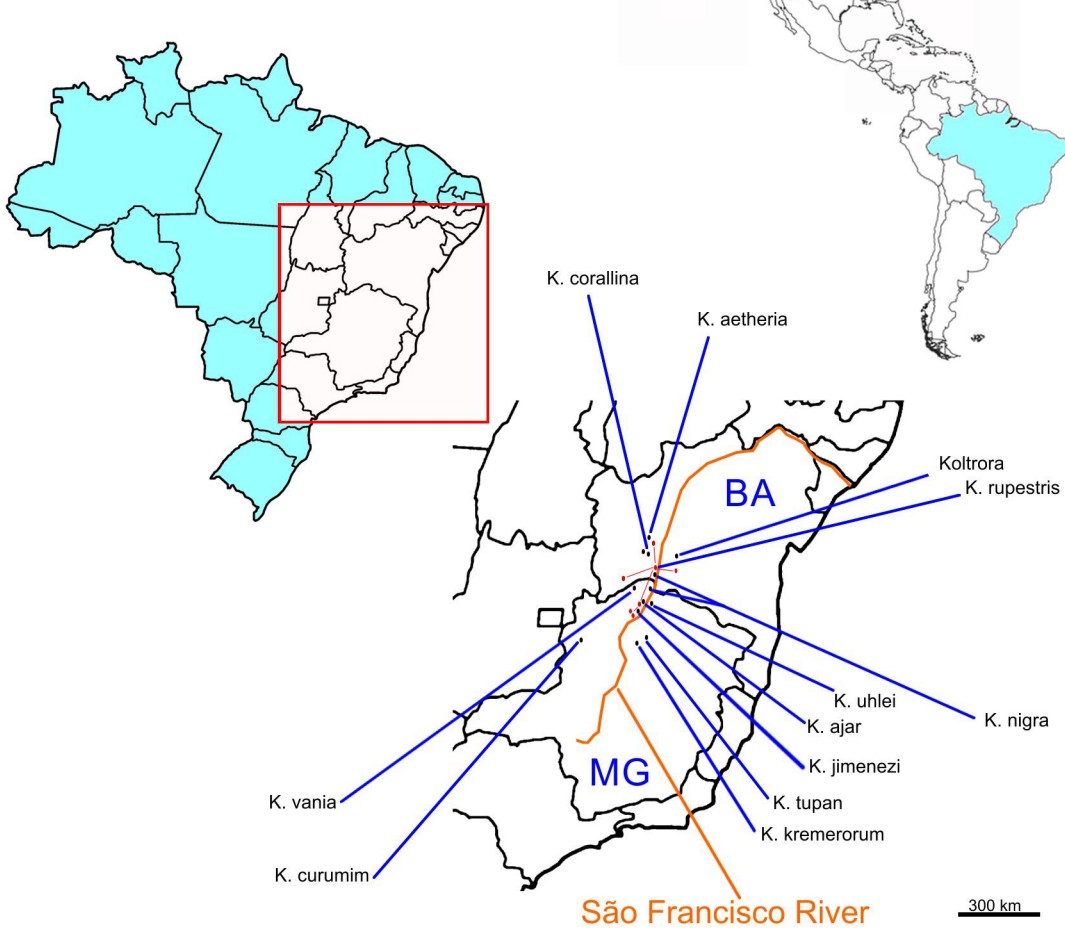

**Fig 60. Self-produced maps with countries and states outlined.** (top-right) Central and South America, Brazil painted blue. (top-left) Brazil with mark of region enlarged below. (center-below) region of studied samples indicated by black dots, except for *K. rupestris* (red). São Francisco River represented in orange. Abbreviations: BA, Bahia; K, *Kora*; MG, Minas Gerais.

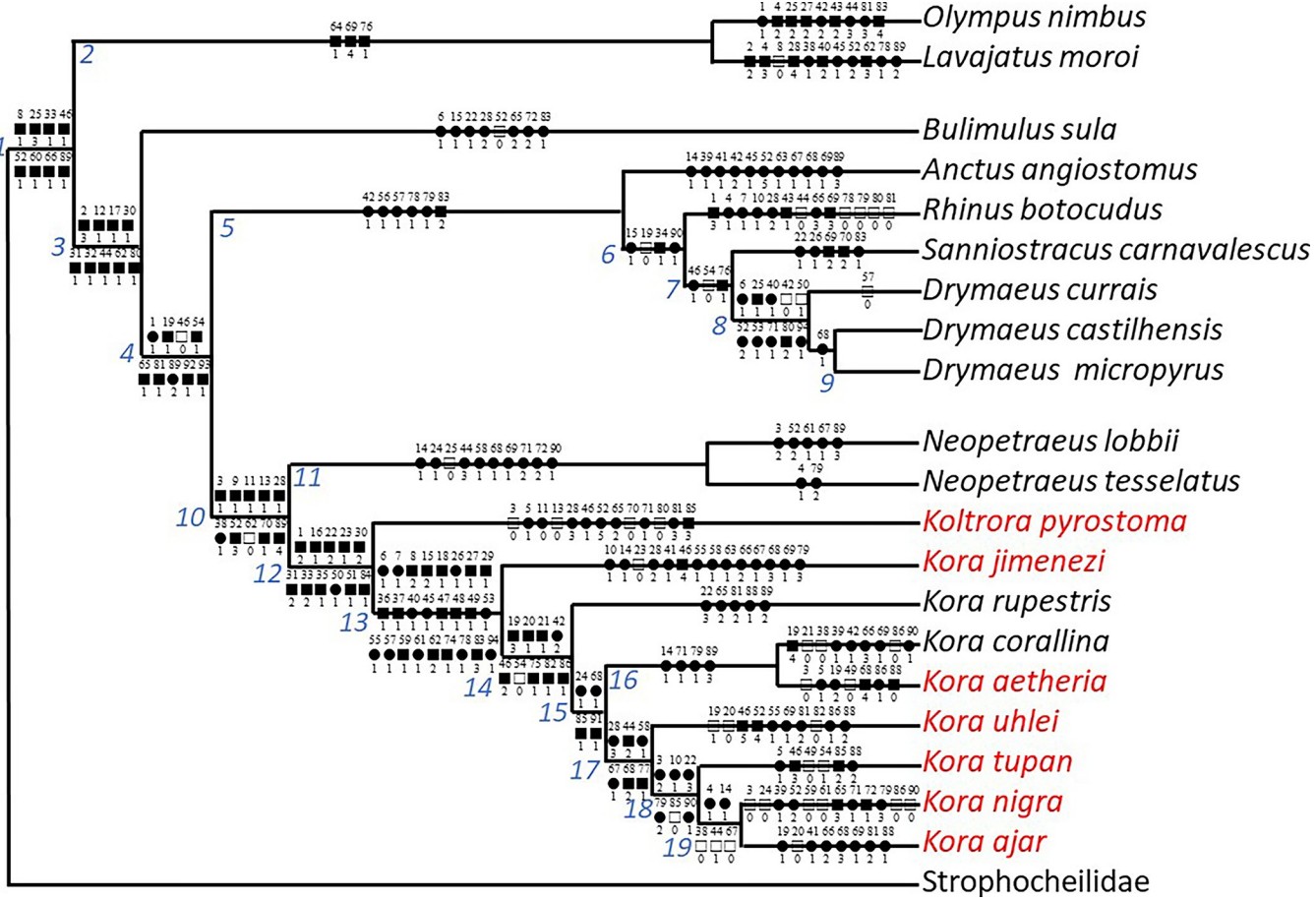

**Fig 61. Single cladogram obtained analyzing the matrix (S2 Appendix) of the species assembly reported in Material and Methods (written black), plus the species studied herein (new taxa written red).** Branch below is the far outgroup (rooting). Each node has the set of synapomorphies shown with symbols, indicating: black square = non-homoplastic synapomorphy; white square = reversion; black circle = convergence. Above number each symbol is the character, number below is the state in that node. Blue number in italics are branches numeration. Method: parsimony; L = 391; CI = 54; RI = 76.

## Discussion

### Taxonomic discussion

The present taxonomic discussion is organized into genus level discussions and, after, a species level one:

At the genus level, *Kora* exhibits much more uniformity of characters than *Neopetraeus*. Its shell characters are relatively monotonous, with almost monochromatic color, except for a small paler subsutural band, and sometimes a darker pre-apertural region. The general shape is relatively similar, with some species slightly more elongated (e.g., *K. rupestris*), while others are more globose (e.g., *K. ajar*). *Neopetraeus*, on the other hand, displays a much wider range of features in its 15 species [6, 11], varying in color, shape, profile, and even sculpture. *Koltrora* is notable for its translucent, fragile shell in its (so far) single species. *Neopetraeus* occurs in the Andean region of South America, well separated from the other two genera, which occur in the mid-east region of Brazil, in the area of the São Francisco River. On the other hand, the three genera share shell features that bring them closer and set them apart from other orthalicoideans, including the smooth protoconch with ~2 whorls, possessing some axial sculpture in the last half whorl in some species (some *Neopetraeus* are exceptions, with a reticulate

protoconch–Fig 54J); and a well-developed umbilicus resulting from a hollow tube zigzagging along the columella. This feature confused the initial identification of *Kora* samples, which were initially attributed to *Neopetraeus* [10], despite the geographic distance.

With respect to the anatomy, the three genera have at least the following distinctions: *Kora* is alone in having a medial differentiated branch in secondary columellar muscles on both sides (e.g., Fig 18B), absent in the other known orthalicoideans; it also has a characteristic white pallial gland in the mantle edge (e.g., Fig 36A: gp); the radular sac bulging in the posterior end of odontophore, which is covered by a transparent membrane (e.g., Fig 3C: mr); the odontophore pair of muscles m8 (e.g., Fig 4C); a very muscular basal region of bursa copulatrix (e.g., Fig 6A: bu); a pair of longitudinal large folds along the inner penis surface (e.g., Fig 38D: pf); and an especially elaborated spermatophore (e.g., Fig 17D). *Koltrora* is the only known orthalicoidean that possesses two ducts to anterior lobe of digestive gland (Fig 46C: dd). *Neopetraeus* is distinctive in having both pairs of dorsal tensor muscles of radula (m4 and m5) as an indistinct single mass (Fig 56C). *Kora* and *Neopetraeus* share the accessory albumen chamber (e.g., Fig 6B: as), which was not found in *Koltrora*. The ureter is totally closed (tubular) in *Kora* and *Koltrora*, as it is a common pattern of orthalicoideans, but *Neopetraeus* has the ureter opened in a larger degree (~50%–Fig 56A); both genera also lost the pair of ventral tensor muscles of the radula (m11), which is present in *Neopetraeus* (Fig 51C). Both genera also share the calcified epiphragm (Figs 2J and 45D), which was not confirmed in *Neopetraeus*, neither in other orthalicoideans. Both genera also have a thick muscular basal region in the penis wall, which is absent in *Neopetraeus*. Other distinctions are present in phylogenetic analysis below.

Although *Koltrora* shares more characters with *Kora*, particularly in the radula, than with *Neopetraeus*, its recognition as a new genus was necessary for several reasons. Phylogenetic analysis demonstrated that *Koltrora* possesses 13 autapomorphies, which together distinguish it from the closely related taxa. Additionally, it lacks the 25 synapomorphies that define *Kora* (Fig 61, node 13). *Koltrora* differs from *Kora* by having a translucent, unpigmented shell with fragile walls, whereas *Kora* has thick-walled, consistently brown-pigmented shells. Furthermore, *Koltrora* is smaller than typical *Kora* species (Fig 58L). Anatomically, *Koltrora* is primarily distinguished from *Kora* by the absence of both the medial, more posteriorized branches of the secondary columellar muscles (Fig 46B: cr, cl); by the complete fusion of both odontophore cartilages (Fig 47B: oc); by having two ducts leading to the anterior lobe of the digestive gland; by the absence of an accessory albumen chamber; by the thick glandular uterine walls (Fig 47E: ut); by the epiphallus being widely open to the penis; and by the presence of a transverse fold in the middle of the penis chamber, with an adjacent arrangement. Additionally, *Koltrora pyrostoma* inhabits an isolated region in Bom Jesus da Lapa, Bahia, northeastern far from the typical range of *Kora* species. It is found at a relatively high altitude of approximately 520 meters, indicating a different environment compared to *Kora* species, which occur at lower altitudes in regions closer to the São Francisco River (as discussed below).

The genus *Kora* has been solidified as a distinct genus with the addition of new species and anatomical characters. In addition to having a shell with a relatively smooth protoconch and some axial sculpture on its last whorl, a brown color with the usual paler subsutural band, and an open umbilicus covered by an oblique columellar fold, *Kora* can be distinguished from its closest genera, *Neopetraeus* and *Koltrora*, by a set of anatomical features. *Kora* possesses a pallial gland; a wide, flaccid translucent membrane (mr) covering the radular sac, which bulges posteriorly to the odontophore; and a radula with numerous hook-like teeth (a trait shared with *Koltrora*). Additionally, *Kora* has muscular walls at the base of the duct of the bursa copulatrix; a well-developed accessory albumen chamber (as) (shared with *Neopetraeus*); a penis divided into compartments, always with a strong pair of longitudinal inner folds; a spermatophore with a chitinous basal tube; and a calcified epiphragm (shared with *Koltrora*).

The genus *Kora*, thus, has consolidated as a different genus with the addition of the new species and the anatomical characters. Beyond the shell with relatively smooth protoconch, with some axial sculpture in the last whorl; brown color with usual subsutural paler band; open umbilicus covered by oblique columellar fold; there is a set of anatomical features that can be used to distinguish the genus particularly from *Neopetraeus* and *Koltrora*, its closest genera. *Kora* has a pallial gland; has a wide, flaccid translucent membrane (mr) covering the radular sac bulging posteriorly to odontophore; radula with numerous hook-like teeth (shared with *Koltrora*); has muscular walls of the base of the duct of the bursa copulatrix; has a well-developed accessory albumen chamber (as) (shared with *Neopetraeus*); has the penis divided into compartments, aways with strong with pair of longitudinal inner folds; a spermatophore with chitinous basal tube; and a calcified epiphragm (shared with *Koltrora*).

**At the species level**, all 11 *Kora* species studied here naturally have their shells similar (Fig 58), only small details can distinguish the species. The more important differences are exposed below (Table 2). A relatively intraspecific uniformity, however, was observed, what is important in a so featureless shelled genus. The species can be separated into size categories, those considered large, in the range of the 55 mm (*K. tupan*, *K. ajar*), those medium-sized, in the range of the 45 mm (*K. corallina*, *K. rupestris*, *K. jimenezi*, *K. uhlei*, *K. kremerorum*), and the small ones, in the range of the 30 mm (*K. nigra*, *K. aetheria*, *K. vania*, *K. curumim*). Related to the elongation degree, there is the more elongated, with tax length/width ~2.3 (*K. corallina*, *K. rupestris*), and those more globose, with tax ~1.6–1.7 (*K. nigra*, *K. ajar*, *K. curumim*), with remaining species with the tax ~1.9–2.0. Three species have a weak dorso-ventral shell compression: *K. tupan*, *K. aetheria* and *K. kremerorum*; the remaining species have a rounded transverse section. The superior implantation of the outer lip has a horizontal portion in *K. nigra*, *K. tupan*, *K. ajar*, *K. jimenezi*, *K. kremerorum*, and *K. vania*. The inner lip has usually a middle prominence in most species, almost making a fold, but some species this is reduced, as in *K. aetheria*, and weakly present in *K. nigra*, *K. kremerorum* and *K. curumim*. Related to the color, *K. corallina* and *K. rupestris* have their apical region light yellow to white, darkening in middle level of spire; the remaining species have their apex with uniform color as remaining areas. The subsutural lighter band is not present only in *K. aetheria* and *K. curumim*. The pigmentation of the peristome is usually white, but *K. nigra*, *K. ajar* and *K. uhlei* commonly have brown pigment in it. The sculpture, beyond the usual axial narrow undulations, delicate spiral striae is also usually found, but they are absent in *K. aetheria*, *K. uhlei*, and *K. curumim*; and they are very scanty in *K. vania*. The umbilicus is usually widely opened, but *K. curumim* has it tightly occluded by inner lip; while *K. aetheria*, *K. vania* and *K. jimenezi* have the umbilicus narrowly opened.

Concerning anatomy, the differences between species become more pronounced. Table 3 presents synthesized information on the anatomical features of the species studied here. This discussion focuses only on those considered more important. The data presented in Table 3 is consistent across all examined specimens (Table 1), showing minimal intraspecific variation and providing a compelling basis for their use in taxonomy and phylogeny. Due to space limitations, the interpretation of Table 3 should be correlated with the list of abbreviations and, in some instances, with the information provided in the respective description. Features such as folds in the mantle edge, the type of venation in the lung, and the organization of the kidney lobe serve as intriguing sources of taxonomic data, differing among the species. Similarly, variation was observed in the number of insertions of the secondary columellar muscles, with each species having its arrangement, and they are mostly asymmetrical. This observation extends to the buccal mass and odontophore muscles. As expected, the jugal muscles (m1l, m1v), and the m3 have proven useful in differentiating closely related species [24, 30]. On the other hand, the discovery of m2a in the present context is intriguing (Fig 23F). This pair of muscles is exclusive

to *K. ajar* and *K. jimenezi*, exhibiting an impressive convergence with *Anctus* (see below). This muscle pair, inserted into the m2 basal edge, contains a blood vessel (Fig 30C). The jaw plate exhibits sets of different shapes; some have a central notch in the cutting edge (*K. corallina*, *K. rupestris*, *K. aetheria*), others are simply rectangular (*K. nigra*, *K. tupan*, *K. ajar*, *K. uhlei*, *Koltrora*), while *K. jimenezi* is narrow and arched. The localization of the salivary gland apertures also varies among the species, with some positioned in the middle and others more posterior. Additionally, *K. jimenezi* and *K. uhlei* have these apertures on papillae.

Concerning the odontophore, all species exhibit fused cartilages along the ventral edge, but the degree of fusion varies. *Koltrora* has approximately 100%, while *K. uhlei* has around 90%; *K. nigra* has the smallest fusion value of approximately 60%, and the remaining species are about 75% fused. The intrinsic muscles m4, m5, m7, and m10 also show distinctions among the species (Table 3), while the exclusive *Kora* pair m8 is relatively uniform, except in *K. jimenezi*, where it is relatively reduced. The function of this muscle m8 is unknown, as it only longitudinally covers the dorsal edge of the cartilages (e.g., Fig 4B, 4C).

Genital structures contain a wealth of comparative information. The carrefour region, for example, has a unique arrangement in each species (Figs 6B, 11C, 15A, 20E, 23E, 27E, 33E, 38A, 47C, 53D, 57B). This region, like many others (e.g., odontophore), has been neglected in descriptions and comparative studies, despite being highly informative. The carrefour (fertilization complex) form itself varies, as does its duct; its insertion can be in the duct of the albumen gland (ad) (*K. corallina*, *Neopetraeus*), between this duct and the albumen chamber (*K. nigra*, *K. tupan*, *K. uhlei*), at the tip of the spermoviduct (eo) (*K. rupestris*, *Koltrora*), directly in the albumen chamber (ac) (*K. ajar*, *K. jimenezi*), and between the duct of the albumen gland and spermoviduct (*K. aetheria*). The albumen chamber is a blind sac, in the usual fashion (e.g., Fig 11C: ac), in *K. nigra*, *K. rupestris*, *K. tupan*, *K. aetheria*, and *Koltrora*; however, in the remaining species, *Neopetraeus* included, it has the strange form of a wide, bulged curve (e.g., Fig 57B: ac). A second diverticulum-like projection, usually balloon-like, also filled with albumen and interpreted as an accessory albumen chamber (as), is a novelty in the presently studied species, being absent only in *Koltrora*. Another feature is the presence of a thick muscular region in the basal portion of the duct of the bursa copulatrix (e.g., Fig 6A: bu). It is present only in *Kora*, with the notable exception of *K. jimenezi*.

The characteristics of the penis also exhibit exclusivities for each species, showing weak intraspecific variation in adult forms. This renders the structure relatively reliable for species identification. The relative size (compared to the spermoviduct), the proportion of the epiphallus, and the internal (external during copula) arrangement of folds and structures are all important sources of comparative data (see Table 3). All *Kora* species (except for *K. jimenezi*) and *Koltrora* have the basal region of the penis with very muscular walls (e.g., Fig 20D: mp). Additionally, all *Kora* species have the internal organization of the penis divided into regions, but a pair of main longitudinal folds is always present (e.g., Fig 20D: pf). This is possibly linked to the formation of the long rod of the spermatophore, a characteristic feature of the genus (e.g., Fig 17D). However, details of this pair of main folds (pf) and the arrangement of the remaining regions are exclusive to each species (Table 3). An important feature of some *Kora* species is the transverse umbrella-like fold (e.g., Fig 20: um). This peculiar fold is only found in *K. rupestris*, *K. tupan*, *K. ajar*, *K. aetheria*, and *K. uhlei*. Each of these species also has a different number of rods sustaining the thin fold (Table 3). As no *Kora* mating behavior has been observed so far, the function and even the appearance of this umbrella-like fold during copulation can only be imagined. Another distinctive feature is the insertion of the penis muscle (pm); in most species, it is at the tip of the epiphallus (terminal) (e.g., Fig 20A). However, in *K. rupestris*, *K. ajar*, and *K. uhlei*, the insertion is at the base of the epiphallus, with additional

insertions along the lateral wall of the epiphallus (e.g., Fig 23B: pm). The insertion is subterminal in *Koltrora* (Fig 48C).

Regarding both species described by Pena (2024) [4], they do not belong to any species described herein. *Kora arnaldoi* clearly belongs to what can be called the 'K. rupestris complex,' characterized by species with very elongated shells, and long, conic, and pointed spire. It differs from *K. rupestris*, with which it shares a similar outline, by having a much more intense and coarser axial sculpture. Anatomically, both species further differ in the shape of the jaw plate and the tips of the radular teeth, as discussed below. The paper [4] also described the anatomy of what was identified as *K. rupestris*. However, comparing the shells (that paper: fig 2) with the true *K. rupestris* (Fig 12A–12H) makes it clear that it is a misidentification. The globose shell of that described *K*. cf. *rupestris* [4] resembles that of four species: (1) *K. jimenezi*, but it differs by having an even more globous shell and lacking a multilobed gonad; (2) *K. nigra*, which has a more obese shell, a narrower jaw, a different arrangement of folds on the mantle edge, and lacks the bulged carrefour region; (3) *K. tupan*, which has a much smaller, wider, and smoother shell, a narrower jaw, lacks a prominent bulb in the carrefour, and has a much shorter bursa; and (4) *K. ajar*, which has a smaller and smoother shell, a narrower jaw, and a penis muscle inserted at the tip of the epiphallus (not at its base). Moreover, both species described in that paper [4] have a feature not found in any herein studied species–a subterminal canalicular groove at the tip of each radular tooth (that paper: Figs 4B, 4C and 5E–5G). All of the species studied herein have only a subterminal wrinkle. These data confirm the validity of *K. arnaldoi* and suggest that the reported *K*. cf. *rupestris* may belong to another new species.

## Distribution

As can be seen in Fig 60, all Brazilian species studied in the present paper were collected in areas surrounding the São Francisco River, as it crosses through the states of Minas Gerais and Bahia. This region, primarily north of Minas Gerais up to South Bahia, is characterized by calcareous soil, abundant caves, and a remarkable richness of malacofauna. As depicted on the map (Fig 60), species of *Kora* exhibit a high degree of endemicity, often separated by distances of less than 100 km. They are found on both sides of the river, with an equal prevalence of species on each side. The only species with a broader distribution is *K. rupestris* (indicated by red dots), occurring in an area of approximately 300 km and present on both sides of the river. All species also inhabit regions near the river; the farthest species is *K. curumim*, located approximately 250 km from the river.

Other pulmonates, such as *Habeas* [7, 8], and northern species of *Anthinus* [9], exhibit similar geographical patterns to those observed in *Kora*. The intricacies of the Brazilian region, both in biological and geological terms, are still under investigation—a project to which this paper contributes. It is anticipated that a more comprehensive explanation for the distribution patterns will emerge in the near future. *K. rupestris* stands out as an exception, in which anthropic influence is not excluded.

## Phylogenetic analysis

The three genera studied in this paper resulted in a monophyletic branch (node 10), supported by 10 synapomorphies. Such as the shell with slightly dislocated peristome (character 9) and strong umbilicus going further along columella (ch. 11, 13), higher fusion of odontophore cartilages (ch. 52), the accessory albumen chamber (ch. 70), and the elongation of the epiphallus (ch. 89).

Both *Neopetraeus* species resulted as first branch (node 11), also supported by 10 synapomorphies. The more remarkable are: the ureter ~50% opened (as a groove) (character 25),

which, in the present context, resulted as a reversion; the odontophore m4 and 5 pairs as a single bundle (ch. 44); the insertion of the carrefour in the duct of albumen gland (ch. 68); the prostate fulfilling only ~20% of the spermoviduct (ch. 71); and the oviduct lacking spermoduct folds (ch. 72).

The other branch (node 12) gathers *Koltrora* and a monophyletic *Kora*. The branch is supported by 11 synapomorphies. Such as the shell monochromatic color (character 1); the calcified epiphragm (ch. 16); the folds of the mantle edge (ch. 22); the intercalated pair of vessels ate left from pulmonary vein in lung (ch. 23); the similarities in the radula (ch. 31, 32, 33, 35); the loss of the odontophore pair m11 (ch. 50); the jaw and peribuccal muscles (mj) forming a ventral platform (ch. 51); and the pair of longitudinal folds at least in the basal region of the penis (ch. 84). Conversely, *Koltrora pyrostoma* has nothing less than 13 homoplastic autapomorphies, being 4 of them reversions in the present context; two of them are noteworthy: the loss of the accessory albumen chamber (ch. 70) and of the penis shield (ch. 80). This set of autapomorphies, and the lack of the *Kora* synapomorphies, are sufficient reasons for the generic separations of both taxa, despite they share more characters than *Neopetraeus*. This genus is geographically more distant, but it has the overall shape more similar to *Kora* than *Koltrora* does. But some details of the shell (e.g., the smooth protoconch) and of the anatomy showed that *Koltrora* is closer to *Kora*, but sufficiently different to deserve its own genus. The nearly identical radula between *Koltrora* and *Kora*, in contrast to *Neopetraeus* which exhibits the typical radula patterns of most orthalicoideans (Figs 52 and 55G–55I), appears to be the primary factor influencing the phylogenetic proximity between both genera. Despite *Kora* and *Neopetraeus* sharing an overall shell shape and the auxiliary/secondary albumen chamber, the loss of this latter character is attributed to *Koltrora* in this study.

The *Kora* branch (node 13) has the important support of 25 synapomorphies. Such as: the shell sculpture of axial undulations and spiral micro-stripes (characters 6, 7); the axial cords in end of protoconch (ch. 15); the medial detached branch of secondary columellar muscles (ch. 18); the robustness of the jaw folds (ch. 29); the radular sac widely bulging posteriorly in buccal mass covered by special membrane (ch. 36, 37); the loss of the odontophore horizontal muscle (m6) (ch. 45) and the presence of the pair m8 (ch. 47); the duct to anterior lobe of the digestive gland inserted in stomach, having only right branches (ch. 57, 59); the digitiform seminal receptacle (ch. 62); a confluence of inner folds originating the vas deferens in spermoviduct (ch. 74); the filiform penis internally divided into compartments (ch. 78, 83); and the short cerebral commissure (ch. 94).

*Kora jimenezi*, with its 14 homoplastic autapomorphies, is the first *Kora* branch. It is separated from the other congeners (**node 14**) by 9 synapomorphies. The noteworthiest are: 7–8 insertions of the left, and 6–7 of the right secondary columellar muscles (characters 19, 20); the pair of buccal mass dorso-ventral m3 (ch. 42); the odontophore pair m7 as a single, narrow bundle (ch. 46); the strongly muscular basal region of the walls of the bursa copulatrix duct (ch. 75), and of the penis (ch. 82). The node 14 is divided into *K. rupestris* and **node 15**, which is supported by 4 synapomorphies: pulmonary vein anteriorly branched (ch. 24); the carrefour insertion in duct of albumen gland (ch. 68); the imbricated pair of penial longitudinal folds (ch. 85); and converging folds from vas deferens coming from epiphallus (ch. 91). The node 15 is a dichotomy of 2 branches: nodes 16 and 17. **Node 16** gathers *K. corallina* and *K aetheria*, and is supported by 4 synapomorphies: shell umbilicus widely opened (ch. 14); prostate occupying ~35% of the spermoviduct (ch. 71); penis ~75% of spermoviduct length (ch. 79); and epiphallus ~35% of penis length (ch. 89). **Node 17** has the remaining *Kora*, being supported by 6 synapomorphies, such as jaw plate rectangular (ch. 28); odontophore pair m5 as continuation of the pair m4 (ch. 44); a bulger projection in the carrefour (ch. 67); insertion of carrefour between albumen gland duct and accessory albumen chamber (ch. 68); and vas deferens

inserted in penis strongly curved (ch. 77). This node is divided into *K. uhlei* and **node 18**, which is supported by 6 synapomorphies, such as shell peristome with outer lip superior insertion rather horizontal (ch. 10); mantle edge with pointed secondary folds (ch. 22); penis 85–90% of spermoviduct length (ch. 79); large longitudinal fold in epiphallus (ch. 90). Node 18 includes *K. tupan* and **node 19**, supported by 5 synapomorphies, like loss of the pair m1v (ch. 38); the odontophore pair m5 covering the pair m4 (ch. 44); and the loss of the bulges portion of the carrefour (ch. 67). It is remarkable to observe that all *Kora* species have their considerable set of autapomorphies. Almost all of them are homoplastic, i.e., reversions or convergences with other branches. These set of characters double-reinforce the species distinctions, some of them are included in their diagnoses.

Going more amply, it is important to check that what up to recently was Bulimulidae (**node 3**), resulted monophyletic, supported by 9 synapomorphies. Some of them are a narrow umbilicus in the shell covered by inner lip (character 12); the pair of secondary columellar muscles (ch. 17); the similarity between the radular rachidian, lateral and marginal teeth (ch. 30, 31, 32); the odontophore pair m5 covering the pair m4, inserted in it instead of in the cartilage (ch. 44); the digitiform-cylindric form of seminal receptacle (ch. 62); and the penis shield restricted to the penis base (ch. 80). More recently, based on molecular studies, what was only Bulimulidae now is a set of several families. The present assembly, for example, there is Simpulopsidae (*Rhinus*), Megaspiridae (*Kora*), and truly Bulimulidae (*Bulimulus*, *Anctus*, *Sanniostracus*, *Drymaeus*, *Neopetraeus*) [5, 6]. The presently shown arrangement does not agree with this recent taxonomy, as will be discussed below. All these genera are gathered in the superfamily Orthalicoidea.

Node 3 is divided into *Bulimulus sula* and **node 4**, supported by 9 synapomorfies, such as colorful shell (character 1); 3–4 insertions of the left secondary columellar muscle (ch. 19); posterior location of the salivary aperture in oral cavity (ch. 54); the conic form of the carrefour (ch. 65); insertion of penis muscle at tip of epiphallus (ch. 81); epiphallus ~10–15% of penis length (ch. 89); and nerve ring with pleural ganglia close to pedal pair, and bearing pleural commissure (ch. 92, 93).

Node 4 is a dichotomy, one of its branches is the node 10 already discussed above. The other is the **node 5**, which is supported by 6 synapomorphies. Those more remarkable are: the buccal mass pair of muscles m3 close to esophagus (ch. 42); the stomach as a simple curve, with anterior duct inserted in it (ch. 56, 57); the slightly filiform penis (ch. 78); and the inner penial organization with a special mosaic of low folds (ch. 83). Node 5 gathers *Anctus angiostomus* plus **node 6**, supported by 4 synapomorphies: reticulated protoconch (ch. 15); left secondary columellar muscle with 2 insertions (ch. 19); multicuspid radular marginal teeth (ch. 34); and epiphallus with large longitudinal inner fold (ch. 90). This node, in turn, is divided into *Rhinus botocudus* and **node 7**, having 3 synapomorphies, such as odontophore pair m7 as 2 separated muscles (ch. 46); and uterus wall thick glandular (ch. 73). Node 7 has *Sanniostracus carnavalescus* as a branch, and **node 8** as the other, representing the genus *Drymaeus*, supported by 10 synapomorphies. The more notable are: ureter opened ~15% inside pulmonary cavity (ch. 25); buccal mass pair m2 as a single bundle (ch. 40); loss of m3 and m11 (ch. 42, 50); odontophore cartilages ~50% fused (ch. 52); pair of salivary glands forming a single mass (ch. 53); prostate occupying ~35% of the spermoviduct (ch. 71); penis shield with ~1/3 of the penis length (ch. 80); and a short cerebral commissure (ch. 94). The 3 species of *Drymaeus* (node 8) have an internal arrangement, with *D. castilhensis* closer to *D. micropyrus*, in the **node 9**, supported by a single synapomorphy (insertion of the carrefour in the duct of the albumen gland–ch. 68), separating them from *D. currais*.

Going further ampler, two more remote outgroups resulted, as expected, outside the orthalicoideans, in the **node 2**. This node gathers together the Solaropsidae *Olympus nimbus*

(superfamily Sagdoidea) [6], and the Achatinidae/Subulininae *Lavajatus moroi* (Achatinoidea). The node is supported by 3 synapomorphies: Insertion of hermaphrodite duct in the tip of seminal receptacle (character 64); the lack of the albumen chamber (ch. 69); and the bursa copulatrix shorter than 25% of the spermoviduct (ch. 76). Of course, it is important to keep in mind that, as these taxa are not the focus of the present paper, the survey of characteristics is not as detailed as in the orthalicoideans. Anyway, both being reunited in a branch (node 2) is a surprise. This arrangement serves at least as first step of an investigation to define Orthalicoidea phylogenetically based on phenotypic features, what is still missing.

The genus *Olympus* Simone, 2010, has been considered synonym with *Solaropsis* Beck, 1837 [6]. It is appropriate to clarify a taxonomic misconception in this case, which is based on a molecular approach [32]. In the referenced paper, *Olympus* is placed in a final polytomy in the main cladogram of that paper (see [32] fig 1), mixed with species of *Solaropsis* and *Psadara* Miller, 1878. At the base of this polytomy, two species of Solaropsis precede it. The authors, based on this result, synonymized all genera, rendering Solaropsidae monogeneric. However, this overlooks considerable phenotypic differences between *Solaropsis* and, at least, *Psadara* and *Olympus* [13, 19]. The premature consideration of all these genera as synonyms resulted from the low resolution of the cladogram, the non-inclusion of the type species of *Psadara* and *Solaropsis*, and a disregard for the important set of morphologic differences among the three genera [19]. The synonymy can be easily challenged after a total resolution of the polytomies, with the inclusion of the type species, and a supraspecific taxonomic rearrangement of the species. Therefore, for taxonomic stability, a more conservative approach should be taken in maintaining the validity of all three genera until a better resolution is achieved, along with a deeper analysis of the generic differences, which is still lacking. Possibly, after a better resolution, more species can be added to *Olympus*, instead of being an invalid genus belonging to a large monogeneric family. Confirming that, a recently described species, *Solaropsis penthesileae* Salvador, 2021 [33] has the shell very similar to *O. nimbus*, and the molecular analysis also resulted both species as sister taxa ([33]: fig 1). Certainly *S. penthesileae* is an *Olympus*. Applying these justifications, *Olympus* is maintained valid in this paper.

## Brief comparison with molecular approaches

While the primary focus of this study pertains to the current assembly of studied taxa, the broader range of included taxa in the phenotypic phylogenetic analysis allows for some inferences and comparisons with molecular studies. Molecular approaches have become much more prevalent in recent times, nearly rendering other methodologies obsolete. Conversely, these approaches seem to evolve without apparent concern for anchoring the resulting taxa with recognizable morphological attributes. There is little regard for prior arrangements, whether based on morphological or molecular datasets, as the results of each published study rarely align fully with those of previous ones.

For instance and example, within the orthalicoideans, the positioning of *Drymaeus* in relation to *Bulimulus* and *Bostryx* is subject to variation. In some studies, *Drymaeus* is found to be closer to *Bulimulus* than *Bostryx* [34]; in others, *Bostryx* is closer to *Bulimulus* than *Drymaeus* [35]. At times, *Drymaeus* is identified as the sister group to a combined cluster of *Bostryx* and *Bulimulus* species [5]. With such fluidity, maintaining taxonomic stability becomes challenging, and it poses a significant obstacle to achieving mental coherence in the classification trustworthiness.

A recent study [5] marks a significant milestone in the field of orthalicoideans, both in terms of the extensive range of studied taxa and the profound taxonomic changes it introduces. That paper provides a comprehensive history and analysis of previous molecular studies, many

of which were conducted by the authors themselves. Among the species examined in that study is *Kora rupestris*, which emerged as a monophyletic branch with *Thaumastus* positioned at the base of a larger branch containing other orthalicoideans. The authors connected this branch with a preceding one that encompasses *Megaspira*, and coined the term 'Megaspiridae' to describe this recognized paraphyletic group. However, 'Megaspiridae' appears to be a very heterogeneous taxon, as the three genera share little beyond being snails. *Megaspira* was previously associated with Achatinidae/Subulininae due to its turriform shell until a recent reevaluation, with its transfer to Orthalicoidea being solely based on molecular evidence [34], as its anatomy remains unknown.

*Neopetraeus tessellatus* is also featured in the aforementioned study [5: fig 6]. It emerged as the first branch of *Drymaeus*, alongside *D. expansus*. In that study, the authors proposed placing both genera, along with three others, in the new subfamily Peltellinae within Bulimulidae. Therefore, the proximity observed between *Kora* and *Neopetraeus* in this analysis, supported by 10 synapomorphies (Fig 61: node 10), is not corroborated by the findings in that particular paper. In this study, the representative of *Bulimulus* was identified as the first branch in the orthalicoidean lineage (node 3), while *Drymaeus* emerged as the last branch (node 8). In contrast, ([8]: fig 6) depicts a very different scenario, wherein both genera are practically sister taxa in an almost terminal branch.

In the present paper, between *Bulimulus* and *Drymaeus*, the genera *Anctus*, *Rhinus* and *Sanniostracus* are successively allocated (Fig 61: nodes 5–7). In that paper [5], *Bulimulus*, *Sanniostracus*, *Drymaeus* and *Anctus* are Bulimulidae, while *Rhinus* is Simpulopsidae. That arrangement is not compatible with the present one, as a simpulopsid is allocated between bulimulids, in a paraphyletic arrangement.

The discrepancies observed in phenotypic and molecular arrangements, as well as among different molecular studies, could potentially be attributed to the limited morphological framework in Mollusca when compared to other zoological groups such as vertebrates and arthropods [30]. This highlights the urgent need to promote research in comparative morphology, parallel to the progress seen in molecular studies.

## Main evolutive processes

Orthalicoideans and strophocheiloideans constitute the most speciose groups among South American land snails [14]. However, they may not be closely related, as they are separated at the infraorder level [6], with the former belonging to Orthalicoidei and the latter to Rhytidoidei. According to a previous study [9], the phylogeny revealed a clear trend among strophocheilids to increase in body size, consume a wide range of vegetal matter, and reproduce through a strategy of few and large eggs, interpreted as adaptations of the digestive and genital systems.

In contrast, Orthalicoideans maintain a plesiomorphic pattern in terms of size and egg-laying, but different genera and species exhibit significant meal specializations. They are generally challenging to maintain in captivity and display a wide range of digestive conformations, particularly in buccal structures and radula. Nevertheless, node 3 (Fig 61) presents, for the first time, a set of synapomorphies for Orthalicoidea. While this is preliminary, given that several important subgroups are still missing, it serves as an initial step and provides a framework for future studies at a similar level of detail.

The first orthalicoidean branch is *Bulimulus*, characterized by a fragile, featureless shell. This contrasts with its sister branch (node 4), where shells are predominantly colorful and thick-walled. Additionally, there is an increment in the epiphallus, with the penis muscle connected to it rather than being inserted into the penis body, along with the appearance of a commissure of the pair of pleural ganglia.

This branch (node 4) exhibits a dichotomy, giving rise to nodes 5 and 10. Node 10 encompasses all species focused on in this paper, predominantly characterized by large, thick-walled shells, a robust peristome, and a notable umbilicus that forms the tip of a hollow fold running along the columella. The internal structure reveals the presence of an accessory albumen chamber, although the mechanism of egg laying in these animals remains elusive.

On the other branch (node 5), a mix of bulimulids and simpulopsid is observed. This branch demonstrates a trend towards an increase in midgut size, with specializations in the gastric region, and a simplistically organized yet highly elongated penis.

Despite being preliminary, this marks the first attempt to construct an orthalicoid phylogeny based solely on phenotypic features—a crucial initial step toward a broader understanding. This framework will serve as the foundation for ongoing studies encompassing other branches within this superfamily. Remarkably, this paper introduces newly discovered structures, including the odontophore pair of muscles (m8), resembling cartilage tensors, so far exclusive to *Kora*. The accessory albumen chamber, shared by *Kora* and *Neopetraeus*, is noteworthy, with a similar feature found in *Sanniostracus carnavalescus* [17]. Additionally, a distinct model of spermatophore (e.g., Fig 8F, 8G) with a multifolded tubular basal rod has been identified in certain *Kora* species. The calcified epiphragm, commonly found in non-South American snails (e.g., helicinids), is practically absent on this mainland, except for instances in *Kora* and *Koltrora*. The unusual radula conformation observed in these two genera further contributes to the morphologic and taxonomic novelties presented. These examples underscore the gaps in our knowledge of South American land snails, emphasizing the need for further comprehensive studies.

## Environmental and conservation comments

The region adjacent to the São Francisco River, which flows south to north along the eastern side of Minas Gerais and Bahia states in Brazil, is rich in calcareous substrate. This type of soil is rare in Brazilian territory, which is mostly composed of acidic soils where continental mollusks struggle to thrive [35, 36]. The calcareous region is relatively highly exploited for agriculture but has an almost complete absence of malacological studies. Recent expeditions to this area have revealed a high diversity of mollusk species, significantly higher than in other Brazilian ecosystems. This has been demonstrated in the present study and a few other recent ones [1–4, 8, 9]. In those studies, several new taxa, species, and genera have been introduced.

As this new scenario is only beginning to be understood, and since the São Francisco River region is relatively well-exploited commercially, additional studies on the conservation of regional ecosystems and endemic species are urgently needed. As demonstrated by the present study and others mentioned herein, most species in that region exhibit restricted distributions, confined by rivers, valleys, and mountains. The detected endemicity and limited geographic distributions already highlight the vulnerability of these newly discovered species. Moreover, the number of species yet to be discovered can only be imagined. Additionally, this region contains clusters of caves along it, which provide further habitats for endemic mollusks.

Papers like this one have the additional goal of serving as a persuasive factor for environmental preservation, as type localities are protected by Brazilian law. The study of local fauna is, therefore, fundamental for supporting such protective policies.

## Supporting information

**S1 Appendix.**
(DOCX)

**S2 Appendix.**
(DOCX)

## Acknowledgments

I am especially grateful to the people that sponsored the collecting expeditions and donated the samples to MZSP, particularly José Coltro Jr, Mauricio Uhle, Weslley Vailant-Mattos, and David Jimenez. Thanks to Frederico Gutierrez for the kind donation of both Peruvian *Neopetraeus* samples studied herein. To Andrea Salvador, NHMUK, I thank for permission to publish the photo of the *N. lobbi* lectotype. For the SEM examination, I thank to Lara Guimarães, Natan Carvalho Pedro, and Fernanda Santos Silva (MZSP).

## Author Contributions

**Conceptualization:** Luiz Ricardo L. Simone.

**Data curation:** Luiz Ricardo L. Simone.

**Formal analysis:** Luiz Ricardo L. Simone.

**Funding acquisition:** Luiz Ricardo L. Simone.

**Investigation:** Luiz Ricardo L. Simone.

**Methodology:** Luiz Ricardo L. Simone.

**Project administration:** Luiz Ricardo L. Simone.

**Resources:** Luiz Ricardo L. Simone.

**Software:** Luiz Ricardo L. Simone.

**Supervision:** Luiz Ricardo L. Simone.

**Validation:** Luiz Ricardo L. Simone.

**Visualization:** Luiz Ricardo L. Simone.

**Writing – original draft:** Luiz Ricardo L. Simone.

**Writing – review & editing:** Luiz Ricardo L. Simone.

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
