## [Decision Letter · Decision Letter 0]

21 May 2024

PONE-D-23-41760Review of the genus Kora
from Brazil, with description of eight new species and a new related genus Koltrora,
including comparison with two Andean Neopetraeus species (Gastropoda, Eupulmonata,
Orthalicoidea).PLOS ONE

Dear Dr. Simone,

Thank you for submitting your manuscript to PLOS ONE. After careful consideration, we
feel that it has merit but does not fully meet PLOS ONE’s publication criteria as it
currently stands. Therefore, we invite you to submit a revised version of the
manuscript that comprehensively addresses the points raised during the review
process. **Note
that all three reviewers have voiced serious concerns with the study in its
current form. To be acceptable for publication in PLOS ONE, the manuscript will
thus have to be carefully revised, which will require novel analyses and
extensive rewriting of the manuscript text. Specifically, serious flaws have
been identified with the general design of the study, in particular concerning
the phylogenetic section of the paper, as well as with the usage of English
language in the manuscript text.**

Please submit your revised manuscript by Jul 05 2024 11:59PM. If you will need more
time than this to complete your revisions, please reply to this message or contact
the journal office at plosone@plos.org. When
you're ready to submit your revision, log on to https://www.editorialmanager.com/pone/ and select the 'Submissions
Needing Revision' folder to locate your manuscript file.

Please include the following items when submitting your revised
manuscript:A rebuttal letter that responds to each point raised by the academic
editor and reviewer(s). You should upload this letter as a separate file
labeled 'Response to Reviewers'.A marked-up copy of your manuscript that highlights changes made to the
original version. You should upload this as a separate file labeled
'Revised Manuscript with Track Changes'.An unmarked version of your revised paper without tracked changes. You
should upload this as a separate file labeled 'Manuscript'.

If you would like to make changes to your financial disclosure, please include your
updated statement in your cover letter. Guidelines for resubmitting your figure
files are available below the reviewer comments at the end of this letter.

We look forward to receiving your revised manuscript.

Kind regards,

Michael Schubert

Academic Editor

PLOS ONE

2. Please take this opportunity to be sure you have met all of our guidelines for new
species. For proper registration of a new zoological taxon, we require two specific
statements to be included in your manuscript.

a. In the Results section, the globally unique identifier (GUID), currently in the
form of a Life Science Identifier (LSID), should be listed under the new species
name, for example:

Anochetus boltoni Fisher sp. nov.
urn:lsid:zoobank.org:act:B6C072CF-1CA6-40C7-8396-534E91EF7FBB

Another LSID for the manuscript itself should also appear within the Nomenclature
statement. You will need to contact Zoobank (zoobank.org/About) to obtain a GUID (LSID). You should receive one
LSID for your manuscript and a separate, unique LSID for the new species. 

b. Please also insert the following text into the Methods section, in a sub-section
to be called "Nomenclatural Acts":

The electronic edition of this article conforms to the requirements of the amended
International Code of Zoological Nomenclature, and hence the new names contained
herein are available under that Code from the electronic edition of this article.
This published work and the nomenclatural acts it contains have been registered in
ZooBank, the online registration system for the ICZN. The ZooBank LSIDs (Life
Science Identifiers) can be resolved and the associated information viewed through
any standard web browser by appending the LSID to the prefix "http://zoobank.org/". The LSID for this publication is:
urn:lsid:zoobank.org:pub: XXXXXXX. The electronic edition
of this work was published in a journal with an ISSN, and has been archived and is
available from the following digital repositories: PubMed Central, LOCKSS [author to
insert any additional repositories].

All PLOS ONE articles are deposited in PubMed Central and LOCKSS. If your institute,
or those of your co-authors, has its own repository, we recommend that you also
deposit the published online article there and include the name in your article.

Following a recent ruling by the International Commission on Zoological Nomenclature,
electronic journals are now a valid format for publication of new zoological taxa.
In order to ensure the valid publication of your new species, please be sure to
include the updated version of Nomenclatural Acts (above). A complete explanation of
our guidelines for publishing new species can be found on our website: http://www.plosone.org/static/guidelines#zoological.

4. We note that Figure 60 in your submission contain map images which may be
copyrighted. All PLOS content is published under the Creative Commons Attribution
License (CC BY 4.0), which means that the manuscript, images, and Supporting
Information files will be freely available online, and any third party is permitted
to access, download, copy, distribute, and use these materials in any way, even
commercially, with proper attribution. For these reasons, we cannot publish
previously copyrighted maps or satellite images created using proprietary data, such
as Google software (Google Maps, Street View, and Earth). For more information, see
our copyright guidelines: http://journals.plos.org/plosone/s/licenses-and-copyright.

We require you to either present written permission from the copyright holder to
publish these figures specifically under the CC BY 4.0 license, or remove the
figures from your submission:

a. You may seek permission from the original copyright holder of Figure 60 to publish
the content specifically under the CC BY 4.0 license.  

Reviewers' comments:

Reviewer's Responses to Questions

**Comments to the Author**

1. Is the manuscript technically sound, and do the data support the conclusions?

Reviewer #1: No

Reviewer #2: Yes

Reviewer #3: Partly

2. Has the statistical analysis been performed
appropriately and rigorously? 

Reviewer #1: No

Reviewer #2: N/A

Reviewer #3: No

3. Have the authors made all data underlying the
findings in their manuscript fully available?

Reviewer #1: Yes

Reviewer #2: Yes

Reviewer #3: Yes

4. Is the manuscript presented in an intelligible
fashion and written in standard English?

Reviewer #1: Yes

Reviewer #2: No

Reviewer #3: No

5. Review Comments to the Author

**Reviewer #1:** This work contains very detailed anatomical studies of what
is supposed to be a number of species of the genus Kora It contains descriptions of
several new species and a new genus. Relationships of this taxa are investigated by
analyses based on morphological characters; I believe the author calls these
'phenotypic analyses'.

I appreciate the exceptionally detailed documentation of anatomical/morphological
features of the studied snails, which provide a large dataset. I disagree with the
main conclusion of the author to have identified 8 new species and one new genus.
Let me try to explain why ( i will use a single example, but I believe it to be
representative of the general assessment of data in this work, which I reject)

This is an unnecessarily complex study, which to understand is challenging, because
the reader is overwhelmed with large amounts of unimportant or uninformative data
while at the same time relevant information is often withheld or has not been used.
My review would need to be as long as the manuscript because nearly every sentence
contains questionable statements, conclusions, misleading formulations, and a
mishmash of unnecessary or confusing information. Sifting through all this to
distill the objective truth and separate that from the interpretation by the author
is too cumbersome and time consuming a task. Sorry, I can't. My live is too short
... So, I need to give a rather generic evaluation.

The English language is understandable, but it is often grammatically or
stylistically imperfect. This does usually not impede the ability to understand the
facts, but clarity suffers to some degree.

I've contemplated commenting on language but then I decided generally not to comment
on linguistics because I do not have time for that.

I try to summarize the findings of the study:

Snails collected at 12 different sites (locations: see map in Fig. 60) have been
studied and detailed recordings of anatomical characteristics have been made and
collated in a data matrix. The data matrix was analyzed, and the investigated
samples have been arranged in a tree shared characteristic (a presume some kind of a
cladistic analysis). So, relationships between examined samples have been
hypothesized based on shared characters. Equally, differences between species can be
understood as diagnostic characters that underpin their distinctiveness.

My first criticism is that many recorded characters are of dubious systematic value.
Now, based on my understanding of cladistics (not sure if cladistic methods were
used here) is that it may not matter if some characters are uninformative, but that
an analysis of all characters simultaneously will still return a plausible
hypothesis on the degree of similarity / dissimilarity between the examined samples.
I accept that.

My conclusion then is that irrespective of the possible flaws in some of the
characters (as they have been recorded and coded), the tree shown in Fig. 61 is a
plausible hypothesis of the real relationships between the examined samples as based
on morphology.

Great!

Looking at this tree, I find that the sample identified as - for example - Kora ajar
is most closely related to the sample identified as Kora nigra based on many shared
characteristics, some of which may be synapomorphies (shared only by these two
samples). Let's dive into the details:

One supposed synapomorphies of K.nigra+K.ajar is:

4 ("Length/width ratio of shell = 1.5) as opposed to the other two possible character
states (2 and 3).

There is the first problem. The author has grouped continuous characters, such as
size and shape variables, into classes. I think this is wrong and creates
potentially misleading results. For example, for all Kora species the author
classifies shells as either in the 30 mm range or the 45 mm range (Table 2). Thats
so strange! Why not use real measurements? Firstly, the individuals of a species are
not all identical in size. So, species will occupy a size range. I find it hard to
believe that all species in a genus should be clustering around two arbitrary size
classes, 30 and 45 mm each. If that was the case, I think, the most likely
interpretation is that across all measured shells, we can identify two groups, not
12! In this context I also criticize the small number of measured individuals. The
author has used measurements of 3 to 5 individuals per supposed species. That number
is insufficient to capture the entire range of variation in a species especially as
it appears that individuals of each species have only been found at single sites
each. Again, it puzzles me why the author hasn't measured more shells. For example,
106 shells have been 'examined' of K. corallina, but 5 have been measured only. So,
what does 'examined' actually mean? Looked at?

I conclude that all size-related shell characters are insufficiently sampled and
documented.

Back to K.nigra+K. ajar.

The next supposed synapomorphy is:

14 (Umbilicus = "widely opened" ['wide' would be sufficient wording]) as opposed to
alternative character "closed to narrow".

Strangely, umbilical anatomy is coded in three different characters 12 to 14 - which
in my view record partly redundant features. Why not simply say "Umbilicus:
0=closed, 1=narrow, 2=wide" . Again, the problem is that a continuous character is
classified. When exactly is the umbilics narrow and when is it wide? This is a
completely subjective assessment. Looking through the photographs of shells, the
umbilicus is often not shown or hard to recognize, so I have to rely on the author
here. However, umbilical anatomy is often variable within a single species and
individuals of the same species may have closed to open umbilici.

Next character:

38 (pair m1l - some buccal muscle [one out of 10]: absent). As opposed to present in
all other Kora species and all outgroup taxa. Really? This is such a freaky detail.
I have difficulties to believe that this is a real character. What is the likelihood
that this pair of muscles is not absent but has not been observed for some other
reason? Buccal muscle anatomy is highly complex and intricate, and I am not even
sure that all these 10 muscles that the author proposes actually exist. I consider
this a dubious character, one where I have to have blind trust in the observations
of the author, which are also not sufficiently documented.

Next character:

44 (m5 - 'covering m4'); as opposed to alternative character states 'separated from
M4' and 'continuation of m4'). Yet, another buccal mass muscle... same applies to
what I have said on character 38 - this is an incredible detail on a very complex
and intricate structure, where I consider that observations are impossible to verify
while having a high possibility of subjective error.

Next character:

67 (Carrefour bulged distal region by side of receptacle: 0= absent). As opposed to
'present'. Really? The difficulty with saying that there is a bulge next to the
receptacle in the carrefour is that all the internal organs are soft tissues that
are shaped by muscle contraction, preservation, dissection. I would doubt that this
is a reliable character at all.

One cannot dissect the reproductive organs of a dead, preserved snail, spread them
out in a petri dish and treat a little bump somewhere as a plausible anatomical
feature! Anyway, not while looking at probably just a handful of specimens. This is
nonsense!

SO, the aim of my exercise is that large parts of the information that 'inform' the
analysis of relationships are spurious characters. These characters may be partly
prone to error due to the complexity and intricacy of the structures studied. Many
other characters are probably unsuitable to underpin systematic conclusions as they
likley represent just variations on the level of the individual.

Let's be clear, we do not study humans this way because it makes no sense. I cannot
use hair colour, skin colour, nose shape, penis size or whatever to study the
relationships of human populations. The reason is that the observable variability in
any of these characters is intraspecific. I could still go on and create a tree, but
the tree would be an artifact of incomplete character sampling. The same applies to
the snails. Many characters, which have been meticulously recorded are
uninformative. I went through the whole list of characters, and I consider my
statement generally true. So, the tree is at best hypothetical.

Nonetheless, let's say the tree shown in fig. 61 is correct.

There is no reason whatsoever to treat the tips of the tree as species. Basically,
the tips of the tree represent individuals. The tree is entirely phenetic and
records the anatomy as observed by the author. However, there is no way to tell
where the species boundaries are in this tree. Essentially, what we found is that
there are 12 individuals (or groups of individuals - but the author said that the
individuals of each 'species' are so similar that there is virtually no variation
between them, so they have all been merged into single figures). These could all be
members of the same species! Or they could represent anything from between one and
12 species. To decide how many species there may be requires criteria (basically one
has to conclude how much variation one allows within a species). The author has
decided to not allow any variation in a species.

That means the authors is a typologist.

Typology is not science because it rejects the existence of variation. But variation
is the fundamental prerequisite for evolution to happen. All the so-called species
examined herein are all descendants of a single ancestor, and over time have
acquired a certain amount of variation. But variation has to be there in the first
place and many of the ancestral variation is likely shared by more closely related
species. To not accept reasonable amounts of variation is unnecessary and unhelpful
as it denies the most fundamental condition of evolution.

I consider that the author has just meticulously (and to a large extent
unnecessarily) recorded random individual differences and shown how different
individuals can be lined up with respect to the degree of similarity. To decide
where the species limits in this tree are, one would need to try to group tips
together and examine where two groups differ from each other in certain characters
with statistical significance. Shell characters could be the easiest to use for
that, but the author has not attempted to find a more objective way of analyzing
shell characters (using real measurements rather than artificial size classes).

In biology, species are not the smallest recognizable units. The smallest
recognizable unit is the individual. Individuals form populations (already a
difficult term which we fail to properly define and delimitate) and species consist
of populations. To delimit species correctly, is not to show that two individuals
differ, but to show that populations differ in their means of certain characters.
Even the sheer existence of differences in itself is not a good argument for
specific distinction, in as much as the different mean hair colour of Germans and
Ethiopians is not a good argument for specific distinction either! In fact, there
needs to be a criterion that the differences are indicative of some sort of
reproductive or genetic isolation.

The author has stopped at documenting differences between individuals. He has not
done the second and third task necessary to delimit species in a plausible and
scientific way.

Of course, he can still describe these species because the rules of nomenclature do
not care for science. He'd be in very good company as creating meaningless species
descriptions is a common occupation especially in malacology. However, while doing
this we fail to adequately document biodiversity while simply inflating taxonomic
catalogues that become increasingly useless records of biological diversity for
their exceedingly high subjectivity and extremely limited reproducibility.

Sorry for rambling, back to the data. I tried to show what I perceive to be the main
flaw of the morphological analysis. The manuscript also contains species
descriptions. Yet, the challenge is to find the few informative characters in the
swamp of generic descriptions. Honestly, it is confusing and unnecessary to describe
the entire anatomy of an organism in the context of a systematic treatment.

Perhaps describe the anatomy in more detail at the genus level but restrict the
species descriptions to the informative bits! Here you need to document the range of
variation, you cannot simply treat species in a typological manner as if there was
no variation. If this is the case, your so-called species is not a species! It's
simply a single individual or a small group of individuals that are closely related
with each other. Given that all species are closely related it is to be expected
that they share many similarities, so please avoid describing the same all over
again if it's just generic.

What is a 'distinctive description'? I have no idea. Your descriptions are
uninformative. Where you observe differences with other species, you make
statements, such as ", remarks and distinctions following" Where? It is completely
impossible to follow the logic of your comparisons. Why don't you provide all the
relevant information under each species instead of referring the reader to another
part of the paper? The descriptions have similar statements of over the place, such
as "general structures similar to preceding species" - look, it's not helpful nor
necessary to swamp the few diagnostic characters in an ocean of uninformative
descriptions.

Thankfully you have a Table 2. However, corresponding to what I said already, many of
the features that are stated to be diagnostic for species, based on my own
experience as a land snail taxonomist, are likely more variable within a species
than allowed by you. Based on three existing 'size classes' (if these are actually
real), you may deal with just three species. Many of the characters listed are
likely variable within a single species (bumps here and there, umbilicus closed-open
etc etc). Whatever the matter, you need to demonstrate the suitability of characters
that you propose to be taxonomically informative instead of simply amassing a lot of
documented variation.

Bulk variation may be informative regardless of whether I consider a feature to be
possibly informative, or not, but only if you have analyzed a large number of
individuals from more than a single location simply because you need to sample the
existing variability in each character sufficiently before you can try to recognize
groups in terms of different character distribution. What you do is typology (a
comparison of individuals, with each individual being treated as the essence of a
species). These species exist in your imagination only. To get them out into the
real world as scientifically tested hypotheses requires proper statistical analyses
of objective variation.

I have made some comments in the manuscript file, but not consistently because I got
tired. For what's worth check them to find some contentious lines.

These comments are by no means complete.

**Reviewer #2:** This is a tremendous piece of work and a benefit to
Malacology. I recommend its publication, however, the writing is too difficult to
understand without substantial effort and there should be a review of the English
phrasing and grammar to make sure that it is as clear as possible. I would recommend
a revision of the writing followed by another round of review. At present certain
parts are difficult to give a fair review because it isn't entirely clear what the
author is trying to say. Please see the edits and suggestions in the version of your
manuscript attached. I did not attempt to revise all of the English in the paper but
made my best attempt in the first 1/3.

**Reviewer #3**: The new species descriptions and morphological work
presented in this manuscript are excellent and nicely figured. This part of the
manuscript is generally clear and well written and is a very important addition to
our knowledge of this group. I would very much like to see this portion of the
manuscript published. My only problem with this section is that many of the species
comparisons that should be part of the systematics section are presented in the
Discussion section making direct species comparisons more difficult.

The phylogenetic section is much less focused and not very complete. The taxon
sampling for the phylogenetic section is too incomplete to make the comparisons this
manuscript attempts to make and leads to comparisons which seem to make less sense.
Neopetraeus is portrayed as sister to or closely related to Kora and Koltrora, but
most of the taxa needed to judge this are missing. It is not clear to me why Kora
and Koltrora are compared to Neopetraeus rather than to Thaumastus. Pena et al. 2011
published at least some anatomical data for Thaumastus, which falls within the same
clade as the target taxa in phylogenies generated with sequence data. Why was this
group not included? The choice of outgroups also seems unnecessarily distant from
the group of interest. The discussion also spends too much time talking about
relationships within one of the outgroups which are not relevant to this
manuscript.

The focus of the manuscript seems very narrow, especially for a journal with a
diverse audience. I would like to see a little more information about the importance
of the discovery of so many new species in one group. I see these discoveries as
suggesting the group is diverse and relatively poorly sampled. In fact, a second
paper describing a new species of Kora was published while this paper was being
reviewed. Sampling is desperately needed. Unfortunately, land snails also have some
of the highest extinction rates of all animals, making species discovery imperative
while there is still a chance for conservation actions. I would like to see some
mention of this added to the manuscript.

As mentioned above a paper describing a species of Kora was published very recently
and reference to it will need to be added to this manuscript: Pena 2024. Notes on
Kora and description of a new species from Minas Gerais, Southeast Brazil (Mollusca:
Gastropoda: Stylommatophora) Zoologia 41: e23059 https://doi.org/10.1590/S1984-4689.v41.e23059 .

The manuscript needs some proofreading to improve English, clarity and focus. See
some examples below along with other minor comments:

Page 2, paragraph 2: line 1, delete ‘the’ before ‘recent’; line 3, delete ‘truly’;
line 4, ‘lands nails’ should be land snails; line 7, ‘stophocheilid’ should be
strophocheilid.

Page 2, paragraph 3: line 2, change to the four described species including the type
species,; line 4, add ‘are’ between ‘which also’. The second half of this paragraph
needs to be edited for clarity and conciseness. Is the information about lost
specimens that were studied by deceased workers that never published on them
necessary?

Page 3, line 21, change exteriorized to everted and ‘the copula’ to ‘copulation’.

Page 4, line 3-4, this sentence is not clear. What do scenario, style, model and
disposal refer to?

Page 6, beginning of the Systematics section. Include some higher taxonomic
levels:

Class Gastropoda Cuvier, 1795

Order Stylommatophora A. Schmidt, 1855

Superfamily Orthalicoidea E. von Martens, 1860

Family ? Give family and/or explain ambiguity or why you are not giving the
family.

Page 14, Distinctive Redescription. Not sure what Distinctive means here. Why not
just Redescription? Also ‘Proper description in [3]. Complement:’ is not clear to
me. I think 3 is a reference to the original description. This might be clearer as:
‘Complement to original description [3]:’.

Page 15, line 11, can delete ‘(as)’.

Page 16, first paragraph, penultimate sentence is not clear.

Page 31, last line. Local should be locality.

Page 32, Measurements section. Which measure is which, add height, width or some
other explanation.

Page 32-33, This new genus is similar to Kora. Make clearer how you would
differentiate it from Kora.

Page 33, Etymology, change contributed ‘with’ to ‘to’. Change ‘collects’ to
‘collections’ and ‘donation’ to donations; or whole phrase to ‘by collecting and
donating material’.

Page 43. The first paragraph of the Discussion describes how the previous
descriptions were written. This is too late since the reader has already read the
descriptions. Move this to the Methods section.

Page 44. Notorious has a negative connotation. I think you mean notable.

Page 49. The last two paragraphs are not directly related to this paper and can be
deleted.

Page 52. Acknowledgements. Change ‘the collect expedition’ to collecting expeditions
or the collecting expedition, depending on if singular or pleural.

Figure 60 caption. Self produced is not necessary.

Appendix 1 title is not clear, could edit to List of characters used in the
phylogenetic analysis

Appendix 2 title is not clear, could edit to: Matrix of characters used in the
phylogenetic analysis

In figure 61 Kota [sic] carolina should be Kora carolina.

6. PLOS authors have the option to publish the peer
review history of their article (what does this mean?). If published, this will
include your full peer review and any attached files.

If you choose “no”, your identity will remain anonymous but your review may still be
made public.

**Do you want your identity to be public for this peer review?** For
information about this choice, including consent withdrawal, please see our
Privacy Policy.

Reviewer #1: **Yes: **Frank Koehler

Reviewer #2: No

Reviewer #3: No

---

## [Author Response · Author response to Decision Letter 0]

8 Jun 2024

In the annexed document "Answers to reviewers" I have provided responses and
additional explanations for each comment made by the reviewers in RED. I have sent a
completely reformulated manuscript based on their criticisms and suggestions, which
have significantly improved the document, and I am very grateful for their
input.

As you can see in the attached documents, I addressed every point highlighted by the
three reviewers. The vast majority of their suggestions were accepted and
incorporated into the revised manuscript. For the few points where there was
disagreement, I have provided detailed explanations. Even in these cases, most of
the text was modified to some extent.

Overall, the paper has been revised in almost every paragraph. Its language has been
double-checked by colleagues and AI tools to ensure no significant errors remain.
The phylogenetic section was fully accepted by two of the reviewers without any
criticisms, and I have chosen to follow their recommendations. Although I seriously
considered the criticisms from Reviewer 1, the only one to address this section,
their requirements were impractical to implement, as explained in that document. 

It is important to emphasize that the phylogenetic methodology applied in this paper
is widely used. This methodology has been applied in all my previous papers,
including those published in, e.g., Plos One and Cladistics. The current paper
provides comprehensive information for verification, including the data matrix, the
list of characters that generated it, the programs used to process it, the resulting
cladogram (which can be reproduced using the same matrix and programs), and the
synapomorphies supporting each node (Fig. 61) that are further discussed. The
methodology follows the classic Hennigian approach, which has been used for more
than half a century, ensuring clarity and transparency.

Abstracting, practically everything was accepted and corrected upon reviewers
comments, with the exception of the collapse of the taxonomic and phylogenetic
conclusions, which was only addressed by referee 1. However, this issue is properly
and thoroughly explained.

---

## [Decision Letter · Decision Letter 1]

3 Sep 2024

PONE-D-23-41760R1

Review of the land snails of the genus Kora from Brazil, with description of eight
new species and a new related genus Koltrora, including comparison with two Andean
Neopetraeus species (Gastropoda, Eupulmonata, Orthalicoidea).

PLOS ONE

Dear Dr. Simone,

Thank you for submitting your manuscript to PLOS ONE. After careful consideration, we
have decided that your manuscript does not meet our criteria for publication and
must therefore be rejected.

Specifically, it was noted during previous rounds of peer review that the design of
the study is flawed and that there are serious issues with the usage of English
language in the manuscript text. Based on the latest reviewer reports these
shortcomings have not been addressed sufficiently to improve the overall quality of
the work. The manuscript does thus not meet publication criteria 3 and 4 of PLOS
ONE in that that the analyses are not performed to a high technical standard and
that the conclusions are not supported by the reported data. In addition, given the
shortcomings with English language usage, the manuscript text does not adhere to
publication criterion 5 of PLOS ONE, stating that the article needs to be presented
in an intelligible fashion.

I am sorry that we cannot be more positive on this occasion, but hope that you
appreciate the reasons for this decision.

Kind regards,

Michael Schubert

Academic Editor

PLOS ONE

Reviewers' comments:

Reviewer's Responses to Questions

**Comments to the Author**

1. If the authors have adequately addressed your comments raised in a previous round
of review and you feel that this manuscript is now acceptable for publication, you
may indicate that here to bypass the “Comments to the Author” section, enter your
conflict of interest statement in the “Confidential to Editor” section, and submit
your "Accept" recommendation.

Reviewer #1: (No Response)

Reviewer #3: (No Response)

2. Is the manuscript technically sound, and do the data
support the conclusions?

Reviewer #1: No

Reviewer #3: No

3. Has the statistical analysis been performed
appropriately and rigorously? 

Reviewer #1: No

Reviewer #3: No

4. Have the authors made all data underlying the
findings in their manuscript fully available?

Reviewer #1: Yes

Reviewer #3: Yes

5. Is the manuscript presented in an intelligible
fashion and written in standard English?

Reviewer #1: No

Reviewer #3: No

6. Review Comments to the Author

Reviewer #1: None of my previous points of criticism have been addressed by the
author. I am not surprised because I essentially questioned that the work is
scientific. The work has not changed other than for some superficial cosmetics and I
uphold my assessment that the work is essentially typological and therefore
unscientific. Unfortunately, this is not an unusual case in systematic taxonomy, a
field which is teaming with pattern recognizers that work in a conceptual vacuum.
The mistake that the author makes, is to presume that simply distinguishing
something is the same as describing species. It is not! The something that the
author distinguishes can be anything from a group of individuals to a population,
perhaps even a species. But the complete lack of an intellectual assessment of what
the documented variation means in a biological sense makes it impossible to come to
a sensible conclusion.

The author goes to incredible lengths to amass information on differences and
similarities between groups of individuals of land snails collected at many
different sites in Brazil. My compliments for the diligence in collecting data.
However, that a comparison of different individuals yields observable differences
cannot be translated into a taxonomic classification without an assessment of what
these differences mean. There are characters that may well vary within populations.
For example, humans may have black, blond, or brown hair. To code hair colour in a
matrix that is designed to find the closest relative of humans is nonsense. And even
if it was not, it cannot inform us about the taxonomic rank that the closest
relative should be assigned to. For that, some sort conceptual framework is
required.

Now, the logic of the author is that using many, many characters will somehow
miraculously filter the wheat from the chaff. Yet, looking at fig. 61 any character
will do in justifying the 'distinctiveness of a species'. But this does not need to
be true. I don't even question that there are differences. The tree in Fig. 61
simply shows which studied individuals are more similar with one another, and which
are not. But as any other tree, it cannot inform about where the boundaries between
taxa are. For that we need additional criteria. These need to be formulated and they
need to make biological sense. That the individual called Kora ajar differs in a
number of characters from the one labelled as Kora nigra, in itself is not an
argument that they represent different species. In my previous review I provided a
detailed assessment of each of these characters and concluded that they do not
require that assumption to be made. I am not going to repeat myself here.

In general terms, we cannot know if the reported differences require the assumption
of specific distinction in any of these cases. That can only be decided when there
is at least an implicit species concept. The implicit species concept of the present
work is: If there is a difference, it is a different species.

Yet, the only objectively distinguishable unit in biology is the individual. From
that level on biological systems reveal fractal patterns of similarities and
differences that are shaped by different evolutionary processes. In this continuum,
we artificially create a hierarchical classification (an almost nonsensical idea!).
The created artificial classes should be reasonably broad and allow for variation,
or we run out of taxonomic categories very soon. Note also: Artificial does not mean
arbitrary. We need to accept that classifications are models of the objective truth,
but we need to adhere to sensible, generally accepted criteria to create this model,
otherwise we go out of business.

Above the level of an individual, the first layer is probably a family. The members
of a family are closely related, yet not morphologically identical. Using the method
of the author on ten human siblings and two cousins would create essentially the
same tree. Because we will find shared similarities between these individuals as
well as differences. (Look at those noses!) So why do we not classify these
individuals as distinct species?! Of course, this is utter nonsense and exaggerated.
But my point is, this essentially is what the author does - piling up observed
differences with no regard of whether these differences are useful to address the
problem at hand. This mix of similarities and differences continues to shape
populations, metapopulations, which eventually become perhaps genetically isolated
so we can call them species. The whole process is utterly arbitrary and inadequate
since we try to please our human desire to categorize stuff. In reality there are no
distinct borders unless shaped by extinction of intermediate forms.

Whatever conceptual consideration there may be, it has all been relegated to the
discussion chapter. Here, the observed differences are summarized for each taxon. I
suspect, however, that each description is based on a single individual (and even if
not, it is based on few individuals from the same population). Who says that (for
example) "the radular sac bulging in posterior end" is a useful character to
distinguish anything? Any character is influenced by individual variation relating
to genetics, nutritional status, reproductive status, age, preservation of the
animal etc. So, a bulging radular sac may mean NOTHING! Unless you confirm this
character is consistent in several specimens (more than 5 please) - and equally
consistently different in what you perceive to be a different species. Yet, even
then, this finding would not necessitate distinct species status. But at least, I
could understand where you are coming from. My previous assessment of how evidence
was treated was more comprehensive and has remained completely ignored. I won't
provide further details to support my claim but refer you to my previous review.

Now, as inevitably the taxonomic endeavor is highly subjective, one may argue that
the present work is just another way of distinguishing stuff and that stuff is
called by THIS author species and by MYSELF just a group of individuals. However, if
we accept that level of subjectivity, we have left the realm of science and entered
the realm of populism.

To avoid of getting to a place where taxonomy is completely uncoupled from scientific
theory (the work makes a big step in this direction) we need to have a concept of
what constitutes a species. That means, we need to make a reasoned assessment of
which differences are meaningful and which are not. Many of the differences that are
compiled in tables 1 and 2 are not biologically meaningful. That is because the
author does not allow for any variation among a species. Starting with shell
dimensions. A species does not have 45 mm long shells - no variation present. The
author claims to have studied 146 specimens, but does not document the range of
variation among these specimens even in such basic measures as shell dimensions.
Similarly, no variation is documented in any character. Now, I wouldn't expect a
huge deal of variation among individuals collected at exactly the same site. But I
would start to question my own work (and in fact I have done this!) when every snail
from a new place belongs to a new species. Species have ranges. They are not
restricted to a single site. Even if the geographic spread of the work is huge, it
is impossible that all species are single-site endemics. If this is the case, what
you call species are at best populations. If every population you study appears to
represent a distinct species then there are two possible scenarios. Either your
species are not species but simply populations, or you have entered an area that is
utterly understudied. It may well be that the second scenario is true. But then you
are prematurely elevating everything to species. What you need to do in this case is
work much harder. Go back and collect and spend another 5-10 years in documenting
biodiversity patterns. You just take the easy way out and leave it to the those that
come after you to validate your shamanic taxonomy.

How should it be done? Let's say that to distinguish species means at the least to
search for discontinuous distributions in character states. However, discontinuous
distributions can only be identified if the sampling allows for that (you need to
have a continuous sampling, so you can distinguish between discontinuity in
characters based on discontinuous sampling [your case] and discontinuity that is
real). So, for each presumed species, you would need to sample additional
populations to study the amount of variation between close populations and how this
variation spreads across spatial and environmental gradients. Ideally, we would seek
to identify regions where purported species overlap or contact and document the
discontinuity in the distribution of characteristics that are unlikely to be varying
a lot between individuals, such as hair colour.

If we cannot do that because our sampling is sparse, we cannot just go ahead and
speculate that every population we study belongs to a different species, simply
because we observed some sort of difference.

Anyway.. why do I feel the need to explain how systematic research should be done to
an accomplished author, who will simply be annoyed by my condemnation and shrug it
off? I know these taxa will be described simply because the way 'scientific
publishing works. It seems that everybody is entitled to their own truth. But not if
I can help it. Hence, I assess this work as NOT WORTH BEING PUBLISHED in its present
form. what is definitely worth being published is the rich data. Just take away the
new names, and I congratulate the author to a very important piece of work! Also,
you need to pay much more attention to document observed variation. Variation is the
food for evolutionary change. At some stage evolutionary change deserves recognition
in a taxonomic classification, but not at any level.

Reviewer #3: This manuscript provides a great deal of morphological information on
this relatively poorly known group of snails and includes useful data that is worthy
of publication. However, the author has failed to address the more important
critiques of this manuscript. Most of my criticisms focus on the ease of comparison
of species in the taxonomic section and the sparse taxon coverage in the
phylogenetic portion.

As mentioned in my earlier review. I dislike the organization of the taxonomic
discussion. The descriptions are very hard to compare between species because they
do not contain the same information. The manuscript seems to avoid clear statements
like Koltrora differs from Koro in x, y, z or this species differs from that species
in x, y , z. Clear, unambiguous comparisons need to be presented in a taxonomic
remarks section under every species. Each description should include the exact same
data for each taxon in the same order to allow readers to make easy comparisons. The
tables of species comparisons include measurement data without any measures of
variation making the comparisons between species meaningless. I do not see strong
data for recognizing Koltrora as a separate genus from Kora. The number of
morphological differences between Koltrora and Koro do not seem very different from
the differences within Kora species. I don’t think it makes sense to further
subdivide a group which is clearly not well sampled and where relationships are not
clearly resolved.

The author makes claims about the monophyly of Neopetraeus, Koro and Koltrora but has
not sampled Thaumastus even though a recent analysis based on sequence data suggests
Thaumastus may be sister to Kora and a fair amount of morphological data is
available for Thaumastus. The lack of sampling does not really allow the authors to
make strong claims about relationships.

The English usage while much better, still needs editing. I have not copy edited
extensively. A few examples are given below along with some other mostly minor
comments.

Abstract line 1, The definite arcticle ‘The’ should be deleted from the beginning of
the second sentence.

Third sentence change north and south to ‘northern Gerais state to southern
Bahia’.

Delete the sentence that starts ‘However, I remember…’ from the last paragraph of the
Introduction.

At the end of page 3 change ‘shell last whorl’ to ‘last whorl of the shell’.

At end of page 3 what do you mean by material? Make this clearer. Radulae were
extracted from buccal masses using potassium hypochlorite…

Beginning of page 4. Only the penis is everted. Also, the sentence including ‘despite
it is recognized that’ can be simplified to make it clearer.

Table 2. The shell measurements are not meaningful without some measure of
variation.

Table 2, ‘longer than wide’? All are longer than wide, I think this should be height
to width ratio.

Page 37 Distribution section. Sentence needs editing “Known only from the for
region”.

Page 37. Etymology section. Last sentence is not clear. Could rewrite as Color easily
seen through translucent shell.

7. PLOS authors have the option to publish the peer
review history of their article (what does this mean?). If published, this will
include your full peer review and any attached files.

If you choose “no”, your identity will remain anonymous but your review may still be
made public.

**Do you want your identity to be public for this peer review?** For
information about this choice, including consent withdrawal, please see our
Privacy Policy.

Reviewer #1: No

Reviewer #3: No

- - - - -

---

## [Author Response · Author response to Decision Letter 1]

9 Oct 2024

Response to reviewers

Prologue

After nearly a year of rigorous analytical work, I must admit that receiving a
negative decision on my manuscript is somewhat disappointing. I understand the
challenges faced by researchers, but I would like to emphasize how difficult it is
for someone from the Third World to secure full financial support for publication
fees, especially for a journal like PLOS. I was able to gather the necessary funds
by leveraging the prestige I have earned through over four decades of hard work,
resulting in more than 500 published papers, 300 of which are in international,
peer-reviewed journals (please see the curriculum website http://lattes.cnpq.br/5020466945403853).

Given this effort and to avoid wasting this opportunity, I hope you consider this
additional submission, as I am fully addressing Reviewer 3’s comments and most of
Reviewer 1’s concerns. Below, in an overview, I will briefly address some of the key
points raised, but I will answer every assertion from both reviewers after this
prologue:

Reviewer 3, like Reviewer 2 in the previous round, raised some contentious points. In
the previous round, I spent over a month revising the manuscript to address their
criticisms and suggestions, while also responding to some misunderstandings. Some of
the issues that Reviewer 3 raised as missing in this round, I corrected promptly in
this third phase, and I am confident that I can submit a revised version that fully
addresses their concerns. It is important to emphasize that reviewers 2 and 3 did
not share the same concerns as reviewer 1. They neither questioned the validity of
the taxa nor doubted the appropriateness of the methodologies I used to reach my
taxonomic and phylogenetic conclusions.

Thus, reviewer 1 presented more significant challenges. While I can modify some
aspects of the manuscript to meet his suggestions, other requests are impossible to
accommodate. As more minutely explained below, reviewer 1's main demand is to merge
an assemblage of three genera (two of which are already established in the
literature) and 14 species (five of which are also recognized in the literature,
occurring in a wide area, from Andes up to Bahia, Brazil) into a single species.
Additionally, he suggested disregarding a substantial body of data, including
conchological, morpho-anatomical, histological, and biogeographical evidence, as
well as forthcoming molecular results that strongly support the conclusions
presented. All of this data, including a well-supported phylogeny based on orthodox
(reproductible) methods, is included in the current manuscript, leaving no room for
doubt.

Moreover, reviewer 1 continues to accuse the paper of being "unscientific," which I
perceive as a personal attack. My entire career and all of my previous publications,
of my team and dozens of former students have followed the same methodological
approach, as do most papers in general Zoological Taxonomy. Reviewer 1 seems to have
a dogmatic perspective on science. It is important to note that not all science is
experimental. Taxonomy, like, e.g., astronomy, belongs to the observational
sciences, where phenomena are described rather than reproduced through experiments.
In taxonomy and anatomy, reproducibility is somewhat possible when another
researcher examines the same set of samples, but it cannot be reduced to personal
opinions or biases.

Additionally, the use of anatomical data to draw taxonomic or phylogenetic
conclusions is undoubtedly a valid scientific practice, one that can and should be
subject to criticism and potential falsification. However, such criticism must be
grounded in the study of the material itself, after understanding the used ordinary
methodology, not merely based on opinions drawn from different experiences, and with
other ecosystems. Moreover, the new referee's focus on individual specimens is
unsupported. This approach is neither aligned with current international taxonomic
practices nor addressed in the International Code of Zoological Nomenclature (ICZN),
which this paper strictly adheres to.

A perplexing situation arises because this manuscript mirrors a previous one, I
published in PLOS ONE (https://doi.org/10.1371/journal.pone.0273067), which follows the
same model, methodology, and philosophy. The only difference lies in the taxon. How,
then, can PLOS ONE have previously published a paper using this approach, only for
it to now be deemed "unscientific"?

The present documentation demonstrates that I fully addressed the Reviewer 3 (and
Reviewer 2’s) concerns and partially addressed Reviewer 1’s. I also believe that
Reviewer 1's report is biased, stemming from a personal issue, and I have responded
to each of their specific criticisms in this latest round of review.

It is important to emphasize that I cannot incorporate the molecular approach into
this paper at this stage for several reasons. First, it involves a different set of
authors. Second, the molecular study is still in progress and will require several
more months to complete. Finally, the current paper needs to be published first, as
it analyzes organic variation and establishes a foundational scenario that the
molecular data will later confront.

I would greatly appreciate the opportunity to submit the revised version of the
manuscript, and I am confident that you will not be disappointed with the outcome,
as demonstrated below.

Response to Reviewers' comments:

(Answers in RED)

Reviewer's Responses to Questions

Comments to the Author

3. Has the statistical analysis been performed appropriately and rigorously? 

Reviewer #1: No

Reviewer #3: No

The paper does not employ statistical analyses. The primary basis for obtaining data
on taxonomy and phylogeny in it was morpho-anatomy. This approach was sufficient, as
evidenced by the fact that the final result was a single cladogram. Therefore,
additional methods, such as morphometric or statistical analysis, were deemed
unnecessary. However, since both referees raised concerns regarding the
measurements, the revised version now includes more measured specimens (reported in
Table 1), with further explanations provided in the ‘Materials and Methods’ section
and throughout the Tables (more details below).

4. Have the authors made all data underlying the findings in their manuscript fully
available?

Reviewer #1: Yes

Reviewer #3: Yes

5. Is the manuscript presented in an intelligible fashion and written in standard
English?

PLOS ONE does not copyedit accepted manuscripts, so the language in submitted
articles must be clear, correct, and unambiguous. Any typographical or grammatical
should be corrected at revision, so please note any specific errors here.

Reviewer #1: No

Reviewer #3: No

As mentioned in the previous correction, the English in the paper was reviewed by two
individuals highly proficient in the language: Mauricio Uhle and José Coltro.
Additionally, the entire text was processed through three successive Artificial
Intelligence (AI) software tools. The final revision was completed using ChatGPT.
While I understand the text may not be flawless, it certainly meets the standard for
English. The only sections not reviewed by ChatGPT are the descriptions, as the
software does not handle telegraphic language. However, these parts were still
reviewed by both individuals and AI tools to catch any typographical or grammatical
errors.

6. Review Comments to the Author

Reviewer #3: This manuscript provides a great deal of morphological information on
this relatively poorly known group of snails and includes useful data that is worthy
of publication. However, the author has failed to address the more important
critiques of this manuscript. Most of my criticisms focus on the ease of comparison
of species in the taxonomic section and the sparse taxon coverage in the
phylogenetic portion.

As mentioned in my earlier review. I dislike the organization of the taxonomic
discussion. The descriptions are very hard to compare between species because they
do not contain the same information. The manuscript seems to avoid clear statements
like Koltrora differs from Koro in x, y, z or this species differs from that species
in x, y , z. Clear, unambiguous comparisons need to be presented in a taxonomic
remarks section under every species. Each description should include the exact same
data for each taxon in the same order to allow readers to make easy comparisons.

To address this request, a 'Taxonomical Remarks' section has been added for all Kora
species as well as Koltrora, as can be seen in the new version. This addition
required nearly two extra pages per species, increasing the length of the paper by
over 20 pages. These remarks were not included previously because all distinctions
had already been detailed in various sections of the paper, such as the diagnoses,
the distinctive descriptions, the comparative tables, the final Discussion, and the
phylogenetic analysis. In that original context, adding 'Taxonomical Remarks' seemed
redundant. However, this has been rectified in the new version, which now includes a
thorough differentiation of each species based on shell characteristics, general
anatomy, and the genital system, as reviewer 3 wants.

 The tables of species comparisons include measurement data without any measures of
variation making the comparisons between species meaningless.

Regarding the measurements, I have increased the number of shells measured (see Table
1, column 4), and calculated averages to address the referee’s concerns. In Table 2,
column 2, these issues have been specifically addressed, although the categories in
which each species was arbitrarily classified remain unchanged. These categories
were necessary for the phylogenetic analysis, as it does not accommodate continuous
parameters. This is explained in more detail in the Materials and Methods section,
as well as in the tables that include any measurements.

 I do not see strong data for recognizing Koltrora as a separate genus from Kora. The
number of morphological differences between Koltrora and Koro do not seem very
different from the differences within Kora species. I don’t think it makes sense to
further subdivide a group which is clearly not well sampled and where relationships
are not clearly resolved.

The number and degree of differences between Kora and Koltrora are too significant to
justify merging the two taxa. To reinforce the case for this generic separation, I
added a nearly one-page 'Taxonomical Discussion' at the end of the genus description
and expanded the generic comparison in the Discussion section. In the former, I
highlight key conchological and anatomical differences that make it impossible to
treat both taxa as one. I also point out unique characteristics in Koltrora that are
absent in Kora or related taxa. I am confident that, upon reading these sections,
the referee will be convinced that these are indeed distinct genera. Additionally,
Koltrora does not fit within any known genus.

The author makes claims about the monophyly of Neopetraeus, Koro and Koltrora but has
not sampled Thaumastus even though a recent analysis based on sequence data suggests
Thaumastus may be sister to Kora and a fair amount of morphological data is
available for Thaumastus. The lack of sampling does not really allow the authors to
make strong claims about relationships.

 The case of Thaumastus, if it is related to Kora, is actually addressed in two
sections of the paper: the Introduction and the Discussion (under the molecular
approaches subsection). This is indeed a valuable point, and there are plans to
include Thaumastus in a future paper for a more detailed analysis. However, it is
not highly essential to the current study, because both genera are very
distinct.

There is some anatomical information on Thaumastus, primarily from Meire Pena’s
papers. However, as with the most recently described Kora by her (reported in this
new version), only a gross anatomical overview is available, with insufficient
detail for inclusion in this study. Fundamental anatomical details are still
lacking. Thus, in the present level of knowledge, the inclusion of Thaumastus is not
possible.

The English usage while much better, still needs editing. I have not copy edited
extensively. A few examples are given below along with some other mostly minor
comments.

Abstract line 1, The definite arcticle ‘The’ should be deleted from the beginning of
the second sentence. Done

Third sentence change north and south to ‘northern Gerais state to southern Bahia’.
Done

Delete the sentence that starts ‘However, I remember…’ from the last paragraph of the
Introduction. Done

At the end of page 3 change ‘shell last whorl’ to ‘last whorl of the shell’. Done

At end of page 3 what do you mean by material? Make this clearer. Radulae were
extracted from buccal masses using potassium hypochlorite… Not exactly. The radula
was extracted during the odontophore dissection for its understanding, the
hypochlorite is used only for cleaning the isolated radula only. That part was
rephrased.

Beginning of page 4. Only the penis is everted. Also, the sentence including ‘despite
it is recognized that’ can be simplified to make it clearer. Done

Table 2. The shell measurements are not meaningful without some measure of variation.
As reported above, to address this criticism, (1) I increased the number of measured
shells (as reported in Table 1). (2) I included the average and range values as
absolute measures, which form the basis for the size range categorization in column
2 of Table 2. (3) In all data columns of Table 2, I clarified that the reported
values are averages, rounded to the nearest whole number. Since this paper does not
focus on statistical morphometric conclusions, these approximations do not affect
the final results, but they are crucial for supporting the character categorization
in the phylogenetic analysis, which is a key part of the study.

Table 2, ‘longer than wide’? All are longer than wide, I think this should be height
to width ratio. Replaced.

Page 37 Distribution section. Sentence needs editing “Known only from the for
region”. Done

Page 37. Etymology section. Last sentence is not clear. Could rewrite as Color easily
seen through translucent shell. Done

Reviewer #1: None of my previous points of criticism have been addressed by the
author. I am not surprised because I essentially questioned that the work is
scientific. The work has not changed other than for some superficial cosmetics and I
uphold my assessment that the work is essentially typological and therefore
unscientific. Unfortunately, this is not an unusual case in systematic taxonomy, a
field which is teaming with pattern recognizers that work in a conceptual vacuum.
The mistake that the author makes, is to presume that simply distinguishing
something is the same as describing species. It is not! The something that the
author distinguishes can be anything from a group of individuals to a population,
perhaps even a species. But the complete lack of an intellectual assessment of what
the documented variation means in a biological sense makes it impossible to come to
a sensible conclusion. 

The entire paper, now comprising 89 pages and 61 plates, is p

---

## [Decision Letter · Decision Letter 2]

31 Oct 2024

PONE-D-23-41760R2Review of the land snails
of the genus Kora from Brazil, with description of eight new species and a new
related genus Koltrora, including comparison with two Andean Neopetraeus species
(Gastropoda, Eupulmonata, Orthalicoidea).PLOS ONE

Dear Dr. Simone,

Thank you for submitting your manuscript to PLOS ONE. After careful consideration, we
feel that it has merit but does not fully meet PLOS ONE’s publication criteria as it
currently stands. Therefore, we invite you to submit a revised version of the
manuscript that addresses the points raised during the review process.

I have carefully reviewed the submission history of your manuscript, including the
suggestions made by various reviewers and the editor in previous rounds. First and
foremost, I would like to apologize for the lengthy and demanding review process, as
well as for the time it has required on your part. In recognition of the rigor of
this process, I requested two additional reviewers, who had no prior knowledge of
your work, to assess the manuscript anew. For this current round of review, both the
reviewers and I, as the editor, are distinct from those previously involved with the
manuscript.

With the completion of these two reviews, along with my own assessment, I am pleased
to inform you that I believe your manuscript is indeed suitable for publication.
However, it is clear that there are still some revisions and suggestions that need
to be addressed, with particular attention to the comments made by Reviewer 4.
Reviewer 4 has provided critical feedback that, in my assessment, should be
implemented to meet PLOS ONE’s publication standards.

Please let me know if you have any questions regarding the feedback or the next steps
in the revision process. I look forward to seeing the revised version of your
work.

Please submit your revised manuscript by Dec 15 2024 11:59PM. If you will need more
time than this to complete your revisions, please reply to this message or contact
the journal office at plosone@plos.org. When
you're ready to submit your revision, log on to https://www.editorialmanager.com/pone/ and select the 'Submissions
Needing Revision' folder to locate your manuscript file.

Please include the following items when submitting your revised manuscript:A rebuttal letter that responds to each point raised by the academic
editor and reviewer(s). You should upload this letter as a separate file
labeled 'Response to Reviewers'.A marked-up copy of your manuscript that highlights changes made to the
original version. You should upload this as a separate file labeled
'Revised Manuscript with Track Changes'.An unmarked version of your revised paper without tracked changes. You
should upload this as a separate file labeled 'Manuscript'.

If you would like to make changes to your financial disclosure, please include your
updated statement in your cover letter. Guidelines for resubmitting your figure
files are available below the reviewer comments at the end of this letter.

We look forward to receiving your revised manuscript.

Kind regards,

Wesley Dondoni Colombo

Academic Editor

PLOS ONE

Journal Requirements:

Additional Editor Comments (if provided):

Reviewers' comments:

Reviewer's Responses to Questions

**Comments to the Author**

1. If the authors have adequately addressed your comments raised in a previous round
of review and you feel that this manuscript is now acceptable for publication, you
may indicate that here to bypass the “Comments to the Author” section, enter your
conflict of interest statement in the “Confidential to Editor” section, and submit
your "Accept" recommendation.

Reviewer #4: (No Response)

Reviewer #5: All comments have been addressed

2. Is the manuscript technically sound, and do the data
support the conclusions?

Reviewer #4: Partly

Reviewer #5: Yes

3. Has the statistical analysis been performed
appropriately and rigorously? 

Reviewer #4: No

Reviewer #5: N/A

4. Have the authors made all data underlying the
findings in their manuscript fully available?

Reviewer #4: Yes

Reviewer #5: Yes

5. Is the manuscript presented in an intelligible
fashion and written in standard English?

Reviewer #4: Yes

Reviewer #5: Yes

6. Review Comments to the Author

Reviewer #4: The manuscript deals with the revision of a previously created genus of
land snails with the addition of a set of new species and the creation of another
new genus with a single species. All this information is presented in a cladistic
analysis with a particular methodology. This work follows the same scheme and
methodology as another contribution by the same author previously published in PLOS
ONE. Although this is an indisputable fact, contrary to what the author suggests,
criticisms can still be made of the method used and the format employed. The author
of the present manuscript is a malacologist renowned for his extensive experience in
marine gastropods. On this occasion, the manuscript he wrote deals with species of
terrestrial pulmonate gastropods and this is evident in the framework used and the
selection of characters for the description of species.

As has been stated for some time and only to cite a single author on this subject:
Morphology-based cladistics, the birthplace of cladistic methods, has been attacked
by molecular systematists as hopelessly ambiguous and inevitably dispensable in an
era increasingly awash in molecular data (Sereno, 2009). In complete agreement with
Sereno, I consider that morphological studies as well as cladistics based on
morphology are enormously useful tool and should not be discredited to achieve the
advancement of science.

At the same time, this kind of study must be rigorous and repeatable, as well as they
have to provide predictable hypotheses. The present manuscript, after having already
gone through revisions, has been improved since its initial version. It denotes a
detailed study of the species involved, with extensive descriptions of organs not
generally used after the publication of Tiller (1989). Tillier studied morphological
characters from the nervous, pallial and digestive systems in a comparative way of
different Pulmonate gastropod groups. This mentioned work is a classic, but received
many criticisms for the phylogenetic method used to address its results, in addition
to revealing that these characters are highly homoplasic, so in general, only those
of the reproductive system continued to be used in taxonomy and phylogeny because
are more informative.

Following there is a list of some points for the author to address:

Abstract:

-“a calcified epiphragm, unique in South American snails.” This is incorrect as there
are other genera with calcified epiphragm inhabiting South America, please change
the sentence.

Results:

All the species descriptions are very detailed, but I found the following
problems:

1. The diameter of glandular organs is cited as a distinguishing species
characteristic. However, glandular organs undergo significant changes due to
seasonality, making this trait unreliable for species comparisons—unless a
possibility would be that all specimens were collected and preserved during the same
season, which is not specified in this work.

2. The Sperm inner longitudinal fold is a character described in different species
and shown through a drawing of a transverse section of the spermoviduct. However,
since histology has not been carried out I wonder how he managed to see such a
detailed aspect in the interior glandular organ. Not only its morphology but also
how it is inserted and connected to the albumen gland.

3. In the same way, there is apparently a problem in the identification/delimitation
of organs and the terminology used by the author since he describes and equals the
“Carrefour” with the fertilization pouch-spermathecal complex organ. The Carrefour
is a portion basal to that structure and usually it is embedded into the albumen
gland, for which only with histological methods is possible to see how it is
connected either to the hermaphroditic duct or to the albumen gland/spermividuct.
See Tompa (1984) for a better description of this area. Figure 11C for example shows
a fertilization pouch-spermathecal complex term as “Carrefour” by the author with a
long basal tube described as “simple, wide, ~twice longer than receptacle and
3-times its width”. It is not clear to me if this an inner duct (embedded into the
albumen gland) or if it is external. Is it consider as part of the “carrefour” in
the present manuscript?

4. Taxonomic remarks are excessively long, often repeating characters already
described for each species. They should be more concise highlighting the differences
of main characters.

5. Pag. 46: author said: …because it lacks the reticulated sculpturing seen in the
protoconchs of Bulimulus. This is incorrect since species of Bulimulus lack of a
reticulate Protoconch.

6. It is not explained why or for what purpose two cladistic softwares such as TNT
and PAUP were used or if differences were found between them. Nor is it indicated
which commands and strategy were used for the phylogenetic analysis performed.
Regardless of establishing that a method already published is used, the author must
indicate which particular strategy was used for the analysis of this data.

7. Another interesting feature is the presence of a thick muscular region in the
basal portion of the duct of the bursa copulatrix (e.g., Fig. 6A: bu)” A better
drawing or photo is needed here because Fig. 6A is not clear in indicating the basal
area of the duct.

8. “A second diverticulum-like projection, usually balloon-like, also filled with
albumen and interpreted as an accessory albumen chamber (as), is a novelty in the
presently studied species, being absent only in Koltrora”. The terminology used is
confusing since this area of union between the albumen gland and the spermatoviduct
is present in many Pulmonate snails, but does not have a projection shape nor is it
a diverticulum.

Discussion:

Taxonomic discussion

The first and second paragraphs of the discussion should be compacted because they
are repetitive and seem to be more of a justification of what was done in the work.
Throughout the text, there are also many repetitions of what was already said in
Results and at the same time in the same Discussion they are repeated at least
twice. The text should be more concise discussing the results obtained concerning
what was previously known. In turn, I find it difficult to follow the excessively
detailed comparisons of the musculature of the buccal mass and odontophore whose
importance is lost when the information is not hierarchical.

The author sustained that “Importantly, no exclusive diagnostic character was
obtained, neither at the genus nor at the species levels”, Does it refer to not
having identified autapomorphies? How do you interpret the set of diagnostic
characters detailed in each case? This is contradictory to the cladistic analysis
carried out as well as to the identification of a set of particular characters for
each genus.

Please avoid using the following words as they are totally subjective and only
meaningful in a narrative context: Importantly, interesting (used several times),
relatively, mostly, extraordinary, impressively, Notably, enigmatic.

-“The ureter is totally closed (tubular) in Kora and Koltrora, as apparently is the
usual pattern of Orthalicoideans but Neopetraeus has the ureter opened in a larger
degree”. This is incorrect since there are other genera classified in Orthalicoidea
with this same characteristic.

-“Koltrora differs from Kora….by the thick glandular uterine walls (Fig. 47E: ut); by
the epiphallus being widely open to the penis”. The first character is inappropriate
since a glandular reproductive organ varies seasonally in its diameter/thickness.
The second mentioned character is not clear since the relationship of the epiphallus
with the penis is not clarified nor is it detectable in the figures.

-In diagnosing Kora, the author includes characters shared with Koltrora and
Neopetraeus. Only those characters unique to Kora that justify its difference from
the other genera should be mentioned.

-“Concerning the odontophore, all species exhibit fused cartilages along the ventral
edge, but the degree of fusion varies. Koltrora has approximately 100%, while K.
uhlei has around 90%; K. nigra has the smallest fusion value of approximately 60%,
and the remaining species are about 75% fused.” How is the degree of fusion in % of
the cartilages of the odontophore measured? The differences in % are very subtle and
merit an explanation of how they were obtained.

-What does it mean that the m8 muscles are relatively uniform??

-The author sustained that “The carrefour region, for example, has a unique
arrangement in each species (Figs 6B, 11C, 15A, 20E, 23E, 27E, 33E, 38A, 47C, 53D,
57B).” However, the mentioned figures show significant similarities in the
morphology of the organ between Fig.11C and 15A, between 20E and 33E, also between
Fig 23E and 27E making this assertion doubtful or at least these figures are not
sufficiently clear to show the difference that the author maintains.

The title “Biogeographic analysis” should be changed to “Distribution” since no
analysis was performed.

Cladistic Analysis

The goal in cladistics is not always to find a single tree but to identify the most
likely and robust relationships, which may involve accepting that there are multiple
equally valid phylogenetic hypotheses. Finding multiple equally parsimonious trees
is a common situation in cladistics, and although it can present some challenges,
this does not imply that the analysis is flawed or useless. Parsimony is one of the
criteria for evaluating phylogeny, and finding multiple parsimonious trees means
that there are several equally valid hypotheses based on the current data. It may
reflect the reality of the evolutionary data. The key is how to interpret and handle
this situation.

The tree obtained in the phylogenetic analysis is described in the Discussion, not in
the Results section. This should be migrated to Results and only a discussion of the
relationships found in the proposed hypothesis should be made.

Some characters that result in synapomorphies in the analysis carried out are
doubtful due to what was previously explained about "carrefour" and its
delimitation, for example. Character64.

The author sustained that in node 12, 11 synapomorphies support its monophyly and
then mentioned the “interesting” ones which in fact are the total set of
synapomorphies. Why they are “interestings”? It is not informative to indicate that
they are interesting, since having resulted in synapomorphic characters they support
the monophyly of the clade. If you wanted to say something more about each of them,
you should specify it in another way.

In page 62 the author mentioned that Koltrora pyrostoma has “13 homoplastic
autapomorphies, being 4 of them reversions in the present context; two of them are
noteworthy: the loss of the accessory albumen chamber (ch. 70) and of the penis

shield (ch. 80). This set of autapomorphies, and the lack of the Kora synapomorphies,
are

sufficient reasons for the generic separations of both taxa, despite they share more
characters

than Neopetraeus. Please explain why the absence of characters are enough to define
the monophyly of a genus. Again in the same page, the author says: The Kora branch
(node 13) has the extraordinary support of 25 synapomorphies. The more interesting
are: characters 6,7,15,18,29, 36,37,45,47,57,59,62,74,78,83,94”. Why is that? Are
there synapomorphies? Which criteria follows the author to consider them
“interesting” characters?

Explanations are given in the Discussion section of phylogenetic relationships of
non-Orthalicoidean groups (such as Solaropsidae) considered as outgroups in the
analysis performed, although the author himself clarifies that his phylogeny does
not attempt to resolve relationships outside his ingroup. Inferences are also made
about relationships of genera such as Drymaeus and Bulimulus that have few or single
representatives in the cladogram. These inferences should be removed because they
may be speculative and based on little evidence. The author should focus on
discussing only his results with previous phylogenies.

In page 65 the author sustains that: “With such fluidity, maintaining taxonomic
stability becomes challenging, and it poses a significant obstacle to achieving
mental coherence in the classification trustworthiness”. I would like to suggest to
the author another point of view explained by Dominguez & Wheeler (1987) in
their paper “Taxonomic stability is Ignorance”. They argued that stability should
not be prioritised over accuracy and that sticking to outdated classifications for
the sake of consistency reflects ignorance of evolving scientific knowledge.
Therefore, the ultimate goal of taxonomy should be to reflect the best possible
understanding of evolutionary relationships, even if it means frequent
revisions.

Main evolutive processes

Page 66: “Despite being preliminary, this marks the first attempt to construct an
orthalicoid

phylogeny based solely on phenotypic features—a crucial initial step toward a
broader

understanding” Although I agree with the disconnection of morphology with the latest
published molecular analyses, considerations of phylogenetic relationships cannot be
made using a restricted morphological phylogeny of such an hyperdiverse superfamily
as Orthalicoidea using a low number of representative species.

I would suggest to eliminate this title from the Discussion section.

Reviewer #5: I commend the author for the work done, the detail in the descriptions,
and the composure in responding to each of the reviewers' suggestions.

7. PLOS authors have the option to publish the peer
review history of their article (what does this mean?). If published, this will
include your full peer review and any attached files.

If you choose “no”, your identity will remain anonymous but your review may still be
made public.

**Do you want your identity to be public for this peer review?** For
information about this choice, including consent withdrawal, please see our
Privacy Policy.

Reviewer #4: No

Reviewer #5: No

---

## [Author Response · Author response to Decision Letter 2]

15 Nov 2024

6. Review Comments to the Author

Reviewer #4: The manuscript deals with the revision of a previously created genus of
land snails with the addition of a set of new species and the creation of another
new genus with a single species. All this information is presented in a cladistic
analysis with a particular methodology. This work follows the same scheme and
methodology as another contribution by the same author previously published in PLOS
ONE. Although this is an indisputable fact, contrary to what the author suggests,
criticisms can still be made of the method used and the format employed. The author
of the present manuscript is a malacologist renowned for his extensive experience in
marine gastropods. On this occasion, the manuscript he wrote deals with species of
terrestrial pulmonate gastropods and this is evident in the framework used and the
selection of characters for the description of species. It is important to emphasize
that I am not opposed to criticism. No scientific procedure is immune to it, and
when criticisms are well-intentioned and constructive, they are very welcome. Over
more than 40 years, they have contributed significantly to shaping the professional
I am today. However, Reviewer 1’s comments fall far from this standard. I won’t
reiterate all the arguments supporting this perspective here; please refer to the
previous documentation, where I thoroughly countered his claims, including those
labeling my work as unscientific.

Furthermore, the above assertion suggests that I am a neophyte in the study of land
mollusks, with more extensive experience in marine gastropods. This impression is
incorrect. As detailed on my curriculum website (http://lattes.cnpq.br/5020466945403853), I have authored 640
published works, including 300 papers in peer-reviewed international journals, many
of which are highly regarded, such as PLOS, Zoological Journal, and Zootaxa. I have
published extensively across all branches of Mollusca, with marine and terrestrial
gastropods comprising the majority of my work in equal measure.

A genuine challenge, as Reviewer 6 correctly points out, lies in the phylogenetic
complexities of land snails, most of which are Heterobranchia. Their phylogeny is
indeed less developed compared to other gastropod groups. In the current paper, I
could not delve as deeply into phylogenetic analysis as I typically do with, for
instance, caenogastropods. Nonetheless, we can advance our understanding by doing
our best to achieve meaningful results, sufficient to address the taxonomy level
discussed in the paper.

As has been stated for some time and only to cite a single author on this subject:
Morphology-based cladistics, the birthplace of cladistic methods, has been attacked
by molecular systematists as hopelessly ambiguous and inevitably dispensable in an
era increasingly awash in molecular data (Sereno, 2009). In complete agreement with
Sereno, I consider that morphological studies as well as cladistics based on
morphology are enormously useful tool and should not be discredited to achieve the
advancement of science. It is a pleasure to hear that. I totally agree with
this.

At the same time, this kind of study must be rigorous and repeatable, as well as they
have to provide predictable hypotheses. The present manuscript, after having already
gone through revisions, has been improved since its initial version. It denotes a
detailed study of the species involved, with extensive descriptions of organs not
generally used after the publication of Tiller (1989). Tillier studied morphological
characters from the nervous, pallial and digestive systems in a comparative way of
different Pulmonate gastropod groups. This mentioned work is a classic, but received
many criticisms for the phylogenetic method used to address its results, in addition
to revealing that these characters are highly homoplasic, so in general, only those
of the reproductive system continued to be used in taxonomy and phylogeny because
are more informative. Tillier (1989) remains a foundational reference for my work.
However, because it is quite general and does not focus extensively on
orthalicoideans—a group only weakly represented in that paper—I did not cite it in
the present study. The notion that non-genital structures are highly homoplastic and
of limited use in phylogenetic analysis is, in my view, a misconception or dogma
created to justify the shift towards a purely sequencing-based approach. My research
has consistently demonstrated otherwise.

Following there is a list of some points for the author to address:

Abstract:

-“a calcified epiphragm, unique in South American snails.” This is incorrect as there
are other genera with calcified epiphragm inhabiting South America, please change
the sentence. Okay, it was rephrased[*correction001]. However, neither I, nor any
colleagues I have consulted, know of any other group of native South American snails
that possess a calcified epiphragm. While several groups, such as the
Epiphragmophoridae, do produce an epiphragm, it is, however, typically only
minimally calcified or non-calcified. I thank any more precise information.

Results:

All the species descriptions are very detailed, but I found the following
problems:

1. The diameter of glandular organs is cited as a distinguishing species
characteristic. However, glandular organs undergo significant changes due to
seasonality, making this trait unreliable for species comparisons—unless a
possibility would be that all specimens were collected and preserved during the same
season, which is not specified in this work. I understand that glandular organs and
other structures can vary with seasonal changes. However, I cannot omit details
about their size and proportions in the descriptions. To address potential issues
arising from this variability, I typically describe the average size of these organs
across specimens. Furthermore, I either avoid using size and proportions in
comparative analyses or use them minimally, and I never rely on these
characteristics as diagnostic factors.

2. The Sperm inner longitudinal fold is a character described in different species
and shown through a drawing of a transverse section of the spermoviduct. However,
since histology has not been carried out I wonder how he managed to see such a
detailed aspect in the interior glandular organ. Not only its morphology but also
how it is inserted and connected to the albumen gland. The detailed anatomical
investigation protocols used in this paper are the same as those I have applied in
all my previous studies, as explained in the references cited in the Material and
Methods section. Although proper histological analysis was not performed (as that
constitutes a different type of study), however, serial sections were used to reveal
certain minute and complex structures. In this paper, serial sections were only made
for a few specimens, as the specimens were large enough to be thoroughly examined
through dissection. With this protocol, it is entirely possible to observe and
accurately illustrate these structures.

Regarding the folds in the spermoviduct, they can be identified through detailed
dissection, as demonstrated in the attached photo of Kora nigra (indicated by
arrows), which bears two distinct folds. Serial sections are not required to observe
these features; the folds are even visible running along the length of the
spermoviduct when it is opened longitudinally. For example, see Fig. 6A (sp) in the
paper.

3. In the same way, there is apparently a problem in the identification/delimitation
of organs and the terminology used by the author since he describes and equals the
“Carrefour” with the fertilization pouch-spermathecal complex organ. The Carrefour
is a portion basal to that structure and usually it is embedded into the albumen
gland, for which only with histological methods is possible to see how it is
connected either to the hermaphroditic duct or to the albumen gland/spermividuct.
See Tompa (1984) for a better description of this area. Figure 11C for example shows
a fertilization pouch-spermathecal complex term as “Carrefour” by the author with a
long basal tube described as “simple, wide, ~twice longer than receptacle and
3-times its width”. It is not clear to me if this an inner duct (embedded into the
albumen gland) or if it is external. Is it consider as part of the “carrefour” in
the present manuscript? The terminology used for pulmonate snails is not uniform
across different schools of thought, resulting in significant variation. I have
consistently used the terminology established in my previous papers, which has been
influenced by Leme, Scott, Paraense, and others. In this context, "carrefour" refers
to the "region of fertilization," “talon,” or "crossing," as described by some
authors. It is important to maintain terminological consistency based on my previous
work. Additionally, yes, most of the carrefour is located within the albumen gland
and requires dissection of it to access it; serial sections or histological analysis
are not necessary. The detailed dissection protocols I applied are sufficient to
fully expose it, as shown in Fig. 11C. Fig. 11A provides a natural view of the
carrefour still immersed in the albumen gland (ag). I have numerous photos
documenting these dissections if the reviewer requires further evidence. Annexed, I
have included a photo that illustrates a stage of the dissection that produced Fig.
11C, showing the exposed carrefour (arrow) and its connection to the beginning of
the spermoviduct.

Above, I presented my rationale for using the term "carrefour" instead of
"fertilization pouch-spermathecal complex organ," to maintain consistency with my
previous papers. However, I understand that this is a request, and if the reviewer
believes it is important to make this terminological change, I am open to it at this
stage of the review. I only ask for permission to document this terminological
change in the Material and Methods section.

4. Taxonomic remarks are excessively long, often repeating characters already
described for each species. They should be more concise highlighting the differences
of main characters. I fully agree with this point. The taxonomic remarks were not
present in the first two versions of the paper but were introduced in the third
version to address the requirements of Reviewer 3. In my opinion, the species
comparisons and discussions were already thoroughly addressed, ad nauseam, in the
Discussion section, as well as in the tables and phylogenetic analyses. Therefore, I
believe they can be completely omitted. However, I defer to the Editor and Reviewer
on this matter, and I am fine with their deletion.

Therefore, I leave it to the Editor to decide whether I should address Reviewer 3's
request and keep the taxonomical remarks or follow Reviewer 6's suggestion and
remove them. I prefer the solution proposed by Reviewer 6.

5. Pag. 46: author said: …because it lacks the reticulated sculpturing seen in the
protoconchs of Bulimulus. This is incorrect since species of Bulimulus lack of a
reticulate Protoconch. Ok, rephrased[*correction002]. ‘Wrinkles’ was introduced as a
character of Bulimulus’ protoconch based on Breure (1979).

6. It is not explained why or for what purpose two cladistic softwares such as TNT
and PAUP were used or if differences were found between them. Nor is it indicated
which commands and strategy were used for the phylogenetic analysis performed.
Regardless of establishing that a method already published is used, the author must
indicate which particular strategy was used for the analysis of this data. Ok, I
introduced an additional paragraph at the end of the item “Phylogenetic analysis” in
the Material and Methods to attend this requirement[*correction003].

7. Another interesting feature is the presence of a thick muscular region in the
basal portion of the duct of the bursa copulatrix (e.g., Fig. 6A: bu)” A better
drawing or photo is needed here because Fig. 6A is not clear in indicating the basal
area of the duct. Yes, that character is very interesting, and it appears clearly in
Fig. 6A, where the cut basal walls of the bursa duct (indicated by “bu”) are much
thicker than those of the preceding section. Additionally, this thick region (bu) is
also illustrated in Figs. 11A, 12N (a photograph), 20B, 23C, and 38B.

8. “A second diverticulum-like projection, usually balloon-like, also filled with
albumen and interpreted as an accessory albumen chamber (as), is a novelty in the
presently studied species, being absent only in Koltrora”. The terminology used is
confusing since this area of union between the albumen gland and the spermatoviduct
is present in many Pulmonate snails, but does not have a projection shape nor is it
a diverticulum. In the region between the albumen gland and the beginning of the
spermoviduct, there is typically an albumen chamber, a sac where the albumen gland
stores albumen. The Kora also possesses this structure. The “accessory albumen
chamber” is an additional feature, located slightly more anteriorly and connected by
a different duct. In the figures, the usual albumen chamber, found in most
pulmonates, is labeled as ‘ac,’ while the accessory one, a novelty of Kora, is
labeled ‘as.’ The term “balloon-like” describes its shape, as it usually features a
narrow duct leading to a bulged, swollen distal portion that resembles a flying
balloon. It is classified as a diverticulum because any blind sac that projects from
a tube in anatomy is termed a “diverticulum.” This terminology is commonly used in
anatomy, to my knowledge. These terms seem clear to me, but I am happy to make
changes if any clarification is requested.

Discussion:

Taxonomic discussion

The first and second paragraphs of the discussion should be compacted because they
are repetitive and seem to be more of a justification of what was done in the work.
Throughout the text, there are also many repetitions of what was already said in
Results and at the same time in the same Discussion they are repeated at least
twice. The text should be more concise discussing the results obtained concerning
what was previously known. In turn, I find it difficult to follow the excessively
detailed comparisons of the musculature of the buccal mass and odontophore whose
importance is lost when the information is not hierarchical. These paragraphs were
included as responses to previous assertions made by the previous referees. In light
of the current criticisms, they have been deleted [*correction004]. Relevant
information from these sections has been transferred to the Material and Methods
section[*correction005].

The author sustained that “Importantly, no exclusive diagnostic character was
obtained, neither at the genus nor at the species levels”, Does it refer to not
having identified autapomorphies? How do you interpret the set of diagnostic
characters detailed in each case? This is contradictory to the cladistic analysis
carried out as well as to the identification of a set of particular characters for
each genus. Despite complementary, the taxonomical and phylogenetic scenarios are
different in papers. Yes, there are indeed several autapomorphies present among all
examined taxa. The translucent shell of Koltrora serves as one example. It is indeed
a diagnostic character, however, autapomorphies are not utilized in phylogenetic
analyses because they do not contribute to resolving evolutionary relationships and
are typically excluded in the final stages to prevent index inflation. The only
autapomorphies retained—and those are depicted in Fig. 61—are those that turned out
to be homoplastic or transformational. The intent of the explanatory statement is to
convey that no character remains entirely exclusive; instead, characters show
convergence with other orthalicoideans. For example, shell mentioned translucency,
while characteristic of Koltrora, also appears in the unrelated simpulopsid
Leiostracus perlucidus.

Even the acce

to Reviewers.docx

---

## [Editor Report · Decision Letter 3]

25 Nov 2024

Review of the land snails of the genus Kora from Brazil, with description of eight
new species and a new related genus Koltrora, including comparison with two Andean
Neopetraeus species (Gastropoda, Eupulmonata, Orthalicoidea).

PONE-D-23-41760R3

Dear Dr. Simone,

We’re pleased to inform you that your manuscript has been judged scientifically
suitable for publication and will be formally accepted for publication once it meets
all outstanding technical requirements.

Kind regards,

Wesley Dondoni Colombo

Academic Editor

PLOS ONE

Additional Editor Comments (optional):

I have carefully analyzed your revised manuscript, particularly given its long and
complex revision history. This in-depth assessment allowed me to appreciate the
scientific rigor and persistence you have demonstrated throughout the review
process.

I wholeheartedly agree with your statement that morphological studies, along with
morphology-based cladistics, remain invaluable tools for advancing scientific
understanding. Despite the increasing focus on molecular data, your work highlights
the critical role that morphological and anatomical studies play in deepening our
knowledge of biodiversity. This is especially significant in taxonomic and cladistic
frameworks, which often face unwarranted criticism in contemporary scientific
discourse.

Upon review, I find that the majority of the reviewers' comments have been addressed
comprehensively. For the points where you diverged from the reviewers' suggestions,
your responses and justifications were clear, well-reasoned, and sufficient to
substantiate your choices.

Regarding the taxonomic remarks, which you left to my discretion, I agree with
Reviewer 3. I find these remarks to be a valuable addition to your paper and
consider them important for maintaining the depth and comprehensiveness of your
taxonomic work. I recommend that they be retained in the final manuscript.

In conclusion, I am pleased to inform you that I consider your manuscript ready for
publication in PLOS ONE. Congratulations on this accomplishment, which represents a
significant contribution to malacology and the study of land snails.
---

## [Editor Report · Acceptance letter]

29 Nov 2024

PONE-D-23-41760R3 

PLOS ONE

Dear Dr. Simone, 

I'm pleased to inform you that your manuscript has been deemed suitable for
publication in PLOS ONE. Congratulations! Your manuscript is now being handed over
to our production team.

Kind regards, 

on behalf of

Dr. Wesley Dondoni Colombo 

Academic Editor

PLOS ONE